# Text-to-Image Diffusion Models Cannot Count, and Prompt Refinement Cannot Help

## Abstract

Generative modeling is widely regarded as one of the most essential problems in today's AI community, with text-to-image generation having gained unprecedented real-world impacts. Among various approaches, diffusion models have achieved remarkable success and have become the de facto solution for text-to-image generation. However, despite their impressive performance, these models exhibit fundamental limitations in adhering to numerical constraints in user instructions, frequently generating images with an incorrect number of objects. While several prior works have mentioned this issue, a comprehensive and rigorous evaluation of this limitation remains lacking. To address this gap, we introduce **T2ICountBench**, a novel benchmark designed to rigorously evaluate the counting ability of state-of-the-art text-to-image diffusion models. Our benchmark encompasses a diverse set of generative models, including both open-source and private systems. It explicitly isolates counting performance from other capabilities, provides structured difficulty levels, and incorporates human evaluations to ensure high reliability. Extensive evaluations with T2ICountBench reveal that all state-of-the-art diffusion models fail to generate the correct number of objects, with accuracy dropping significantly as the number of objects increases. Additionally, an exploratory study on prompt refinement demonstrates that such simple interventions generally do not improve counting accuracy. Our findings highlight the inherent challenges in numerical understanding within diffusion models and point to promising directions for future improvements.

## 1 Introduction

Generative modelling is widely regarded as one of the most essential problems in today's AI community, encompassing tasks such as natural language generation (Brown et al., 2020; Achiam et al., 2023; Liu et al., 2024), image synthesis (Donahue & Simonyan, 2019; Dhariwal & Nichol, 2021; Yang et al., 2023), video generation (Tulyakov et al., 2018; Ho et al., 2022; Singer et al., 2023), and speech synthesis (Oord et al., 2018; Radford et al., 2023; Tan et al., 2024). Among various generative approaches, Diffusion Models (DMs) have demonstrated remarkable success across multiple domains, particularly in text-to-image and text-to-video generation (Ruiz et al., 2023; Wu et al., 2023a; Yang et al., 2024c). Notable models like Diffusion Transformers (DiTs) (Peebles & Xie, 2023) and Video LDM (Blattmann et al., 2023) have been shown to produce high-resolution and realistic images and videos, forming the foundation of advanced generative AI tools, including OpenAI Sora (OpenAI, 2024) and Kling (Kuaishou, 2024).

Despite these advancements, diffusion-based models exhibit fundamental limitations in adhering to numerical constraints in user instructions. Prior empirical studies have shown that text-to-image diffusion models often struggle with basic object counting tasks (Saharia et al., 2022; Huang et al., 2023; Petsiuk et al., 2022). Specifically, when given prompts specifying an exact number of objects (e.g., "generate an image with 7 apples on a wooden table"), the generated content frequently fails to match the requested quantity. These limitations become even more pronounced in complex scenarios, such as "generate an image with 7 apples on a table, separated by 3 oranges." Such failures raise concerns about the reliability of such generative models and highlight their inherent difficulty in following precise numerical constraints.

However, existing empirical studies on the counting ability of text-to-image models suffer from key limitations. Many benchmark studies evaluate only a small number of possibly outdated generative models (Saharia et al., 2022; Petsiuk et al., 2022), with most models dating back to 2022–2023. Additionally, some benchmarks are too general and fail to disentangle counting ability from other factors such as adherence to style and shape constraints (Huang et al., 2023; Peng et al., 2024; Wu et al., 2024). These shortcomings necessitate the need for a comprehensive, up-to-date, and specialized benchmark dedicated to evaluating the counting ability of text-to-image models.

To address this gap, we introduce **T2ICountBench**, a novel benchmark designed to rigorously assess the counting ability of state-of-the-art text-to-image models in 2025. Our benchmark covers a diverse set of generative models, including both open-source and private image generation systems (Podell et al., 2024; Baldridge et al., 2024; Yang et al., 2024b). Unlike prior works, T2ICountBench explicitly isolates counting performance from other capabilities and provides structured difficulty levels, spanning object counts from 1 to 15. Additionally, our benchmark incorporates human evaluations to ensure high reliability and robustness.

With the proposed T2ICountBench, we conduct a comprehensive evaluation to determine whether diffusion-based text-to-image models can accurately generate objects under numerical constraints. Our results show that most existing models exhibit significant failures in simple counting tasks, frequently generating the wrong number of objects. To highlight the non-trivial nature of this limitation, we also explore whether simple prompt refinements—decomposing a difficult counting task (e.g., generating 15 objects) into smaller subtasks—can improve performance. Our contributions are summarized as follows:

- We present a comprehensive and rigorous benchmark, T2ICountBench, for evaluating the counting ability of text-to-image diffusion models. This benchmark effectively exposes the inherent limitations of these models in generating the exact number of objects.

- We conduct extensive ablation studies on various factors influencing counting performance, including the number of objects, scene type, and style. Our findings indicate that as the number of objects increases from 1 to 15, model accuracy significantly drops, reaching around 10% for higher counts. We also find that complex background scenes will further adversely affect counting ability.

- We performed an exploratory study to investigate whether simple prompt refinements could alleviate counting limitations. Our results indicate that such refinements generally do not improve counting performance, highlighting the inherent challenge of text-to-image diffusion models in counting.

**Roadmap.** In Section 3, we introduce our new benchmark to evaluate the counting capability of text-to-image diffusion models. In Section 4, we show the main findings from our counting benchmark. In Section 5, we discuss the possibility of improving text-to-image diffusion models with prompt refinement. In Section 6, we show the conclusion of this paper.

## 2 RELATED WORKS

**Benchmarks on Text-to-Image Generation.** The rapid advancement and real-world impact of text-to-image models have driven the development of evaluation benchmarks, particularly following the emergence of diffusion models. Early benchmarks (Ramesh et al., 2022; Cho et al., 2023; Hu et al., 2023) primarily relied on captions sourced from well-established datasets such as MS COCO, focusing on generating simple objects and scenes that could be automatically evaluated using pre-trained vision models. For instance, DALL-Eval (Cho et al., 2023) employs a 3D renderer to generate synthetic scenes for training text-to-image models, subsequently assessing them with object detection models. It also incorporates fairness considerations by evaluating social biases such as gender and skin tone. GenEval (Ghosh et al., 2023) as an object-focused automatic evaluation framework that uses object detection and related vision models to assess fine-grained compositional and text-to-image alignment. Addressing DALL-Eval's limited scope, TIFA (Hu et al., 2023) expands evaluation criteria by leveraging a pretrained visual question-answering (VQA) model, enabling assessments beyond synthetic captions and 3D-rendered scenes to include more diverse conditions such as geolocation and weather variations.

More recent benchmarks have shifted toward evaluating advanced capabilities of text-to-image models. HPDv2 (Wu et al., 2023b) and Gecko (Wiles et al., 2024) incorporate human preference-based ranking to assess alignment with aesthetic preferences. Another key research direction focuses on compositional text-to-image generation, which involves associating arbitrary attributes with objects beyond predefined datasets like COCO and reasoning about complex object relationships. Representative benchmarks in this area include T2I-CompBench (Huang et al., 2023), ConceptMix (Wu et al., 2024), and GenAI-Bench (Li et al., 2024a). Additionally, Commonsense-T2I (Fu et al., 2024) and PhyBench (Meng et al., 2024) further extend these evaluations by incorporating real-world commonsense reasoning, such as physical constraints. Despite the progress in benchmarking various aspects of text-to-image models, ranging from basic object recognition to complex compositional and commonsense reasoning, the fundamental ability of these models to accurately count objects still requires a rigorous evaluation. This paper aims to address this gap through a rigorous evaluation of the counting capability of state-of-the-art text-to-image models.

**Diffusion Models for Text-to-Image Generation.**    As a fundamental paradigm shift in generative AI, diffusion models have substantially enhanced the quality and resolution of generated images, surpassing earlier approaches such as Variational Autoencoders (VAEs) (Kingma & Welling, 2014; Razavi et al., 2019) and Generative Adversarial Networks (GANs) (Goodfellow et al., 2014; Xu et al., 2018). Recent diffusion-based backbone models (Ho et al., 2020; Song et al., 2021b;a; Lipman et al., 2023) have achieved impressive results in high-fidelity image synthesis without control conditions. However, the challenge of precisely controlling image content via language prompts has motivated the development of more controllable text-to-image generation methods (Rombach et al., 2022; Ramesh et al., 2022).

Text-to-image diffusion models can be broadly classified into two categories: pixel space models (Nichol et al., 2022; Saharia et al., 2022; Chen et al., 2023) and latent space models (Rombach et al., 2022; Samuel et al., 2023; Podell et al., 2024). Pixel space models directly perturb image pixels with noise and iteratively denoise them. For example, GLIDE (Nichol et al., 2022) adapts class-conditioned diffusion models by replacing class labels with text tokens and employs both classifier guidance and classifier-free guidance to align images with text. Imagen (Saharia et al., 2022) similarly leverages classifier-free guidance but utilizes a pretrained large language model for text encoding to enhance image fidelity and text alignment. Re-Imagen (Chen et al., 2023) further augments this approach by incorporating Retrieval-Augmented Generation (RAG) to improve image quality by grounding from multi-modal knowledge bases. In contrast, DALL·E 2 (Ramesh et al., 2022) uses a diffusion decoder that inverts a CLIP image encoder, effectively bridging text embeddings and image generation in a semantically rich manner.

Owing to the substantial computational demands of pixel space models for high-resolution synthesis, latent space models have emerged as a more efficient alternative. These models perform the diffusion process in a compressed latent space derived from pretrained autoencoders such as VQ-VAE (Van Den Oord et al., 2017), which reduces computational load while maintaining image quality. A well-known example is Stable Diffusion (Podell et al., 2024), which builds on the latent diffusion framework to generate high-resolution images efficiently. Additionally, NAO (Samuel et al., 2023) investigates the structure of the latent space to further enhance performance, especially in long-tail and few-shot scenarios. Despite these advances, a rigorous evaluation of these models' ability to accurately count objects in generated images remains largely unexplored, motivating the empirical studies in this paper. Our findings in this paper may also inspire future directions for enhancing current text-to-image and text-to-video diffusion models, particularly regarding controllability (Wang et al., 2024c;a; Cheng et al., 2025; Cao et al., 2025a) and expressiveness (Cao et al., 2025c; Chen et al., 2025; Gong et al., 2025; Cao et al., 2025b), thereby providing novel insights into the synthesis process and benchmark performance.

## 3 THE T2I COUNTBENCH

In this section, we first introduce the baseline models used in our benchmark in Section 3.1, followed by the prompts designed to evaluate the counting ability of text-to-image diffusion models in Section 3.2. We then describe our evaluation protocol in Section 3.3.

Table 1: **Basic information of the Evaluated Text-to-Image Diffusion Models.**

| Model Name | Organization | Year | # Params | Open |
|---|---|---|---|---|
| Recraft V3 (AI, 2024a) | Recraft AI | 2024 | N/A | No |
| Imagen-3 (Baldridge et al., 2024) | Google | 2024 | N/A | No |
| Grok 3 (xAI, 2025) | xAI | 2025 | N/A | No |
| Gemini 2.0 Flash (Google, 2025) | Google | 2025 | N/A | No |
| FLUX 1.1 (Labs, 2024) | Black Forest | 2024 | N/A | No |
| Firefly 3 (Adobe, 2024) | Adobe | 2024 | N/A | No |
| Dall·E 3 (Betker et al., 2023) | OpenAI | 2024 | N/A | No |
| SD 3.5 Large Turbo (AI, 2024b) | Stability AI | 2024 | 8.1B | Yes |
| Doubao (Team, 2025) | Bytedance | 2023 | N/A | No |
| Qwen2.5-Max (Yang et al., 2024a) | Alibaba | 2025 | N/A | No |
| WanX2.1 (Cloud, 2025) | Alibaba | 2025 | 14B | Yes |
| Kling (Kuaishou, 2024) | Kwai | 2024 | N/A | No |
| Star-3 Alpha (LiblibAI, 2024) | LiblibAI | 2024 | N/A | No |
| Hunyuan (Li et al., 2024b) | Tencent | 2024 | 1.5B | Yes |
| GLM-4 (GLM et al., 2024) | ZhipuAI | 2024 | 9B | Yes |

## 3.1 BASELINE MODELS

A rigorous evaluation of the counting ability of text-to-image diffusion models requires a diverse and up-to-date selection of models. However, existing benchmarks often fall short in this issue. For instance, a human evaluation benchmark that includes counting tasks (Petsiuk et al., 2022) considers only Stable Diffusion (Rombach et al., 2022) and DALL·E 2 (Ramesh et al., 2022), both released in 2022, covering a limited subset of available models. Similarly, several recent benchmarks (Li et al., 2024a; Meng et al., 2024; Fu et al., 2024) evaluate at most ten text-to-image diffusion models, failing to provide a comprehensive assessment of counting capabilities across the latest systems.

To address these limitations, our benchmark includes 15 state-of-the-art text-to-image diffusion models, encompassing both open-source and privately owned commercial models. This selection ensures broad coverage of models widely used in generative AI research and applications, most of which have been introduced after 2024. By incorporating a more extensive set of models, we provide a trustworthy and representative evaluation of counting performance. Basic information on the selected models is presented in Table 1, and further implementation details on baseline model evaluation (e.g., model type, length-to-width ratio) are presented in Appendix B.

## 3.2 GENERATION PROMPTS

The design of generation prompts is the key to effectively evaluating text-to-image models. Although counting is a fundamental capability of diffusion models, many existing benchmarks (e.g., ConceptMix (Wu et al., 2024), Commonsense-T2I (Fu et al., 2024), and PhyBench (Meng et al., 2024)) do not include object quantity in their prompts. Moreover, previous studies on evaluating the counting ability of diffusion models have offered only preliminary explorations without a comprehensive, multi-level evaluation (Saharia et al., 2022; Li et al., 2024a). For instance, while GenAI-Bench (Li et al., 2024a) provides a broad evaluation of text-to-image generation, only 339 of its prompts address counting. These prompts are also combined with a wide range of additional conditions, limited to numbers below 10, and often generate fewer than 3 objects.

In contrast, our approach uses a simple yet effective prompt design that directly tests the counting ability while minimizing irrelevant factors. Our prompt template used in most experiments is:

> **Prompt Template 1**: Generate <number> <object> in/on <scene> in <style>.

Here, <number> denotes an integer between 1 and 15 in Arabic numeral form, providing a more comprehensive range than those used in previous benchmarks. The <object> field covers 6 common categories: fruit, human, animal, abstract shape, furniture, and plant. In addition, we vary the scene and style by including 3 different types for each to assess the models' performance under different conditions. Overall, our benchmark evaluates 525 prompts for each baseline model. These prompts cover all 15 numbers, 7 object categories, and combinations of 3 scenes and 3 styles. For example:

> **Example Prompt 1.1**: Generate 13 chairs on a wooden floor in a watercolor style.

## 3.3 Evaluation Protocols

To ensure a rigorous and thorough evaluation, we adopt a full human evaluation process. Five graduate students with expertise in AI and visual perception assess each generated image. An image is marked as "correct" if it contains exactly the number of objects specified in the prompt; otherwise, it is labeled as "incorrect". To ensure a fair comparison, we have each model generate four images per prompt, and we consider the task successful if at least one of the four images is correct. This comprehensive human evaluation offers more reliable results than previous approaches that rely on object detection (Cho et al., 2023) or visual question answering models (Hu et al., 2023), both of which may introduce biases.

Our primary evaluation metric is counting accuracy, which considers only whether the generated images contain the correct number of objects. Each unique combination of object, scene, and style is treated as a distinct task, and overall accuracy is computed from correct outputs across all 15 numbers and relevant prompts. This design allows us to more directly and intuitively compare the counting capabilities of different text-to-image models.

## 4 Experiments

In this section, we present our experimental results using the proposed T2ICountBench. Section 4.1 reports the overall counting performance of all baseline models, while Section 4.2 investigates the impact of various factors on the counting ability of text-to-image diffusion models. Finally, Section 4.3 presents our analysis of variance across human annotators.

### 4.1 Overall Counting Results

To evaluate the fundamental counting ability of diffusion models, we employ the general prompt described as Prompt Template 1 in Section 3.2. Specifically, the four key elements in the prompt template are instantiated as follows:

- $<$number$>$: $1, 2, 3, \ldots, 15$;
- $<$object$>$: `'fruit'`, `'human'`, `'animal'`, `'shape'`, `'furniture'`, `'plant'`;
- $<$scene$>$: `'home'`, `'nature'`, `'city'`;
- $<$style$>$: `'plain'`, `'watercolor'`, `'cartoon'`.

For each model, we generate outputs for all possible combinations of these properties and record the number of cases in which the generated image contains the correct quantity of objects. All counting results are evaluated through a full human evaluation process as described in Section 3.3. We then categorize the results by object class and present them in Table 2.

The overall results lead to several observations. First, when considering both per-category and overall average accuracy, all state-of-the-art text-to-image diffusion models struggle to generate objects in the correct quantities. No model achieves an average accuracy above $50\%$, and for each category, accuracy does not exceed $60\%$. Additionally, the variance across different object categories is minimal, indicating that models consistently perform poorly across all categories. These findings highlight a significant gap in the counting ability of diffusion models.

Furthermore, a comparison among models reveals a large disparity in performance. For instance, the strongest models, such as Imagen-3 (with an average accuracy of $43\%$) and Gemini 2.0 Flash (with an average accuracy of $39\%$), significantly outperform models like Recraft V3 and SD 3.5, which achieve average accuracies of $25\%$ and $26\%$, respectively. This represents nearly a $150\%$ improvement in accuracy between the best and worst performing models.

**Observation 4.1.** *Overall, state-of-the-art models exhibit a significant gap in accurately counting objects, and the performance difference between the strongest and weakest models is notable.*

Table 2: **Counting Accuracy Across Different Object Categories.** The models are sorted in ascending order based on their average accuracy across six object categories.

| Model | Fruit | Human | Animal | Shape | Furniture | Plant | Avg. Acc. |
|---|---|---|---|---|---|---|---|
| Recraft V3 | 0.23 | 0.36 | 0.27 | 0.23 | 0.24 | 0.20 | 0.25 |
| SD 3.5 | 0.14 | 0.33 | 0.35 | 0.27 | 0.27 | 0.23 | 0.26 |
| Grok 3 | 0.21 | 0.55 | 0.23 | 0.35 | 0.23 | 0.17 | 0.29 |
| DALL·E 3 | 0.29 | 0.31 | 0.43 | 0.23 | 0.31 | 0.23 | 0.30 |
| GLM-4 | 0.27 | 0.33 | 0.44 | 0.25 | 0.37 | 0.24 | 0.32 |
| Qwen2.5-Max | 0.35 | 0.29 | 0.41 | 0.33 | 0.33 | 0.19 | 0.32 |
| Firefly 3 | 0.27 | 0.41 | 0.51 | 0.29 | 0.33 | 0.20 | 0.34 |
| FLUX 1.1 | 0.31 | 0.43 | 0.40 | 0.27 | 0.36 | 0.31 | 0.35 |
| Kling | 0.30 | 0.51 | 0.45 | 0.19 | 0.33 | 0.35 | 0.35 |
| Doubao | 0.35 | 0.43 | 0.4 | 0.35 | 0.39 | 0.33 | 0.37 |
| Hunyuan | 0.33 | 0.36 | 0.49 | 0.37 | 0.36 | 0.32 | 0.37 |
| Star-3 Alpha | 0.38 | 0.39 | 0.44 | 0.39 | 0.39 | 0.27 | 0.37 |
| WanX2.1 | 0.37 | 0.52 | 0.47 | 0.32 | 0.36 | 0.20 | 0.37 |
| Gemini 2.0 Flash | 0.28 | 0.48 | 0.48 | 0.51 | 0.29 | 0.32 | 0.39 |
| Imagen-3 | 0.31 | 0.53 | 0.51 | 0.44 | 0.43 | 0.33 | 0.43 |

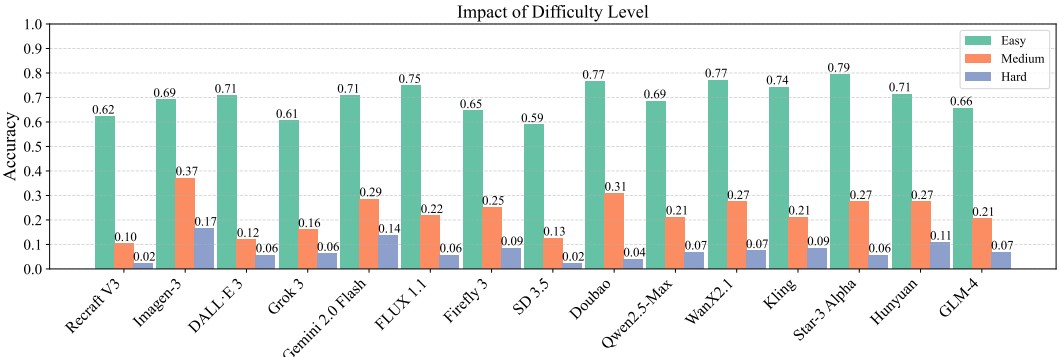

Figure 1: **Impact of Difficulty Levels**. This figure presents the comparison of the accuracy of various models across three difficulty levels (Easy, Medium, Hard). The horizontal axis lists the models, while the vertical axis represents accuracy. Each bar in the figure represents the accuracy for a specific model under the corresponding prompt difficulty level.

## 4.2 ABLATION STUDY

In this section, we present several ablation studies to examine the impact of different factors on the counting ability of text-to-image diffusion models, including difficulty levels, scenes, and styles.

**Impact of Difficulty Levels.** To assess the models' ability to handle counting tasks of varying difficulty, we leverage our wide range of counting numbers (1–15). Specifically, we use the same prompt template and evaluation process as described in Section 4.1 to compute the overall accuracy of each model across different difficulty levels. We define three levels: (i) `Easy`: counting tasks with numbers 1, 2, 3, 4, 5; (ii) `Medium`: counting tasks with numbers 6, 7, 8, 9, 10; (iii) `Hard`: counting tasks with numbers 11, 12, 13, 14, 15.

Figure 1 clearly shows a significant gap in counting accuracy across the three difficulty levels. For all 15 models, the `Easy` level yields accuracies between approximately 60% and 80%, while the `Medium` level drops to between 10% and 30%. The most striking results are observed at the `Hard` level, where almost all models achieve accuracies below 10%. Only Imagen-3, Gemini 2.0 Flash, and Hunyuan exceed 10%, with some models such as Recraft V3 and SD 3.5 achieving accuracies as low as 2%. This indicates that these models nearly fail the counting task at higher difficulty levels. We thus make the following observation:

**Observation 4.2.** *As the counting task becomes more difficult (i.e., as the number of objects increases from 1 to 15), the models' accuracies drop drastically. For tasks involving 11–15 objects, nearly all models exhibit accuracies below 10%.*

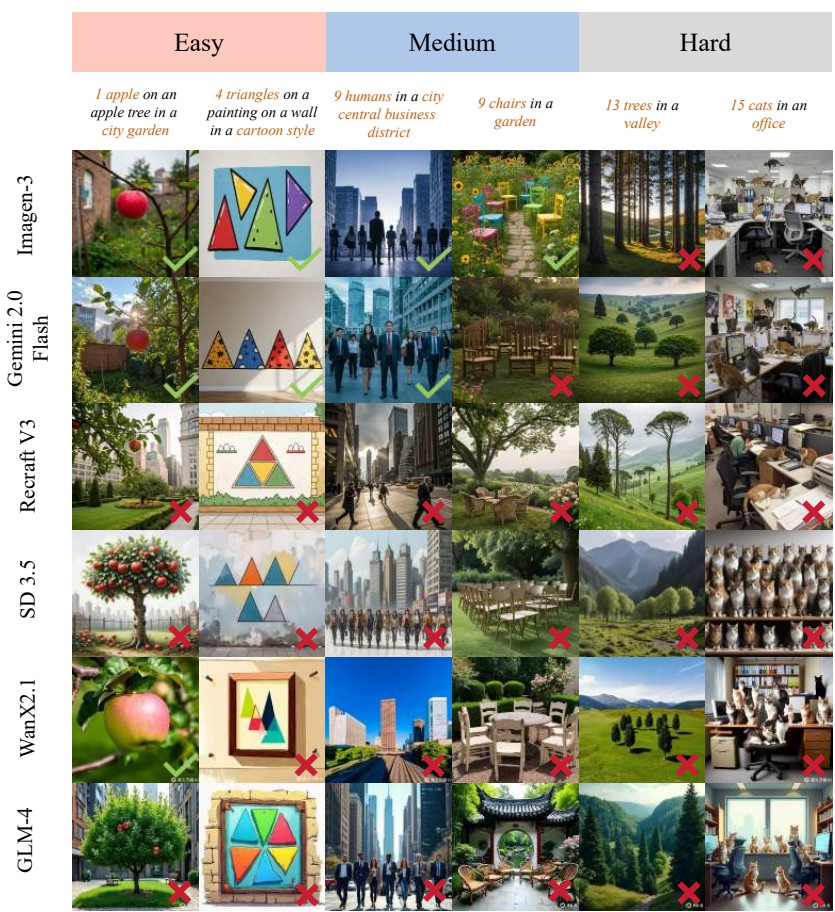

Figure 2: **Qualitative Study of Different Difficulty Levels**. A high-resolution version of this image is available in Figure 8 in Appendix D.

To further support our observations, we present qualitative results on the impact of difficulty levels in Figure 2. We observe that at higher difficulty levels, all models make significant mistakes, with the generated object counts deviating markedly from the target. In contrast, images generated at lower difficulty levels sometimes succeed (e.g., Imagen-3 achieved perfect counts in both easy and medium prompts). Additionally, higher difficulty levels tend to result in reduced image detail and fidelity; for example, in the "15 cats" prompt on GLM-4, the generated image depicts a cat with two tails. This indicates that increased task difficulty not only exacerbates counting errors but also adversely affects overall image quality, which resonates with our quantitative results in this study. Due to space limitations, we moved the statement on the impact of the scene and the impact of style to the Appendix C.

## 4.3 HUMAN ANNOTATOR VARIANCE ANALYSIS

For each prompt and model, four images were generated and independently evaluated by five annotators. An annotator considered the model's result correct if at least one of the four images had a correct count; otherwise, it was marked as incorrect. To assess consistency among annotators, we computed Fleiss' Kappa using Eq. (1). The resulting value of 0.58 indicates moderate inter-annotator agreement.

$$\kappa = \frac{P - P_e}{1 - P_e},\tag{1}$$

where $P = \frac{1}{N} \sum_{i=1}^{N} \frac{n_{i0}(n_{io}-1)+n_{i1}(n_{i1}-1)}{n(n-1)}$, $P_e = \left(\frac{1}{Nn} \sum_{i=1}^{N} n_{i0}\right)^2 + \left(\frac{1}{Nn} \sum_{i=1}^{N} n_{i1}\right)^2$, $N$ represents the number of evaluated image groups, $n$ is the number of five annotators. $n_{i0}$ indicates that the $i$ th sample is marked as an incorrect count, and $n_{i1}$ indicates that the $i$ th sample is marked as a correct count.

# 5 PROMPT REFINEMENT

In this section, we address the counting limitations of text-to-image diffusion models through prompt refinement. Specifically, we first introduce our proposed prompt refinements in Section 5.1, followed by experimental results in Section 5.2. Finally, in Section 5.3 we discuss several open questions and conjectures regarding the counting ability of text-to-image models.

## 5.1 THE PROPOSED PROMPTS

Due to the poor performance observed when directly generating a large number of objects (as shown in Section 4), we adopt a simple work-around by refining the prompts, which verifies whether such counting limitations can be solved by straightforward improvements. Our exploratory study takes a task-decomposition approach by breaking the generation task into smaller subtasks, which mirrors how humans draw many objects on a single canvas. We consider four types of prompt refinement mechanisms: Multiplicative Decomposition, Additive Decomposition, Grid Prior, and Position Guidance.

**Multiplicative Decomposition.** For example, when drawing 15 apples on a table, a human might consider drawing 5 apples in a row and repeating this process 3 times. In this prompt refinement, we decompose the task by instructing the model to generate a large number of objects as smaller groups. Specifically, let the number of objects to be generated be $N$, and let $a$ be a factor of $N$ smaller than $\sqrt{N/2}$, and $b$ be a factor larger than $\sqrt{N/2}$, satisfying $N = ab$. When $N$ is prime, its only factors are 1 and $N$, so we set $a = 1$ and $b = N$. Our prompt can be shown as follows:

**Prompt Template 2**: Generate $a$ times $b$ <object> in/on <scene> in <style>.

For example, considering the task of generating 12 watermelons on a wooden table in a cartoon style, the refined prompt would be:

**Example Prompt 2.1**: Generate 3 times 4 watermelons on a wooden table in cartoon style.

Besides the basic prompt refinement introduced above, we also explore three mechanisms, namely **Additive Decomposition**, **Grid Prior**, and **Position Guidance**. Due to space limitations, their detailed descriptions and examples are provided in Appendix C.

## 5.2 PROMPT REFINEMENT RESULTS

Building on the prompt refinement approaches introduced in the previous subsection, we systematically evaluate whether these refinements can mitigate the counting limitations of text-to-image diffusion models. In this study, we consider the prompt templates in Prompts 2–5 from Section 5.1 and fill in the properties as follows:

<number>: $1, 2, 3, \ldots, 15$; <object>: `'fruit'`, `'human'`, `'animal'`, `'shape'`, `'furniture'`, `'plant'`.

In order to focus on the simplest generation scenarios and eliminate the impact of extraneous factors, we fix the <scene> and <style> to `'Home'` and `'Plain'`, respectively. Specifically, we compute the average accuracy for each model across all six object types under each prompt refinement strategy, and our results are presented in Table 3.

The results in Table 3 reveal that all four types of prompt refinement lead to worse performance compared with the original prompt. The performance drop is particularly pronounced for multiplicative decomposition, additive decomposition, and position guidance, where the average accuracy across 15 models decreases by more than 40% relative to the original accuracy (dropping from 42% to 26%, 23%, and 20%, respectively). In some cases, such as with Firefly 3, the reduction is as steep as

75% (from 42% to 10% under multiplicative decomposition). Among the refinement strategies, grid prior shows the most promise, as its performance drop is relatively marginal compared with other methods.

**Observation 5.1.** *For most models and in most cases, prompt refinement degrades the counting performance of text-to-image diffusion models, with particularly severe drops observed for multiplicative decomposition, additive decomposition, and position guidance.*

Table 3: **Prompt Refinement Results.** Each entry represents the average accuracy across all object categories for a specific prompt refinement method.

| Model | Original | Multiplicative | Additive | Grid | Position | Avg. Acc. |
|---|---|---|---|---|---|---|
| Recraft V3 | 0.37 | 0.29 | 0.26 | 0.26 | 0.15 | 0.26 |
| Imagen-3 | 0.58 | 0.33 | 0.29 | 0.49 | 0.26 | 0.39 |
| DALL·E 3 | 0.36 | 0.38 | 0.20 | 0.33 | 0.16 | 0.29 |
| Grok 3 | 0.34 | 0.26 | 0.35 | 0.26 | 0.22 | 0.29 |
| Gemini 2.0 Flash | 0.46 | 0.39 | 0.30 | 0.36 | 0.33 | 0.37 |
| FLUX 1.1 | 0.38 | 0.16 | 0.24 | 0.32 | 0.13 | 0.25 |
| Firefly 3 | 0.42 | 0.10 | 0.16 | 0.39 | 0.17 | 0.25 |
| SD 3.5 | 0.29 | 0.18 | 0.15 | 0.30 | 0.13 | 0.21 |
| Doubao | 0.48 | 0.20 | 0.12 | 0.40 | 0.08 | 0.26 |
| Qwen2.5-Max | 0.41 | 0.27 | 0.27 | 0.35 | 0.19 | 0.30 |
| WanX2.1 | 0.50 | 0.35 | 0.19 | 0.34 | 0.30 | 0.34 |
| Kling | 0.40 | 0.17 | 0.10 | 0.33 | 0.07 | 0.22 |
| Star-3 Alpha | 0.35 | 0.25 | 0.15 | 0.29 | 0.22 | 0.25 |
| Hunyuan | 0.45 | 0.28 | 0.26 | 0.21 | 0.26 | 0.29 |
| GLM-4 | 0.46 | 0.36 | 0.35 | 0.39 | 0.28 | 0.37 |
| **Avg. Acc.** | 0.42 | 0.26 | 0.23 | 0.34 | 0.20 | 0.29 |

## 5.3 DISCUSSION

We discuss possible reasons for the observed counting failures in text-to-image diffusion models. One key reason for poor counting performance is that several early text-to-image models (e.g., Stable Diffusion (Rombach et al., 2022), SDXL (Podell et al., 2024), unCLIP (Ramesh et al., 2022)) use CLIP (Radford et al., 2021) as their text encoder. Previous studies have demonstrated that CLIP has inherent counting issues (Paiss et al., 2023; Jiang et al., 2023; Zhang et al., 2024). This limitation also contributes to the failure of prompt refinement, as CLIP is not designed to process complex, instruction-based prompts. In contrast, models such as Imagen (Saharia et al., 2022) and DALL·E 3 (Betker et al., 2023) employ large language models like T5 (Raffel et al., 2020) for prompt processing, which offer improved language understanding. Nonetheless, their counting failures may stem from insufficient alignment with human preferences, preventing strict adherence to detailed instructions.

To improve the counting capability in existing text-to-image diffusion models, there are several open directions, including CLIP counting ability improvement, automatic prompt refinement, and human preference alignment. Due to the space limitation, we defer the more details of the potential directions are presented in Appendix A.

## 6 CONCLUSION

In this paper, we introduced T2ICountBench, a comprehensive benchmark to rigorously evaluate the counting ability of text-to-image diffusion models. Our extensive evaluations reveal that even state-of-the-art models struggle to adhere to numerical constraints, with accuracy dropping sharply as the number of objects increases and under complex scene conditions. We also show that simple prompt refinements generally fail to improve counting performance, underscoring inherent challenges in numerical understanding within these models. These findings motivate further research to address these limitations and enhance the reliability of diffusion-based generative systems.

## ETHIC STATEMENT

This paper does not involve human subjects, personally identifiable data, or sensitive applications. We do not foresee direct ethical risks. We follow the ICLR Code of Ethics and affirm that all aspects of this research comply with the principles of fairness, transparency, and integrity.

## REPRODUCIBILITY STATEMENT

We ensure the reproducibility of our empirical findings. For all experiments, we describe the sources of the LLM models, datasets, evaluation metrics, and experiment setup in the main text. All prompt templates used are also provided to support the reproducibility of our results.

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

# Appendix

In Section A, we discuss some future directions. In Section B, we present the details of the evaluated generative models. In Section C, we show the details of two prompt refinement mechanisms and some additional quantitative experiments. In Section D, we present more qualitative studies. In Section E, we discuss the potential risks of this paper. In Section F, we show a full list of the results for every single experiment. In Section G, we present all the generated images in this benchmark study.

## A  FUTURE WORKS

To enhance counting ability in text-to-image models, one direction is to improve CLIP-based models by incorporating recent advances that address CLIP's counting shortcomings (Paiss et al., 2023; Jiang et al., 2023). Another promising approach is automatic prompt refinement (Mo et al., 2024; Wang et al., 2024b), which translates complex human instructions into simpler forms that diffusion models can more reliably interpret. For models leveraging large language models, reinforcement learning techniques may further align generated images with human preferences and improve the processing of task-decomposition prompts (Fan et al., 2023).

## B  IMPLEMENTATION DETAILS

In this section, we provide the implementation details for generating images using the baseline models outlined in Table 1. Specifically, the details for all 15 models are listed as follows:

- Model 1: **Recraft V3** (AI, 2024a). Recraft V3 is a close-sourced text-to-image model from Recraft AI company, released in 2024. We use default mode of this model for experiment.8

- Model 2: **Imagen-3** (Baldridge et al., 2024). Imagen-3 is a close-sourced text-to-image model from Google company, released in 2024. We use best quality mode of this model for experiment. Since the default setting for landscape ratio is 16:9, we change it to 1:1 to ensure fair comparison.

- Model 3: **Grok 3** (xAI, 2025). Grok 3 is a close-sourced multi-modal model from xAI company, released in 2025. We use default mode of this model for experiment.

- Model 4: **Gemini 2.0 Flash** (Google, 2025). Gemini 2.0 Flash is a close-sourced multi-modal model from Google company, released in 2025. We use default mode of this model for experiment.

- Model 5: **FLUX 1.1** (Labs, 2024). FLUX 1.1 is a close-sourced text-to-image model from Black Forest Labs company, released in 2024. We use default mode of this model for experiment. Since the default setting for landscape ratio is 4:3, we change it to 1:1 to ensure fair comparison.

- Model 6: **Firefly 3** (Adobe, 2024). Firefly 3 is a close-sourced multi-modal model from Adobe company, released in 2024. We use fast mode of this model for experiment.

- Model 7: **Dall·E 3** (Betker et al., 2023). Dall·E 3 is a close-sourced text-to-image model from OpenAI company, released in 2024. We use default mode of this model for experiment.

- Model 8: **Stable Diffusion 3.5 Large Turbo** (AI, 2024b). Stable Diffusion 3.5 Large Turbo is an open-sourced text-to-image model from Stability AI company, released in 2024. We use default mode of this model for experiment.

- Model 9: **Doubao** (Team, 2025). Doubao is a close-sourced multi-modal model from Bytedance company, released in 2023. We use default mode of this model for experiment.

- Model 10: **Qwen2.5-Max** (Yang et al., 2024a). Qwen2.5-Max is a close-sourced multi-modal model from Alibaba Cloud company, released in 2025. We use default mode of this model for experiment.

- Model 11: **WanX2.1** (Cloud, 2025). WanX2.1 is a close-sourced multi-modal model from Alibaba Cloud company, released in 2025. We use default mode of this model for experiment.

- Model 12: **Kling** (Kuaishou, 2024). Kling is a close-sourced multi-modal model from Kwai company, released in 2024. We use default mode of this model for experiment.

- Model 13: **Star-3 Alpha** (LiblibAI, 2024). Star-3 Alpha is a close-sourced text-to-image model from LiblibAI company, released in 2024. We use default mode of this model for experiment. Since the default setting for landscape ratio is 4:3, we change it to 1:1 to ensure fair comparison.

- Model 14: **Hunyuan** (Li et al., 2024b). Hunyuan is an open-sourced multi-modal model from Tencent company, released in 2024. We use default mode of this model for experiment.

- Model 15: **GLM-4** (GLM et al., 2024). GLM-4 is an open-sourced multi-modal model from ZhipuAI company, released in 2024. We use default mode of this model for experiment.

## C  ADDITIONAL EXPERIMENTS

**Additive Decomposition.** Another human-inspired approach to generating a large number of objects is to first create a subset of objects and then generate the remainder. Unlike the multiplicative decomposition, this method breaks the task into two smaller parts that are subsequently combined. Specifically, let the number of objects to be generated be $N$, where $N \geq 2$, and let $\lfloor x \rfloor$ denote the floor function, which returns the largest integer less than or equal to $x$. We define our prompt as follows:

> **Prompt Template 3**: Generate $\lfloor N/2 \rfloor$ plus $N - \lfloor N/2 \rfloor$ <object> in/on <scene> in <style>.

An example for such prompt refinement on generating 11 objects would be:

> **Example Prompt 3.1**: Generate 5 plus 6 triangles on a painting on a wall.

**Grid Prior.** An extension of the multiplicative decomposition is to provide an explicit spatial arrangement for the objects. Without such guidance, the model might be uncertain about where to place the generated objects. Therefore, we use a grid layout to structure the output. This process resembles a chain-of-thought strategy by breaking down the task into simpler, sequential steps, in which the first step determines the positions, and the second step puts the objects. Specifically, let the number of objects to be generated be $N$, and let $a$ be its largest factor smaller than $N/2$, and $b$ be the smallest factor larger than $N/2$, so that $N = ab$. Our prompt can be shown as follows:

> **Prompt Template 4**: Generate <number> <object> in/on <scene> in <style>, with a $a$ row $b$ column grid.

Extending the example of 12 watermelons from the multiplicative decomposition, we have the following instance:

> **Example Prompt 4.1**: Generate 12 watermelons on a wooden table in cartoon style, with a 3 row 4 column grid.

**Position Guidance.** A further extension of the additive decomposition approach is to provide explicit positional guidance. In this method, the two groups of objects are placed in designated areas on the canvas, which reduces the cognitive load on the model and provides clearer instructions. In our template, the first group is positioned on the left and the second group on the right. We designed the prompt carefully to ensure that the positional instructions integrate seamlessly with the scene and style constraints:

> **Prompt Template 5**: Generate $\lfloor N/2 \rfloor$ <object> on the left, $N - \lfloor N/2 \rfloor$ <object> on the right, in/on <scene> in <style>.

Extending the triangles example from the additive decomposition, an example for position guidance would be:

> **Example Prompt 5.1**: Generate 5 triangles on the left, 6 triangles on the right, on a painting on a wall.

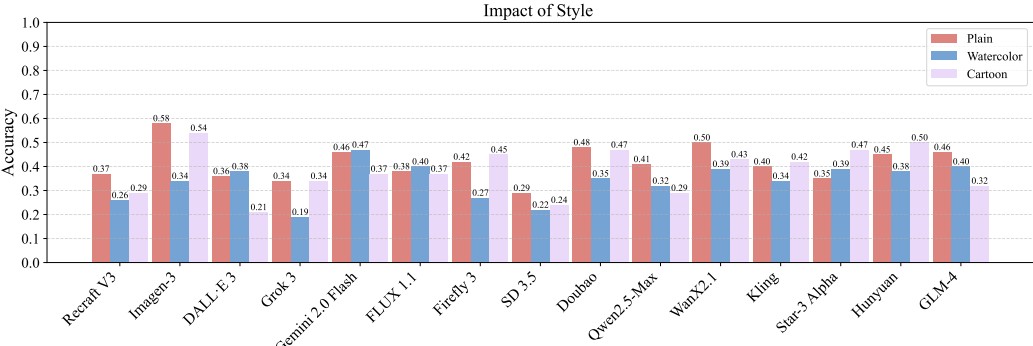

Figure 3: **Impact of Style**. This figure presents the comparison of the accuracy of various models across three styles (Plain, Watercolor, Cartoon). The horizontal axis lists the models, while the vertical axis represents accuracy. Each bar in the figure represents the accuracy for a specific model under corresponding prompt style setting.

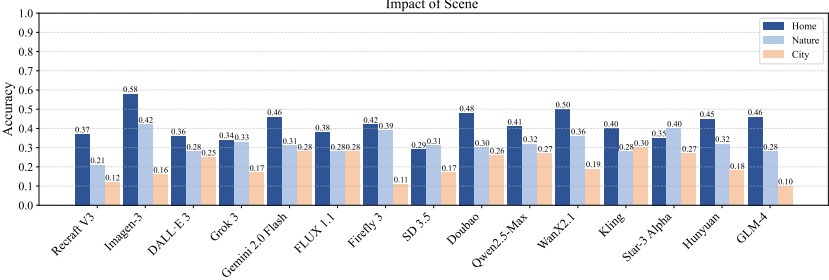

Figure 4: **Impact of Scene**. This figure presents the comparison of the accuracy of various models across three scenes (Home, Nature, City). The horizontal axis lists the models, while the vertical axis represents accuracy. Each bar in the figure represents the accuracy for a specific model under the corresponding prompt scene setting.

**Impact of Scene.** In this study, we investigate how the scene in which objects are presented affects counting performance. The intuition is that complex environments, such as cityscapes with multiple irrelevant elements, may pose a greater challenge compared to simpler settings like a simple wooden table in a home environment. We use the general prompt described in Prompt Template 1 with the following settings:

<number>: $1, 2, 3, \ldots, 15$; <object>: 'fruit', 'human', 'animal', 'shape', 'furniture','plant'; <scene>:'home','nature','city'.

- <number>: $1, 2, 3, \ldots, 15$;
- <object>: 'fruit','human','animal','shape','furniture','plant';
- <scene>: 'home','nature','city';

The <style> keyword is fixed to 'plain' to exclude the effect of styles and focus on the effect of scene modifications. All generation results are evaluated by human annotators, and the results for each scene are summarized in Figure 4 and Figure 5.

The experimental results reveal a significant variance in counting accuracy across different scenes. When averaging the results of all 15 models, the home scene achieves an average accuracy of

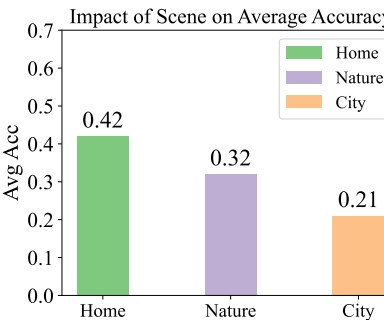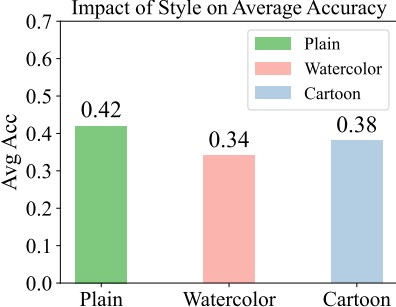

Figure 5: **Impact of Scene and Style on Average Accuracy**. **Left**: This figure presents a comparison of the average accuracy across three scenes (Home, Nature, City) for 15 models. Each bar represents the average accuracy of the 15 models under the corresponding prompt scene setting. **Right**: This figure presents a comparison of the average accuracy across three styles (Plain, Watercolor, Cartoon) for 15 models. Each bar represents the average accuracy of the 15 models under the corresponding prompt style setting.

42%, whereas the `city` scene falls to 21%—a reduction of nearly 50%. This indicates that the compositional complexity of a scene strongly influences a model's counting ability. Moreover, the variation in accuracy for individual models across scenes can be even more pronounced than the average difference across all models. For example, Imagen-3 achieves an accuracy of 58% in the `home` scene but only 16% in the `city` scene, while GLM-4 scores 46% in `home` compared to just 10% in `city`. This leads us to the following observation:

**Observation C.1.** *The models' counting ability is significantly affected by the scene. Complex scenes such as* `city` *and* `nature` *lead to a drop in counting performance.*

Another interesting finding is that a model performing well in one scene does not necessarily excel in other scenes. For instance, in the `home` scene, FLUX 1.1 ranks among the worst in counting accuracy; however, in the `city` scene, its accuracy rises to 28%, making it the second best in that category. Similarly, the best model in the `home` and `nature` scenes, Imagen-3, shows relatively poor performance in the `city` scene compared to other models. This variability suggests a notable instability in the counting ability of text-to-image models across different scenes, indicating a potential direction for future research. We summarize this observation as follows:

**Observation C.2.** *Models that perform well in one scene may not maintain high performance in other scenes, highlighting an instability in counting ability under varying scene conditions.*

**Impact of Style.** In this study, we examine the effect of image style on the counting ability of text-to-image diffusion models. Unlike the scene, which can introduce many irrelevant objects, style is an important property while imposing less generation burden on the generative models. We use the previously used prompt described in Prompt Template 1 and follow the same human evaluation protocols as in our other experiments. Specifically, we use the following property composition to fill in the prompt template:

- <number>: $1, 2, 3, \ldots, 15$;
- <object>: `'fruit'`,`'human'`,`'animal'`,`'shape'`,`'furniture'`,`'plant'`;
- <style>: `'plain'`,`'watercolor'`,`'cartoon'`.

To exclude the effect of scenes and focus on the style categories, the <scene> keyword is fixed to `'home'`. By aggregating the accuracy results into three style categories, we present the findings in Figure 3 and Figure 5.

The results indicate that style has a less significant impact on the counting performance compared to the scene. For example, the average accuracy across all 15 models for the styles `plain`, `watercolor`, and `cartoon` are 42%, 34%, and 38%, respectively, which are on a similar scale. Furthermore, for specific models such as FLUX 1.1 and SD 3.5, the variance in accuracy across different style classes is minimal. Thus, we summarize the following observation:

**Observation C.3.** *Style categories have a less significant impact on models' counting abilities.*

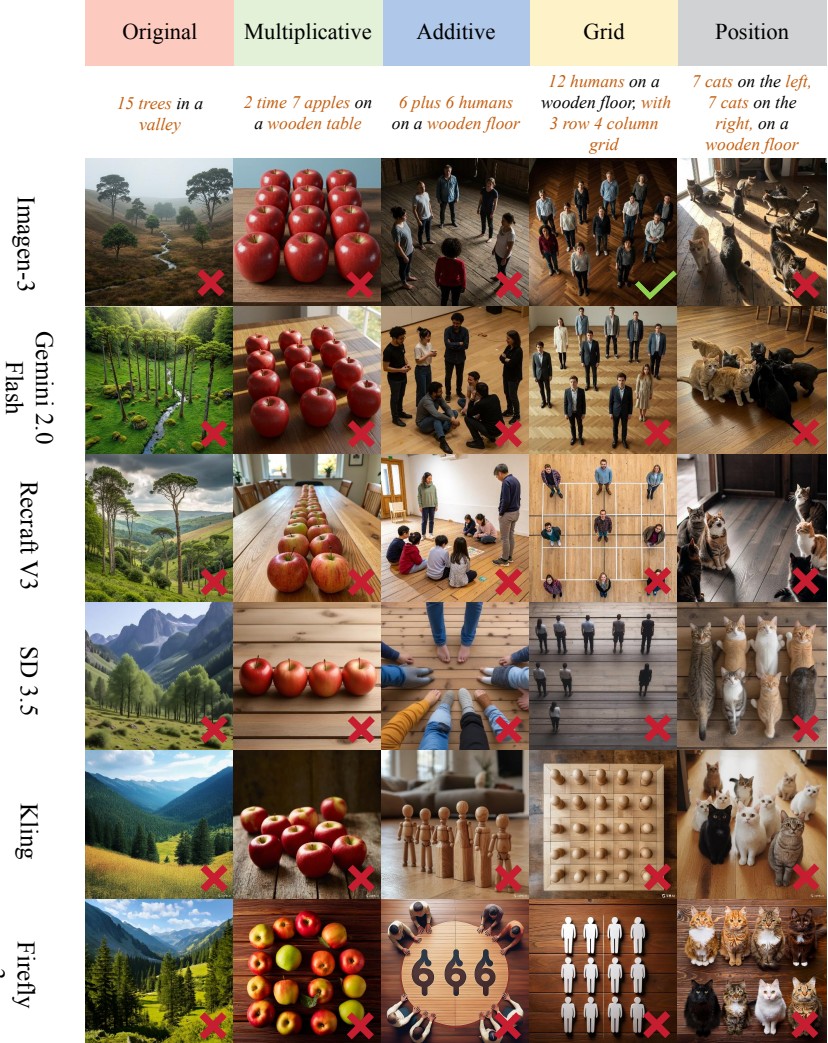

Figure 6: **Qualitative Study of Prompt Refinement Results**. This figure presents the qualitative study of the prompt refinement results in Section 5.1. A high-resolution version of this image is available in Figure 11 in Appendix D.

To support our observations on prompt refinement, we present a qualitative study in Figure 6. The figure shows that for the multiplicative refinement prompt "2 times 7 apples," almost all models fail to adhere to both numbers (2 and 7), completely disregarding the instruction; only Recraft V3 manages to generate two columns, while still failing to produce the correct number of rows. For the additive prompt, models appear to misinterpret "6 + 6" as simply 6. Furthermore, with more complex prompts such as grid prior or position guidance, nearly all models struggle with the subtasks—they fail to correctly interpret directions (e.g., left and right) or generate a grid with the correct number of rows and columns, and in some cases, do not generate a grid at all. These diverse failure cases further reinforce our quantitative findings that prompt refinement does not overcome the counting limitations of text-to-image diffusion models.

## D  QUALITATIVE STUDY

In this section, we introduce the qualitative study based on our experiments across all models.

**Qualitative Study on Main Results.** We present a qualitative study on the main results in Figure 7. The images generated by most models exhibit satisfactory fidelity and aesthetics, with minimal distortions or incorrect spatial relationships (e.g., misplaced eyes or noses on human and cat faces). One notable negative example is the fruit result of the "5 watermelons" prompt on SD 3.5 in Figure 7, where the watermelons are irregularly cut and arranged messily. This issue may be attributable to the relatively small number of parameters of the model. Despite the overall high fidelity, many models still encounter counting errors, as demonstrated by the "5 flowers" prompt on Gemini 2.0 Flash in the plant result. This observation suggests that fidelity and counting accuracy are not necessarily correlated—a model that produces high-fidelity images may still fail to count objects correctly. Interestingly, some models misinterpret the word "earthy." For instance, in the shape results, the "3 triangles" prompt on Doubao produced an output in which an Earth-like model is depicted on land with triangles superimposed on its surface. Although the counting outcome is correct, this example indicates that these models may benefit from additional human preference alignment to better follow user instructions.

**Qualitative Study on the Impact of Different Difficulty Levels.** We present a qualitative study on the impact of different difficulty levels in Figure 8. Our observations indicate that as the difficulty level increases, the quality of the generated images deteriorates. For example, at the medium difficulty level, the "9 humans" prompt on Imagen-3 demonstrates that the model can generate the correct number of objects; however, at higher difficulty levels, the "15 cats" prompt on Imagen-3 reveals that the model tends to produce unsatisfactory results, underscoring the inherent limitations of diffusion-based text-to-image models. Furthermore, we observe that higher difficulty levels are associated with diminished image detail and fidelity. For instance, in the "15 cats" prompt on GLM-4, the generated image features a cat with two tails. This not only indicates counting difficulties but also suggests that increased task difficulty adversely affects other aspects of the model's performance in certain cases.

**Qualitative Study on the Impact of Scene.** We present a qualitative study on the impact of scene in Figure 9. We observe that the scene context can adversely influence the models' counting ability. For instance, when prompted with "8 trees," the models often generate more trees than specified, frequently relegating many trees to the background. This behavior may stem from a conflict between the models' large-scale prior knowledge (e.g., that many trees typically line streets) and the instruction to produce only a limited number of trees. Additionally, scene context can impact image fidelity; in the "8 trees" example with Dall·E 3, trees are placed in the middle of the road, which contradicts common sense.

**Qualitative Study on the Impact of Style.** We present a qualitative study on the impact of scene in Figure 10. The results indicate that altering the style does not overcome the inherent counting limitations of text-to-image diffusion models. Specifically, for the "8 flowers" prompt in a cartoon style, all models fail to produce the correct number of flowers. Moreover, in the "10 humans" example rendered in cartoon style, the image generated by GLM-4 exhibits noticeable facial distortions. These findings suggest that while style variations can modify visual aesthetics, they may also affect the overall fidelity of the generated images.

**Qualitative Study on Prompt Refinement Results.** To support our observations on prompt refinement, we present a qualitative study in Figure 11. This figure shows that for the multiplicative refinement prompt "2 times 7 apples," almost all models fail to adhere to both numbers (2 and 7), completely disregarding the instruction; only Recraft V3 manages to generate 2 columns, while still failing to generate the correct number of objects. For the additive prompt, models appear to misinterpret "6 + 6" as simply 6. Furthermore, with more complex prompts such as grid prior or position guidance, nearly all models struggle with the subtasks—they fail to correctly interpret directions (e.g., left and right) or generate a grid with the correct number of rows and columns, and in some cases, do not generate a grid at all. These diverse failure cases further reinforce our quantitative findings that prompt refinement does not overcome the counting limitations of text-to-image diffusion models.

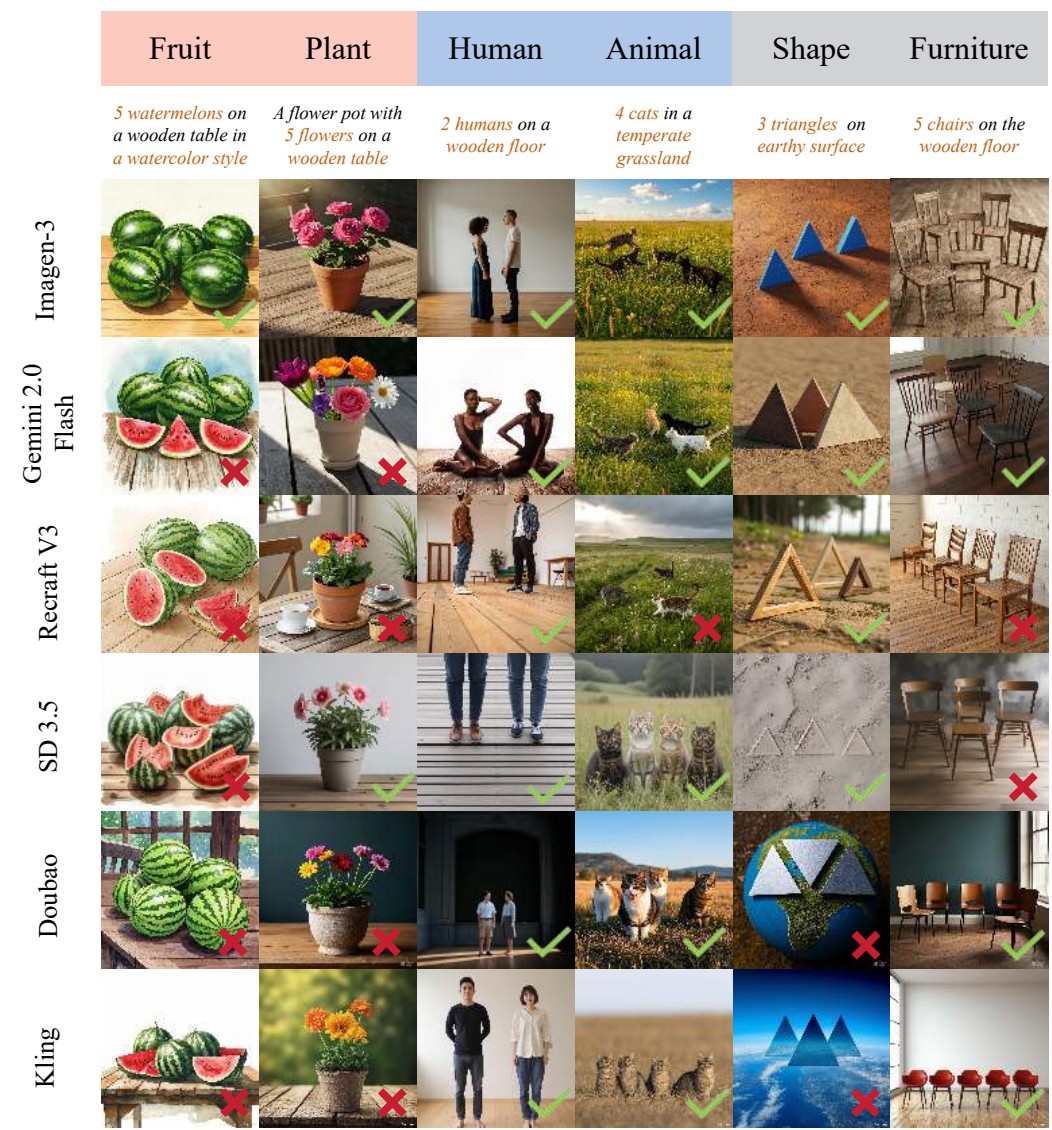

Figure 7: **Qualitative Study on Main Results.** This figure presents the qualitative study of the main results in Section 4.1. We selected the two best models (top two rows) with the highest average accuracy in Table 2, the two worst models (middle two rows) with the lowest average accuracy in Table 2, and two additional models (bottom two rows) that exhibit distinct behaviors. Correct images are marked with a tick, whereas erroneous images are indicated with a cross.

## E POTENTIAL RISKS

One potential risk of our work is that the suggested directions for improving counting abilities in diffusion-based text-to-image models may lead to more realistic image generation, which could be misused to mislead the public. However, we believe that existing safeguard mechanisms for diffusion models remain effective for mitigating such risks. Moreover, our work focuses solely on benchmarking and does not involve releasing any new large pretrained models. Therefore, we do not foresee any negative societal impact resulting from this study.

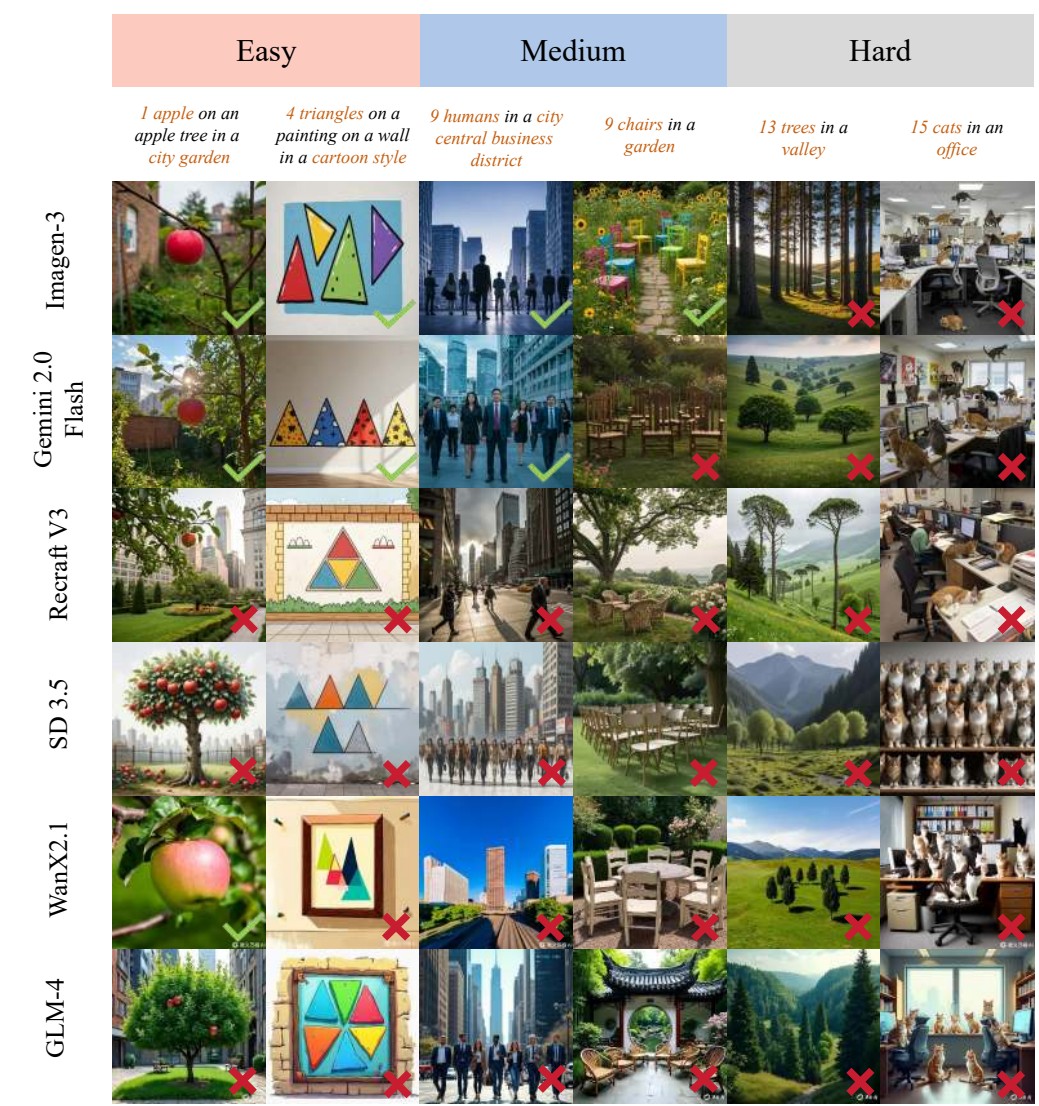

Figure 8: **Qualitative Study on Different Difficulty Levels.** This figure presents the qualitative study on the different difficulty levels in Section 4.2. We selected the two best models (top two rows in this figure) with the highest average accuracy in Table 2, the two worst models (middle two rows in this figure) with the lowest average accuracy in Table 2, and two additional models (bottom two rows in this figure) that exhibit distinct behaviors. Correct images are marked with a tick, whereas erroneous images are indicated with a cross.

## F    DETAILED RESULTS

In this section, we present the detailed results for each prompt used in our experiments across all models in tables 4–39. The models' capability in generating exactly 1-15 objects is thoroughly demonstrated here, instead of focusing on difficulty levels.

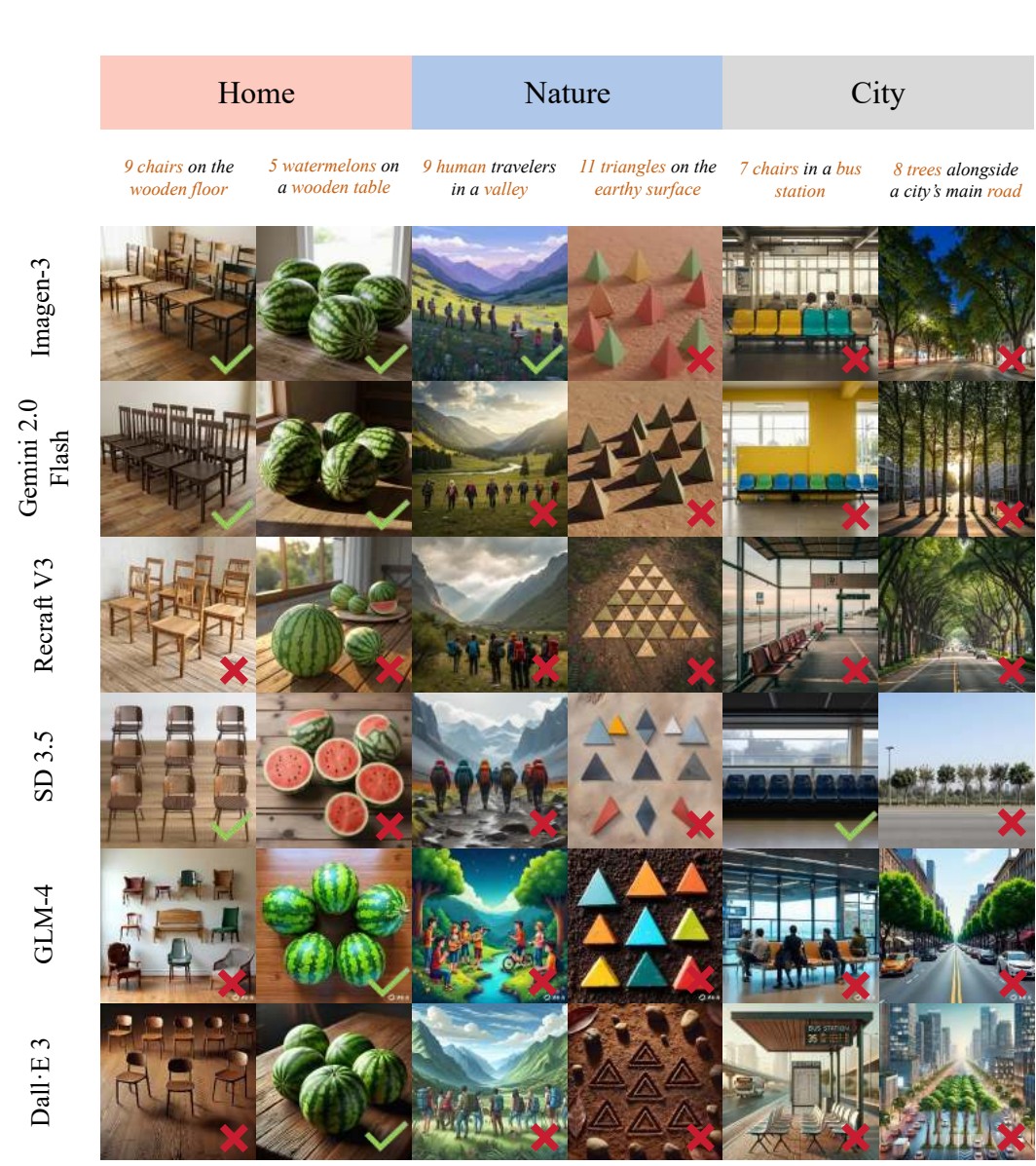

Figure 9: **Qualitative Study on the Impact of Scene.** This figure presents the qualitative study of the impact of scene in Section 4.2. We selected the two best models (top two rows in this figure) with the highest average accuracy in Table 2, the two worst models (middle two rows) with the lowest average accuracy in Table 2, and two additional models (bottom two rows) that exhibit distinct behaviors. Correct images are marked with a tick, whereas erroneous images are indicated with a cross.

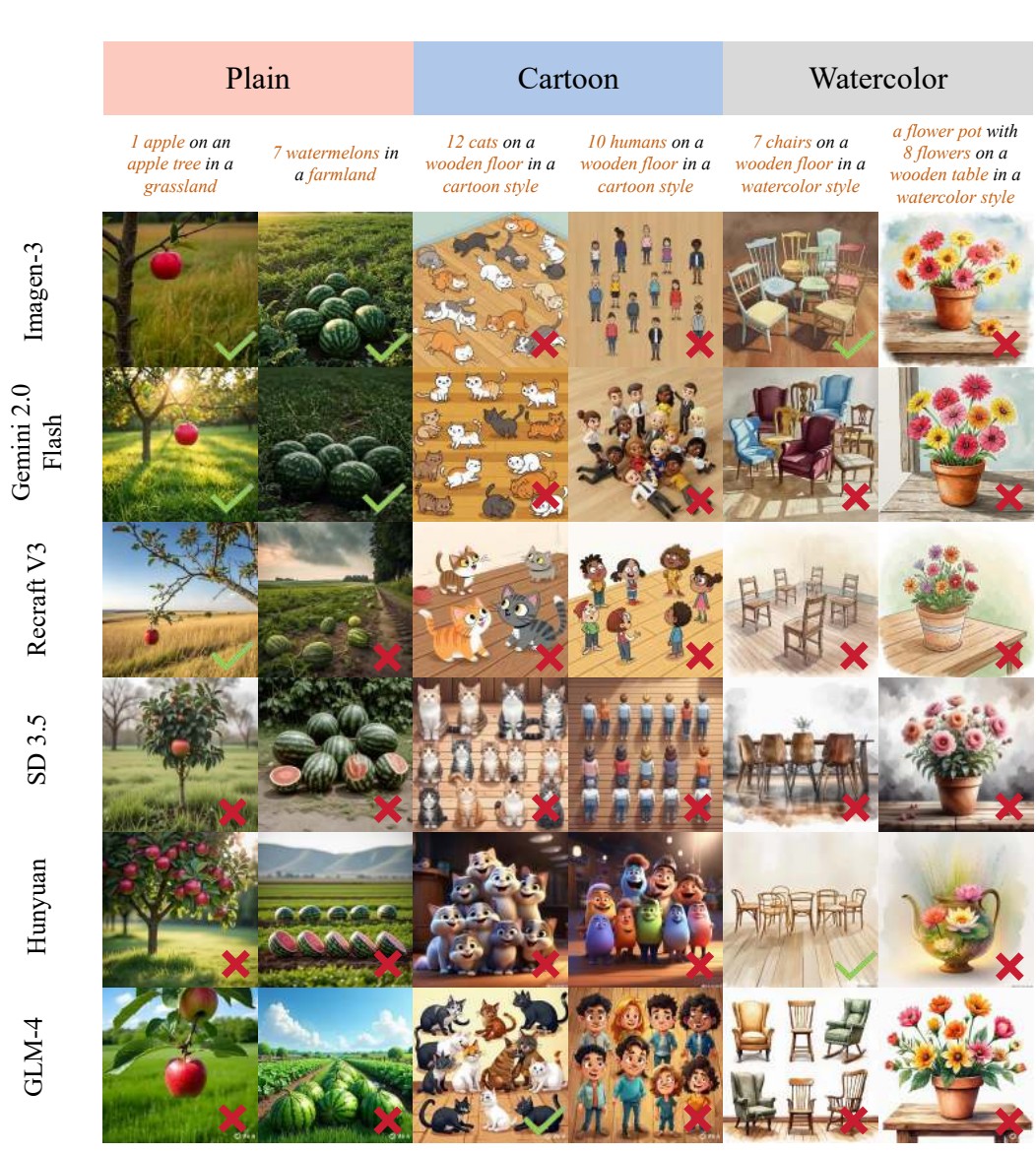

Figure 10: **Qualitative Study on the Impact of Style.** This figure presents the qualitative study on the impact of style in Section 4.2. We selected the two best models (top two rows in this figure) with the highest average accuracy in Table 2, the two worst models (middle two rows in this figure) with the lowest average accuracy in Table 2, and two additional models (bottom two rows in this figure) that exhibit distinct behaviors. Correct images are marked with a tick, whereas erroneous images are indicated with a cross.

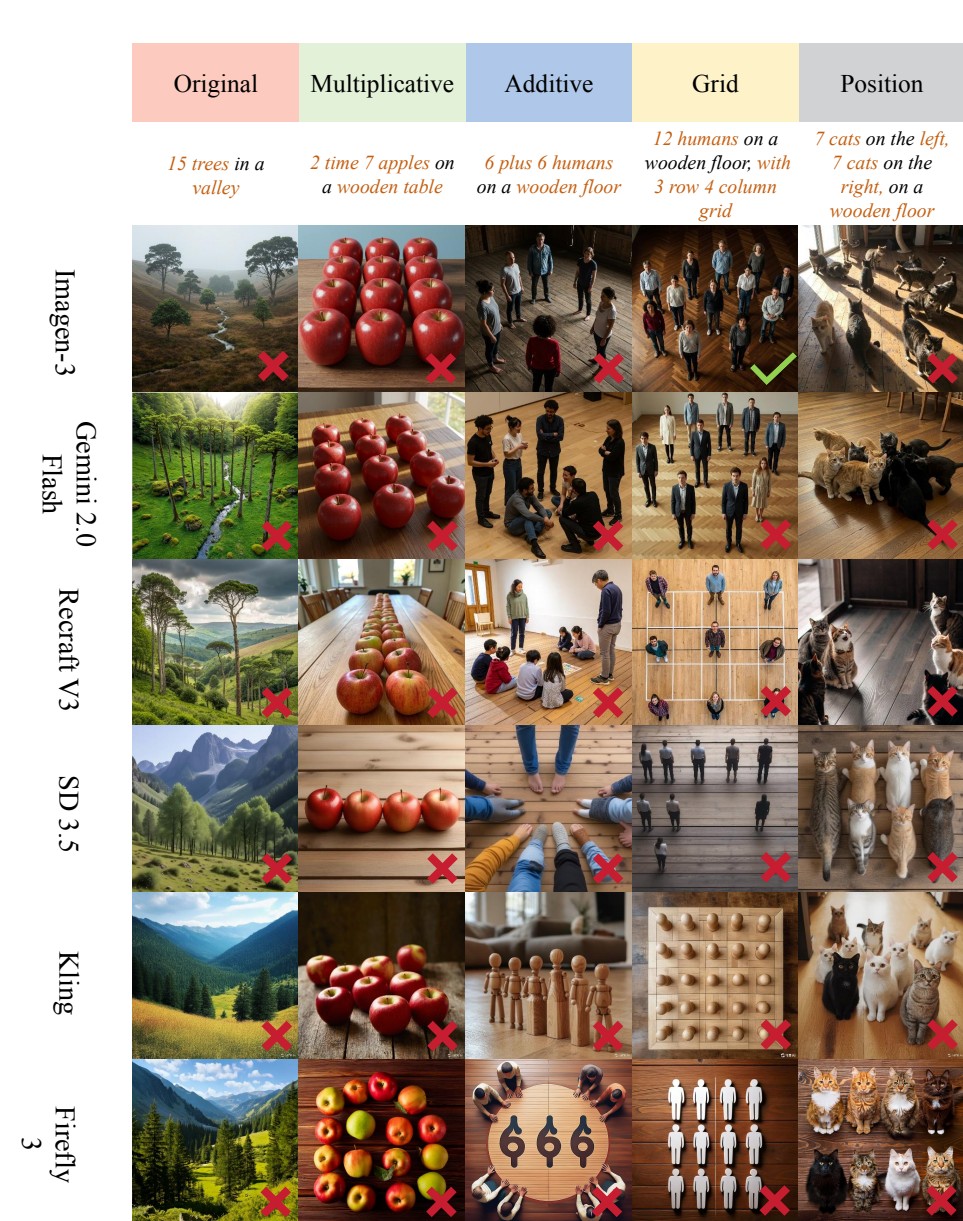

Figure 11: **Qualitative Study on Prompt Refinement Results.** This figure presents the qualitative study of the prompt refinement results in Section 5.1. We selected the two best models (top two rows in this figure) with the highest average accuracy in Table 2, the two worst models (middle two rows in this figure) with the lowest average accuracy in Table 2, and two additional models (bottom two rows in this figure) that exhibit distinct behaviors. Correct images are marked with a tick, whereas erroneous images are indicated with a cross.

| Objects | Model | 1 | 2 | 3 | 4 | 5 | 6 | 7 | 8 | 9 | 10 | 11 | 12 | 13 | 14 | 15 | Correct | Wrong |
|---|---|---|---|---|---|---|---|---|---|---|---|---|---|---|---|---|---|---|
| Apples | Recraft V3 | 1 | 0 | 1 | 1 | 1 | 0 | 0 | 1 | 1 | 0 | 0 | 0 | 1 | 0 | 0 | 7 | 8 |
| | Imagen-3 | 1 | 1 | 1 | 1 | 1 | 1 | 0 | 1 | 0 | 1 | 1 | 1 | 0 | 0 | 0 | 10 | 5 |
| | Dall·E 3 | 1 | 1 | 1 | 1 | 0 | 0 | 0 | 0 | 0 | 0 | 0 | 0 | 0 | 0 | 0 | 4 | 11 |
| | Grok 3 | 1 | 1 | 1 | 1 | 0 | 0 | 0 | 0 | 0 | 0 | 0 | 1 | 0 | 0 | 0 | 5 | 10 |
| | Gemini 2.0 Flash | 1 | 1 | 1 | 1 | 1 | 1 | 1 | 0 | 0 | 0 | 0 | 1 | 0 | 0 | 0 | 8 | 7 |
| | FLUX 1.1 | 1 | 1 | 1 | 1 | 1 | 0 | 0 | 0 | 1 | 0 | 0 | 1 | 0 | 1 | 0 | 7 | 8 |
| | Firefly 3 | 1 | 1 | 1 | 1 | 0 | 0 | 0 | 0 | 1 | 0 | 0 | 1 | 0 | 0 | 0 | 6 | 9 |
| | SD 3.5 | 1 | 1 | 0 | 1 | 1 | 1 | 0 | 1 | 1 | 0 | 0 | 0 | 0 | 0 | 0 | 7 | 8 |
| | Doubao | 1 | 1 | 1 | 1 | 1 | 0 | 1 | 0 | 1 | 0 | 0 | 0 | 0 | 0 | 0 | 7 | 8 |
| | Qwen2.5-Max | 1 | 1 | 1 | 1 | 1 | 1 | 0 | 0 | 1 | 0 | 0 | 0 | 0 | 0 | 0 | 7 | 8 |
| | WanX2.1 | 1 | 1 | 1 | 1 | 1 | 1 | 1 | 0 | 1 | 0 | 0 | 0 | 0 | 0 | 0 | 8 | 7 |
| | Kling | 1 | 1 | 1 | 1 | 0 | 0 | 0 | 1 | 0 | 0 | 0 | 1 | 0 | 0 | 0 | 6 | 9 |
| | Star-3 Alpha | 1 | 1 | 1 | 1 | 1 | 0 | 1 | 1 | 1 | 1 | 0 | 0 | 0 | 0 | 0 | 9 | 6 |
| | Hunyuan | 1 | 0 | 1 | 1 | 1 | 1 | 1 | 0 | 0 | 0 | 0 | 1 | 0 | 0 | 0 | 8 | 7 |
| | GLM-4 | 1 | 1 | 1 | 1 | 1 | 0 | 0 | 1 | 1 | 0 | 0 | 0 | 0 | 0 | 0 | 7 | 8 |
| Watermelons | Recraft V3 | 1 | 1 | 1 | 1 | 0 | 1 | 0 | 0 | 0 | 0 | 0 | 0 | 0 | 0 | 0 | 5 | 10 |
| | Imagen-3 | 1 | 1 | 1 | 1 | 1 | 1 | 1 | 1 | 0 | 0 | 1 | 1 | 1 | 1 | 0 | 12 | 3 |
| | Dall·E 3 | 1 | 1 | 1 | 1 | 1 | 0 | 0 | 0 | 0 | 0 | 0 | 0 | 0 | 0 | 0 | 5 | 10 |
| | Grok 3 | 1 | 1 | 1 | 1 | 0 | 0 | 0 | 0 | 0 | 0 | 0 | 0 | 0 | 0 | 0 | 4 | 11 |
| | Gemini 2.0 Flash | 1 | 1 | 1 | 1 | 1 | 1 | 0 | 0 | 1 | 0 | 0 | 1 | 0 | 0 | 0 | 8 | 7 |
| | FLUX 1.1 | 1 | 1 | 1 | 1 | 0 | 0 | 0 | 0 | 0 | 0 | 0 | 1 | 0 | 1 | 0 | 6 | 9 |
| | Firefly 3 | 1 | 1 | 1 | 1 | 0 | 0 | 0 | 1 | 1 | 0 | 0 | 1 | 0 | 0 | 0 | 7 | 8 |
| | SD 3.5 | 0 | 0 | 0 | 0 | 0 | 0 | 0 | 0 | 0 | 0 | 0 | 0 | 0 | 0 | 0 | 0 | 15 |
| | Doubao | 1 | 1 | 1 | 1 | 1 | 1 | 1 | 0 | 1 | 0 | 0 | 0 | 0 | 0 | 0 | 8 | 7 |
| | Qwen2.5-Max | 1 | 1 | 1 | 1 | 1 | 1 | 0 | 0 | 0 | 0 | 0 | 0 | 0 | 0 | 0 | 6 | 9 |
| | WanX2.1 | 1 | 1 | 1 | 1 | 1 | 0 | 0 | 0 | 1 | 0 | 0 | 0 | 1 | 0 | 0 | 7 | 8 |
| | Kling | 1 | 1 | 1 | 1 | 1 | 0 | 1 | 0 | 0 | 0 | 0 | 0 | 0 | 0 | 0 | 6 | 9 |
| | Star-3 Alpha | 0 | 1 | 0 | 0 | 0 | 0 | 1 | 0 | 0 | 0 | 0 | 1 | 0 | 0 | 0 | 3 | 12 |
| | Hunyuan | 1 | 1 | 0 | 1 | 1 | 1 | 0 | 0 | 0 | 1 | 0 | 1 | 0 | 0 | 0 | 7 | 8 |
| | GLM-4 | 1 | 1 | 1 | 1 | 1 | 0 | 1 | 0 | 1 | 0 | 0 | 0 | 0 | 0 | 0 | 7 | 8 |

Table 4: **Counting Apples and Watermelons in Home Scene Results.**

| Objects | Model | 1 | 2 | 3 | 4 | 5 | 6 | 7 | 8 | 9 | 10 | 11 | 12 | 13 | 14 | 15 | Correct | Wrong |
|---|---|---|---|---|---|---|---|---|---|---|---|---|---|---|---|---|---|---|
| Humans | Recraft V3 | 1 | 1 | 1 | 1 | 1 | 1 | 1 | 1 | 0 | 1 | 0 | 0 | 0 | 0 | 0 | 9 | 6 |
| | Imagen-3 | 1 | 1 | 1 | 1 | 1 | 1 | 1 | 1 | 1 | 1 | 1 | 0 | 0 | 0 | 0 | 11 | 4 |
| | Dall·E 3 | 1 | 1 | 1 | 1 | 1 | 0 | 0 | 1 | 0 | 1 | 0 | 0 | 0 | 0 | 0 | 7 | 8 |
| | Grok 3 | 1 | 1 | 1 | 1 | 0 | 1 | 1 | 1 | 0 | 1 | 1 | 1 | 1 | 0 | 0 | 11 | 4 |
| | Gemini 2.0 Flash | 1 | 1 | 1 | 1 | 1 | 1 | 1 | 1 | 0 | 0 | 0 | 0 | 0 | 0 | 0 | 8 | 7 |
| | FLUX 1.1 | 1 | 1 | 1 | 1 | 1 | 1 | 1 | 0 | 0 | 0 | 0 | 0 | 0 | 0 | 0 | 7 | 8 |
| | Firefly 3 | 1 | 1 | 1 | 1 | 1 | 1 | 0 | 0 | 0 | 0 | 0 | 0 | 0 | 0 | 0 | 6 | 9 |
| | SD 3.5 | 1 | 1 | 1 | 1 | 1 | 0 | 0 | 0 | 0 | 0 | 0 | 0 | 0 | 0 | 0 | 5 | 10 |
| | Doubao | 1 | 1 | 1 | 1 | 1 | 1 | 1 | 1 | 0 | 1 | 1 | 1 | 0 | 0 | 0 | 11 | 4 |
| | Qwen2.5-Max | 1 | 1 | 1 | 0 | 1 | 0 | 1 | 1 | 1 | 0 | 1 | 0 | 0 | 0 | 0 | 8 | 7 |
| | WanX2.1 | 1 | 1 | 1 | 1 | 1 | 1 | 1 | 1 | 1 | 1 | 0 | 1 | 0 | 0 | 0 | 11 | 4 |
| | Kling | 1 | 1 | 1 | 1 | 1 | 1 | 0 | 1 | 0 | 0 | 1 | 0 | 0 | 1 | 0 | 9 | 6 |
| | Star-3 Alpha | 1 | 1 | 1 | 0 | 1 | 0 | 0 | 0 | 0 | 0 | 0 | 0 | 1 | 0 | 0 | 5 | 10 |
| | Hunyuan | 1 | 1 | 1 | 1 | 1 | 0 | 0 | 0 | 1 | 0 | 1 | 0 | 0 | 0 | 0 | 7 | 8 |
| | GLM-4 | 1 | 1 | 1 | 1 | 1 | 0 | 1 | 0 | 0 | 0 | 0 | 0 | 0 | 0 | 0 | 6 | 9 |
| Cats | Recraft V3 | 1 | 1 | 1 | 1 | 1 | 0 | 0 | 0 | 0 | 0 | 0 | 0 | 0 | 0 | 0 | 5 | 10 |
| | Imagen-3 | 1 | 1 | 1 | 1 | 1 | 0 | 1 | 1 | 1 | 0 | 0 | 0 | 1 | 0 | 0 | 9 | 6 |
| | Dall·E 3 | 1 | 1 | 1 | 1 | 1 | 0 | 0 | 0 | 0 | 0 | 0 | 0 | 0 | 0 | 0 | 5 | 10 |
| | Grok 3 | 1 | 1 | 1 | 1 | 0 | 0 | 0 | 0 | 0 | 0 | 0 | 0 | 0 | 0 | 0 | 4 | 11 |
| | Gemini 2.0 Flash | 1 | 1 | 1 | 1 | 0 | 1 | 0 | 1 | 0 | 0 | 0 | 0 | 1 | 0 | 0 | 7 | 8 |
| | FLUX 1.1 | 1 | 1 | 1 | 1 | 1 | 1 | 1 | 0 | 0 | 0 | 0 | 0 | 0 | 0 | 0 | 7 | 8 |
| | Firefly 3 | 1 | 1 | 1 | 1 | 1 | 1 | 1 | 0 | 0 | 0 | 0 | 0 | 1 | 0 | 0 | 8 | 7 |
| | SD 3.5 | 1 | 1 | 1 | 1 | 1 | 0 | 1 | 0 | 0 | 0 | 0 | 0 | 0 | 0 | 0 | 6 | 9 |
| | Doubao | 1 | 1 | 1 | 1 | 1 | 0 | 0 | 0 | 0 | 1 | 0 | 0 | 0 | 0 | 0 | 6 | 9 |
| | Qwen2.5-Max | 1 | 1 | 1 | 1 | 0 | 1 | 0 | 0 | 0 | 1 | 1 | 0 | 0 | 0 | 0 | 6 | 9 |
| | WanX2.1 | 1 | 1 | 1 | 1 | 1 | 1 | 1 | 1 | 1 | 0 | 0 | 0 | 0 | 0 | 0 | 9 | 6 |
| | Kling | 1 | 1 | 1 | 1 | 1 | 1 | 0 | 0 | 0 | 1 | 0 | 0 | 0 | 0 | 1 | 8 | 7 |
| | Star-3 Alpha | 1 | 1 | 1 | 0 | 1 | 1 | 0 | 1 | 0 | 0 | 0 | 0 | 0 | 0 | 0 | 6 | 8 |
| | Hunyuan | 1 | 1 | 1 | 1 | 1 | 1 | 1 | 0 | 1 | 1 | 1 | 1 | 1 | 1 | 0 | 13 | 2 |
| | GLM-4 | 1 | 1 | 1 | 1 | 1 | 0 | 1 | 0 | 0 | 0 | 0 | 0 | 0 | 0 | 0 | 6 | 9 |

Table 5: **Counting Human and Animals in Home Scene Results.**

| Objects | Model | 1 | 2 | 3 | 4 | 5 | 6 | 7 | 8 | 9 | 10 | 11 | 12 | 13 | 14 | 15 | Correct | Wrong |
|---|---|---|---|---|---|---|---|---|---|---|---|---|---|---|---|---|---|---|
| Triangles | Recraft V3 | 1 | 1 | 1 | 1 | 0 | 0 | 0 | 0 | 0 | 1 | 1 | 0 | 0 | 0 | 0 | 6 | 9 |
| | Imagen-3 | 1 | 1 | 1 | 0 | 1 | 0 | 1 | 0 | 1 | 1 | 0 | 1 | 0 | 0 | 0 | 8 | 7 |
| | Dall·E 3 | 1 | 1 | 1 | 1 | 1 | 0 | 0 | 1 | 0 | 0 | 0 | 1 | 0 | 0 | 0 | 7 | 8 |
| | Grok 3 | 1 | 1 | 0 | 0 | 0 | 1 | 0 | 1 | 1 | 0 | 0 | 0 | 0 | 0 | 0 | 5 | 10 |
| | Gemini 2.0 Flash | 1 | 1 | 1 | 1 | 1 | 1 | 0 | 1 | 1 | 1 | 0 | 0 | 0 | 1 | 0 | 10 | 5 |
| | FLUX 1.1 | 1 | 1 | 1 | 1 | 1 | 0 | 1 | 0 | 0 | 0 | 0 | 1 | 0 | 0 | 0 | 7 | 8 |
| | Firefly 3 | 1 | 1 | 1 | 1 | 1 | 0 | 0 | 0 | 1 | 0 | 0 | 0 | 0 | 0 | 0 | 6 | 9 |
| | SD 3.5 | 1 | 1 | 1 | 0 | 0 | 0 | 1 | 1 | 0 | 0 | 0 | 0 | 0 | 0 | 0 | 6 | 9 |
| | Doubao | 1 | 1 | 1 | 1 | 0 | 0 | 1 | 0 | 0 | 0 | 0 | 0 | 0 | 0 | 0 | 6 | 9 |
| | Qwen2.5-Max | 1 | 1 | 0 | 0 | 1 | 1 | 0 | 0 | 1 | 0 | 0 | 0 | 0 | 0 | 0 | 4 | 11 |
| | WanX2.1 | 1 | 1 | 1 | 1 | 1 | 0 | 0 | 1 | 0 | 0 | 0 | 0 | 0 | 0 | 0 | 6 | 9 |
| | Kling | 1 | 1 | 1 | 1 | 1 | 0 | 0 | 0 | 1 | 0 | 0 | 0 | 0 | 0 | 0 | 6 | 9 |
| | Star-3 Alpha | 1 | 1 | 1 | 0 | 0 | 0 | 0 | 0 | 1 | 1 | 0 | 0 | 0 | 0 | 0 | 5 | 10 |
| | Hunyuan | 1 | 1 | 1 | 1 | 1 | 0 | 0 | 0 | 1 | 0 | 0 | 0 | 0 | 0 | 0 | 6 | 9 |
| | GLM-4 | 1 | 1 | 1 | 1 | 1 | 1 | 0 | 0 | 0 | 0 | 0 | 1 | 0 | 0 | 1 | 8 | 7 |
| Chairs | Recraft V3 | 1 | 1 | 0 | 0 | 0 | 0 | 0 | 0 | 0 | 0 | 0 | 0 | 0 | 0 | 0 | 2 | 13 |
| | Imagen-3 | 1 | 1 | 1 | 0 | 0 | 1 | 1 | 1 | 0 | 0 | 0 | 0 | 0 | 0 | 0 | 6 | 9 |
| | Dall·E 3 | 1 | 1 | 1 | 1 | 0 | 0 | 0 | 0 | 0 | 0 | 0 | 0 | 0 | 0 | 0 | 4 | 11 |
| | Grok 3 | 1 | 1 | 1 | 0 | 1 | 0 | 0 | 0 | 0 | 0 | 0 | 1 | 0 | 0 | 0 | 5 | 10 |
| | Gemini 2.0 Flash | 1 | 1 | 0 | 0 | 1 | 0 | 0 | 0 | 0 | 0 | 0 | 0 | 0 | 0 | 0 | 3 | 12 |
| | FLUX 1.1 | 1 | 1 | 0 | 0 | 0 | 0 | 0 | 0 | 0 | 0 | 0 | 0 | 0 | 0 | 0 | 2 | 13 |
| | Firefly 3 | 1 | 1 | 1 | 1 | 0 | 1 | 0 | 1 | 1 | 0 | 0 | 0 | 0 | 1 | 0 | 8 | 7 |
| | SD 3.5 | 0 | 1 | 0 | 1 | 0 | 0 | 0 | 0 | 0 | 0 | 0 | 0 | 0 | 0 | 0 | 2 | 13 |
| | Doubao | 1 | 1 | 1 | 0 | 1 | 1 | 1 | 0 | 0 | 0 | 0 | 0 | 0 | 0 | 0 | 6 | 9 |
| | Qwen 2.5-Max | 1 | 0 | 1 | 1 | 1 | 1 | 0 | 0 | 1 | 0 | 0 | 0 | 1 | 1 | 0 | 8 | 7 |
| | WanX2.1 | 1 | 1 | 0 | 1 | 0 | 0 | 0 | 0 | 0 | 0 | 0 | 0 | 0 | 1 | 0 | 4 | 11 |
| | Kling | 1 | 1 | 0 | 1 | 0 | 0 | 0 | 0 | 0 | 0 | 0 | 0 | 0 | 0 | 0 | 3 | 12 |
| | Star-3 Alpha | 1 | 1 | 0 | 0 | 1 | 0 | 0 | 0 | 1 | 0 | 1 | 0 | 0 | 0 | 0 | 5 | 10 |
| | Hunyuan | 1 | 1 | 1 | 0 | 0 | 0 | 0 | 0 | 1 | 0 | 0 | 0 | 0 | 0 | 0 | 4 | 11 |
| | GLM-4 | 1 | 1 | 1 | 1 | 1 | 1 | 0 | 0 | 1 | 0 | 0 | 0 | 0 | 0 | 0 | 7 | 8 |

Table 6: **Counting Triangles and Chairs in Home Scene Results.**

| Objects | Model | 1 | 2 | 3 | 4 | 5 | 6 | 7 | 8 | 9 | 10 | 11 | 12 | 13 | 14 | 15 | Correct | Wrong |
|---|---|---|---|---|---|---|---|---|---|---|---|---|---|---|---|---|---|---|
| Flowers | Recraft V3 | 1 | 1 | 1 | 1 | 0 | 0 | 0 | 0 | 0 | 0 | 0 | 0 | 0 | 1 | 0 | 5 | 10 |
| | Imagen-3 | 1 | 1 | 0 | 0 | 1 | 0 | 0 | 0 | 0 | 1 | 0 | 0 | 0 | 0 | 1 | 5 | 10 |
| | Dall·E 3 | 1 | 1 | 1 | 1 | 1 | 0 | 1 | 0 | 0 | 0 | 0 | 0 | 0 | 0 | 0 | 6 | 9 |
| | Grok 3 | 1 | 1 | 0 | 0 | 0 | 0 | 0 | 0 | 0 | 0 | 0 | 0 | 0 | 0 | 0 | 2 | 13 |
| | Gemini 2.0 Flash | 1 | 1 | 1 | 0 | 0 | 0 | 0 | 1 | 0 | 0 | 0 | 0 | 0 | 0 | 0 | 4 | 11 |
| | FLUX 1.1 | 1 | 1 | 0 | 1 | 0 | 0 | 0 | 0 | 0 | 0 | 0 | 0 | 0 | 0 | 0 | 4 | 11 |
| | Firefly 3 | 1 | 1 | 1 | 0 | 0 | 0 | 0 | 0 | 0 | 0 | 0 | 0 | 0 | 0 | 0 | 3 | 12 |
| | SD 3.5 | 1 | 1 | 1 | 0 | 1 | 0 | 0 | 0 | 0 | 0 | 0 | 0 | 0 | 0 | 0 | 4 | 11 |
| | Doubao | 1 | 1 | 1 | 1 | 0 | 1 | 0 | 1 | 0 | 0 | 0 | 0 | 0 | 0 | 0 | 6 | 9 |
| | Qwen2.5-Max | 1 | 0 | 1 | 0 | 0 | 0 | 0 | 0 | 0 | 0 | 0 | 0 | 0 | 0 | 0 | 2 | 13 |
| | WanX2.1 | 1 | 1 | 1 | 1 | 1 | 0 | 0 | 0 | 0 | 0 | 0 | 1 | 0 | 0 | 0 | 7 | 8 |
| | Kling | 1 | 1 | 1 | 0 | 0 | 1 | 0 | 0 | 0 | 0 | 0 | 0 | 0 | 0 | 0 | 4 | 11 |
| | Star-3 Alpha | 1 | 1 | 0 | 1 | 0 | 1 | 0 | 0 | 0 | 0 | 0 | 0 | 0 | 0 | 0 | 4 | 11 |
| | Hunyuan | 1 | 1 | 1 | 0 | 0 | 0 | 0 | 0 | 0 | 0 | 0 | 0 | 0 | 0 | 0 | 3 | 12 |
| | GLM-4 | 1 | 1 | 1 | 0 | 0 | 1 | 1 | 1 | 1 | 0 | 0 | 0 | 0 | 0 | 0 | 7 | 8 |

Table 7: **Counting Flowers in Home Scene Results.**

| Objects | Model | 1 | 2 | 3 | 4 | 5 | 6 | 7 | 8 | 9 | 10 | 11 | 12 | 13 | 14 | 15 | Correct | Wrong |
|---|---|---|---|---|---|---|---|---|---|---|---|---|---|---|---|---|---|---|
| Apples | Recraft V3 | 1 | 0 | 0 | 1 | 0 | 0 | 0 | 0 | 0 | 0 | 0 | 0 | 0 | 0 | 0 | 2 | 13 |
| | Imagen-3 | 1 | 1 | 1 | 1 | 0 | 0 | 0 | 0 | 0 | 0 | 0 | 0 | 0 | 0 | 0 | 4 | 11 |
| | Dall·E 3 | 1 | 1 | 1 | 1 | 0 | 0 | 0 | 0 | 1 | 0 | 0 | 0 | 0 | 0 | 0 | 5 | 10 |
| | Grok 3 | 1 | 1 | 0 | 1 | 0 | 0 | 0 | 0 | 0 | 0 | 0 | 0 | 0 | 0 | 0 | 3 | 12 |
| | Gemini 2.0 Flash | 1 | 1 | 1 | 0 | 0 | 0 | 0 | 0 | 0 | 0 | 0 | 0 | 0 | 0 | 0 | 3 | 12 |
| | FLUX 1.1 | 1 | 1 | 0 | 0 | 0 | 0 | 0 | 0 | 0 | 0 | 0 | 0 | 0 | 0 | 0 | 2 | 13 |
| | Firefly 3 | 1 | 0 | 0 | 0 | 0 | 0 | 0 | 0 | 0 | 0 | 0 | 0 | 0 | 0 | 0 | 1 | 14 |
| | SD 3.5 | 0 | 0 | 0 | 1 | 0 | 0 | 0 | 0 | 0 | 0 | 0 | 0 | 0 | 0 | 0 | 1 | 14 |
| | Doubao | 1 | 1 | 1 | 0 | 0 | 1 | 1 | 0 | 0 | 0 | 0 | 0 | 0 | 0 | 0 | 5 | 10 |
| | Qwen2.5-Max | 1 | 1 | 1 | 1 | 1 | 0 | 0 | 0 | 0 | 1 | 0 | 0 | 0 | 0 | 0 | 6 | 9 |
| | WanX2.1 | 1 | 1 | 1 | 1 | 0 | 0 | 0 | 0 | 0 | 0 | 0 | 0 | 0 | 0 | 0 | 4 | 11 |
| | Kling | 0 | 0 | 0 | 0 | 0 | 0 | 1 | 0 | 1 | 0 | 0 | 0 | 0 | 0 | 0 | 2 | 13 |
| | Star-3 Alpha | 1 | 1 | 1 | 1 | 0 | 1 | 1 | 0 | 0 | 0 | 0 | 0 | 0 | 0 | 0 | 6 | 9 |
| | Hunyuan | 0 | 0 | 0 | 0 | 0 | 0 | 0 | 0 | 0 | 0 | 0 | 0 | 0 | 0 | 0 | 0 | 15 |
| | GLM-4 | 0 | 1 | 0 | 0 | 0 | 0 | 0 | 0 | 0 | 0 | 0 | 0 | 0 | 0 | 0 | 1 | 14 |
| Watermelons | Recraft V3 | 1 | 0 | 1 | 0 | 0 | 0 | 0 | 0 | 0 | 0 | 0 | 0 | 0 | 0 | 0 | 2 | 13 |
| | Imagen-3 | 1 | 0 | 1 | 0 | 1 | 1 | 0 | 0 | 0 | 0 | 0 | 0 | 0 | 0 | 0 | 4 | 11 |
| | Dall·E 3 | 1 | 1 | 1 | 1 | 0 | 0 | 0 | 0 | 0 | 0 | 1 | 0 | 0 | 0 | 0 | 5 | 10 |
| | Grok 3 | 1 | 1 | 1 | 1 | 0 | 1 | 0 | 0 | 1 | 0 | 0 | 0 | 0 | 0 | 0 | 6 | 9 |
| | Gemini 2.0 Flash | 1 | 1 | 1 | 0 | 0 | 1 | 0 | 0 | 0 | 0 | 0 | 0 | 0 | 0 | 0 | 4 | 11 |
| | FLUX 1.1 | 1 | 1 | 1 | 0 | 0 | 0 | 0 | 0 | 0 | 0 | 0 | 0 | 0 | 0 | 0 | 3 | 12 |
| | Firefly 3 | 1 | 1 | 1 | 1 | 1 | 1 | 0 | 1 | 0 | 1 | 0 | 0 | 0 | 0 | 0 | 8 | 7 |
| | SD 3.5 | 1 | 1 | 1 | 0 | 1 | 0 | 0 | 0 | 0 | 0 | 0 | 0 | 0 | 0 | 0 | 4 | 15 |
| | Doubao | 1 | 1 | 1 | 0 | 0 | 1 | 0 | 1 | 0 | 0 | 0 | 0 | 0 | 0 | 0 | 5 | 10 |
| | Qwen2.5-Max | 1 | 1 | 1 | 1 | 1 | 0 | 0 | 0 | 0 | 0 | 1 | 0 | 0 | 0 | 0 | 6 | 9 |
| | WanX2.1 | 1 | 1 | 1 | 1 | 1 | 1 | 0 | 0 | 1 | 0 | 0 | 0 | 0 | 0 | 0 | 7 | 8 |
| | Kling | 1 | 1 | 1 | 1 | 1 | 0 | 0 | 0 | 0 | 0 | 0 | 0 | 0 | 0 | 0 | 5 | 10 |
| | Star-3 Alpha | 1 | 1 | 1 | 1 | 1 | 1 | 0 | 0 | 0 | 0 | 0 | 0 | 0 | 0 | 0 | 6 | 9 |
| | Hunyuan | 1 | 1 | 1 | 1 | 1 | 1 | 0 | 0 | 1 | 0 | 0 | 1 | 0 | 0 | 0 | 7 | 8 |
| | GLM-4 | 1 | 1 | 0 | 1 | 1 | 0 | 0 | 0 | 0 | 0 | 0 | 0 | 0 | 0 | 0 | 4 | 11 |

Table 8: **Counting Apples and Watermelons in Nature Scene Results.**

| Objects | Model | 1 | 2 | 3 | 4 | 5 | 6 | 7 | 8 | 9 | 10 | 11 | 12 | 13 | 14 | 15 | Correct | Wrong |
|---|---|---|---|---|---|---|---|---|---|---|---|---|---|---|---|---|---|---|
| Humans | Recraft V3 | 1 | 1 | 1 | 1 | 0 | 0 | 0 | 0 | 0 | 0 | 0 | 0 | 0 | 0 | 0 | 4 | 11 |
| | Imagen-3 | 1 | 1 | 1 | 1 | 1 | 1 | 1 | 1 | 1 | 0 | 0 | 0 | 1 | 0 | 0 | 10 | 5 |
| | Dall·E 3 | 1 | 1 | 1 | 0 | 1 | 0 | 0 | 0 | 0 | 0 | 0 | 0 | 1 | 0 | 0 | 5 | 10 |
| | Grok 3 | 1 | 1 | 1 | 1 | 1 | 1 | 1 | 1 | 1 | 0 | 1 | 1 | 0 | 0 | 0 | 11 | 4 |
| | Gemini 2.0 Flash | 1 | 1 | 1 | 1 | 0 | 0 | 0 | 0 | 0 | 0 | 0 | 0 | 0 | 0 | 0 | 4 | 11 |
| | FLUX 1.1 | 1 | 1 | 1 | 1 | 0 | 1 | 1 | 0 | 0 | 0 | 0 | 0 | 0 | 0 | 0 | 6 | 9 |
| | Firefly 3 | 1 | 1 | 1 | 1 | 1 | 0 | 0 | 0 | 0 | 0 | 0 | 0 | 0 | 0 | 0 | 5 | 10 |
| | SD 3.5 | 1 | 1 | 1 | 1 | 1 | 0 | 1 | 0 | 1 | 0 | 0 | 0 | 0 | 0 | 0 | 7 | 8 |
| | Doubao | 1 | 1 | 1 | 1 | 1 | 0 | 0 | 0 | 0 | 0 | 0 | 0 | 0 | 0 | 0 | 5 | 10 |
| | Qwen2.5-Max | 1 | 0 | 0 | 1 | 0 | 0 | 0 | 0 | 0 | 0 | 0 | 0 | 0 | 0 | 0 | 2 | 13 |
| | WanX2.1 | 1 | 1 | 1 | 1 | 1 | 0 | 0 | 0 | 0 | 0 | 0 | 0 | 0 | 0 | 0 | 5 | 10 |
| | Kling | 1 | 1 | 0 | 1 | 1 | 0 | 1 | 0 | 1 | 0 | 0 | 0 | 0 | 0 | 0 | 4 | 11 |
| | Star-3 Alpha | 1 | 1 | 1 | 1 | 1 | 0 | 0 | 1 | 0 | 1 | 0 | 0 | 0 | 0 | 0 | 7 | 8 |
| | Hunyuan | 1 | 1 | 1 | 1 | 0 | 0 | 1 | 1 | 0 | 0 | 0 | 0 | 0 | 0 | 0 | 6 | 9 |
| | GLM-4 | 1 | 1 | 0 | 0 | 1 | 1 | 0 | 0 | 0 | 0 | 0 | 0 | 0 | 0 | 0 | 4 | 11 |
| Cats | Recraft V3 | 1 | 1 | 1 | 0 | 0 | 0 | 0 | 0 | 0 | 0 | 0 | 0 | 0 | 0 | 0 | 3 | 12 |
| | Imagen-3 | 1 | 1 | 1 | 1 | 0 | 0 | 0 | 0 | 0 | 1 | 0 | 0 | 0 | 0 | 0 | 5 | 10 |
| | Dall·E 3 | 1 | 1 | 1 | 1 | 1 | 0 | 0 | 0 | 0 | 0 | 0 | 0 | 0 | 0 | 0 | 5 | 10 |
| | Grok 3 | 1 | 1 | 1 | 0 | 0 | 0 | 0 | 0 | 1 | 0 | 0 | 0 | 0 | 0 | 0 | 4 | 11 |
| | Gemini 2.0 Flash | 1 | 1 | 1 | 1 | 1 | 0 | 0 | 0 | 0 | 0 | 0 | 0 | 0 | 0 | 0 | 5 | 10 |
| | FLUX 1.1 | 1 | 1 | 1 | 0 | 0 | 1 | 1 | 0 | 0 | 0 | 0 | 0 | 0 | 0 | 0 | 6 | 9 |
| | Firefly 3 | 1 | 1 | 1 | 1 | 1 | 1 | 1 | 1 | 1 | 0 | 0 | 0 | 0 | 0 | 0 | 9 | 6 |
| | SD 3.5 | 1 | 1 | 1 | 1 | 0 | 0 | 0 | 0 | 1 | 0 | 0 | 0 | 0 | 0 | 0 | 5 | 10 |
| | Doubao | 1 | 1 | 1 | 1 | 1 | 1 | 0 | 0 | 0 | 0 | 0 | 0 | 0 | 0 | 0 | 6 | 9 |
| | Qwen2.5-Max | 1 | 1 | 1 | 1 | 0 | 0 | 1 | 0 | 1 | 0 | 0 | 0 | 0 | 0 | 0 | 6 | 9 |
| | WanX2.1 | 1 | 1 | 1 | 1 | 1 | 1 | 1 | 0 | 0 | 1 | 0 | 0 | 0 | 0 | 0 | 8 | 7 |
| | Kling | 1 | 1 | 1 | 1 | 1 | 0 | 1 | 0 | 0 | 0 | 0 | 0 | 0 | 0 | 0 | 6 | 9 |
| | Star-3 Alpha | 1 | 1 | 1 | 1 | 1 | 0 | 1 | 0 | 0 | 0 | 0 | 0 | 0 | 1 | 0 | 7 | 8 |
| | Hunyuan | 1 | 1 | 1 | 0 | 1 | 1 | 0 | 0 | 0 | 0 | 0 | 0 | 0 | 0 | 0 | 5 | 10 |
| | GLM-4 | 1 | 1 | 1 | 1 | 1 | 1 | 0 | 0 | 1 | 0 | 0 | 0 | 0 | 0 | 0 | 7 | 8 |

Table 9: **Counting Humans and Cats in Nature Scene Results.**

| Objects | Model | 1 | 2 | 3 | 4 | 5 | 6 | 7 | 8 | 9 | 10 | 11 | 12 | 13 | 14 | 15 | Correct | Wrong |
|---|---|---|---|---|---|---|---|---|---|---|---|---|---|---|---|---|---|---|
| Chairs | Recraft V3 | 1 | 1 | 1 | 0 | 0 | 0 | 0 | 0 | 0 | 0 | 0 | 0 | 0 | 0 | 0 | 3 | 12 |
| | Imagen-3 | 1 | 1 | 1 | 1 | 1 | 0 | 0 | 0 | 1 | 1 | 0 | 0 | 0 | 1 | 0 | 8 | 7 |
| | Dall·E 3 | 1 | 1 | 1 | 1 | 0 | 0 | 0 | 0 | 0 | 0 | 0 | 0 | 0 | 0 | 0 | 4 | 11 |
| | Grok 3 | 1 | 1 | 1 | 0 | 0 | 0 | 0 | 0 | 0 | 0 | 0 | 0 | 0 | 0 | 0 | 3 | 12 |
| | Gemini 2.0 Flash | 1 | 1 | 1 | 0 | 0 | 0 | 0 | 0 | 0 | 0 | 0 | 1 | 0 | 0 | 0 | 4 | 11 |
| | FLUX 1.1 | 1 | 1 | 1 | 1 | 0 | 0 | 0 | 1 | 1 | 0 | 0 | 0 | 0 | 0 | 0 | 6 | 9 |
| | Firefly 3 | 1 | 1 | 1 | 1 | 1 | 1 | 0 | 0 | 0 | 0 | 0 | 0 | 0 | 0 | 0 | 6 | 9 |
| | SD 3.5 | 1 | 1 | 1 | 1 | 1 | 0 | 0 | 0 | 0 | 0 | 0 | 0 | 0 | 0 | 0 | 5 | 10 |
| | Doubao | 1 | 1 | 1 | 1 | 0 | 1 | 0 | 0 | 0 | 1 | 0 | 0 | 0 | 0 | 0 | 6 | 9 |
| | Qwen2.5-Max | 1 | 1 | 1 | 0 | 1 | 0 | 0 | 0 | 0 | 0 | 0 | 0 | 0 | 0 | 0 | 4 | 11 |
| | WanX2.1 | 1 | 1 | 1 | 1 | 0 | 1 | 0 | 0 | 0 | 1 | 0 | 0 | 0 | 0 | 0 | 6 | 9 |
| | Kling | 1 | 1 | 1 | 1 | 0 | 0 | 0 | 1 | 0 | 0 | 0 | 1 | 0 | 1 | 0 | 8 | 7 |
| | Star-3 Alpha | 1 | 1 | 0 | 0 | 0 | 1 | 0 | 0 | 1 | 0 | 0 | 0 | 0 | 0 | 0 | 4 | 11 |
| | Hunyuan | 1 | 1 | 1 | 1 | 1 | 0 | 0 | 0 | 0 | 0 | 0 | 0 | 0 | 0 | 0 | 5 | 10 |
| | GLM-4 | 1 | 1 | 1 | 1 | 1 | 0 | 0 | 0 | 0 | 0 | 0 | 0 | 0 | 0 | 0 | 5 | 10 |
| Triangles | Recraft V3 | 1 | 1 | 1 | 0 | 1 | 1 | 0 | 0 | 1 | 0 | 0 | 0 | 0 | 0 | 0 | 6 | 9 |
| | Imagen-3 | 1 | 1 | 1 | 1 | 1 | 0 | 0 | 0 | 1 | 1 | 0 | 0 | 0 | 0 | 1 | 6 | 9 |
| | Dall·E 3 | 1 | 0 | 1 | 0 | 0 | 0 | 0 | 0 | 0 | 0 | 0 | 0 | 0 | 0 | 0 | 2 | 13 |
| | Grok 3 | 1 | 1 | 1 | 0 | 0 | 1 | 0 | 0 | 0 | 0 | 0 | 0 | 0 | 0 | 0 | 4 | 11 |
| | Gemini 2.0 Flash | 1 | 1 | 1 | 0 | 1 | 0 | 1 | 0 | 1 | 0 | 0 | 0 | 0 | 1 | 0 | 7 | 8 |
| | FLUX 1.1 | 1 | 1 | 0 | 0 | 0 | 0 | 0 | 0 | 0 | 0 | 0 | 0 | 0 | 0 | 0 | 2 | 13 |
| | Firefly 3 | 1 | 1 | 0 | 1 | 0 | 0 | 0 | 0 | 1 | 0 | 0 | 1 | 1 | 0 | 0 | 6 | 9 |
| | SD 3.5 | 1 | 0 | 1 | 1 | 1 | 0 | 1 | 0 | 0 | 0 | 0 | 0 | 0 | 1 | 0 | 6 | 9 |
| | Doubao | 0 | 0 | 0 | 0 | 0 | 0 | 0 | 0 | 1 | 0 | 0 | 0 | 0 | 0 | 0 | 1 | 14 |
| | Qwen2.5-Max | 1 | 1 | 1 | 0 | 0 | 1 | 1 | 0 | 0 | 0 | 0 | 0 | 0 | 0 | 0 | 5 | 10 |
| | WanX2.1 | 1 | 1 | 1 | 1 | 0 | 0 | 0 | 0 | 1 | 0 | 0 | 0 | 0 | 0 | 0 | 5 | 10 |
| | Kling | 0 | 0 | 0 | 0 | 0 | 0 | 0 | 0 | 0 | 0 | 0 | 0 | 0 | 0 | 0 | 0 | 15 |
| | Star-3 Alpha | 1 | 1 | 1 | 0 | 0 | 0 | 1 | 0 | 1 | 0 | 0 | 1 | 1 | 0 | 0 | 7 | 8 |
| | Hunyuan | 1 | 1 | 1 | 0 | 0 | 0 | 0 | 0 | 0 | 1 | 0 | 1 | 0 | 0 | 0 | 5 | 10 |
| | GLM-4 | 1 | 0 | 0 | 1 | 0 | 0 | 0 | 0 | 1 | 0 | 0 | 1 | 0 | 0 | 0 | 4 | 11 |

Table 10: **Counting Triangles and Chairs in Nature Scene Results.**

| Objects | Model | 1 | 2 | 3 | 4 | 5 | 6 | 7 | 8 | 9 | 10 | 11 | 12 | 13 | 14 | 15 | Correct | Wrong |
|---|---|---|---|---|---|---|---|---|---|---|---|---|---|---|---|---|---|---|
| Trees | Recraft V3 | 1 | 1 | 0 | 0 | 0 | 0 | 0 | 0 | 0 | 0 | 0 | 0 | 0 | 0 | 0 | 2 | 13 |
| | Imagen-3 | 1 | 1 | 1 | 0 | 1 | 1 | 0 | 0 | 0 | 0 | 0 | 0 | 0 | 0 | 0 | 5 | 10 |
| | Dall·E 3 | 1 | 1 | 1 | 0 | 0 | 0 | 0 | 0 | 0 | 0 | 0 | 0 | 0 | 0 | 0 | 3 | 12 |
| | Grok 3 | 1 | 1 | 0 | 1 | 0 | 1 | 0 | 0 | 0 | 0 | 0 | 0 | 0 | 0 | 0 | 4 | 11 |
| | Gemini 2.0 Flash | 1 | 1 | 0 | 1 | 0 | 1 | 1 | 0 | 0 | 1 | 0 | 0 | 0 | 0 | 0 | 6 | 9 |
| | FLUX 1.1 | 1 | 1 | 1 | 1 | 1 | 0 | 0 | 0 | 0 | 0 | 0 | 0 | 0 | 0 | 0 | 5 | 10 |
| | Firefly 3 | 1 | 1 | 1 | 1 | 0 | 1 | 0 | 1 | 0 | 0 | 0 | 0 | 0 | 0 | 0 | 6 | 9 |
| | SD 3.5 | 1 | 1 | 1 | 1 | 0 | 1 | 0 | 0 | 0 | 0 | 0 | 0 | 0 | 0 | 0 | 5 | 10 |
| | Doubao | 1 | 1 | 1 | 1 | 0 | 0 | 0 | 0 | 0 | 0 | 0 | 0 | 0 | 0 | 0 | 4 | 11 |
| | Qwen2.5-Max | 1 | 1 | 1 | 0 | 1 | 1 | 0 | 0 | 0 | 0 | 0 | 0 | 0 | 0 | 0 | 5 | 10 |
| | WanX2.1 | 1 | 1 | 0 | 0 | 0 | 0 | 0 | 0 | 0 | 0 | 0 | 1 | 0 | 0 | 0 | 3 | 12 |
| | Kling | 1 | 1 | 1 | 0 | 0 | 0 | 0 | 0 | 0 | 0 | 0 | 0 | 0 | 0 | 0 | 3 | 12 |
| | Star-3 Alpha | 1 | 1 | 1 | 1 | 1 | 0 | 0 | 0 | 0 | 0 | 0 | 0 | 0 | 0 | 0 | 5 | 10 |
| | Hunyuan | 1 | 1 | 1 | 1 | 1 | 0 | 0 | 0 | 0 | 0 | 0 | 0 | 0 | 0 | 0 | 5 | 10 |
| | GLM-4 | 1 | 1 | 1 | 1 | 0 | 0 | 0 | 0 | 0 | 0 | 0 | 0 | 0 | 0 | 0 | 4 | 11 |

Table 11: **Counting Trees in Nature Scene Results.**

| Objects | Model | 1 | 2 | 3 | 4 | 5 | 6 | 7 | 8 | 9 | 10 | 11 | 12 | 13 | 14 | 15 | Correct | Wrong |
|---|---|---|---|---|---|---|---|---|---|---|---|---|---|---|---|---|---|---|
| Apples | Recraft V3 | 1 | 1 | 0 | 0 | 0 | 0 | 0 | 0 | 0 | 0 | 0 | 0 | 0 | 0 | 0 | 2 | 13 |
| | Imagen-3 | 1 | 0 | 0 | 0 | 0 | 0 | 0 | 0 | 0 | 0 | 0 | 0 | 0 | 0 | 0 | 1 | 14 |
| | Dall·E 3 | 1 | 1 | 1 | 1 | 0 | 1 | 0 | 0 | 0 | 0 | 0 | 0 | 0 | 0 | 0 | 5 | 10 |
| | Grok 3 | 0 | 0 | 0 | 0 | 1 | 0 | 0 | 0 | 0 | 0 | 0 | 0 | 0 | 0 | 0 | 1 | 14 |
| | Gemini 2.0 Flash | 1 | 1 | 0 | 1 | 0 | 0 | 1 | 0 | 0 | 0 | 0 | 0 | 0 | 0 | 1 | 5 | 10 |
| | FLUX 1.1 | 1 | 1 | 1 | 0 | 0 | 1 | 0 | 0 | 0 | 0 | 0 | 1 | 0 | 0 | 0 | 5 | 10 |
| | Firefly 3 | 1 | 1 | 0 | 0 | 0 | 0 | 0 | 0 | 0 | 1 | 0 | 0 | 0 | 0 | 0 | 3 | 12 |
| | SD 3.5 | 0 | 0 | 0 | 0 | 0 | 0 | 0 | 0 | 0 | 0 | 0 | 0 | 0 | 0 | 0 | 0 | 15 |
| | Doubao | 1 | 1 | 1 | 1 | 0 | 1 | 0 | 0 | 0 | 0 | 0 | 0 | 0 | 0 | 0 | 5 | 10 |
| | Qwen2.5-Max | 1 | 1 | 1 | 0 | 0 | 0 | 0 | 0 | 0 | 0 | 0 | 0 | 0 | 0 | 1 | 4 | 11 |
| | WanX2.1 | 1 | 1 | 1 | 1 | 0 | 0 | 0 | 0 | 0 | 0 | 0 | 0 | 0 | 0 | 0 | 4 | 11 |
| | Kling | 0 | 0 | 1 | 0 | 0 | 1 | 0 | 1 | 0 | 0 | 0 | 0 | 0 | 0 | 0 | 3 | 12 |
| | Star-3 Alpha | 1 | 1 | 1 | 1 | 1 | 1 | 0 | 0 | 0 | 0 | 0 | 0 | 0 | 0 | 0 | 6 | 9 |
| | Hunyuan | 0 | 1 | 1 | 1 | 0 | 1 | 0 | 0 | 0 | 0 | 0 | 0 | 0 | 0 | 0 | 4 | 11 |
| | GLM-4 | 0 | 0 | 0 | 0 | 0 | 0 | 0 | 0 | 0 | 0 | 0 | 0 | 0 | 0 | 0 | 0 | 15 |
| Watermelons | Recraft V3 | 1 | 1 | 0 | 0 | 0 | 0 | 0 | 0 | 0 | 0 | 0 | 0 | 0 | 0 | 0 | 2 | 13 |
| | Imagen-3 | 1 | 0 | 0 | 0 | 0 | 0 | 0 | 0 | 0 | 0 | 0 | 0 | 0 | 0 | 0 | 1 | 14 |
| | Dall·E 3 | 1 | 1 | 1 | 1 | 1 | 1 | 0 | 0 | 0 | 0 | 0 | 0 | 0 | 0 | 0 | 6 | 9 |
| | Grok 3 | 1 | 0 | 0 | 0 | 0 | 0 | 0 | 0 | 0 | 0 | 0 | 0 | 0 | 0 | 0 | 1 | 14 |
| | Gemini 2.0 Flash | 1 | 1 | 0 | 0 | 0 | 0 | 0 | 0 | 0 | 0 | 0 | 0 | 0 | 0 | 0 | 2 | 13 |
| | FLUX 1.1 | 1 | 1 | 1 | 1 | 1 | 1 | 0 | 0 | 0 | 0 | 0 | 0 | 0 | 0 | 0 | 6 | 9 |
| | Firefly 3 | 1 | 0 | 0 | 0 | 0 | 0 | 0 | 0 | 0 | 0 | 0 | 0 | 0 | 0 | 0 | 1 | 14 |
| | SD 3.5 | 0 | 0 | 0 | 0 | 0 | 0 | 0 | 0 | 0 | 0 | 0 | 0 | 0 | 0 | 0 | 0 | 15 |
| | Doubao | 0 | 1 | 1 | 1 | 1 | 1 | 0 | 0 | 0 | 0 | 0 | 0 | 0 | 0 | 0 | 5 | 10 |
| | Qwen2.5-Max | 1 | 1 | 0 | 0 | 0 | 1 | 1 | 1 | 0 | 0 | 0 | 0 | 0 | 0 | 0 | 5 | 10 |
| | WanX2.1 | 0 | 0 | 0 | 0 | 1 | 0 | 1 | 0 | 0 | 0 | 0 | 0 | 0 | 0 | 0 | 2 | 13 |
| | Kling | 1 | 1 | 1 | 0 | 0 | 0 | 0 | 1 | 0 | 0 | 0 | 0 | 0 | 0 | 0 | 4 | 11 |
| | Star-3 Alpha | 1 | 1 | 1 | 0 | 1 | 0 | 0 | 0 | 0 | 0 | 0 | 0 | 0 | 0 | 0 | 4 | 11 |
| | Hunyuan | 0 | 1 | 0 | 0 | 0 | 0 | 0 | 0 | 0 | 0 | 0 | 0 | 0 | 0 | 0 | 1 | 14 |
| | GLM-4 | 1 | 1 | 0 | 0 | 0 | 1 | 0 | 0 | 0 | 0 | 0 | 0 | 0 | 0 | 0 | 3 | 12 |

Table 12: **Counting Apples and Watermelons in City Scene Results.**

| Objects | Model | 1 | 2 | 3 | 4 | 5 | 6 | 7 | 8 | 9 | 10 | 11 | 12 | 13 | 14 | 15 | Correct | Wrong |
|---|---|---|---|---|---|---|---|---|---|---|---|---|---|---|---|---|---|---|
| Humans | Recraft V3 | 1 | 1 | 1 | 1 | 0 | 0 | 0 | 0 | 0 | 0 | 0 | 0 | 0 | 0 | 0 | 4 | 11 |
| | Imagen-3 | 1 | 0 | 0 | 0 | 0 | 0 | 0 | 0 | 1 | 0 | 0 | 0 | 0 | 0 | 0 | 2 | 13 |
| | Dall·E 3 | 0 | 0 | 0 | 0 | 0 | 0 | 0 | 0 | 0 | 0 | 0 | 0 | 0 | 0 | 0 | 0 | 15 |
| | Grok 3 | 1 | 0 | 1 | 0 | 1 | 1 | 1 | 0 | 0 | 1 | 0 | 0 | 0 | 0 | 0 | 6 | 9 |
| | Gemini 2.0 Flash | 1 | 0 | 0 | 0 | 1 | 1 | 0 | 0 | 1 | 0 | 0 | 0 | 0 | 0 | 0 | 4 | 11 |
| | FLUX 1.1 | 0 | 1 | 0 | 1 | 0 | 0 | 0 | 0 | 0 | 0 | 0 | 0 | 0 | 0 | 0 | 2 | 13 |
| | Firefly 3 | 1 | 1 | 1 | 0 | 0 | 0 | 1 | 0 | 0 | 0 | 0 | 0 | 0 | 0 | 0 | 4 | 11 |
| | SD 3.5 | 0 | 0 | 0 | 0 | 0 | 0 | 0 | 0 | 0 | 0 | 0 | 0 | 0 | 0 | 0 | 0 | 15 |
| | Doubao | 0 | 0 | 0 | 1 | 0 | 0 | 0 | 0 | 0 | 0 | 0 | 0 | 0 | 0 | 0 | 1 | 14 |
| | Qwen2.5-Max | 1 | 1 | 0 | 0 | 0 | 0 | 1 | 0 | 0 | 0 | 0 | 0 | 0 | 0 | 0 | 3 | 12 |
| | WanX2.1 | 1 | 1 | 0 | 1 | 0 | 0 | 0 | 0 | 0 | 0 | 0 | 0 | 0 | 0 | 0 | 3 | 12 |
| | Kling | 1 | 1 | 1 | 1 | 1 | 0 | 0 | 0 | 1 | 0 | 0 | 0 | 0 | 0 | 0 | 6 | 9 |
| | Star-3 Alpha | 1 | 1 | 1 | 0 | 0 | 0 | 0 | 0 | 0 | 0 | 0 | 0 | 0 | 0 | 0 | 3 | 12 |
| | Hunyuan | 0 | 0 | 1 | 0 | 1 | 1 | 0 | 0 | 0 | 0 | 0 | 0 | 0 | 0 | 0 | 3 | 12 |
| | GLM-4 | 0 | 0 | 0 | 0 | 0 | 0 | 0 | 0 | 0 | 0 | 1 | 0 | 0 | 0 | 0 | 1 | 14 |
| Cats | Recraft V3 | 1 | 1 | 1 | 0 | 1 | 0 | 0 | 0 | 0 | 0 | 0 | 0 | 0 | 0 | 0 | 4 | 11 |
| | Imagen-3 | 1 | 0 | 1 | 1 | 0 | 1 | 0 | 0 | 0 | 0 | 0 | 1 | 1 | 0 | 0 | 6 | 9 |
| | Dall·E | 1 | 1 | 1 | 1 | 0 | 0 | 1 | 0 | 1 | 0 | 0 | 1 | 0 | 0 | 0 | 7 | 8 |
| | Grok 3 | 1 | 1 | 1 | 0 | 0 | 0 | 0 | 0 | 0 | 0 | 0 | 0 | 0 | 0 | 0 | 3 | 12 |
| | Gemini 2.0 Flash | 1 | 1 | 1 | 0 | 0 | 1 | 1 | 1 | 0 | 1 | 1 | 0 | 0 | 0 | 0 | 8 | 7 |
| | FLUX 1.1 | 1 | 1 | 1 | 1 | 0 | 0 | 0 | 0 | 0 | 0 | 0 | 0 | 0 | 0 | 0 | 4 | 11 |
| | Firefly 3 | 0 | 1 | 1 | 0 | 1 | 0 | 0 | 0 | 0 | 0 | 0 | 0 | 0 | 0 | 0 | 3 | 12 |
| | SD 3.5 | 1 | 1 | 1 | 1 | 1 | 0 | 0 | 0 | 0 | 0 | 0 | 0 | 0 | 0 | 0 | 5 | 10 |
| | Doubao | 1 | 1 | 1 | 0 | 0 | 0 | 0 | 1 | 0 | 0 | 0 | 0 | 0 | 0 | 0 | 4 | 11 |
| | Qwen2.5-Max | 1 | 1 | 0 | 1 | 1 | 0 | 1 | 0 | 0 | 0 | 0 | 0 | 0 | 0 | 0 | 5 | 10 |
| | WanX2.1 | 1 | 1 | 0 | 0 | 0 | 0 | 1 | 1 | 0 | 0 | 0 | 0 | 0 | 0 | 0 | 4 | 11 |
| | Kling | 1 | 1 | 1 | 1 | 1 | 0 | 0 | 0 | 0 | 0 | 0 | 1 | 0 | 0 | 0 | 6 | 9 |
| | Star-3 Alpha | 1 | 1 | 1 | 1 | 1 | 0 | 0 | 0 | 0 | 0 | 0 | 0 | 0 | 0 | 0 | 5 | 10 |
| | Hunyuan | 0 | 1 | 0 | 1 | 1 | 0 | 1 | 0 | 0 | 0 | 0 | 0 | 0 | 0 | 0 | 4 | 11 |
| | GLM-4 | 1 | 1 | 1 | 1 | 0 | 0 | 0 | 0 | 0 | 0 | 0 | 1 | 0 | 0 | 0 | 5 | 10 |

Table 13: **Counting Humans and Cats in City Scene Results.**

| Objects | Model | 1 | 2 | 3 | 4 | 5 | 6 | 7 | 8 | 9 | 10 | 11 | 12 | 13 | 14 | 15 | Correct | Wrong |
|---|---|---|---|---|---|---|---|---|---|---|---|---|---|---|---|---|---|---|
| Triangles | Recraft V3 | 0 | 0 | 0 | 0 | 0 | 0 | 0 | 0 | 0 | 0 | 0 | 0 | 0 | 0 | 0 | 0 | 15 |
| | Imagen-3 | 1 | 0 | 1 | 0 | 0 | 1 | 0 | 0 | 0 | 0 | 0 | 0 | 0 | 0 | 0 | 3 | 12 |
| | Dall·E 3 | 1 | 0 | 1 | 0 | 0 | 0 | 1 | 0 | 0 | 0 | 0 | 0 | 0 | 0 | 0 | 3 | 12 |
| | Grok 3 | 1 | 1 | 0 | 0 | 1 | 0 | 1 | 0 | 0 | 0 | 0 | 0 | 0 | 0 | 0 | 4 | 11 |
| | Gemini 2.0 Flash | 1 | 1 | 1 | 0 | 0 | 0 | 0 | 0 | 0 | 0 | 0 | 1 | 1 | 0 | 0 | 5 | 10 |
| | FLUX 1.1 | 1 | 1 | 1 | 0 | 0 | 0 | 0 | 0 | 0 | 0 | 0 | 0 | 0 | 0 | 0 | 3 | 12 |
| | Firefly 3 | 0 | 0 | 1 | 0 | 0 | 0 | 0 | 0 | 0 | 0 | 0 | 0 | 0 | 0 | 0 | 1 | 14 |
| | SD 3.5 | 0 | 1 | 1 | 1 | 0 | 0 | 0 | 0 | 0 | 0 | 0 | 0 | 0 | 0 | 0 | 3 | 12 |
| | Doubao | 0 | 0 | 0 | 0 | 0 | 1 | 0 | 1 | 1 | 0 | 0 | 0 | 0 | 0 | 0 | 3 | 12 |
| | Qwen2.5-Max | 1 | 0 | 1 | 1 | 0 | 1 | 0 | 1 | 0 | 0 | 0 | 0 | 0 | 0 | 0 | 5 | 10 |
| | WanX2.1 | 1 | 1 | 1 | 0 | 0 | 0 | 0 | 0 | 0 | 0 | 0 | 0 | 0 | 0 | 0 | 3 | 12 |
| | Kling | 1 | 0 | 0 | 0 | 0 | 0 | 0 | 0 | 0 | 0 | 0 | 0 | 0 | 0 | 0 | 1 | 14 |
| | Star-3 Alpha | 1 | 1 | 1 | 0 | 0 | 0 | 0 | 0 | 0 | 0 | 0 | 0 | 0 | 0 | 0 | 3 | 12 |
| | Hunyuan | 1 | 0 | 1 | 0 | 1 | 0 | 0 | 0 | 0 | 0 | 0 | 0 | 0 | 0 | 0 | 3 | 12 |
| | GLM-4 | 0 | 0 | 0 | 0 | 0 | 0 | 0 | 0 | 0 | 0 | 0 | 0 | 0 | 0 | 0 | 0 | 15 |
| Chairs | Recraft V3 | 1 | 0 | 0 | 0 | 0 | 0 | 0 | 0 | 0 | 0 | 0 | 0 | 0 | 0 | 0 | 1 | 14 |
| | Imagen-3 | 1 | 0 | 0 | 1 | 0 | 1 | 0 | 0 | 1 | 0 | 0 | 0 | 0 | 0 | 0 | 4 | 11 |
| | Dall·E 3 | 1 | 0 | 1 | 0 | 0 | 0 | 0 | 0 | 0 | 0 | 0 | 0 | 1 | 0 | 0 | 3 | 12 |
| | Grok 3 | 1 | 0 | 0 | 0 | 0 | 0 | 0 | 0 | 0 | 0 | 0 | 0 | 0 | 0 | 0 | 1 | 14 |
| | Gemini 2.0 Flash | 1 | 0 | 1 | 0 | 0 | 0 | 0 | 0 | 0 | 0 | 0 | 0 | 0 | 0 | 0 | 2 | 13 |
| | FLUX 1.1 | 0 | 1 | 1 | 1 | 0 | 1 | 0 | 0 | 1 | 0 | 0 | 0 | 0 | 0 | 0 | 5 | 10 |
| | Firefly 3 | 0 | 0 | 0 | 0 | 0 | 0 | 0 | 0 | 0 | 0 | 0 | 0 | 0 | 0 | 0 | 0 | 15 |
| | SD 3.5 | 0 | 1 | 1 | 1 | 0 | 0 | 1 | 0 | 1 | 0 | 1 | 0 | 0 | 0 | 0 | 6 | 9 |
| | Doubao | 0 | 1 | 1 | 1 | 1 | 0 | 1 | 0 | 0 | 0 | 0 | 0 | 0 | 0 | 0 | 5 | 10 |
| | Qwen2.5-Max | 0 | 1 | 1 | 1 | 1 | 0 | 0 | 0 | 0 | 0 | 0 | 0 | 0 | 0 | 0 | 4 | 11 |
| | WanX2.1 | 0 | 1 | 1 | 1 | 1 | 0 | 0 | 0 | 0 | 0 | 0 | 0 | 0 | 0 | 0 | 4 | 11 |
| | Kling | 0 | 1 | 1 | 1 | 1 | 0 | 1 | 0 | 0 | 0 | 0 | 0 | 0 | 0 | 0 | 5 | 10 |
| | Star-3 Alpha | 1 | 1 | 1 | 1 | 1 | 0 | 1 | 0 | 0 | 0 | 0 | 0 | 0 | 0 | 0 | 6 | 9 |
| | Hunyuan | 0 | 1 | 0 | 0 | 1 | 1 | 0 | 0 | 0 | 0 | 0 | 0 | 0 | 0 | 0 | 3 | 12 |
| | GLM-4 | 0 | 0 | 0 | 0 | 1 | 0 | 0 | 0 | 0 | 0 | 0 | 0 | 0 | 0 | 0 | 1 | 15 |

Table 14: **Counting Triangles and Chairs in City Scene Results.**

| Objects | Model | 1 | 2 | 3 | 4 | 5 | 6 | 7 | 8 | 9 | 10 | 11 | 12 | 13 | 14 | 15 | Correct | Wrong |
|---|---|---|---|---|---|---|---|---|---|---|---|---|---|---|---|---|---|---|
| Trees | Recraft V3 | 0 | 0 | 0 | 0 | 0 | 0 | 0 | 0 | 0 | 0 | 0 | 0 | 0 | 0 | 0 | 0 | 15 |
| | Imagen-3 | 0 | 0 | 0 | 0 | 0 | 0 | 0 | 0 | 0 | 0 | 0 | 0 | 0 | 0 | 0 | 0 | 15 |
| | Dall·E 3 | 1 | 0 | 1 | 0 | 0 | 0 | 0 | 0 | 0 | 0 | 0 | 0 | 0 | 0 | 0 | 2 | 13 |
| | Grok 3 | 1 | 0 | 0 | 0 | 1 | 0 | 0 | 0 | 0 | 0 | 0 | 0 | 0 | 0 | 0 | 2 | 13 |
| | Gemini 2.0 Flash | 1 | 1 | 1 | 0 | 0 | 0 | 0 | 0 | 0 | 0 | 0 | 0 | 0 | 0 | 0 | 3 | 12 |
| | FLUX 1.1 | 1 | 1 | 1 | 1 | 0 | 0 | 0 | 0 | 0 | 0 | 0 | 0 | 0 | 0 | 0 | 4 | 11 |
| | Firefly 3 | 0 | 0 | 0 | 0 | 0 | 0 | 0 | 0 | 0 | 0 | 0 | 0 | 0 | 0 | 0 | 0 | 15 |
| | SD 3.5 | 0 | 0 | 1 | 1 | 1 | 1 | 0 | 0 | 0 | 0 | 0 | 0 | 0 | 0 | 0 | 4 | 11 |
| | Doubao | 0 | 1 | 1 | 1 | 0 | 0 | 0 | 0 | 1 | 0 | 0 | 0 | 0 | 0 | 0 | 4 | 11 |
| | Qwen2.5-Max | 0 | 1 | 0 | 0 | 1 | 0 | 0 | 0 | 0 | 0 | 0 | 0 | 0 | 0 | 0 | 2 | 13 |
| | WanX2.1 | 0 | 0 | 0 | 0 | 0 | 0 | 0 | 0 | 0 | 0 | 0 | 0 | 0 | 0 | 0 | 0 | 15 |
| | Kling | 1 | 1 | 0 | 1 | 1 | 1 | 1 | 0 | 0 | 0 | 0 | 0 | 0 | 0 | 0 | 6 | 9 |
| | Star-3 Alpha | 0 | 1 | 0 | 0 | 0 | 0 | 0 | 0 | 0 | 0 | 0 | 0 | 0 | 0 | 0 | 1 | 14 |
| | Hunyuan | 0 | 1 | 0 | 0 | 0 | 0 | 0 | 0 | 0 | 0 | 0 | 0 | 0 | 0 | 0 | 1 | 14 |
| | GLM-4 | 0 | 0 | 0 | 0 | 0 | 0 | 0 | 0 | 0 | 0 | 0 | 0 | 0 | 0 | 0 | 0 | 15 |

Table 15: **Counting Trees in City Scene Results.**

| Objects | Model | 1 | 2 | 3 | 4 | 5 | 6 | 7 | 8 | 9 | 10 | 11 | 12 | 13 | 14 | 15 | Correct | Wrong |
|---|---|---|---|---|---|---|---|---|---|---|---|---|---|---|---|---|---|---|
| Apples | Recraft V3 | 1 | 1 | 1 | 1 | 0 | 0 | 0 | 0 | 0 | 0 | 0 | 0 | 0 | 0 | 0 | 4 | 11 |
| | Imagen-3 | 1 | 1 | 1 | 1 | 1 | 1 | 1 | 1 | 1 | 0 | 0 | 0 | 0 | 0 | 0 | 9 | 6 |
| | Dall·E 3 | 1 | 1 | 1 | 1 | 0 | 1 | 1 | 0 | 0 | 0 | 0 | 0 | 0 | 0 | 0 | 6 | 9 |
| | Grok 3 | 1 | 1 | 0 | 0 | 0 | 0 | 0 | 0 | 0 | 0 | 0 | 0 | 0 | 0 | 0 | 2 | 13 |
| | Gemini 2.0 Flash | 1 | 1 | 1 | 1 | 0 | 1 | 1 | 0 | 0 | 0 | 0 | 0 | 0 | 0 | 0 | 6 | 9 |
| | FLUX 1.1 | 0 | 1 | 1 | 1 | 1 | 1 | 0 | 0 | 1 | 0 | 0 | 0 | 0 | 0 | 0 | 6 | 9 |
| | Firefly 3 | 0 | 1 | 1 | 1 | 0 | 0 | 0 | 0 | 1 | 1 | 0 | 1 | 0 | 0 | 0 | 6 | 9 |
| | SD 3.5 | 1 | 1 | 1 | 1 | 0 | 0 | 0 | 0 | 0 | 0 | 0 | 0 | 0 | 0 | 0 | 3 | 12 |
| | Doubao | 1 | 1 | 1 | 1 | 0 | 0 | 1 | 0 | 0 | 0 | 0 | 0 | 0 | 0 | 0 | 5 | 10 |
| | Qwen2.5-Max | 1 | 1 | 1 | 1 | 0 | 0 | 1 | 0 | 0 | 0 | 0 | 0 | 0 | 0 | 1 | 6 | 9 |
| | WanX2.1 | 1 | 1 | 1 | 1 | 0 | 0 | 1 | 0 | 0 | 1 | 0 | 0 | 0 | 0 | 0 | 6 | 9 |
| | Kling | 1 | 1 | 1 | 0 | 0 | 0 | 0 | 0 | 0 | 0 | 0 | 0 | 0 | 0 | 0 | 3 | 12 |
| | Star-3 Alpha | 1 | 1 | 1 | 1 | 0 | 0 | 0 | 0 | 0 | 0 | 0 | 0 | 0 | 0 | 0 | 5 | 10 |
| | Hunyuan | 1 | 1 | 1 | 1 | 0 | 1 | 0 | 0 | 1 | 1 | 0 | 0 | 0 | 0 | 0 | 7 | 8 |
| | GLM-4 | 1 | 1 | 1 | 1 | 1 | 0 | 0 | 1 | 1 | 0 | 0 | 0 | 0 | 0 | 0 | 7 | 8 |
| Watermelons | Recraft V3 | 1 | 0 | 1 | 0 | 0 | 0 | 0 | 0 | 0 | 0 | 0 | 0 | 0 | 0 | 0 | 2 | 13 |
| | Imagen-3 | 0 | 0 | 0 | 0 | 0 | 0 | 0 | 0 | 0 | 0 | 0 | 0 | 0 | 0 | 0 | 0 | 15 |
| | Dall·E 3 | 1 | 0 | 0 | 1 | 0 | 0 | 0 | 0 | 0 | 0 | 0 | 1 | 0 | 0 | 0 | 3 | 12 |
| | Grok 3 | 1 | 0 | 1 | 0 | 0 | 0 | 0 | 0 | 0 | 0 | 0 | 0 | 0 | 0 | 0 | 2 | 13 |
| | Gemini 2.0 Flash | 1 | 0 | 0 | 0 | 0 | 0 | 0 | 0 | 0 | 0 | 0 | 0 | 0 | 0 | 0 | 1 | 14 |
| | FLUX 1.1 | 1 | 0 | 1 | 1 | 0 | 0 | 0 | 0 | 1 | 0 | 0 | 0 | 0 | 0 | 0 | 4 | 11 |
| | Firefly 3 | 0 | 0 | 0 | 0 | 0 | 0 | 0 | 0 | 0 | 0 | 0 | 0 | 0 | 0 | 0 | 0 | 15 |
| | SD 3.5 | 1 | 0 | 0 | 0 | 0 | 0 | 0 | 0 | 0 | 0 | 0 | 0 | 0 | 0 | 0 | 1 | 14 |
| | Doubao | 1 | 0 | 0 | 0 | 1 | 0 | 0 | 0 | 0 | 0 | 0 | 0 | 0 | 0 | 0 | 2 | 13 |
| | Qwen2.5-Max | 1 | 1 | 1 | 0 | 0 | 1 | 0 | 0 | 0 | 0 | 0 | 1 | 0 | 0 | 0 | 5 | 10 |
| | WanX2.1 | 1 | 1 | 0 | 0 | 0 | 0 | 0 | 0 | 0 | 0 | 0 | 0 | 0 | 0 | 0 | 2 | 13 |
| | Kling | 1 | 1 | 0 | 0 | 0 | 0 | 1 | 0 | 0 | 0 | 0 | 0 | 0 | 0 | 0 | 4 | 12 |
| | Star-3 Alpha | 0 | 1 | 1 | 1 | 0 | 0 | 0 | 0 | 0 | 0 | 0 | 0 | 0 | 0 | 0 | 3 | 12 |
| | Hunyuan | 1 | 1 | 0 | 1 | 0 | 1 | 0 | 0 | 0 | 0 | 0 | 0 | 0 | 0 | 0 | 4 | 11 |
| | GLM-4 | 0 | 0 | 0 | 1 | 0 | 0 | 0 | 0 | 0 | 0 | 0 | 1 | 0 | 0 | 0 | 2 | 13 |

Table 16: **Counting Apples and Watermelons With Cartoon style Results.**

| Objects | Model | 1 | 2 | 3 | 4 | 5 | 6 | 7 | 8 | 9 | 10 | 11 | 12 | 13 | 14 | 15 | Correct | Wrong |
|---|---|---|---|---|---|---|---|---|---|---|---|---|---|---|---|---|---|---|
| Humans | Recraft V3 | 1 | 1 | 1 | 1 | 0 | 0 | 0 | 0 | 0 | 0 | 0 | 0 | 0 | 0 | 0 | 4 | 11 |
| | Imagen-3 | 1 | 1 | 1 | 1 | 1 | 0 | 1 | 0 | 0 | 1 | 0 | 0 | 0 | 0 | 0 | 7 | 8 |
| | Dall·E 3 | 1 | 1 | 1 | 1 | 1 | 0 | 1 | 0 | 0 | 0 | 1 | 0 | 0 | 0 | 0 | 7 | 8 |
| | Grok 3 | 1 | 1 | 1 | 0 | 1 | 0 | 0 | 1 | 0 | 0 | 0 | 0 | 0 | 0 | 0 | 5 | 10 |
| | Gemini 2.0 Flash | 1 | 1 | 1 | 1 | 1 | 1 | 1 | 1 | 1 | 0 | 1 | 1 | 0 | 0 | 0 | 11 | 4 |
| | FLUX 1.1 | 1 | 1 | 1 | 1 | 0 | 1 | 1 | 1 | 1 | 0 | 0 | 1 | 0 | 0 | 0 | 9 | 6 |
| | Firefly 3 | 1 | 1 | 1 | 1 | 1 | 1 | 0 | 0 | 0 | 0 | 1 | 0 | 0 | 1 | 0 | 8 | 7 |
| | SD 3.5 | 1 | 1 | 1 | 1 | 0 | 0 | 0 | 0 | 0 | 0 | 0 | 0 | 1 | 0 | 0 | 6 | 9 |
| | Doubao | 1 | 1 | 1 | 1 | 1 | 0 | 1 | 1 | 0 | 1 | 0 | 0 | 0 | 0 | 0 | 8 | 7 |
| | Qwen2.5-Max | 1 | 1 | 1 | 0 | 1 | 1 | 0 | 1 | 0 | 0 | 0 | 0 | 0 | 0 | 0 | 6 | 9 |
| | WanX2.1 | 1 | 1 | 1 | 1 | 1 | 1 | 1 | 1 | 1 | 0 | 1 | 1 | 0 | 1 | 0 | 12 | 3 |
| | Kling | 1 | 1 | 1 | 1 | 0 | 0 | 0 | 1 | 1 | 1 | 1 | 1 | 0 | 0 | 0 | 9 | 6 |
| | Star-3 Alpha | 1 | 1 | 1 | 1 | 0 | 0 | 0 | 1 | 0 | 0 | 0 | 0 | 0 | 0 | 1 | 6 | 9 |
| | Hunyuan | 0 | 1 | 1 | 1 | 1 | 0 | 0 | 1 | 0 | 0 | 0 | 0 | 0 | 0 | 0 | 5 | 10 |
| | GLM-4 | 1 | 1 | 1 | 1 | 1 | 0 | 0 | 1 | 0 | 0 | 0 | 0 | 1 | 0 | 0 | 7 | 8 |
| Cats | Recraft V3 | 1 | 1 | 1 | 1 | 1 | 0 | 0 | 0 | 0 | 0 | 0 | 0 | 0 | 0 | 0 | 5 | 10 |
| | Imagen-3 | 1 | 0 | 1 | 0 | 0 | 0 | 1 | 0 | 0 | 0 | 0 | 0 | 0 | 1 | 1 | 5 | 10 |
| | Dall·E 3 | 1 | 1 | 1 | 1 | 1 | 1 | 0 | 0 | 1 | 0 | 0 | 1 | 0 | 0 | 0 | 8 | 7 |
| | Grok 3 | 1 | 1 | 0 | 0 | 1 | 0 | 0 | 0 | 0 | 0 | 0 | 0 | 0 | 0 | 0 | 3 | 12 |
| | Gemini 2.0 Flash | 1 | 0 | 1 | 1 | 0 | 1 | 1 | 1 | 0 | 0 | 1 | 0 | 0 | 0 | 0 | 7 | 8 |
| | FLUX 1.1 | 0 | 1 | 1 | 1 | 1 | 1 | 1 | 0 | 0 | 1 | 0 | 1 | 0 | 0 | 0 | 8 | 7 |
| | Firefly 3 | 0 | 1 | 1 | 1 | 0 | 1 | 0 | 1 | 1 | 0 | 0 | 0 | 0 | 0 | 0 | 6 | 9 |
| | SD 3.5 | 0 | 1 | 1 | 1 | 0 | 1 | 0 | 0 | 0 | 0 | 0 | 0 | 0 | 0 | 0 | 4 | 11 |
| | Doubao | 1 | 1 | 1 | 1 | 0 | 0 | 0 | 0 | 0 | 1 | 0 | 1 | 0 | 0 | 0 | 6 | 9 |
| | Qwen2.5-Max | 1 | 1 | 1 | 1 | 0 | 1 | 0 | 1 | 0 | 0 | 0 | 0 | 1 | 1 | 0 | 8 | 7 |
| | WanX2.1 | 1 | 1 | 1 | 1 | 1 | 0 | 1 | 0 | 0 | 0 | 0 | 0 | 0 | 0 | 0 | 6 | 9 |
| | Kling | 1 | 1 | 1 | 1 | 1 | 0 | 0 | 0 | 0 | 0 | 0 | 1 | 0 | 0 | 0 | 6 | 9 |
| | Star-3 Alpha | 1 | 1 | 1 | 0 | 1 | 1 | 1 | 1 | 1 | 1 | 0 | 0 | 0 | 0 | 0 | 9 | 6 |
| | Hunyuan | 1 | 1 | 1 | 1 | 1 | 0 | 0 | 0 | 0 | 1 | 1 | 0 | 0 | 0 | 0 | 7 | 8 |
| | GLM-4 | 1 | 1 | 1 | 1 | 1 | 1 | 1 | 0 | 1 | 0 | 0 | 1 | 0 | 0 | 0 | 9 | 6 |

Table 17: **Counting Humans and Cats With Cartoon style Results.**

| Objects | Model | 1 | 2 | 3 | 4 | 5 | 6 | 7 | 8 | 9 | 10 | 11 | 12 | 13 | 14 | 15 | Correct | Wrong |
|---|---|---|---|---|---|---|---|---|---|---|---|---|---|---|---|---|---|---|
| Triangles | Recraft V3 | 0 | 0 | 0 | 1 | 0 | 1 | 1 | 0 | 0 | 0 | 0 | 0 | 0 | 0 | 0 | 3 | 12 |
| | Imagen-3 | 1 | 1 | 1 | 1 | 0 | 0 | 1 | 0 | 0 | 1 | 0 | 0 | 0 | 0 | 0 | 6 | 9 |
| | Dall·E 3 | 0 | 1 | 1 | 1 | 0 | 0 | 0 | 0 | 0 | 0 | 0 | 0 | 0 | 0 | 0 | 3 | 12 |
| | Grok 3 | 1 | 1 | 0 | 1 | 0 | 0 | 0 | 0 | 0 | 0 | 0 | 0 | 0 | 0 | 0 | 3 | 12 |
| | Gemini 2.0 Flash | 1 | 1 | 1 | 1 | 1 | 1 | 1 | 0 | 1 | 0 | 0 | 0 | 1 | 0 | 1 | 10 | 5 |
| | FLUX 1.1 | 1 | 1 | 1 | 0 | 0 | 0 | 1 | 0 | 0 | 0 | 0 | 0 | 0 | 0 | 0 | 4 | 11 |
| | Firefly 3 | 0 | 1 | 1 | 0 | 0 | 0 | 0 | 0 | 1 | 0 | 0 | 0 | 0 | 0 | 0 | 3 | 12 |
| | SD 3.5 | 1 | 1 | 1 | 0 | 0 | 0 | 0 | 0 | 1 | 0 | 0 | 0 | 0 | 0 | 0 | 4 | 11 |
| | Doubao | 1 | 1 | 1 | 1 | 0 | 0 | 1 | 0 | 1 | 0 | 0 | 1 | 0 | 0 | 0 | 7 | 8 |
| | Qwen2.5-Max | 1 | 1 | 0 | 1 | 0 | 1 | 0 | 0 | 0 | 0 | 0 | 0 | 0 | 0 | 0 | 4 | 11 |
| | WanX2.1 | 1 | 1 | 1 | 0 | 0 | 1 | 0 | 0 | 0 | 0 | 0 | 0 | 0 | 0 | 0 | 4 | 11 |
| | Kling | 1 | 1 | 1 | 0 | 0 | 1 | 0 | 0 | 0 | 0 | 0 | 0 | 0 | 0 | 0 | 4 | 11 |
| | Star-3 Alpha | 1 | 1 | 1 | 0 | 1 | 0 | 1 | 0 | 1 | 0 | 0 | 0 | 0 | 0 | 1 | 7 | 8 |
| | Hunyuan | 1 | 1 | 1 | 0 | 1 | 0 | 0 | 0 | 0 | 0 | 0 | 0 | 0 | 0 | 0 | 4 | 11 |
| | GLM-4 | 1 | 1 | 0 | 0 | 0 | 0 | 0 | 0 | 1 | 0 | 0 | 0 | 0 | 0 | 0 | 3 | 12 |
| Chairs | Recraft V3 | 1 | 1 | 1 | 1 | 1 | 1 | 1 | 0 | 0 | 0 | 0 | 0 | 0 | 0 | 0 | 7 | 8 |
| | Imagen-3 | 1 | 1 | 1 | 0 | 0 | 0 | 0 | 0 | 0 | 0 | 0 | 0 | 0 | 0 | 0 | 3 | 12 |
| | Dall·E 3 | 1 | 1 | 1 | 1 | 1 | 0 | 1 | 0 | 0 | 0 | 0 | 0 | 1 | 0 | 0 | 7 | 8 |
| | Grok 3 | 1 | 1 | 0 | 1 | 0 | 0 | 0 | 0 | 0 | 0 | 0 | 0 | 0 | 0 | 0 | 3 | 12 |
| | Gemini 2.0 Flash | 1 | 1 | 1 | 0 | 1 | 0 | 0 | 0 | 0 | 0 | 1 | 0 | 1 | 1 | 0 | 7 | 8 |
| | FLUX 1.1 | 1 | 1 | 1 | 1 | 1 | 1 | 0 | 1 | 0 | 0 | 0 | 0 | 0 | 0 | 0 | 7 | 8 |
| | Firefly 3 | 0 | 1 | 1 | 0 | 1 | 0 | 1 | 0 | 1 | 0 | 0 | 0 | 0 | 0 | 0 | 5 | 10 |
| | SD 3.5 | 1 | 1 | 1 | 0 | 0 | 1 | 0 | 0 | 0 | 0 | 0 | 0 | 0 | 0 | 0 | 4 | 11 |
| | Doubao | 1 | 1 | 1 | 0 | 1 | 1 | 0 | 0 | 0 | 0 | 0 | 0 | 0 | 0 | 0 | 5 | 10 |
| | Qwen2.5-Max | 1 | 1 | 1 | 0 | 0 | 0 | 0 | 1 | 0 | 0 | 0 | 0 | 0 | 0 | 0 | 4 | 11 |
| | WanX2.1 | 1 | 1 | 1 | 1 | 1 | 0 | 0 | 0 | 1 | 0 | 0 | 0 | 0 | 1 | 0 | 7 | 8 |
| | Kling | 1 | 1 | 1 | 1 | 1 | 0 | 0 | 0 | 0 | 0 | 0 | 0 | 0 | 0 | 0 | 5 | 10 |
| | Star-3 Alpha | 1 | 1 | 1 | 1 | 0 | 0 | 0 | 1 | 1 | 0 | 0 | 0 | 0 | 0 | 0 | 6 | 9 |
| | Hunyuan | 1 | 1 | 1 | 1 | 1 | 0 | 0 | 1 | 1 | 0 | 0 | 0 | 0 | 0 | 0 | 7 | 8 |
| | GLM-4 | 1 | 1 | 1 | 1 | 1 | 1 | 0 | 1 | 1 | 1 | 0 | 0 | 0 | 0 | 1 | 10 | 5 |

Table 18: **Counting Triangles and Chairs With Cartoon style Results.**

| Objects | Model | 1 | 2 | 3 | 4 | 5 | 6 | 7 | 8 | 9 | 10 | 11 | 12 | 13 | 14 | 15 | Correct | Wrong |
|---|---|---|---|---|---|---|---|---|---|---|---|---|---|---|---|---|---|---|
| Flowers | Recraft V3 | 1 | 0 | 1 | 0 | 0 | 0 | 0 | 0 | 0 | 0 | 0 | 0 | 0 | 0 | 0 | 2 | 13 |
| | Imagen-3 | 1 | 1 | 1 | 0 | 1 | 1 | 0 | 0 | 0 | 0 | 0 | 0 | 0 | 0 | 1 | 6 | 9 |
| | Dall·E 3 | 1 | 1 | 1 | 1 | 0 | 1 | 0 | 0 | 0 | 0 | 1 | 0 | 0 | 0 | 0 | 6 | 9 |
| | Grok 3 | 1 | 0 | 0 | 0 | 0 | 0 | 1 | 0 | 0 | 0 | 0 | 0 | 0 | 0 | 0 | 2 | 13 |
| | Gemini 2.0 Flash | 1 | 1 | 1 | 1 | 0 | 0 | 1 | 1 | 0 | 0 | 0 | 1 | 0 | 0 | 0 | 7 | 8 |
| | FLUX 1.1 | 1 | 1 | 1 | 0 | 0 | 0 | 1 | 0 | 0 | 0 | 0 | 0 | 0 | 0 | 0 | 4 | 11 |
| | Firefly 3 | 0 | 0 | 0 | 0 | 0 | 0 | 0 | 0 | 0 | 0 | 0 | 0 | 0 | 0 | 0 | 0 | 15 |
| | SD 3.5 | 0 | 0 | 1 | 0 | 0 | 0 | 0 | 0 | 0 | 0 | 0 | 0 | 0 | 0 | 0 | 1 | 14 |
| | Doubao | 1 | 1 | 1 | 0 | 0 | 1 | 0 | 0 | 0 | 0 | 0 | 0 | 0 | 0 | 0 | 4 | 11 |
| | Qwen2.5-Max | 0 | 0 | 0 | 0 | 0 | 0 | 1 | 0 | 0 | 0 | 0 | 0 | 0 | 0 | 0 | 1 | 14 |
| | WanX2.1 | 1 | 1 | 1 | 0 | 0 | 0 | 1 | 0 | 0 | 0 | 0 | 0 | 0 | 0 | 0 | 4 | 11 |
| | Kling | 0 | 0 | 1 | 0 | 1 | 1 | 0 | 0 | 0 | 0 | 1 | 1 | 0 | 0 | 0 | 5 | 10 |
| | Star-3 Alpha | 1 | 1 | 1 | 0 | 1 | 0 | 0 | 1 | 0 | 0 | 0 | 0 | 0 | 0 | 0 | 5 | 10 |
| | Hunyuan | 1 | 1 | 1 | 0 | 0 | 0 | 0 | 0 | 1 | 0 | 0 | 0 | 0 | 1 | 1 | 6 | 9 |
| | GLM-4 | 1 | 1 | 1 | 0 | 0 | 0 | 0 | 0 | 0 | 0 | 0 | 0 | 0 | 0 | 1 | 4 | 11 |

Table 19: **Counting Flowers With Cartoon Style Results.**

| Objects | Model | 1 | 2 | 3 | 4 | 5 | 6 | 7 | 8 | 9 | 10 | 11 | 12 | 13 | 14 | 15 | Correct | Wrong |
|---|---|---|---|---|---|---|---|---|---|---|---|---|---|---|---|---|---|---|
| Apples | Recraft V3 | 1 | 1 | 1 | 0 | 1 | 1 | 0 | 1 | 0 | 0 | 0 | 0 | 0 | 0 | 0 | 6 | 9 |
| | Imagen-3 | 1 | 1 | 0 | 0 | 0 | 0 | 0 | 0 | 1 | 0 | 0 | 0 | 0 | 0 | 1 | 4 | 11 |
| | Dall·E 3 | 1 | 1 | 0 | 0 | 0 | 0 | 1 | 0 | 0 | 0 | 0 | 0 | 0 | 0 | 0 | 3 | 12 |
| | Grok 3 | 1 | 1 | 0 | 1 | 0 | 0 | 0 | 0 | 0 | 0 | 0 | 0 | 0 | 0 | 0 | 3 | 12 |
| | Gemini 2.0 Flash | 1 | 1 | 1 | 0 | 0 | 1 | 0 | 0 | 1 | 0 | 0 | 0 | 0 | 0 | 0 | 5 | 10 |
| | FLUX 1.1 | 1 | 1 | 1 | 0 | 1 | 0 | 1 | 0 | 0 | 0 | 0 | 0 | 0 | 0 | 0 | 5 | 10 |
| | Firefly 3 | 1 | 1 | 1 | 1 | 1 | 1 | 0 | 0 | 1 | 0 | 0 | 0 | 0 | 0 | 1 | 8 | 7 |
| | SD 3.5 | 1 | 1 | 0 | 0 | 1 | 0 | 0 | 0 | 0 | 0 | 0 | 0 | 0 | 0 | 0 | 3 | 12 |
| | Doubao | 1 | 1 | 1 | 1 | 0 | 0 | 1 | 0 | 0 | 1 | 0 | 0 | 0 | 0 | 0 | 6 | 9 |
| | Qwen2.5-Max | 1 | 1 | 0 | 0 | 1 | 0 | 0 | 0 | 0 | 1 | 0 | 0 | 0 | 0 | 0 | 4 | 11 |
| | WanX2.1 | 1 | 1 | 1 | 1 | 1 | 1 | 1 | 1 | 1 | 0 | 0 | 1 | 0 | 0 | 0 | 10 | 5 |
| | Kling | 1 | 1 | 1 | 1 | 1 | 1 | 0 | 0 | 0 | 0 | 0 | 0 | 0 | 0 | 0 | 6 | 9 |
| | Star-3 Alpha | 1 | 1 | 1 | 1 | 1 | 1 | 0 | 0 | 0 | 0 | 0 | 0 | 0 | 0 | 0 | 6 | 9 |
| | Hunyuan | 1 | 1 | 1 | 1 | 0 | 0 | 1 | 1 | 1 | 0 | 0 | 1 | 0 | 0 | 0 | 8 | 7 |
| | GLM-4 | 1 | 1 | 1 | 1 | 1 | 0 | 1 | 0 | 0 | 0 | 0 | 0 | 0 | 0 | 1 | 7 | 8 |
| Watermelons | Recraft V3 | 1 | 0 | 0 | 0 | 1 | 0 | 0 | 0 | 0 | 0 | 0 | 0 | 0 | 0 | 0 | 2 | 13 |
| | Imagen-3 | 0 | 0 | 0 | 1 | 1 | 0 | 0 | 0 | 0 | 0 | 0 | 0 | 0 | 0 | 0 | 2 | 13 |
| | Dall·E 3 | 1 | 0 | 0 | 0 | 0 | 0 | 0 | 0 | 0 | 0 | 0 | 0 | 0 | 0 | 0 | 1 | 14 |
| | Grok 3 | 1 | 1 | 1 | 1 | 0 | 0 | 0 | 0 | 0 | 0 | 0 | 0 | 0 | 0 | 0 | 4 | 11 |
| | Gemini 2.0 Flash | 0 | 0 | 0 | 0 | 0 | 0 | 0 | 0 | 0 | 0 | 0 | 0 | 0 | 0 | 0 | 0 | 15 |
| | FLUX 1.1 | 1 | 0 | 0 | 0 | 0 | 0 | 0 | 1 | 0 | 0 | 0 | 0 | 0 | 0 | 0 | 2 | 13 |
| | Firefly 3 | 0 | 0 | 0 | 0 | 0 | 0 | 0 | 0 | 0 | 0 | 0 | 1 | 0 | 0 | 0 | 1 | 14 |
| | SD 3.5 | 0 | 1 | 0 | 0 | 0 | 0 | 0 | 0 | 0 | 0 | 0 | 1 | 0 | 0 | 0 | 2 | 13 |
| | Doubao | 0 | 1 | 1 | 1 | 1 | 1 | 0 | 0 | 0 | 0 | 0 | 0 | 0 | 0 | 0 | 5 | 10 |
| | Qwen2.5-Max | 1 | 0 | 0 | 1 | 0 | 0 | 0 | 0 | 0 | 0 | 0 | 0 | 0 | 0 | 1 | 3 | 12 |
| | WanX2.1 | 1 | 1 | 1 | 1 | 0 | 0 | 1 | 0 | 1 | 0 | 0 | 0 | 0 | 0 | 0 | 6 | 9 |
| | Kling | 1 | 1 | 1 | 1 | 0 | 1 | 0 | 0 | 1 | 0 | 0 | 0 | 0 | 0 | 0 | 6 | 9 |
| | Star-3 Alpha | 1 | 1 | 1 | 1 | 1 | 1 | 1 | 1 | 0 | 0 | 1 | 0 | 0 | 0 | 0 | 9 | 6 |
| | Hunyuan | 0 | 1 | 0 | 0 | 1 | 0 | 0 | 1 | 0 | 0 | 0 | 0 | 0 | 0 | 0 | 3 | 12 |
| | GLM-4 | 0 | 0 | 0 | 1 | 0 | 0 | 1 | 0 | 0 | 0 | 0 | 0 | 0 | 0 | 0 | 2 | 13 |

Table 20: **Counting Apples and Watermelons With Watercolor style Results.**

| Objects | Model | 1 | 2 | 3 | 4 | 5 | 6 | 7 | 8 | 9 | 10 | 11 | 12 | 13 | 14 | 15 | Correct | Wrong |
|---|---|---|---|---|---|---|---|---|---|---|---|---|---|---|---|---|---|---|
| Humans | Recraft V3 | 1 | 1 | 1 | 1 | 1 | 1 | 0 | 0 | 0 | 0 | 0 | 0 | 0 | 0 | 0 | 6 | 9 |
| | Imagen-3 | 1 | 1 | 1 | 1 | 1 | 1 | 1 | 1 | 1 | 0 | 0 | 0 | 0 | 0 | 1 | 10 | 5 |
| | Dall·E 3 | 1 | 1 | 1 | 1 | 0 | 0 | 0 | 0 | 0 | 0 | 0 | 0 | 0 | 0 | 0 | 4 | 11 |
| | Grok 3 | 1 | 1 | 1 | 1 | 1 | 1 | 0 | 1 | 0 | 0 | 0 | 1 | 0 | 0 | 0 | 8 | 7 |
| | Gemini 2.0 Flash | 1 | 1 | 1 | 1 | 1 | 0 | 1 | 1 | 1 | 0 | 0 | 1 | 0 | 0 | 0 | 9 | 6 |
| | FLUX 1.1 | 1 | 1 | 1 | 1 | 1 | 0 | 0 | 1 | 0 | 0 | 1 | 1 | 0 | 0 | 0 | 8 | 7 |
| | Firefly 3 | 1 | 1 | 1 | 1 | 1 | 0 | 0 | 0 | 1 | 1 | 1 | 0 | 0 | 0 | 0 | 8 | 7 |
| | SD 3.5 | 1 | 1 | 1 | 1 | 1 | 1 | 0 | 1 | 0 | 0 | 0 | 0 | 0 | 0 | 0 | 7 | 8 |
| | Doubao | 1 | 1 | 1 | 1 | 1 | 1 | 1 | 0 | 0 | 0 | 0 | 0 | 0 | 0 | 0 | 7 | 8 |
| | Qwen2.5-Max | 1 | 1 | 1 | 0 | 0 | 0 | 0 | 0 | 0 | 0 | 0 | 0 | 0 | 0 | 0 | 3 | 12 |
| | WanX2.1 | 1 | 1 | 1 | 1 | 1 | 1 | 1 | 0 | 0 | 0 | 0 | 0 | 1 | 0 | 0 | 8 | 7 |
| | Kling | 1 | 1 | 1 | 1 | 1 | 1 | 1 | 0 | 0 | 0 | 1 | 0 | 0 | 0 | 0 | 8 | 7 |
| | Star-3 Alpha | 1 | 1 | 1 | 1 | 1 | 1 | 0 | 1 | 1 | 0 | 0 | 0 | 0 | 0 | 0 | 8 | 7 |
| | Hunyuan | 1 | 1 | 1 | 1 | 0 | 0 | 1 | 1 | 0 | 0 | 0 | 0 | 0 | 0 | 0 | 6 | 9 |
| | GLM-4 | 1 | 1 | 1 | 1 | 1 | 1 | 1 | 0 | 0 | 0 | 0 | 0 | 0 | 0 | 0 | 7 | 8 |
| Cats | Recraft V3 | 1 | 1 | 1 | 0 | 0 | 0 | 0 | 0 | 0 | 0 | 0 | 0 | 0 | 0 | 0 | 3 | 12 |
| | Imagen-3 | 1 | 1 | 1 | 1 | 1 | 1 | 1 | 1 | 1 | 1 | 0 | 1 | 0 | 1 | 1 | 13 | 2 |
| | Dall·E 3 | 1 | 1 | 1 | 0 | 1 | 1 | 1 | 0 | 0 | 1 | 0 | 0 | 0 | 0 | 0 | 7 | 8 |
| | Grok 3 | 1 | 1 | 1 | 0 | 0 | 0 | 0 | 0 | 0 | 0 | 0 | 0 | 0 | 0 | 0 | 3 | 12 |
| | Gemini 2.0 Flash | 1 | 1 | 1 | 1 | 1 | 1 | 0 | 0 | 1 | 1 | 0 | 0 | 0 | 0 | 1 | 9 | 6 |
| | FLUX 1.1 | 1 | 1 | 1 | 1 | 1 | 0 | 1 | 0 | 0 | 0 | 0 | 0 | 0 | 0 | 0 | 6 | 9 |
| | Firefly 3 | 1 | 1 | 1 | 1 | 1 | 1 | 1 | 1 | 1 | 0 | 0 | 1 | 1 | 0 | 1 | 12 | 3 |
| | SD 3.5 | 1 | 1 | 1 | 1 | 0 | 1 | 0 | 0 | 1 | 0 | 0 | 0 | 0 | 0 | 0 | 6 | 9 |
| | Doubao | 1 | 1 | 1 | 1 | 1 | 1 | 1 | 0 | 1 | 0 | 0 | 0 | 0 | 0 | 0 | 8 | 7 |
| | Qwen2.5-Max | 1 | 1 | 1 | 0 | 1 | 0 | 0 | 0 | 0 | 1 | 0 | 0 | 0 | 0 | 0 | 5 | 10 |
| | WanX2.1 | 1 | 1 | 1 | 1 | 1 | 1 | 0 | 0 | 1 | 0 | 0 | 0 | 0 | 1 | 0 | 8 | 7 |
| | Kling | 1 | 1 | 1 | 1 | 1 | 1 | 1 | 0 | 1 | 0 | 0 | 0 | 0 | 0 | 0 | 8 | 7 |
| | Star-3 Alpha | 1 | 1 | 1 | 1 | 0 | 1 | 1 | 0 | 0 | 0 | 0 | 0 | 0 | 0 | 0 | 6 | 9 |
| | Hunyuan | 1 | 1 | 1 | 1 | 1 | 1 | 1 | 0 | 1 | 0 | 0 | 0 | 0 | 0 | 0 | 8 | 7 |
| | GLM-4 | 1 | 1 | 1 | 1 | 1 | 0 | 0 | 0 | 0 | 0 | 0 | 1 | 0 | 0 | 0 | 6 | 9 |

Table 21: **Counting Humans and Cats With Watercolor style Results.**

| Objects | Model | 1 | 2 | 3 | 4 | 5 | 6 | 7 | 8 | 9 | 10 | 11 | 12 | 13 | 14 | 15 | Correct | Wrong |
|---|---|---|---|---|---|---|---|---|---|---|---|---|---|---|---|---|---|---|
| Triangles | Recraft V3 | 1 | 0 | 0 | 1 | 0 | 0 | 0 | 0 | 0 | 0 | 0 | 0 | 0 | 0 | 0 | 2 | 13 |
| | Imagen-3 | 1 | 1 | 1 | 1 | 1 | 0 | 0 | 1 | 1 | 0 | 0 | 0 | 0 | 0 | 1 | 8 | 7 |
| | Dall·E 3 | 1 | 1 | 0 | 0 | 0 | 0 | 0 | 0 | 0 | 0 | 0 | 0 | 0 | 0 | 0 | 2 | 13 |
| | Grok 3 | 1 | 1 | 0 | 1 | 1 | 1 | 1 | 0 | 1 | 1 | 1 | 0 | 0 | 0 | 1 | 10 | 5 |
| | Gemini 2.0 Flash | 1 | 1 | 1 | 1 | 0 | 1 | 0 | 0 | 0 | 0 | 0 | 1 | 0 | 0 | 0 | 6 | 9 |
| | FLUX 1.1 | 1 | 1 | 1 | 0 | 1 | 0 | 0 | 0 | 0 | 0 | 0 | 0 | 0 | 0 | 0 | 4 | 11 |
| | Firefly 3 | 1 | 1 | 1 | 1 | 0 | 1 | 0 | 0 | 1 | 0 | 0 | 0 | 0 | 0 | 0 | 6 | 9 |
| | SD 3.5 | 0 | 0 | 1 | 0 | 0 | 0 | 0 | 0 | 0 | 0 | 0 | 0 | 0 | 0 | 0 | 1 | 14 |
| | Doubao | 1 | 1 | 1 | 1 | 1 | 1 | 0 | 0 | 1 | 0 | 0 | 1 | 0 | 0 | 1 | 9 | 6 |
| | Qwen2.5-Max | 1 | 1 | 1 | 1 | 1 | 0 | 0 | 0 | 0 | 0 | 0 | 0 | 0 | 1 | 0 | 6 | 9 |
| | WanX2.1 | 1 | 1 | 1 | 1 | 1 | 0 | 0 | 0 | 0 | 0 | 0 | 0 | 0 | 0 | 0 | 6 | 9 |
| | Kling | 1 | 1 | 1 | 0 | 0 | 0 | 0 | 0 | 0 | 0 | 0 | 0 | 0 | 0 | 0 | 3 | 12 |
| | Star-3 Alpha | 1 | 1 | 1 | 1 | 1 | 0 | 0 | 1 | 0 | 0 | 0 | 0 | 0 | 0 | 1 | 7 | 8 |
| | Hunyuan | 1 | 1 | 0 | 1 | 1 | 0 | 0 | 1 | 1 | 1 | 1 | 1 | 0 | 0 | 1 | 10 | 5 |
| | GLM-4 | 1 | 1 | 0 | 0 | 0 | 1 | 0 | 0 | 1 | 0 | 0 | 0 | 0 | 0 | 0 | 4 | 11 |
| Chairs | Recraft V3 | 1 | 1 | 1 | 1 | 1 | 0 | 0 | 0 | 0 | 0 | 0 | 0 | 0 | 0 | 0 | 5 | 10 |
| | Imagen-3 | 1 | 1 | 1 | 1 | 1 | 1 | 1 | 1 | 1 | 0 | 0 | 0 | 1 | 1 | 0 | 11 | 4 |
| | Dall·E 3 | 1 | 1 | 1 | 1 | 1 | 0 | 0 | 0 | 0 | 0 | 0 | 0 | 0 | 0 | 0 | 5 | 10 |
| | Grok 3 | 1 | 1 | 1 | 0 | 1 | 0 | 0 | 0 | 0 | 0 | 0 | 0 | 1 | 0 | 0 | 5 | 10 |
| | Gemini 2.0 Flash | 1 | 1 | 1 | 1 | 0 | 0 | 0 | 0 | 0 | 0 | 1 | 0 | 0 | 1 | 0 | 6 | 9 |
| | FLUX 1.1 | 1 | 1 | 1 | 1 | 1 | 1 | 0 | 1 | 0 | 0 | 0 | 0 | 0 | 0 | 0 | 7 | 8 |
| | Firefly 3 | 1 | 1 | 1 | 1 | 1 | 0 | 0 | 0 | 1 | 0 | 0 | 0 | 0 | 0 | 0 | 6 | 9 |
| | SD 3.5 | 1 | 1 | 0 | 1 | 0 | 0 | 0 | 0 | 0 | 0 | 0 | 0 | 0 | 0 | 0 | 3 | 12 |
| | Doubao | 1 | 1 | 1 | 1 | 0 | 1 | 1 | 0 | 0 | 1 | 0 | 0 | 0 | 0 | 0 | 7 | 8 |
| | Qwen2.5-Max | 1 | 1 | 1 | 0 | 1 | 1 | 0 | 0 | 0 | 0 | 0 | 0 | 0 | 0 | 0 | 5 | 10 |
| | WanX2.1 | 1 | 1 | 1 | 1 | 0 | 1 | 0 | 0 | 0 | 0 | 0 | 0 | 0 | 0 | 1 | 6 | 9 |
| | Kling | 1 | 1 | 1 | 1 | 1 | 0 | 0 | 0 | 0 | 0 | 0 | 0 | 0 | 0 | 0 | 5 | 10 |
| | Star-3 Alpha | 1 | 1 | 1 | 1 | 1 | 1 | 1 | 1 | 0 | 0 | 0 | 0 | 0 | 0 | 0 | 8 | 7 |
| | Hunyuan | 1 | 1 | 1 | 1 | 1 | 0 | 1 | 1 | 0 | 0 | 0 | 0 | 0 | 0 | 1 | 9 | 6 |
| | GLM-4 | 1 | 1 | 1 | 1 | 0 | 1 | 0 | 0 | 0 | 0 | 0 | 0 | 0 | 0 | 0 | 5 | 10 |

Table 22: **Counting Triangles and Chairs With Watercolor style Results.**

| Objects | Model | 1 | 2 | 3 | 4 | 5 | 6 | 7 | 8 | 9 | 10 | 11 | 12 | 13 | 14 | 15 | Correct | Wrong |
|---|---|---|---|---|---|---|---|---|---|---|---|---|---|---|---|---|---|---|
| Flowers | Recraft V3 | 1 | 1 | 1 | 1 | 0 | 1 | 0 | 0 | 0 | 0 | 0 | 0 | 1 | 0 | 0 | 6 | 9 |
| | Imagen-3 | 1 | 1 | 0 | 1 | 1 | 0 | 1 | 0 | 0 | 1 | 1 | 1 | 0 | 0 | 1 | 9 | 6 |
| | Dall·E 3 | 0 | 0 | 0 | 0 | 0 | 0 | 0 | 0 | 0 | 0 | 0 | 0 | 0 | 0 | 0 | 0 | 15 |
| | Grok 3 | 1 | 1 | 1 | 0 | 0 | 0 | 0 | 0 | 0 | 0 | 0 | 0 | 0 | 0 | 0 | 3 | 12 |
| | Gemini 2.0 Flash | 1 | 1 | 1 | 0 | 0 | 1 | 0 | 0 | 0 | 0 | 0 | 0 | 0 | 0 | 0 | 4 | 11 |
| | FLUX 1.1 | 1 | 1 | 1 | 1 | 1 | 0 | 1 | 1 | 0 | 0 | 0 | 0 | 0 | 0 | 0 | 7 | 8 |
| | Firefly 3 | 1 | 1 | 1 | 1 | 1 | 0 | 0 | 1 | 0 | 0 | 0 | 0 | 0 | 0 | 0 | 6 | 9 |
| | SD 3.5 | 1 | 1 | 1 | 0 | 0 | 0 | 0 | 0 | 0 | 0 | 0 | 0 | 0 | 0 | 0 | 3 | 12 |
| | Doubao | 1 | 1 | 1 | 1 | 1 | 0 | 0 | 0 | 0 | 1 | 1 | 0 | 0 | 0 | 0 | 7 | 8 |
| | Qwen2.5-Max | 1 | 1 | 1 | 1 | 0 | 0 | 0 | 0 | 0 | 0 | 0 | 0 | 0 | 0 | 0 | 4 | 11 |
| | WanX2.1 | 1 | 0 | 0 | 0 | 0 | 0 | 0 | 0 | 0 | 0 | 0 | 0 | 0 | 0 | 0 | 1 | 14 |
| | Kling | 1 | 1 | 1 | 1 | 0 | 1 | 0 | 0 | 1 | 0 | 0 | 0 | 1 | 1 | 0 | 8 | 7 |
| | Star-3 Alpha | 1 | 1 | 1 | 1 | 1 | 0 | 0 | 0 | 0 | 0 | 0 | 0 | 0 | 0 | 0 | 5 | 10 |
| | Hunyuan | 0 | 1 | 1 | 1 | 1 | 1 | 1 | 0 | 1 | 0 | 0 | 1 | 1 | 0 | 0 | 9 | 6 |
| | GLM-4 | 0 | 1 | 1 | 0 | 1 | 0 | 0 | 0 | 0 | 0 | 0 | 0 | 0 | 0 | 0 | 3 | 12 |

Table 23: **Counting Flowers With Watercolor Style Results.**

| Objects | Model | 1 | 2 | 3 | 4 | 5 | 6 | 7 | 8 | 9 | 10 | 11 | 12 | 13 | 14 | 15 | Correct | Wrong |
|---|---|---|---|---|---|---|---|---|---|---|---|---|---|---|---|---|---|---|
| Apples | Recraft V3 | 1 | 1 | 1 | 1 | 0 | 0 | 1 | 0 | 1 | 0 | 0 | 0 | 0 | 0 | 0 | 6 | 9 |
| | Imagen-3 | 1 | 1 | 1 | 1 | 1 | 0 | 0 | 0 | 0 | 0 | 0 | 0 | 0 | 0 | 0 | 5 | 10 |
| | Dall·E 3 | 1 | 1 | 1 | 0 | 1 | 0 | 0 | 0 | 0 | 0 | 0 | 1 | 0 | 0 | 0 | 5 | 10 |
| | Grok 3 | 1 | 1 | 1 | 0 | 0 | 0 | 0 | 0 | 0 | 0 | 0 | 0 | 0 | 0 | 0 | 3 | 12 |
| | Gemini 2.0 Flash | 1 | 1 | 1 | 1 | 0 | 0 | 1 | 0 | 0 | 0 | 0 | 0 | 0 | 0 | 0 | 5 | 10 |
| | FLUX 1.1 | 1 | 1 | 0 | 0 | 0 | 0 | 0 | 0 | 0 | 0 | 0 | 0 | 0 | 0 | 0 | 2 | 13 |
| | Firefly 3 | 1 | 0 | 0 | 1 | 0 | 0 | 0 | 0 | 0 | 0 | 0 | 0 | 0 | 0 | 0 | 2 | 13 |
| | SD 3.5 | 1 | 1 | 0 | 0 | 0 | 0 | 0 | 0 | 0 | 0 | 0 | 0 | 0 | 0 | 0 | 2 | 13 |
| | Doubao | 0 | 1 | 1 | 0 | 0 | 0 | 1 | 0 | 0 | 0 | 0 | 0 | 0 | 0 | 0 | 4 | 11 |
| | Qwen2.5-Max | 1 | 1 | 1 | 0 | 0 | 0 | 0 | 0 | 0 | 0 | 0 | 0 | 0 | 0 | 0 | 3 | 12 |
| | WanX2.1 | 1 | 1 | 1 | 0 | 0 | 0 | 1 | 0 | 1 | 0 | 0 | 0 | 0 | 0 | 0 | 5 | 10 |
| | Kling | 1 | 0 | 1 | 0 | 1 | 0 | 1 | 0 | 0 | 0 | 0 | 0 | 0 | 0 | 0 | 4 | 11 |
| | Star-3 Alpha | 1 | 1 | 1 | 0 | 1 | 0 | 0 | 0 | 0 | 0 | 1 | 0 | 0 | 0 | 0 | 5 | 10 |
| | Hunyuan | 1 | 1 | 1 | 0 | 0 | 0 | 0 | 0 | 0 | 0 | 0 | 0 | 0 | 0 | 0 | 3 | 12 |
| | GLM-4 | 1 | 1 | 1 | 1 | 1 | 1 | 1 | 0 | 0 | 0 | 0 | 0 | 0 | 0 | 0 | 7 | 8 |
| Watermelons | Recraft V3 | 1 | 1 | 1 | 1 | 0 | 0 | 1 | 0 | 1 | 0 | 0 | 0 | 0 | 0 | 0 | 6 | 9 |
| | Imagen-3 | 1 | 1 | 1 | 1 | 1 | 0 | 1 | 0 | 0 | 0 | 0 | 0 | 0 | 0 | 0 | 6 | 9 |
| | Dall·E 3 | 1 | 1 | 1 | 1 | 1 | 0 | 1 | 0 | 0 | 1 | 1 | 1 | 1 | 0 | 0 | 10 | 5 |
| | Grok 3 | 1 | 1 | 1 | 0 | 0 | 0 | 0 | 0 | 0 | 0 | 0 | 0 | 0 | 0 | 0 | 3 | 12 |
| | Gemini 2.0 Flash | 1 | 1 | 1 | 0 | 0 | 0 | 1 | 0 | 0 | 0 | 0 | 0 | 0 | 0 | 0 | 4 | 11 |
| | FLUX 1.1 | 0 | 0 | 1 | 0 | 0 | 0 | 0 | 0 | 0 | 0 | 0 | 0 | 0 | 0 | 0 | 1 | 14 |
| | Firefly 3 | 0 | 0 | 0 | 0 | 0 | 0 | 0 | 0 | 0 | 0 | 0 | 0 | 0 | 0 | 0 | 0 | 15 |
| | SD 3.5 | 0 | 0 | 0 | 0 | 0 | 0 | 0 | 0 | 0 | 0 | 0 | 0 | 0 | 0 | 0 | 0 | 15 |
| | Doubao | 0 | 1 | 1 | 0 | 0 | 0 | 0 | 0 | 0 | 0 | 0 | 0 | 0 | 0 | 0 | 2 | 13 |
| | Qwen2.5-Max | 1 | 1 | 1 | 1 | 0 | 0 | 0 | 0 | 0 | 0 | 0 | 0 | 0 | 0 | 0 | 4 | 11 |
| | WanX2.1 | 1 | 0 | 0 | 0 | 1 | 0 | 0 | 0 | 0 | 0 | 0 | 0 | 1 | 0 | 0 | 3 | 12 |
| | Kling | 0 | 0 | 1 | 0 | 0 | 0 | 0 | 0 | 0 | 0 | 0 | 0 | 0 | 0 | 0 | 1 | 14 |
| | Star-3 Alpha | 1 | 1 | 0 | 0 | 0 | 0 | 1 | 0 | 0 | 0 | 1 | 0 | 0 | 0 | 0 | 4 | 11 |
| | Hunyuan | 1 | 1 | 1 | 0 | 1 | 0 | 0 | 0 | 1 | 0 | 0 | 0 | 0 | 0 | 0 | 5 | 10 |
| | GLM-4 | 0 | 1 | 1 | 1 | 1 | 0 | 0 | 1 | 0 | 0 | 0 | 0 | 0 | 0 | 0 | 5 | 10 |

Table 24: **Counting Apples and Watermelons with Multiplicative Decomposition Results.**

| Objects | Model | 1 | 2 | 3 | 4 | 5 | 6 | 7 | 8 | 9 | 10 | 11 | 12 | 13 | 14 | 15 | Correct | Wrong |
|---|---|---|---|---|---|---|---|---|---|---|---|---|---|---|---|---|---|---|
| Humans | Recraft V3 | 1 | 1 | 1 | 0 | 1 | 0 | 0 | 0 | 0 | 0 | 0 | 0 | 0 | 0 | 0 | 4 | 11 |
| | Imagen-3 | 1 | 1 | 1 | 0 | 1 | 0 | 1 | 0 | 0 | 0 | 1 | 0 | 0 | 0 | 0 | 6 | 9 |
| | Dall·E 3 | 1 | 1 | 1 | 0 | 1 | 0 | 1 | 0 | 0 | 0 | 0 | 0 | 0 | 0 | 0 | 5 | 10 |
| | Grok 3 | 1 | 0 | 0 | 1 | 0 | 1 | 1 | 0 | 0 | 0 | 0 | 0 | 0 | 1 | 0 | 5 | 10 |
| | Gemini 2.0 Flash | 1 | 1 | 1 | 1 | 1 | 0 | 1 | 0 | 0 | 0 | 1 | 0 | 0 | 0 | 0 | 7 | 8 |
| | FLUX 1.1 | 1 | 1 | 0 | 0 | 0 | 0 | 0 | 0 | 0 | 0 | 0 | 0 | 0 | 0 | 0 | 2 | 13 |
| | Firefly 3 | 1 | 0 | 0 | 0 | 0 | 0 | 0 | 0 | 0 | 0 | 0 | 0 | 0 | 0 | 0 | 1 | 14 |
| | SD 3.5 | 1 | 1 | 1 | 0 | 0 | 0 | 0 | 0 | 0 | 0 | 0 | 0 | 0 | 0 | 0 | 3 | 12 |
| | Doubao | 0 | 1 | 1 | 0 | 1 | 0 | 0 | 0 | 0 | 0 | 1 | 0 | 0 | 0 | 0 | 4 | 11 |
| | Qwen2.5-Max | 1 | 1 | 0 | 1 | 0 | 0 | 0 | 1 | 0 | 0 | 1 | 0 | 0 | 0 | 0 | 5 | 10 |
| | WanX2.1 | 1 | 1 | 1 | 1 | 1 | 0 | 0 | 1 | 0 | 0 | 1 | 0 | 0 | 0 | 0 | 7 | 8 |
| | Kling | 0 | 0 | 0 | 0 | 1 | 0 | 0 | 0 | 0 | 0 | 0 | 0 | 0 | 0 | 0 | 1 | 14 |
| | Star-3 Alpha | 0 | 1 | 1 | 0 | 1 | 0 | 0 | 0 | 0 | 0 | 0 | 0 | 0 | 0 | 0 | 3 | 12 |
| | Hunyuan | 1 | 1 | 1 | 0 | 0 | 0 | 1 | 0 | 0 | 0 | 0 | 0 | 0 | 0 | 0 | 4 | 11 |
| | GLM-4 | 1 | 1 | 1 | 1 | 1 | 0 | 0 | 0 | 0 | 0 | 0 | 0 | 0 | 0 | 0 | 5 | 10 |
| Cats | Recraft V3 | 1 | 1 | 1 | 1 | 0 | 1 | 0 | 0 | 0 | 0 | 0 | 0 | 0 | 0 | 0 | 5 | 10 |
| | Imagen-3 | 1 | 1 | 0 | 0 | 1 | 1 | 0 | 1 | 0 | 1 | 0 | 0 | 0 | 0 | 0 | 6 | 9 |
| | Dall·E 3 | 1 | 1 | 1 | 1 | 1 | 1 | 1 | 0 | 0 | 0 | 0 | 0 | 0 | 0 | 0 | 7 | 8 |
| | Grok 3 | 1 | 1 | 1 | 0 | 0 | 0 | 0 | 0 | 0 | 0 | 0 | 0 | 0 | 0 | 0 | 3 | 12 |
| | Gemini 2.0 Flash | 1 | 1 | 1 | 0 | 0 | 1 | 1 | 1 | 0 | 0 | 0 | 0 | 0 | 0 | 0 | 6 | 9 |
| | FLUX 1.1 | 1 | 1 | 1 | 0 | 1 | 0 | 0 | 0 | 0 | 0 | 0 | 0 | 0 | 0 | 0 | 4 | 11 |
| | Firefly 3 | 1 | 1 | 1 | 0 | 0 | 0 | 0 | 0 | 0 | 0 | 0 | 0 | 0 | 0 | 0 | 3 | 12 |
| | SD 3.5 | 1 | 1 | 1 | 0 | 1 | 0 | 0 | 0 | 0 | 0 | 0 | 0 | 0 | 0 | 0 | 4 | 11 |
| | Doubao | 1 | 1 | 1 | 0 | 0 | 0 | 0 | 0 | 0 | 0 | 0 | 0 | 0 | 0 | 0 | 3 | 12 |
| | Qwen2.5-Max | 1 | 1 | 1 | 0 | 0 | 0 | 0 | 0 | 0 | 0 | 1 | 0 | 0 | 0 | 0 | 4 | 11 |
| | WanX2.1 | 1 | 1 | 1 | 1 | 1 | 0 | 1 | 0 | 0 | 0 | 0 | 0 | 0 | 0 | 0 | 6 | 9 |
| | Kling | 1 | 1 | 1 | 0 | 1 | 0 | 0 | 0 | 0 | 0 | 0 | 0 | 0 | 0 | 0 | 4 | 11 |
| | Star-3 Alpha | 1 | 1 | 1 | 0 | 1 | 0 | 1 | 0 | 0 | 0 | 0 | 1 | 0 | 0 | 0 | 6 | 9 |
| | Hunyuan | 1 | 1 | 1 | 0 | 0 | 0 | 1 | 0 | 0 | 0 | 0 | 1 | 0 | 0 | 0 | 5 | 10 |
| | GLM-4 | 1 | 1 | 1 | 0 | 0 | 0 | 1 | 1 | 0 | 0 | 0 | 1 | 0 | 0 | 0 | 6 | 9 |

Table 25: **Counting Humans and Cats with Multiplicative Decomposition Results.**

| Objects | Model | 1 | 2 | 3 | 4 | 5 | 6 | 7 | 8 | 9 | 10 | 11 | 12 | 13 | 14 | 15 | Correct | Wrong |
|---|---|---|---|---|---|---|---|---|---|---|---|---|---|---|---|---|---|---|
| Triangles | Recraft V3 | 0 | 0 | 1 | 0 | 0 | 0 | 0 | 0 | 0 | 0 | 0 | 0 | 0 | 0 | 0 | 1 | 14 |
| | Imagen-3 | 1 | 0 | 0 | 0 | 0 | 0 | 0 | 0 | 0 | 0 | 0 | 0 | 0 | 0 | 0 | 1 | 14 |
| | Dall·E 3 | 1 | 1 | 0 | 0 | 0 | 1 | 0 | 0 | 0 | 0 | 0 | 0 | 0 | 0 | 0 | 3 | 12 |
| | Grok 3 | 1 | 1 | 1 | 1 | 0 | 1 | 0 | 0 | 0 | 0 | 0 | 0 | 0 | 0 | 0 | 5 | 10 |
| | Gemini 2.0 Flash | 1 | 1 | 0 | 1 | 1 | 0 | 1 | 1 | 0 | 0 | 0 | 0 | 0 | 1 | 0 | 7 | 8 |
| | FLUX 1.1 | 1 | 1 | 0 | 0 | 0 | 0 | 0 | 0 | 0 | 0 | 0 | 0 | 0 | 0 | 0 | 2 | 13 |
| | Firefly 3 | 1 | 0 | 1 | 0 | 0 | 0 | 0 | 0 | 0 | 0 | 0 | 0 | 0 | 0 | 0 | 2 | 13 |
| | SD 3.5 | 1 | 0 | 1 | 0 | 0 | 0 | 0 | 0 | 0 | 0 | 0 | 0 | 0 | 0 | 0 | 2 | 13 |
| | Doubao | 0 | 0 | 1 | 0 | 0 | 0 | 0 | 0 | 0 | 0 | 0 | 0 | 0 | 0 | 0 | 1 | 14 |
| | Qwen2.5-Max | 1 | 1 | 0 | 0 | 0 | 0 | 1 | 0 | 0 | 0 | 0 | 0 | 0 | 0 | 0 | 3 | 12 |
| | WanX2.1 | 1 | 0 | 1 | 0 | 1 | 0 | 0 | 0 | 0 | 0 | 0 | 0 | 0 | 0 | 0 | 3 | 12 |
| | Kling | 1 | 1 | 0 | 0 | 0 | 0 | 0 | 0 | 0 | 0 | 0 | 0 | 0 | 0 | 0 | 2 | 13 |
| | Star-3 Alpha | 0 | 0 | 1 | 0 | 0 | 0 | 0 | 0 | 0 | 0 | 0 | 0 | 0 | 0 | 0 | 1 | 14 |
| | Hunyuan | 0 | 1 | 1 | 0 | 0 | 0 | 0 | 0 | 0 | 0 | 0 | 0 | 0 | 0 | 0 | 2 | 12 |
| | GLM-4 | 1 | 0 | 0 | 0 | 0 | 0 | 0 | 0 | 0 | 0 | 0 | 1 | 0 | 0 | 0 | 2 | 13 |
| Chairs | Recraft V3 | 1 | 1 | 1 | 1 | 1 | 0 | 0 | 0 | 0 | 0 | 0 | 0 | 0 | 0 | 0 | 5 | 10 |
| | Imagen-3 | 1 | 1 | 1 | 1 | 1 | 0 | 0 | 0 | 0 | 0 | 0 | 1 | 0 | 0 | 0 | 6 | 9 |
| | Dall·E 3 | 1 | 1 | 1 | 1 | 0 | 0 | 0 | 0 | 0 | 1 | 1 | 0 | 0 | 0 | 0 | 6 | 9 |
| | Grok 3 | 1 | 1 | 1 | 1 | 0 | 0 | 0 | 0 | 0 | 0 | 0 | 0 | 0 | 0 | 0 | 4 | 11 |
| | Gemini 2.0 Flash | 1 | 1 | 1 | 1 | 0 | 0 | 0 | 0 | 0 | 1 | 0 | 0 | 1 | 0 | 0 | 6 | 9 |
| | FLUX 1.1 | 1 | 1 | 1 | 0 | 1 | 0 | 0 | 0 | 0 | 0 | 0 | 0 | 0 | 0 | 0 | 4 | 11 |
| | Firefly 3 | 1 | 0 | 1 | 0 | 0 | 0 | 0 | 0 | 0 | 0 | 0 | 0 | 0 | 0 | 0 | 2 | 13 |
| | SD 3.5 | 1 | 1 | 1 | 0 | 1 | 0 | 0 | 0 | 0 | 0 | 0 | 0 | 0 | 0 | 0 | 4 | 11 |
| | Doubao | 1 | 1 | 1 | 0 | 0 | 0 | 0 | 0 | 0 | 0 | 0 | 0 | 0 | 0 | 0 | 3 | 12 |
| | Qwen2.5-Max | 1 | 1 | 1 | 1 | 1 | 1 | 1 | 0 | 0 | 0 | 0 | 0 | 0 | 0 | 0 | 7 | 8 |
| | WanX2.1 | 1 | 1 | 1 | 1 | 1 | 1 | 1 | 0 | 0 | 0 | 0 | 0 | 0 | 0 | 0 | 7 | 8 |
| | Kling | 1 | 1 | 1 | 0 | 0 | 0 | 0 | 0 | 0 | 0 | 0 | 0 | 0 | 0 | 0 | 3 | 12 |
| | Star-3 Alpha | 1 | 1 | 1 | 0 | 1 | 0 | 0 | 0 | 0 | 0 | 0 | 0 | 0 | 0 | 0 | 4 | 11 |
| | Hunyuan | 1 | 1 | 1 | 1 | 1 | 0 | 0 | 0 | 0 | 0 | 0 | 0 | 0 | 0 | 0 | 5 | 10 |
| | GLM-4 | 1 | 1 | 1 | 1 | 1 | 0 | 0 | 1 | 0 | 0 | 0 | 0 | 0 | 0 | 1 | 7 | 8 |

Table 26: **Counting Triangles and Chairs with Multiplicative Decomposition Results.**

| Objects | Model | 1 | 2 | 3 | 4 | 5 | 6 | 7 | 8 | 9 | 10 | 11 | 12 | 13 | 14 | 15 | Correct | Wrong |
|---|---|---|---|---|---|---|---|---|---|---|---|---|---|---|---|---|---|---|
| Flowers | Recraft V3 | 1 | 1 | 1 | 0 | 0 | 0 | 0 | 0 | 0 | 0 | 0 | 0 | 0 | 0 | 0 | 3 | 12 |
| | Imagen-3 | 1 | 1 | 1 | 0 | 1 | 0 | 0 | 0 | 0 | 0 | 0 | 1 | 0 | 0 | 0 | 5 | 10 |
| | Dall·E 3 | 1 | 1 | 1 | 0 | 0 | 0 | 1 | 0 | 0 | 0 | 0 | 0 | 0 | 0 | 0 | 4 | 11 |
| | Grok 3 | 1 | 1 | 1 | 1 | 0 | 0 | 0 | 0 | 0 | 0 | 0 | 0 | 0 | 0 | 0 | 4 | 11 |
| | Gemini 2.0 Flash | 1 | 0 | 0 | 0 | 1 | 0 | 1 | 0 | 1 | 1 | 0 | 0 | 1 | 0 | 0 | 6 | 9 |
| | FLUX 1.1 | 1 | 0 | 0 | 0 | 0 | 0 | 0 | 1 | 0 | 0 | 0 | 0 | 0 | 0 | 0 | 2 | 13 |
| | Firefly 3 | 0 | 0 | 0 | 0 | 0 | 1 | 0 | 0 | 0 | 0 | 0 | 0 | 0 | 0 | 0 | 1 | 14 |
| | SD 3.5 | 1 | 1 | 1 | 0 | 0 | 0 | 0 | 0 | 0 | 0 | 0 | 0 | 1 | 0 | 0 | 4 | 11 |
| | Doubao | 1 | 1 | 1 | 0 | 0 | 0 | 0 | 0 | 0 | 0 | 1 | 0 | 0 | 0 | 0 | 4 | 11 |
| | Qwen2.5-Max | 1 | 1 | 0 | 0 | 0 | 0 | 0 | 0 | 0 | 0 | 0 | 0 | 0 | 0 | 0 | 2 | 13 |
| | WanX2.1 | 1 | 1 | 1 | 1 | 0 | 0 | 0 | 0 | 1 | 0 | 1 | 0 | 0 | 0 | 0 | 6 | 9 |
| | Kling | 1 | 0 | 1 | 1 | 0 | 0 | 0 | 0 | 0 | 0 | 0 | 0 | 0 | 0 | 0 | 3 | 12 |
| | Star-3 Alpha | 1 | 1 | 0 | 1 | 0 | 0 | 0 | 0 | 0 | 0 | 0 | 0 | 0 | 0 | 0 | 3 | 12 |
| | Hunyuan | 1 | 0 | 1 | 1 | 0 | 0 | 1 | 0 | 0 | 0 | 0 | 1 | 0 | 0 | 0 | 5 | 10 |
| | GLM-4 | 1 | 1 | 0 | 0 | 1 | 1 | 0 | 1 | 0 | 1 | 0 | 0 | 0 | 0 | 0 | 6 | 9 |

Table 27: **Counting Flowers with Multiplicative Decomposition Results.**

| Objects | Model | 1 | 2 | 3 | 4 | 5 | 6 | 7 | 8 | 9 | 10 | 11 | 12 | 13 | 14 | 15 | Correct | Wrong |
|---|---|---|---|---|---|---|---|---|---|---|---|---|---|---|---|---|---|---|
| Apples | Recraft V3 | 1 | 1 | 1 | 1 | 0 | 1 | 0 | 0 | 0 | 0 | 0 | 0 | 0 | 0 | 0 | 5 | 10 |
| | Imagen-3 | 1 | 1 | 1 | 1 | 0 | 1 | 0 | 0 | 0 | 0 | 0 | 0 | 0 | 0 | 0 | 5 | 10 |
| | Dall·E 3 | 0 | 0 | 0 | 1 | 0 | 0 | 0 | 0 | 0 | 0 | 0 | 0 | 0 | 0 | 0 | 1 | 14 |
| | Grok 3 | 1 | 1 | 1 | 0 | 0 | 0 | 0 | 1 | 0 | 1 | 0 | 0 | 1 | 1 | 0 | 7 | 8 |
| | Gemini 2.0 Flash | 1 | 0 | 1 | 0 | 0 | 0 | 0 | 0 | 0 | 0 | 1 | 0 | 0 | 0 | 0 | 3 | 12 |
| | FLUX 1.1 | 1 | 1 | 1 | 1 | 1 | 0 | 1 | 0 | 1 | 0 | 0 | 0 | 0 | 0 | 0 | 7 | 8 |
| | Firefly 3 | 1 | 0 | 0 | 0 | 1 | 0 | 0 | 0 | 0 | 0 | 0 | 0 | 0 | 0 | 0 | 2 | 13 |
| | SD 3.5 | 1 | 1 | 0 | 0 | 1 | 0 | 0 | 0 | 0 | 0 | 0 | 0 | 0 | 0 | 0 | 3 | 12 |
| | Doubao | 1 | 1 | 0 | 0 | 0 | 0 | 0 | 0 | 0 | 0 | 0 | 0 | 0 | 0 | 0 | 2 | 13 |
| | Qwen2.5-Max | 1 | 1 | 1 | 0 | 0 | 0 | 0 | 0 | 0 | 0 | 0 | 0 | 0 | 0 | 1 | 4 | 11 |
| | WanX2.1 | 1 | 1 | 1 | 0 | 0 | 0 | 0 | 0 | 0 | 0 | 0 | 0 | 0 | 0 | 0 | 3 | 12 |
| | Kling | 1 | 0 | 1 | 0 | 0 | 0 | 0 | 0 | 0 | 0 | 0 | 0 | 0 | 0 | 0 | 2 | 13 |
| | Star-3 Alpha | 1 | 1 | 0 | 0 | 0 | 0 | 0 | 0 | 0 | 0 | 0 | 0 | 0 | 0 | 0 | 2 | 13 |
| | Hunyuan | 1 | 1 | 1 | 1 | 1 | 0 | 0 | 0 | 1 | 0 | 0 | 1 | 0 | 0 | 0 | 7 | 8 |
| | GLM-4 | 1 | 1 | 1 | 0 | 1 | 0 | 0 | 0 | 0 | 0 | 0 | 0 | 0 | 1 | 0 | 5 | 10 |
| Watermelons | Recraft V3 | 1 | 0 | 0 | 0 | 0 | 0 | 0 | 0 | 0 | 1 | 0 | 0 | 0 | 0 | 0 | 2 | 13 |
| | Imagen-3 | 1 | 0 | 1 | 1 | 1 | 0 | 0 | 0 | 1 | 0 | 0 | 0 | 0 | 0 | 0 | 5 | 10 |
| | Dall·E 3 | 1 | 1 | 0 | 0 | 0 | 1 | 0 | 0 | 0 | 0 | 0 | 0 | 0 | 0 | 0 | 3 | 12 |
| | Grok 3 | 1 | 1 | 1 | 1 | 0 | 0 | 0 | 0 | 0 | 0 | 0 | 0 | 0 | 0 | 0 | 4 | 11 |
| | Gemini 2.0 Flash | 1 | 1 | 1 | 1 | 1 | 0 | 0 | 0 | 0 | 0 | 0 | 0 | 0 | 1 | 0 | 6 | 9 |
| | FLUX 1.1 | 1 | 0 | 1 | 0 | 1 | 1 | 0 | 0 | 0 | 0 | 0 | 0 | 0 | 0 | 0 | 4 | 11 |
| | Firefly 3 | 1 | 0 | 1 | 0 | 0 | 0 | 0 | 0 | 0 | 0 | 0 | 0 | 0 | 0 | 0 | 2 | 13 |
| | SD 3.5 | 1 | 0 | 0 | 1 | 0 | 1 | 1 | 0 | 0 | 0 | 0 | 0 | 0 | 0 | 0 | 4 | 11 |
| | Doubao | 1 | 1 | 0 | 0 | 0 | 0 | 0 | 0 | 0 | 0 | 0 | 0 | 0 | 0 | 0 | 2 | 13 |
| | Qwen2.5-Max | 1 | 1 | 0 | 0 | 0 | 0 | 0 | 1 | 0 | 0 | 0 | 0 | 0 | 0 | 0 | 3 | 12 |
| | WanX2.1 | 1 | 0 | 1 | 0 | 1 | 0 | 0 | 0 | 0 | 0 | 0 | 0 | 0 | 0 | 0 | 3 | 12 |
| | Kling | 1 | 1 | 1 | 0 | 1 | 0 | 0 | 0 | 0 | 0 | 0 | 0 | 0 | 0 | 0 | 4 | 11 |
| | Star-3 Alpha | 1 | 0 | 0 | 0 | 0 | 0 | 0 | 0 | 0 | 0 | 0 | 0 | 0 | 0 | 0 | 1 | 14 |
| | Hunyuan | 1 | 1 | 0 | 0 | 0 | 0 | 1 | 0 | 0 | 0 | 0 | 0 | 0 | 0 | 0 | 3 | 12 |
| | GLM-4 | 1 | 1 | 1 | 1 | 1 | 0 | 0 | 0 | 0 | 0 | 0 | 0 | 0 | 0 | 0 | 5 | 10 |

Table 28: **Counting Apples and Watermelons With Additive Decomposition Results.**

| Objects | Model | 1 | 2 | 3 | 4 | 5 | 6 | 7 | 8 | 9 | 10 | 11 | 12 | 13 | 14 | 15 | Correct | Wrong |
|---|---|---|---|---|---|---|---|---|---|---|---|---|---|---|---|---|---|---|
| Humans | Recraft V3 | 1 | 1 | 1 | 1 | 1 | 0 | 1 | 0 | 0 | 0 | 0 | 0 | 0 | 0 | 0 | 6 | 9 |
| | Imagen-3 | 1 | 1 | 1 | 1 | 1 | 1 | 1 | 0 | 1 | 0 | 0 | 0 | 0 | 0 | 0 | 8 | 7 |
| | Dall·E 3 | 1 | 1 | 1 | 0 | 0 | 0 | 0 | 0 | 0 | 0 | 0 | 1 | 0 | 0 | 0 | 4 | 11 |
| | Grok 3 | 1 | 1 | 1 | 1 | 1 | 1 | 0 | 1 | 1 | 0 | 0 | 0 | 1 | 1 | 0 | 10 | 5 |
| | Gemini 2.0 Flash | 1 | 1 | 1 | 1 | 0 | 0 | 0 | 0 | 1 | 0 | 1 | 0 | 0 | 0 | 0 | 6 | 9 |
| | FLUX 1.1 | 1 | 1 | 0 | 1 | 0 | 0 | 0 | 0 | 0 | 0 | 0 | 0 | 0 | 0 | 0 | 3 | 12 |
| | Firefly 3 | 1 | 1 | 1 | 0 | 0 | 0 | 0 | 0 | 0 | 0 | 0 | 0 | 0 | 0 | 0 | 3 | 12 |
| | SD 3.5 | 1 | 0 | 0 | 0 | 0 | 0 | 0 | 0 | 0 | 0 | 0 | 0 | 0 | 0 | 0 | 1 | 14 |
| | Doubao | 1 | 0 | 0 | 0 | 0 | 0 | 0 | 0 | 0 | 0 | 0 | 0 | 0 | 0 | 0 | 1 | 14 |
| | Qwen2.5-Max | 1 | 1 | 0 | 0 | 0 | 0 | 0 | 0 | 0 | 0 | 1 | 0 | 0 | 0 | 0 | 3 | 12 |
| | WanX2.1 | 1 | 1 | 1 | 0 | 0 | 0 | 0 | 0 | 0 | 0 | 0 | 0 | 0 | 0 | 0 | 3 | 12 |
| | Kling | 1 | 0 | 0 | 0 | 0 | 0 | 0 | 0 | 0 | 0 | 0 | 0 | 0 | 0 | 0 | 1 | 14 |
| | Star-3 Alpha | 0 | 1 | 0 | 0 | 0 | 0 | 0 | 0 | 0 | 0 | 0 | 0 | 0 | 0 | 0 | 1 | 14 |
| | Hunyuan | 1 | 0 | 0 | 0 | 0 | 0 | 1 | 0 | 0 | 0 | 0 | 0 | 0 | 0 | 0 | 2 | 13 |
| | GLM-4 | 1 | 1 | 1 | 1 | 1 | 0 | 1 | 0 | 0 | 1 | 0 | 0 | 0 | 0 | 0 | 7 | 8 |
| Cats | Recraft V3 | 1 | 1 | 0 | 1 | 0 | 0 | 0 | 0 | 0 | 0 | 0 | 0 | 0 | 0 | 0 | 3 | 12 |
| | Imagen-3 | 1 | 1 | 0 | 0 | 0 | 0 | 0 | 0 | 0 | 0 | 0 | 0 | 0 | 0 | 0 | 2 | 13 |
| | Dall·E 3 | 1 | 1 | 1 | 1 | 0 | 0 | 0 | 0 | 0 | 0 | 1 | 0 | 0 | 0 | 0 | 5 | 10 |
| | Grok 3 | 1 | 1 | 1 | 0 | 1 | 1 | 0 | 0 | 0 | 0 | 0 | 0 | 0 | 0 | 0 | 5 | 10 |
| | Gemini 2.0 Flash | 1 | 1 | 0 | 0 | 1 | 0 | 0 | 0 | 0 | 0 | 0 | 0 | 0 | 0 | 0 | 3 | 12 |
| | FLUX 1.1 | 1 | 1 | 0 | 0 | 0 | 0 | 0 | 0 | 0 | 0 | 0 | 0 | 0 | 0 | 0 | 2 | 13 |
| | Firefly 3 | 1 | 1 | 1 | 1 | 0 | 0 | 0 | 0 | 0 | 0 | 0 | 0 | 0 | 0 | 0 | 4 | 11 |
| | SD 3.5 | 1 | 1 | 1 | 0 | 0 | 0 | 0 | 0 | 0 | 0 | 0 | 0 | 0 | 0 | 0 | 3 | 12 |
| | Doubao | 1 | 1 | 0 | 0 | 0 | 0 | 0 | 0 | 0 | 0 | 0 | 0 | 0 | 0 | 0 | 2 | 13 |
| | Qwen2.5-Max | 1 | 1 | 1 | 1 | 1 | 1 | 0 | 0 | 0 | 0 | 0 | 0 | 1 | 0 | 0 | 7 | 8 |
| | WanX2.1 | 1 | 1 | 0 | 0 | 0 | 0 | 0 | 0 | 0 | 0 | 0 | 0 | 0 | 0 | 0 | 2 | 13 |
| | Kling | 1 | 0 | 1 | 0 | 0 | 0 | 0 | 0 | 0 | 0 | 0 | 0 | 0 | 0 | 0 | 2 | 13 |
| | Star-3 Alpha | 1 | 1 | 1 | 0 | 0 | 0 | 1 | 0 | 1 | 0 | 0 | 0 | 0 | 0 | 0 | 5 | 10 |
| | Hunyuan | 1 | 0 | 1 | 0 | 1 | 1 | 0 | 0 | 0 | 0 | 0 | 0 | 0 | 0 | 0 | 4 | 11 |
| | GLM-4 | 1 | 1 | 1 | 1 | 0 | 0 | 0 | 0 | 0 | 0 | 0 | 0 | 0 | 0 | 0 | 4 | 11 |

Table 29: **Counting Humans and Cats With Additive Decomposition Results.**

| Objects | Model | 1 | 2 | 3 | 4 | 5 | 6 | 7 | 8 | 9 | 10 | 11 | 12 | 13 | 14 | 15 | Correct | Wrong |
|---|---|---|---|---|---|---|---|---|---|---|---|---|---|---|---|---|---|---|
| Triangles | Recraft V3 | 1 | 0 | 1 | 0 | 0 | 0 | 0 | 0 | 0 | 0 | 0 | 0 | 0 | 0 | 0 | 2 | 13 |
| | Imagen-3 | 1 | 0 | 0 | 0 | 0 | 0 | 0 | 1 | 0 | 0 | 0 | 0 | 0 | 0 | 0 | 2 | 13 |
| | Dall·E 3 | 1 | 0 | 0 | 0 | 0 | 0 | 0 | 0 | 0 | 0 | 0 | 0 | 0 | 0 | 0 | 1 | 14 |
| | Grok 3 | 1 | 1 | 1 | 1 | 0 | 0 | 0 | 0 | 0 | 1 | 0 | 0 | 1 | 0 | 0 | 6 | 9 |
| | Gemini 2.0 Flash | 1 | 0 | 1 | 0 | 1 | 0 | 0 | 0 | 0 | 0 | 0 | 0 | 0 | 1 | 0 | 4 | 11 |
| | FLUX 1.1 | 1 | 1 | 0 | 0 | 0 | 0 | 0 | 0 | 0 | 0 | 0 | 0 | 0 | 0 | 0 | 2 | 13 |
| | Firefly 3 | 1 | 0 | 0 | 0 | 0 | 1 | 0 | 0 | 0 | 0 | 0 | 0 | 0 | 0 | 0 | 2 | 13 |
| | SD 3.5 | 0 | 0 | 0 | 1 | 0 | 0 | 0 | 0 | 0 | 0 | 0 | 0 | 0 | 0 | 0 | 1 | 14 |
| | Doubao | 1 | 1 | 0 | 0 | 0 | 0 | 0 | 0 | 0 | 0 | 0 | 0 | 0 | 0 | 0 | 2 | 13 |
| | Qwen2.5-Max | 1 | 1 | 0 | 1 | 0 | 0 | 0 | 0 | 1 | 0 | 0 | 0 | 0 | 0 | 0 | 4 | 11 |
| | WanX2.1 | 1 | 0 | 1 | 0 | 1 | 0 | 0 | 0 | 0 | 0 | 0 | 0 | 0 | 0 | 0 | 3 | 12 |
| | Kling | 0 | 0 | 0 | 0 | 0 | 0 | 0 | 0 | 0 | 0 | 0 | 0 | 0 | 0 | 0 | 0 | 15 |
| | Star-3 Alpha | 1 | 0 | 0 | 0 | 0 | 0 | 0 | 0 | 0 | 0 | 0 | 0 | 0 | 0 | 0 | 1 | 14 |
| | Hunyuan | 1 | 0 | 0 | 1 | 0 | 1 | 0 | 0 | 0 | 0 | 0 | 0 | 0 | 0 | 0 | 3 | 12 |
| | GLM-4 | 1 | 1 | 1 | 1 | 0 | 1 | 1 | 0 | 0 | 0 | 0 | 0 | 0 | 0 | 0 | 6 | 9 |
| Chairs | Recraft V3 | 1 | 1 | 1 | 1 | 0 | 1 | 0 | 0 | 1 | 0 | 0 | 0 | 0 | 0 | 0 | 6 | 9 |
| | Imagen-3 | 1 | 1 | 0 | 1 | 1 | 0 | 0 | 0 | 0 | 0 | 0 | 0 | 0 | 0 | 0 | 4 | 11 |
| | Dall·E 3 | 1 | 1 | 0 | 0 | 0 | 0 | 0 | 0 | 1 | 0 | 0 | 1 | 0 | 0 | 0 | 4 | 11 |
| | Grok 3 | 1 | 1 | 0 | 0 | 0 | 0 | 0 | 0 | 0 | 0 | 0 | 1 | 0 | 0 | 0 | 3 | 12 |
| | Gemini 2.0 Flash | 1 | 1 | 1 | 1 | 1 | 1 | 0 | 0 | 0 | 0 | 0 | 0 | 0 | 0 | 0 | 6 | 9 |
| | FLUX 1.1 | 1 | 1 | 0 | 0 | 0 | 0 | 0 | 0 | 1 | 0 | 0 | 0 | 0 | 0 | 0 | 3 | 12 |
| | Firefly 3 | 1 | 1 | 1 | 0 | 0 | 0 | 0 | 0 | 0 | 0 | 0 | 0 | 0 | 0 | 0 | 3 | 12 |
| | SD 3.5 | 1 | 1 | 0 | 0 | 0 | 0 | 0 | 0 | 0 | 0 | 0 | 0 | 0 | 0 | 0 | 2 | 13 |
| | Doubao | 1 | 1 | 0 | 0 | 0 | 0 | 0 | 0 | 0 | 0 | 0 | 0 | 0 | 0 | 0 | 2 | 13 |
| | Qwen2.5-Max | 1 | 1 | 0 | 0 | 0 | 1 | 0 | 0 | 1 | 0 | 1 | 0 | 0 | 0 | 0 | 5 | 10 |
| | WanX2.1 | 1 | 1 | 0 | 0 | 0 | 0 | 0 | 0 | 0 | 0 | 0 | 0 | 0 | 0 | 0 | 2 | 13 |
| | Kling | 1 | 0 | 0 | 0 | 0 | 0 | 0 | 0 | 0 | 0 | 0 | 0 | 0 | 0 | 0 | 1 | 14 |
| | Star-3 Alpha | 1 | 1 | 0 | 0 | 0 | 0 | 0 | 0 | 0 | 0 | 0 | 0 | 0 | 0 | 0 | 2 | 13 |
| | Hunyuan | 1 | 1 | 1 | 1 | 0 | 0 | 1 | 0 | 0 | 0 | 0 | 1 | 0 | 0 | 0 | 5 | 10 |
| | GLM-4 | 1 | 1 | 1 | 1 | 0 | 1 | 1 | 1 | 0 | 0 | 0 | 0 | 0 | 0 | 1 | 8 | 7 |

Table 30: **Counting Triangles and Chairs With Additive Decomposition Results.**

| Objects | Model | 1 | 2 | 3 | 4 | 5 | 6 | 7 | 8 | 9 | 10 | 11 | 12 | 13 | 14 | 15 | Correct | Wrong |
|---|---|---|---|---|---|---|---|---|---|---|---|---|---|---|---|---|---|---|
| Flowers | Recraft V3 | 1 | 1 | 0 | 1 | 0 | 0 | 0 | 0 | 0 | 0 | 0 | 0 | 0 | 0 | 0 | 3 | 12 |
| | Imagen-3 | 1 | 0 | 0 | 0 | 0 | 0 | 0 | 0 | 0 | 1 | 0 | 1 | 0 | 1 | 0 | 4 | 11 |
| | Dall·E 3 | 1 | 1 | 1 | 0 | 0 | 0 | 0 | 0 | 0 | 0 | 0 | 0 | 0 | 0 | 0 | 3 | 12 |
| | Grok 3 | 1 | 1 | 0 | 0 | 0 | 0 | 0 | 0 | 0 | 0 | 0 | 0 | 0 | 0 | 0 | 2 | 13 |
| | Gemini 2.0 Flash | 1 | 1 | 1 | 0 | 0 | 1 | 0 | 0 | 0 | 0 | 0 | 0 | 0 | 0 | 0 | 4 | 11 |
| | FLUX 1.1 | 1 | 0 | 1 | 0 | 0 | 0 | 0 | 0 | 0 | 1 | 0 | 1 | 0 | 0 | 0 | 4 | 11 |
| | Firefly 3 | 1 | 0 | 0 | 0 | 0 | 0 | 0 | 0 | 0 | 0 | 0 | 0 | 0 | 0 | 0 | 1 | 14 |
| | SD 3.5 | 1 | 0 | 1 | 0 | 0 | 0 | 0 | 0 | 0 | 0 | 0 | 0 | 0 | 0 | 0 | 2 | 13 |
| | Doubao | 1 | 1 | 0 | 0 | 0 | 0 | 0 | 0 | 0 | 0 | 0 | 0 | 0 | 0 | 0 | 2 | 13 |
| | Qwen2.5-Max | 1 | 1 | 0 | 0 | 0 | 0 | 0 | 0 | 0 | 0 | 0 | 0 | 0 | 0 | 0 | 2 | 13 |
| | WanX2.1 | 1 | 1 | 0 | 0 | 0 | 0 | 1 | 0 | 0 | 0 | 0 | 0 | 0 | 1 | 0 | 4 | 11 |
| | Kling | 0 | 0 | 1 | 0 | 0 | 0 | 0 | 0 | 0 | 0 | 0 | 0 | 0 | 0 | 0 | 1 | 14 |
| | Star-3 Alpha | 1 | 1 | 1 | 0 | 0 | 0 | 1 | 0 | 0 | 0 | 0 | 0 | 0 | 0 | 0 | 4 | 11 |
| | Hunyuan | 0 | 0 | 0 | 1 | 0 | 0 | 0 | 0 | 1 | 0 | 0 | 0 | 0 | 0 | 0 | 2 | 13 |
| | GLM-4 | 0 | 1 | 1 | 0 | 0 | 0 | 0 | 0 | 0 | 0 | 0 | 0 | 0 | 0 | 0 | 2 | 13 |

Table 31: **Counting Flowers With Additive Decomposition Results.**

| Objects | Model | 1 | 2 | 3 | 4 | 5 | 6 | 7 | 8 | 9 | 10 | 11 | 12 | 13 | 14 | 15 | Correct | Wrong |
|---|---|---|---|---|---|---|---|---|---|---|---|---|---|---|---|---|---|---|
| Apples | Recraft V3 | 1 | 0 | 0 | 1 | 0 | 0 | 0 | 0 | 1 | 0 | 0 | 0 | 0 | 0 | 0 | 3 | 12 |
| | Imagen-3 | 1 | 0 | 1 | 1 | 1 | 1 | 0 | 1 | 1 | 0 | 0 | 0 | 0 | 0 | 0 | 7 | 8 |
| | Dall·E 3 | 1 | 1 | 1 | 1 | 1 | 0 | 0 | 0 | 0 | 0 | 0 | 0 | 0 | 0 | 0 | 5 | 10 |
| | Grok 3 | 1 | 1 | 1 | 0 | 0 | 0 | 0 | 0 | 0 | 0 | 0 | 0 | 0 | 0 | 0 | 3 | 12 |
| | Gemini 2.0 Flash | 1 | 1 | 0 | 1 | 0 | 1 | 0 | 0 | 1 | 0 | 0 | 1 | 0 | 0 | 0 | 6 | 9 |
| | FLUX 1.1 | 1 | 1 | 1 | 1 | 0 | 0 | 0 | 0 | 1 | 0 | 0 | 0 | 0 | 0 | 0 | 5 | 10 |
| | Firefly 3 | 1 | 1 | 1 | 1 | 1 | 0 | 0 | 0 | 1 | 0 | 0 | 1 | 0 | 0 | 1 | 8 | 7 |
| | SD 3.5 | 1 | 1 | 0 | 0 | 0 | 1 | 0 | 0 | 0 | 0 | 0 | 0 | 0 | 0 | 0 | 3 | 12 |
| | Doubao | 1 | 1 | 1 | 1 | 1 | 0 | 0 | 0 | 1 | 0 | 0 | 0 | 0 | 0 | 0 | 6 | 9 |
| | Qwen2.5-Max | 1 | 1 | 1 | 1 | 1 | 1 | 1 | 1 | 0 | 0 | 1 | 0 | 0 | 0 | 0 | 9 | 6 |
| | WanX2.1 | 1 | 0 | 1 | 1 | 0 | 0 | 0 | 0 | 1 | 0 | 0 | 0 | 0 | 0 | 1 | 5 | 10 |
| | Kling | 1 | 1 | 1 | 1 | 0 | 0 | 0 | 0 | 1 | 0 | 0 | 0 | 1 | 0 | 0 | 6 | 9 |
| | Star-3 Alpha | 1 | 1 | 0 | 1 | 0 | 0 | 0 | 0 | 1 | 0 | 0 | 0 | 0 | 0 | 0 | 4 | 11 |
| | Hunyuan | 1 | 0 | 0 | 1 | 0 | 0 | 0 | 0 | 1 | 0 | 0 | 0 | 0 | 0 | 0 | 3 | 12 |
| | GLM-4 | 1 | 0 | 1 | 1 | 0 | 0 | 0 | 0 | 1 | 0 | 0 | 1 | 0 | 0 | 0 | 5 | 10 |
| Watermelons | Recraft V3 | 0 | 1 | 0 | 1 | 0 | 0 | 0 | 0 | 1 | 0 | 0 | 0 | 0 | 0 | 0 | 3 | 12 |
| | Imagen-3 | 1 | 1 | 1 | 1 | 0 | 1 | 0 | 0 | 1 | 0 | 0 | 1 | 0 | 0 | 0 | 7 | 8 |
| | Dall·E 3 | 1 | 1 | 1 | 1 | 1 | 0 | 0 | 0 | 0 | 0 | 0 | 0 | 0 | 0 | 0 | 5 | 10 |
| | Grok 3 | 1 | 1 | 0 | 1 | 0 | 0 | 0 | 0 | 0 | 0 | 0 | 0 | 0 | 0 | 1 | 4 | 11 |
| | Gemini 2.0 Flash | 1 | 1 | 0 | 1 | 1 | 0 | 0 | 0 | 1 | 0 | 0 | 1 | 0 | 0 | 0 | 6 | 9 |
| | FLUX 1.1 | 1 | 1 | 1 | 1 | 0 | 1 | 0 | 1 | 1 | 0 | 0 | 0 | 0 | 0 | 0 | 7 | 8 |
| | Firefly 3 | 1 | 0 | 1 | 0 | 1 | 0 | 0 | 0 | 1 | 0 | 0 | 0 | 0 | 0 | 0 | 4 | 11 |
| | SD 3.5 | 1 | 1 | 0 | 0 | 0 | 0 | 0 | 0 | 1 | 0 | 0 | 0 | 0 | 0 | 0 | 3 | 11 |
| | Doubao | 0 | 1 | 1 | 1 | 1 | 1 | 0 | 0 | 1 | 0 | 0 | 0 | 0 | 1 | 0 | 7 | 8 |
| | Qwen2.5-Max | 1 | 0 | 1 | 1 | 0 | 0 | 0 | 0 | 0 | 0 | 0 | 0 | 0 | 0 | 0 | 3 | 12 |
| | WanX2.1 | 1 | 0 | 1 | 1 | 0 | 1 | 0 | 0 | 1 | 0 | 0 | 0 | 0 | 0 | 0 | 5 | 10 |
| | Kling | 1 | 0 | 1 | 0 | 1 | 0 | 0 | 1 | 1 | 0 | 0 | 0 | 0 | 0 | 0 | 5 | 10 |
| | Star-3 Alpha | 1 | 1 | 1 | 1 | 0 | 0 | 0 | 0 | 1 | 0 | 0 | 1 | 0 | 0 | 0 | 6 | 9 |
| | Hunyuan | 1 | 0 | 0 | 0 | 0 | 0 | 0 | 0 | 0 | 0 | 0 | 1 | 0 | 0 | 0 | 2 | 13 |
| | GLM-4 | 1 | 1 | 1 | 1 | 0 | 0 | 0 | 0 | 1 | 0 | 0 | 0 | 0 | 0 | 0 | 5 | 10 |

Table 32: **Counting Apples and Watermelons with Grid Prior Results.**

| Objects | Model | 1 | 2 | 3 | 4 | 5 | 6 | 7 | 8 | 9 | 10 | 11 | 12 | 13 | 14 | 15 | Correct | Wrong |
|---|---|---|---|---|---|---|---|---|---|---|---|---|---|---|---|---|---|---|
| Humans | Recraft V3 | 1 | 1 | 1 | 1 | 0 | 0 | 0 | 1 | 0 | 0 | 0 | 0 | 0 | 0 | 1 | 6 | 9 |
| | Imagen-3 | 1 | 1 | 1 | 1 | 1 | 1 | 1 | 0 | 1 | 0 | 0 | 1 | 0 | 1 | 0 | 10 | 5 |
| | Dall·E 3 | 1 | 1 | 1 | 1 | 1 | 0 | 1 | 0 | 0 | 0 | 0 | 0 | 0 | 0 | 0 | 6 | 9 |
| | Grok 3 | 1 | 1 | 0 | 1 | 0 | 0 | 0 | 1 | 1 | 0 | 0 | 0 | 1 | 0 | 0 | 6 | 6 |
| | Gemini 2.0 Flash | 1 | 1 | 1 | 1 | 0 | 1 | 1 | 0 | 1 | 0 | 0 | 0 | 0 | 1 | 0 | 8 | 9 |
| | FLUX 1.1 | 1 | 1 | 1 | 1 | 0 | 0 | 0 | 0 | 0 | 0 | 0 | 0 | 0 | 0 | 0 | 4 | 11 |
| | Firefly 3 | 1 | 1 | 1 | 1 | 1 | 0 | 0 | 0 | 0 | 0 | 0 | 1 | 0 | 0 | 0 | 6 | 9 |
| | SD 3.5 | 1 | 1 | 1 | 1 | 0 | 0 | 1 | 1 | 0 | 0 | 0 | 0 | 0 | 0 | 0 | 6 | 9 |
| | Doubao | 1 | 1 | 1 | 1 | 0 | 0 | 1 | 1 | 0 | 0 | 0 | 0 | 0 | 0 | 0 | 6 | 9 |
| | Qwen2.5-Max | 1 | 0 | 1 | 0 | 0 | 1 | 0 | 1 | 0 | 1 | 0 | 0 | 0 | 0 | 0 | 5 | 10 |
| | WanX2.1 | 1 | 1 | 1 | 1 | 1 | 0 | 1 | 0 | 1 | 0 | 0 | 0 | 0 | 0 | 0 | 7 | 8 |
| | Kling | 0 | 1 | 1 | 1 | 0 | 1 | 0 | 1 | 1 | 0 | 0 | 0 | 1 | 0 | 0 | 7 | 8 |
| | Star-3 Alpha | 1 | 0 | 1 | 0 | 0 | 0 | 1 | 0 | 0 | 0 | 0 | 0 | 0 | 0 | 0 | 3 | 12 |
| | Hunyuan | 1 | 1 | 0 | 1 | 0 | 0 | 0 | 0 | 0 | 0 | 0 | 0 | 0 | 0 | 0 | 4 | 11 |
| | GLM-4 | 1 | 1 | 1 | 1 | 1 | 0 | 0 | 0 | 1 | 1 | 0 | 0 | 0 | 0 | 0 | 7 | 8 |
| Cats | Recraft V3 | 1 | 1 | 1 | 1 | 1 | 0 | 0 | 0 | 0 | 0 | 0 | 0 | 0 | 0 | 0 | 5 | 10 |
| | Imagen-3 | 1 | 1 | 1 | 1 | 1 | 0 | 0 | 0 | 1 | 1 | 0 | 1 | 0 | 0 | 1 | 9 | 6 |
| | Dall·E 3 | 1 | 1 | 1 | 1 | 0 | 0 | 0 | 0 | 1 | 0 | 0 | 0 | 0 | 0 | 0 | 5 | 10 |
| | Grok 3 | 1 | 1 | 0 | 1 | 1 | 0 | 0 | 1 | 1 | 0 | 0 | 0 | 0 | 0 | 0 | 6 | 9 |
| | Gemini 2.0 Flash | 1 | 0 | 1 | 1 | 0 | 0 | 0 | 0 | 1 | 0 | 0 | 1 | 0 | 0 | 0 | 5 | 10 |
| | FLUX 1.1 | 1 | 1 | 0 | 1 | 1 | 1 | 0 | 1 | 0 | 0 | 0 | 0 | 0 | 0 | 0 | 6 | 9 |
| | Firefly 3 | 1 | 1 | 1 | 1 | 1 | 1 | 0 | 1 | 1 | 0 | 0 | 0 | 0 | 0 | 0 | 8 | 7 |
| | SD 3.5 | 1 | 1 | 1 | 1 | 0 | 0 | 0 | 1 | 1 | 0 | 0 | 0 | 0 | 0 | 0 | 6 | 9 |
| | Doubao | 1 | 1 | 1 | 1 | 0 | 1 | 0 | 0 | 1 | 0 | 0 | 0 | 0 | 0 | 0 | 6 | 9 |
| | Qwen2.5-Max | 1 | 1 | 1 | 0 | 0 | 0 | 0 | 0 | 1 | 0 | 0 | 0 | 0 | 0 | 0 | 5 | 10 |
| | WanX2.1 | 1 | 0 | 0 | 1 | 1 | 0 | 0 | 0 | 0 | 0 | 0 | 0 | 0 | 1 | 0 | 4 | 11 |
| | Kling | 1 | 1 | 1 | 1 | 0 | 1 | 0 | 0 | 1 | 0 | 0 | 0 | 0 | 0 | 0 | 6 | 9 |
| | Star-3 Alpha | 1 | 1 | 1 | 1 | 0 | 1 | 0 | 0 | 1 | 0 | 0 | 0 | 0 | 0 | 0 | 6 | 9 |
| | Hunyuan | 1 | 1 | 0 | 1 | 0 | 1 | 0 | 0 | 1 | 0 | 0 | 0 | 0 | 0 | 0 | 5 | 10 |
| | GLM-4 | 1 | 1 | 1 | 1 | 0 | 0 | 0 | 0 | 1 | 0 | 0 | 0 | 0 | 0 | 0 | 5 | 10 |

Table 33: **Counting Humans and Cats with Grid Prior Results.**

| Objects | Model | 1 | 2 | 3 | 4 | 5 | 6 | 7 | 8 | 9 | 10 | 11 | 12 | 13 | 14 | 15 | Correct | Wrong |
|---|---|---|---|---|---|---|---|---|---|---|---|---|---|---|---|---|---|---|
| Triangles | Recraft V3 | 0 | 0 | 0 | 0 | 0 | 0 | 0 | 0 | 0 | 0 | 0 | 0 | 0 | 0 | 0 | 0 | 15 |
| | Imagen-3 | 1 | 0 | 0 | 1 | 0 | 1 | 0 | 0 | 1 | 0 | 0 | 0 | 0 | 0 | 0 | 4 | 11 |
| | Dall·E 3 | 1 | 1 | 1 | 1 | 0 | 0 | 0 | 0 | 0 | 0 | 0 | 0 | 0 | 0 | 0 | 4 | 11 |
| | Grok 3 | 1 | 0 | 0 | 0 | 0 | 1 | 0 | 0 | 0 | 0 | 0 | 0 | 0 | 0 | 0 | 2 | 13 |
| | Gemini 2.0 Flash | 1 | 1 | 1 | 0 | 0 | 1 | 0 | 0 | 0 | 0 | 0 | 0 | 0 | 0 | 0 | 4 | 11 |
| | FLUX 1.1 | 1 | 0 | 0 | 0 | 0 | 0 | 0 | 0 | 0 | 0 | 0 | 0 | 0 | 0 | 0 | 1 | 14 |
| | Firefly 3 | 1 | 0 | 1 | 1 | 0 | 1 | 0 | 1 | 1 | 0 | 0 | 0 | 0 | 0 | 0 | 6 | 9 |
| | SD 3.5 | 0 | 0 | 0 | 1 | 0 | 1 | 0 | 0 | 0 | 0 | 0 | 0 | 0 | 0 | 0 | 2 | 13 |
| | Doubao | 1 | 1 | 1 | 0 | 1 | 0 | 1 | 0 | 0 | 1 | 0 | 0 | 0 | 0 | 0 | 6 | 9 |
| | Qwen2.5-Max | 1 | 1 | 1 | 1 | 1 | 1 | 0 | 0 | 0 | 0 | 0 | 0 | 0 | 0 | 0 | 6 | 9 |
| | WanX2.1 | 1 | 1 | 0 | 1 | 0 | 0 | 0 | 0 | 0 | 0 | 0 | 0 | 0 | 0 | 0 | 3 | 12 |
| | Kling | 1 | 0 | 1 | 0 | 0 | 0 | 0 | 0 | 1 | 0 | 0 | 0 | 0 | 0 | 0 | 3 | 12 |
| | Star-3 Alpha | 1 | 1 | 1 | 0 | 0 | 0 | 0 | 0 | 1 | 0 | 0 | 0 | 0 | 0 | 0 | 4 | 11 |
| | Hunyuan | 1 | 0 | 0 | 1 | 0 | 0 | 0 | 0 | 0 | 0 | 0 | 0 | 0 | 0 | 0 | 2 | 13 |
| | GLM-4 | 1 | 1 | 0 | 0 | 0 | 0 | 0 | 1 | 1 | 0 | 1 | 1 | 0 | 0 | 0 | 6 | 9 |
| Chairs | Recraft V3 | 1 | 1 | 1 | 1 | 0 | 1 | 0 | 1 | 1 | 0 | 0 | 0 | 0 | 1 | 0 | 8 | 7 |
| | Imagen-3 | 1 | 1 | 1 | 1 | 1 | 1 | 0 | 1 | 0 | 0 | 0 | 0 | 0 | 0 | 1 | 8 | 7 |
| | Dall·E 3 | 1 | 1 | 1 | 1 | 1 | 1 | 1 | 0 | 0 | 0 | 0 | 0 | 0 | 0 | 0 | 7 | 8 |
| | Grok 3 | 1 | 1 | 0 | 1 | 0 | 0 | 0 | 0 | 0 | 0 | 0 | 0 | 0 | 0 | 0 | 3 | 12 |
| | Gemini 2.0 Flash | 1 | 1 | 0 | 1 | 0 | 1 | 0 | 0 | 1 | 1 | 0 | 0 | 0 | 0 | 0 | 6 | 9 |
| | FLUX 1.1 | 1 | 1 | 1 | 0 | 0 | 1 | 0 | 1 | 1 | 1 | 0 | 1 | 0 | 0 | 0 | 8 | 7 |
| | Firefly 3 | 1 | 1 | 1 | 1 | 1 | 1 | 0 | 1 | 1 | 0 | 0 | 0 | 0 | 0 | 0 | 8 | 7 |
| | SD 3.5 | 1 | 1 | 1 | 0 | 1 | 0 | 0 | 1 | 0 | 0 | 0 | 1 | 0 | 0 | 0 | 6 | 9 |
| | Doubao | 1 | 1 | 1 | 1 | 1 | 1 | 1 | 0 | 0 | 0 | 0 | 0 | 0 | 0 | 0 | 7 | 8 |
| | Qwen2.5-Max | 1 | 1 | 1 | 0 | 0 | 1 | 1 | 1 | 0 | 0 | 0 | 0 | 0 | 0 | 0 | 6 | 9 |
| | WanX2.1 | 1 | 1 | 1 | 1 | 0 | 0 | 0 | 0 | 0 | 0 | 0 | 0 | 0 | 0 | 0 | 4 | 11 |
| | Kling | 1 | 1 | 1 | 1 | 1 | 0 | 0 | 0 | 1 | 0 | 0 | 0 | 0 | 0 | 0 | 6 | 9 |
| | Star-3 Alpha | 1 | 1 | 0 | 0 | 0 | 1 | 0 | 0 | 1 | 0 | 0 | 0 | 0 | 0 | 0 | 4 | 11 |
| | Hunyuan | 1 | 1 | 0 | 1 | 1 | 0 | 0 | 0 | 1 | 0 | 0 | 0 | 0 | 0 | 0 | 5 | 10 |
| | GLM-4 | 1 | 1 | 1 | 1 | 1 | 1 | 0 | 0 | 1 | 0 | 0 | 0 | 0 | 0 | 0 | 7 | 8 |

Table 34: **Counting Triangles and Chairs with Grid Prior Results.**

| Objects | Model | 1 | 2 | 3 | 4 | 5 | 6 | 7 | 8 | 9 | 10 | 11 | 12 | 13 | 14 | 15 | Correct | Wrong |
|---|---|---|---|---|---|---|---|---|---|---|---|---|---|---|---|---|---|---|
| Flowers | Recraft V3 | 0 | 0 | 0 | 1 | 0 | 0 | 0 | 0 | 1 | 0 | 0 | 0 | 0 | 0 | 0 | 2 | 15 |
| | Imagen-3 | 1 | 0 | 1 | 0 | 0 | 1 | 0 | 1 | 1 | 0 | 1 | 0 | 0 | 0 | 0 | 6 | 9 |
| | Dall·E 3 | 1 | 0 | 1 | 0 | 1 | 0 | 0 | 0 | 0 | 0 | 0 | 0 | 0 | 0 | 0 | 3 | 12 |
| | Grok 3 | 1 | 0 | 1 | 0 | 1 | 0 | 0 | 0 | 0 | 0 | 0 | 0 | 0 | 0 | 0 | 3 | 12 |
| | Gemini 2.0 Flash | 1 | 0 | 0 | 1 | 0 | 0 | 0 | 0 | 1 | 0 | 0 | 0 | 0 | 0 | 0 | 3 | 12 |
| | FLUX 1.1 | 1 | 1 | 1 | 0 | 0 | 0 | 0 | 0 | 0 | 0 | 0 | 0 | 0 | 0 | 0 | 3 | 12 |
| | Firefly 3 | 1 | 0 | 0 | 0 | 0 | 0 | 0 | 0 | 0 | 0 | 0 | 0 | 0 | 0 | 0 | 1 | 14 |
| | SD 3.5 | 1 | 1 | 1 | 1 | 1 | 1 | 0 | 0 | 0 | 0 | 0 | 0 | 0 | 0 | 0 | 6 | 9 |
| | Doubao | 1 | 1 | 1 | 0 | 0 | 0 | 0 | 0 | 0 | 0 | 0 | 0 | 0 | 0 | 1 | 4 | 11 |
| | Qwen2.5-Max | 1 | 1 | 0 | 1 | 0 | 0 | 0 | 0 | 0 | 0 | 0 | 0 | 0 | 0 | 0 | 3 | 12 |
| | WanX2.1 | 1 | 1 | 0 | 1 | 1 | 1 | 0 | 1 | 1 | 0 | 0 | 0 | 1 | 0 | 0 | 8 | 7 |
| | Kling | 1 | 1 | 0 | 0 | 0 | 0 | 0 | 0 | 0 | 0 | 0 | 0 | 0 | 0 | 0 | 2 | 13 |
| | Star-3 Alpha | 1 | 0 | 1 | 0 | 0 | 0 | 1 | 0 | 0 | 0 | 0 | 0 | 0 | 0 | 0 | 3 | 12 |
| | Hunyuan | 0 | 0 | 1 | 0 | 0 | 0 | 0 | 0 | 0 | 0 | 0 | 0 | 0 | 0 | 0 | 1 | 14 |
| | GLM-4 | 1 | 1 | 1 | 0 | 1 | 1 | 0 | 0 | 0 | 0 | 0 | 0 | 1 | 0 | 0 | 6 | 9 |

Table 35: **Counting Flowers with Grid Prior Results.**

| Objects | Model | 1 | 2 | 3 | 4 | 5 | 6 | 7 | 8 | 9 | 10 | 11 | 12 | 13 | 14 | 15 | Correct | Wrong |
|---|---|---|---|---|---|---|---|---|---|---|---|---|---|---|---|---|---|---|
| Apples | Recraft V3 | 0 | 1 | 0 | 1 | 0 | 0 | 0 | 0 | 0 | 0 | 0 | 0 | 0 | 0 | 0 | 2 | 13 |
| | Imagen-3 | 1 | 1 | 1 | 1 | 0 | 1 | 0 | 0 | 0 | 1 | 0 | 0 | 0 | 0 | 0 | 6 | 9 |
| | Dall·E 3 | 1 | 1 | 0 | 0 | 0 | 0 | 0 | 0 | 0 | 0 | 0 | 0 | 0 | 0 | 0 | 2 | 13 |
| | Grok 3 | 0 | 1 | 0 | 0 | 0 | 0 | 1 | 0 | 0 | 1 | 0 | 0 | 0 | 0 | 0 | 3 | 12 |
| | Gemini 2.0 Flash | 1 | 1 | 1 | 1 | 1 | 1 | 1 | 0 | 1 | 0 | 0 | 0 | 0 | 0 | 1 | 9 | 6 |
| | FLUX 1.1 | 0 | 1 | 0 | 0 | 0 | 0 | 0 | 0 | 0 | 0 | 0 | 0 | 0 | 0 | 0 | 1 | 14 |
| | Firefly 3 | 0 | 1 | 1 | 1 | 1 | 0 | 0 | 0 | 0 | 0 | 0 | 0 | 0 | 0 | 0 | 4 | 11 |
| | SD 3.5 | 0 | 1 | 0 | 0 | 0 | 0 | 0 | 0 | 0 | 0 | 0 | 0 | 0 | 0 | 0 | 1 | 14 |
| | Doubao | 0 | 1 | 0 | 0 | 0 | 0 | 0 | 0 | 0 | 0 | 0 | 0 | 0 | 0 | 0 | 1 | 14 |
| | Qwen2.5-Max | 0 | 1 | 1 | 0 | 1 | 1 | 0 | 0 | 0 | 0 | 0 | 1 | 1 | 0 | 0 | 6 | 9 |
| | WanX2.1 | 0 | 1 | 1 | 1 | 0 | 0 | 1 | 1 | 1 | 1 | 0 | 0 | 0 | 0 | 0 | 7 | 8 |
| | Kling | 0 | 1 | 0 | 0 | 0 | 0 | 0 | 0 | 0 | 0 | 0 | 0 | 0 | 0 | 0 | 1 | 14 |
| | Star-3 Alpha | 1 | 1 | 1 | 0 | 0 | 0 | 0 | 1 | 1 | 0 | 0 | 0 | 0 | 0 | 0 | 5 | 10 |
| | Hunyuan | 0 | 1 | 1 | 1 | 1 | 0 | 0 | 0 | 0 | 0 | 0 | 0 | 0 | 0 | 0 | 4 | 11 |
| | GLM-4 | 0 | 1 | 1 | 1 | 0 | 0 | 0 | 1 | 0 | 0 | 0 | 1 | 1 | 0 | 0 | 6 | 9 |
| Watermelons | Recraft V3 | 0 | 1 | 0 | 0 | 0 | 0 | 0 | 0 | 1 | 0 | 0 | 0 | 0 | 0 | 0 | 2 | 13 |
| | Imagen-3 | 0 | 1 | 1 | 0 | 0 | 0 | 1 | 0 | 0 | 0 | 0 | 0 | 0 | 0 | 0 | 3 | 12 |
| | Dall·E 3 | 0 | 0 | 0 | 1 | 0 | 0 | 0 | 0 | 0 | 0 | 0 | 1 | 0 | 0 | 0 | 2 | 13 |
| | Grok 3 | 0 | 0 | 1 | 0 | 0 | 0 | 1 | 1 | 0 | 0 | 0 | 0 | 0 | 0 | 0 | 3 | 12 |
| | Gemini 2.0 Flash | 1 | 1 | 1 | 0 | 1 | 0 | 0 | 0 | 0 | 0 | 0 | 0 | 0 | 0 | 0 | 4 | 11 |
| | FLUX 1.1 | 0 | 1 | 0 | 1 | 1 | 0 | 0 | 0 | 0 | 0 | 0 | 0 | 0 | 0 | 0 | 3 | 12 |
| | Firefly 3 | 0 | 1 | 0 | 0 | 0 | 0 | 0 | 0 | 0 | 0 | 0 | 0 | 0 | 0 | 0 | 1 | 14 |
| | SD 3.5 | 0 | 0 | 1 | 1 | 1 | 0 | 0 | 0 | 0 | 0 | 0 | 0 | 0 | 0 | 0 | 3 | 12 |
| | Doubao | 0 | 0 | 1 | 0 | 0 | 0 | 0 | 0 | 0 | 0 | 0 | 0 | 0 | 0 | 0 | 1 | 12 |
| | Qwen2.5-Max | 0 | 0 | 0 | 0 | 0 | 0 | 0 | 0 | 1 | 0 | 0 | 0 | 0 | 0 | 0 | 1 | 14 |
| | WanX2.1 | 1 | 0 | 0 | 0 | 1 | 0 | 0 | 0 | 0 | 0 | 0 | 0 | 0 | 0 | 0 | 2 | 13 |
| | Kling | 0 | 1 | 0 | 0 | 0 | 0 | 0 | 0 | 0 | 0 | 0 | 0 | 0 | 0 | 0 | 1 | 14 |
| | Star-3 Alpha | 0 | 0 | 1 | 0 | 0 | 0 | 0 | 0 | 0 | 0 | 0 | 0 | 0 | 0 | 0 | 1 | 14 |
| | Hunyuan | 0 | 1 | 1 | 0 | 1 | 0 | 0 | 0 | 0 | 0 | 0 | 0 | 0 | 1 | 0 | 4 | 11 |
| | GLM-4 | 0 | 1 | 1 | 0 | 0 | 0 | 0 | 0 | 0 | 0 | 0 | 0 | 0 | 1 | 1 | 4 | 11 |

Table 36: **Counting Apples and Watermelons with Position Guidance Results.**

| Objects | Model | 1 | 2 | 3 | 4 | 5 | 6 | 7 | 8 | 9 | 10 | 11 | 12 | 13 | 14 | 15 | Correct | Wrong |
|---|---|---|---|---|---|---|---|---|---|---|---|---|---|---|---|---|---|---|
| Humans | Recraft V3 | 1 | 1 | 1 | 0 | 1 | 0 | 0 | 0 | 0 | 0 | 0 | 0 | 0 | 0 | 0 | 4 | 11 |
| | Imagen-3 | 1 | 1 | 0 | 0 | 0 | 0 | 0 | 0 | 0 | 0 | 0 | 0 | 0 | 0 | 0 | 2 | 13 |
| | Dall·E 3 | 1 | 1 | 0 | 0 | 0 | 0 | 0 | 0 | 0 | 0 | 0 | 0 | 0 | 0 | 0 | 2 | 13 |
| | Grok 3 | 0 | 1 | 1 | 0 | 0 | 0 | 0 | 0 | 0 | 0 | 0 | 0 | 0 | 0 | 0 | 2 | 12 |
| | Gemini 2.0 Flash | 1 | 1 | 1 | 1 | 1 | 1 | 1 | 0 | 0 | 0 | 0 | 0 | 0 | 0 | 0 | 7 | 8 |
| | FLUX 1.1 | 0 | 1 | 0 | 0 | 0 | 0 | 0 | 0 | 0 | 0 | 0 | 0 | 0 | 0 | 0 | 1 | 14 |
| | Firefly 3 | 1 | 1 | 1 | 0 | 0 | 0 | 0 | 0 | 0 | 0 | 0 | 0 | 0 | 1 | 0 | 4 | 11 |
| | SD 3.5 | 0 | 1 | 0 | 0 | 0 | 0 | 0 | 0 | 0 | 0 | 0 | 0 | 0 | 0 | 0 | 1 | 14 |
| | Doubao | 0 | 0 | 1 | 1 | 0 | 0 | 0 | 0 | 0 | 0 | 0 | 0 | 0 | 0 | 0 | 2 | 13 |
| | Qwen2.5-Max | 0 | 1 | 0 | 1 | 0 | 0 | 0 | 0 | 0 | 0 | 0 | 0 | 0 | 0 | 0 | 2 | 13 |
| | WanX2.1 | 1 | 1 | 1 | 1 | 0 | 0 | 0 | 0 | 0 | 0 | 1 | 0 | 0 | 0 | 0 | 5 | 10 |
| | Kling | 0 | 1 | 0 | 0 | 0 | 0 | 0 | 0 | 0 | 0 | 0 | 0 | 0 | 0 | 0 | 1 | 14 |
| | Star-3 Alpha | 1 | 1 | 1 | 1 | 1 | 0 | 0 | 0 | 1 | 0 | 0 | 0 | 0 | 0 | 0 | 6 | 9 |
| | Hunyuan | 1 | 0 | 1 | 0 | 1 | 0 | 1 | 1 | 0 | 1 | 0 | 0 | 0 | 0 | 0 | 6 | 9 |
| | GLM-4 | 1 | 1 | 1 | 0 | 0 | 0 | 0 | 0 | 0 | 0 | 0 | 1 | 0 | 0 | 0 | 4 | 11 |
| Cats | Recraft V3 | 0 | 1 | 0 | 0 | 0 | 0 | 0 | 0 | 0 | 0 | 0 | 0 | 0 | 0 | 0 | 1 | 14 |
| | Imagen-3 | 1 | 1 | 0 | 1 | 1 | 0 | 0 | 0 | 0 | 0 | 0 | 0 | 0 | 0 | 0 | 4 | 11 |
| | Dall·E 3 | 0 | 1 | 1 | 0 | 0 | 0 | 0 | 0 | 0 | 0 | 0 | 0 | 0 | 0 | 0 | 2 | 13 |
| | Grok 3 | 1 | 1 | 1 | 0 | 1 | 1 | 1 | 0 | 0 | 0 | 0 | 0 | 0 | 0 | 0 | 6 | 9 |
| | Gemini 2.0 Flash | 1 | 1 | 1 | 0 | 1 | 1 | 1 | 0 | 0 | 0 | 1 | 0 | 0 | 0 | 0 | 7 | 8 |
| | FLUX 1.1 | 0 | 1 | 0 | 0 | 0 | 0 | 0 | 1 | 1 | 0 | 0 | 0 | 0 | 0 | 0 | 3 | 12 |
| | Firefly 3 | 0 | 1 | 1 | 0 | 0 | 0 | 0 | 0 | 0 | 0 | 0 | 0 | 0 | 0 | 0 | 2 | 13 |
| | SD 3.5 | 0 | 1 | 0 | 0 | 0 | 0 | 0 | 0 | 0 | 0 | 0 | 0 | 0 | 0 | 0 | 1 | 14 |
| | Doubao | 0 | 1 | 0 | 0 | 1 | 0 | 0 | 0 | 0 | 0 | 0 | 0 | 0 | 0 | 0 | 2 | 13 |
| | Qwen2.5-Max | 1 | 1 | 0 | 0 | 0 | 0 | 0 | 0 | 0 | 0 | 0 | 0 | 0 | 0 | 0 | 2 | 13 |
| | WanX2.1 | 1 | 1 | 0 | 1 | 0 | 1 | 0 | 0 | 1 | 0 | 0 | 0 | 0 | 0 | 1 | 6 | 9 |
| | Kling | 0 | 1 | 0 | 0 | 0 | 0 | 0 | 0 | 0 | 0 | 0 | 0 | 0 | 0 | 0 | 1 | 14 |
| | Star-3 Alpha | 1 | 1 | 0 | 0 | 0 | 0 | 1 | 0 | 0 | 0 | 0 | 0 | 0 | 1 | 0 | 4 | 11 |
| | Hunyuan | 0 | 1 | 0 | 1 | 1 | 0 | 0 | 0 | 0 | 0 | 0 | 0 | 1 | 1 | 0 | 5 | 10 |
| | GLM-4 | 0 | 1 | 1 | 0 | 1 | 0 | 0 | 0 | 0 | 0 | 0 | 0 | 0 | 0 | 0 | 3 | 12 |

Table 37: **Counting Humans and Cats with Position Guidance Results.**

| Objects | Model | 1 | 2 | 3 | 4 | 5 | 6 | 7 | 8 | 9 | 10 | 11 | 12 | 13 | 14 | 15 | Correct | Wrong |
|---|---|---|---|---|---|---|---|---|---|---|---|---|---|---|---|---|---|---|
| Triangles | Recraft V3 | 0 | 1 | 0 | 0 | 0 | 0 | 1 | 1 | 0 | 1 | 0 | 0 | 0 | 0 | 0 | 4 | 11 |
| | Imagen-3 | 0 | 1 | 1 | 1 | 0 | 0 | 0 | 0 | 0 | 0 | 0 | 0 | 0 | 0 | 0 | 3 | 12 |
| | Dall·E 3 | 1 | 1 | 1 | 0 | 1 | 0 | 0 | 0 | 0 | 0 | 0 | 0 | 0 | 0 | 0 | 4 | 11 |
| | Grok 3 | 1 | 1 | 1 | 0 | 0 | 0 | 0 | 0 | 0 | 0 | 0 | 0 | 0 | 0 | 0 | 3 | 12 |
| | Gemini 2.0 Flash | 0 | 1 | 1 | 0 | 0 | 1 | 0 | 0 | 0 | 0 | 0 | 0 | 0 | 0 | 0 | 3 | 12 |
| | FLUX 1.1 | 0 | 0 | 0 | 0 | 0 | 0 | 0 | 0 | 0 | 0 | 1 | 0 | 0 | 0 | 1 | 2 | 13 |
| | Firefly 3 | 0 | 0 | 0 | 0 | 0 | 1 | 0 | 0 | 0 | 0 | 0 | 0 | 0 | 0 | 0 | 1 | 14 |
| | SD 3.5 | 0 | 0 | 0 | 1 | 0 | 0 | 0 | 0 | 0 | 0 | 0 | 0 | 0 | 0 | 0 | 1 | 14 |
| | Doubao | 0 | 0 | 0 | 0 | 0 | 0 | 0 | 0 | 0 | 0 | 0 | 0 | 0 | 0 | 0 | 0 | 15 |
| | Qwen2.5-Max | 0 | 0 | 0 | 0 | 1 | 0 | 0 | 1 | 0 | 0 | 0 | 0 | 0 | 0 | 0 | 2 | 13 |
| | WanX2.1 | 0 | 1 | 1 | 1 | 0 | 0 | 0 | 0 | 0 | 0 | 0 | 0 | 0 | 0 | 0 | 3 | 12 |
| | Kling | 0 | 0 | 0 | 0 | 0 | 0 | 0 | 0 | 0 | 0 | 0 | 0 | 0 | 0 | 0 | 0 | 15 |
| | Star-3 Alpha | 0 | 0 | 0 | 0 | 1 | 1 | 1 | 0 | 0 | 0 | 0 | 0 | 0 | 0 | 0 | 3 | 12 |
| | Hunyuan | 0 | 0 | 0 | 0 | 0 | 0 | 0 | 0 | 0 | 0 | 0 | 1 | 0 | 0 | 0 | 1 | 14 |
| | GLM-4 | 0 | 0 | 0 | 1 | 0 | 0 | 0 | 0 | 0 | 1 | 1 | 1 | 0 | 1 | 0 | 5 | 10 |
| Chairs | Recraft V3 | 0 | 1 | 0 | 0 | 0 | 0 | 0 | 1 | 0 | 0 | 0 | 0 | 0 | 0 | 0 | 2 | 13 |
| | Imagen-3 | 0 | 0 | 0 | 1 | 0 | 0 | 0 | 0 | 0 | 1 | 0 | 0 | 0 | 0 | 0 | 2 | 13 |
| | Dall·E 3 | 0 | 1 | 0 | 0 | 0 | 0 | 0 | 0 | 0 | 0 | 0 | 0 | 0 | 0 | 0 | 1 | 14 |
| | Grok 3 | 1 | 1 | 1 | 0 | 0 | 0 | 1 | 0 | 0 | 0 | 0 | 0 | 0 | 0 | 0 | 4 | 11 |
| | Gemini 2.0 Flash | 0 | 1 | 1 | 1 | 0 | 0 | 0 | 1 | 0 | 0 | 0 | 0 | 0 | 0 | 0 | 4 | 11 |
| | FLUX 1.1 | 0 | 1 | 0 | 0 | 0 | 0 | 0 | 0 | 0 | 0 | 0 | 0 | 0 | 0 | 0 | 1 | 14 |
| | Firefly 3 | 1 | 1 | 1 | 0 | 0 | 0 | 0 | 0 | 0 | 0 | 0 | 0 | 0 | 0 | 0 | 3 | 12 |
| | SD 3.5 | 0 | 1 | 0 | 0 | 0 | 0 | 0 | 0 | 0 | 0 | 0 | 0 | 0 | 0 | 0 | 1 | 14 |
| | Doubao | 0 | 1 | 0 | 0 | 0 | 0 | 0 | 0 | 0 | 0 | 0 | 0 | 0 | 0 | 0 | 1 | 14 |
| | Qwen2.5-Max | 1 | 1 | 0 | 1 | 0 | 0 | 0 | 1 | 0 | 0 | 0 | 0 | 0 | 1 | 0 | 5 | 10 |
| | WanX2.1 | 1 | 1 | 0 | 1 | 0 | 1 | 0 | 1 | 0 | 0 | 0 | 0 | 0 | 0 | 1 | 6 | 9 |
| | Kling | 1 | 0 | 0 | 0 | 0 | 0 | 0 | 0 | 0 | 0 | 0 | 0 | 0 | 0 | 0 | 1 | 14 |
| | Star-3 Alpha | 1 | 1 | 0 | 1 | 0 | 0 | 0 | 0 | 0 | 0 | 0 | 0 | 0 | 0 | 0 | 3 | 12 |
| | Hunyuan | 1 | 1 | 1 | 1 | 1 | 1 | 0 | 0 | 0 | 0 | 0 | 0 | 0 | 0 | 0 | 6 | 9 |
| | GLM-4 | 0 | 1 | 1 | 1 | 0 | 0 | 0 | 0 | 0 | 0 | 0 | 0 | 0 | 0 | 0 | 3 | 12 |

Table 38: **Counting Triangles and Chairs with Position Guidance Results.**

| Objects | Model | 1 | 2 | 3 | 4 | 5 | 6 | 7 | 8 | 9 | 10 | 11 | 12 | 13 | 14 | 15 | Correct | Wrong |
|---|---|---|---|---|---|---|---|---|---|---|---|---|---|---|---|---|---|---|
| Flowers | Recraft V3 | 1 | 0 | 0 | 0 | 0 | 0 | 0 | 0 | 0 | 0 | 0 | 0 | 0 | 0 | 0 | 1 | 14 |
| | Imagen-3 | 1 | 1 | 1 | 0 | 1 | 0 | 0 | 0 | 0 | 0 | 1 | 0 | 0 | 1 | 1 | 7 | 8 |
| | Dall·E 3 | 1 | 1 | 1 | 1 | 0 | 0 | 0 | 0 | 0 | 0 | 0 | 0 | 0 | 0 | 0 | 4 | 11 |
| | Grok 3 | 0 | 1 | 0 | 0 | 0 | 1 | 0 | 0 | 0 | 0 | 0 | 0 | 0 | 0 | 0 | 2 | 13 |
| | Gemini 2.0 Flash | 1 | 0 | 0 | 0 | 0 | 0 | 0 | 0 | 0 | 0 | 0 | 0 | 0 | 0 | 0 | 1 | 14 |
| | FLUX 1.1 | 0 | 0 | 1 | 0 | 0 | 1 | 0 | 1 | 0 | 0 | 0 | 0 | 0 | 0 | 0 | 3 | 12 |
| | Firefly 3 | 1 | 1 | 1 | 0 | 0 | 0 | 0 | 0 | 0 | 0 | 0 | 0 | 0 | 0 | 0 | 3 | 12 |
| | SD 3.5 | 0 | 0 | 1 | 1 | 1 | 0 | 0 | 0 | 0 | 0 | 0 | 0 | 1 | 1 | 1 | 6 | 9 |
| | Doubao | 0 | 1 | 0 | 0 | 0 | 0 | 0 | 0 | 0 | 0 | 0 | 0 | 0 | 0 | 0 | 1 | 14 |
| | Qwen2.5-Max | 0 | 0 | 0 | 1 | 0 | 1 | 0 | 0 | 0 | 0 | 0 | 0 | 0 | 0 | 0 | 2 | 13 |
| | WanX2.1 | 1 | 0 | 1 | 1 | 0 | 0 | 0 | 0 | 0 | 0 | 0 | 0 | 0 | 0 | 0 | 3 | 12 |
| | Kling | 1 | 0 | 0 | 1 | 0 | 0 | 0 | 0 | 0 | 0 | 0 | 0 | 0 | 0 | 0 | 2 | 13 |
| | Star-3 Alpha | 0 | 0 | 1 | 0 | 0 | 0 | 0 | 0 | 0 | 0 | 0 | 0 | 0 | 0 | 0 | 1 | 14 |
| | Hunyuan | 0 | 0 | 0 | 0 | 1 | 0 | 0 | 0 | 0 | 0 | 0 | 0 | 0 | 0 | 0 | 1 | 14 |
| | GLM-4 | 1 | 1 | 0 | 0 | 0 | 0 | 1 | 0 | 1 | 0 | 0 | 0 | 0 | 0 | 0 | 4 | 11 |

Table 39: **Counting Flowers with Position Guidance Results.**

## G  IMAGE EXAMPLES

In this section, we present all the generated images used in our benchmarking results and prompt refinement experiments in Figures 12–74.

**N = 1~15**

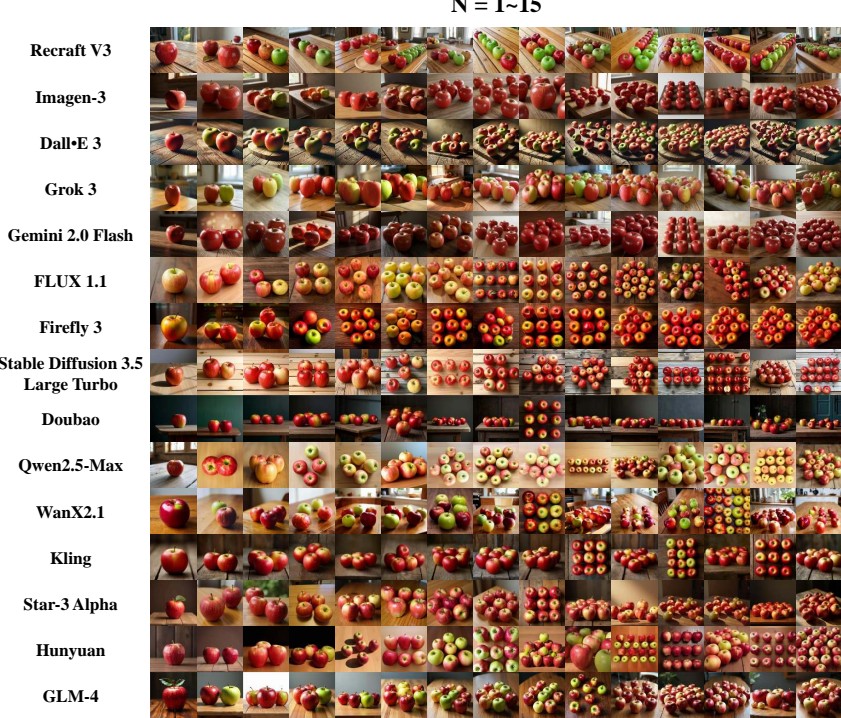

Figure 12: **Counting Apples in Home Scene Results on 15 Models**. This figure presents the generation results of counting apples. We use the Prompt:"$N$ apples on a wooden table.", where $N \in [1, 15]$ denotes the number of objects expected to be generated.

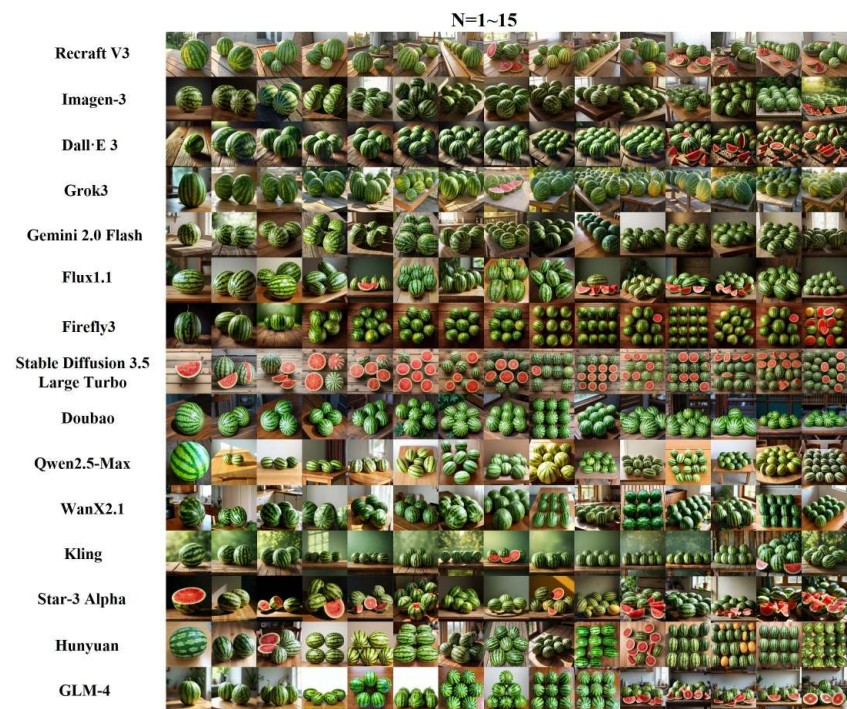

Figure 13: **Counting Watermelons in Home Scene Results on 15 Models**. This figure presents the generation results of counting watermelons. We use the Prompt:"$N$ watermelons on a wooden table.", where $N \in [1, 15]$ denotes the number of objects expected to be generated.

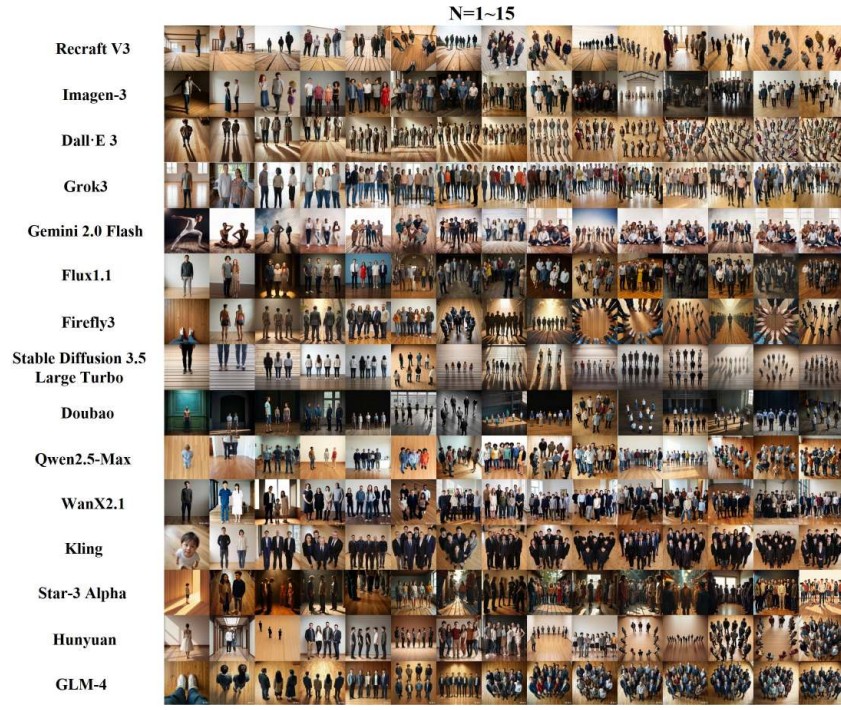

Figure 14: **Counting Humans in Home Scene Results on 15 Models**. This figure presents the generation results of counting Humans. We use the Prompt:"$N$ Humans on the wooden floor.", where $N \in [1, 15]$ denotes the number of objects expected to be generated.

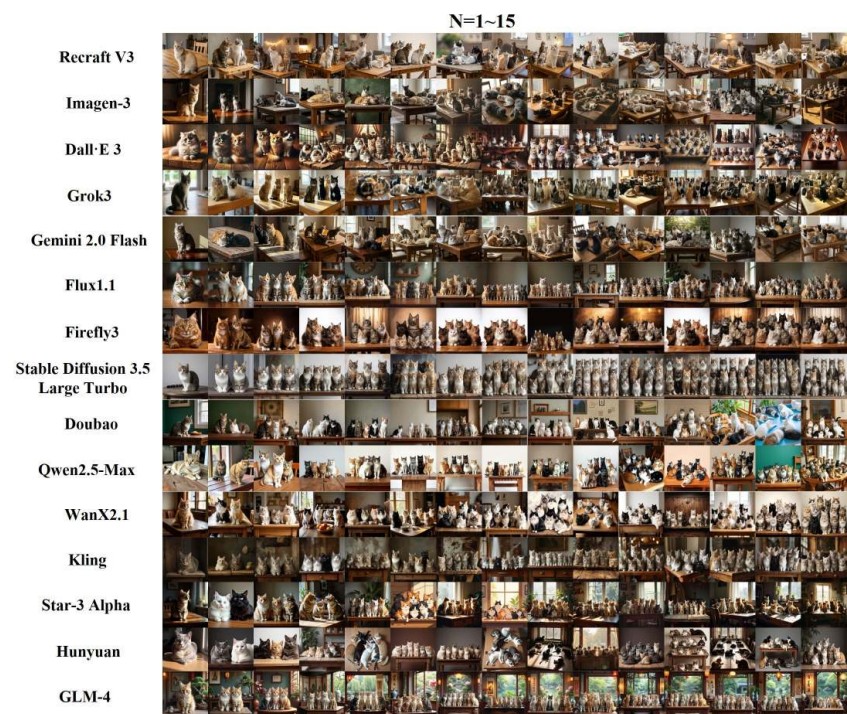

Figure 15: **Counting Cats in Home Scene Results on 15 Models**. This figure presents the generation results of counting cats. We use the Prompt:"$N$ cats on a wooden table.", where $N \in [1, 15]$ denotes the number of objects expected to be generated.

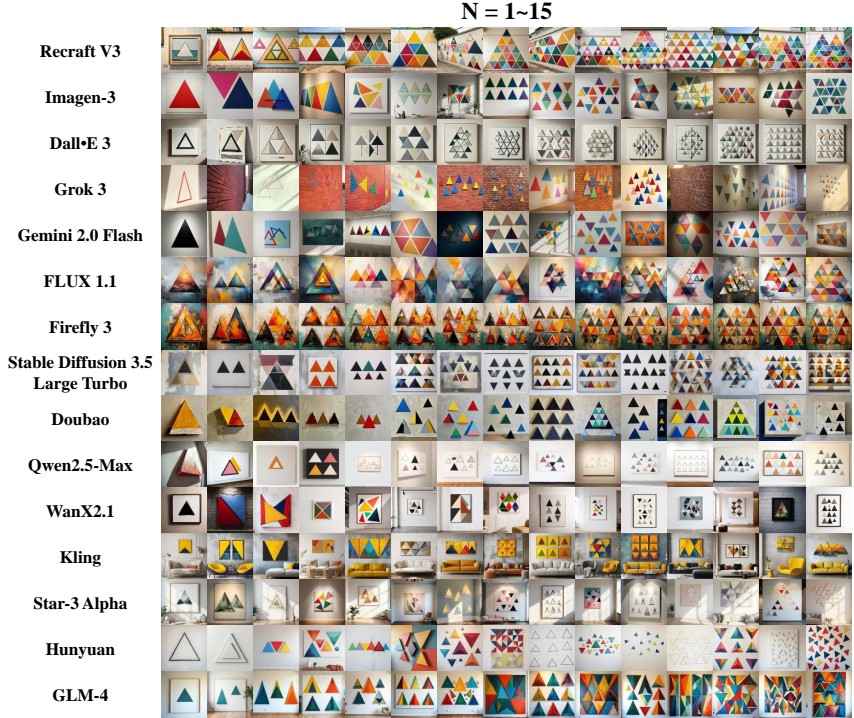

Figure 16: **Counting Triangles in Home Scene Results on 15 Models**. This figure presents the generation results of counting triangles. We use the Prompt:"$N$ triangles on a painting on a wall.", where $N \in [1, 15]$ denotes the number of objects expected to be generated.

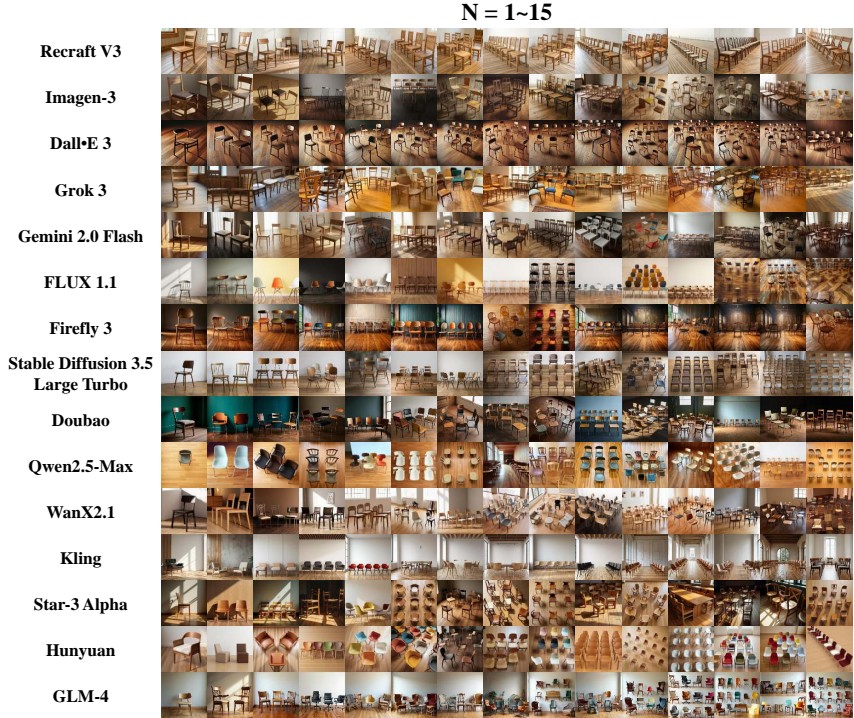

Figure 17: **Counting Chairs in Home Scene Results on 15 Models**. This figure presents the generation results of counting chairs. We use the Prompt:"$N$ chairs on the wooden floor.", where $N \in [1, 15]$ denotes the number of objects expected to be generated.

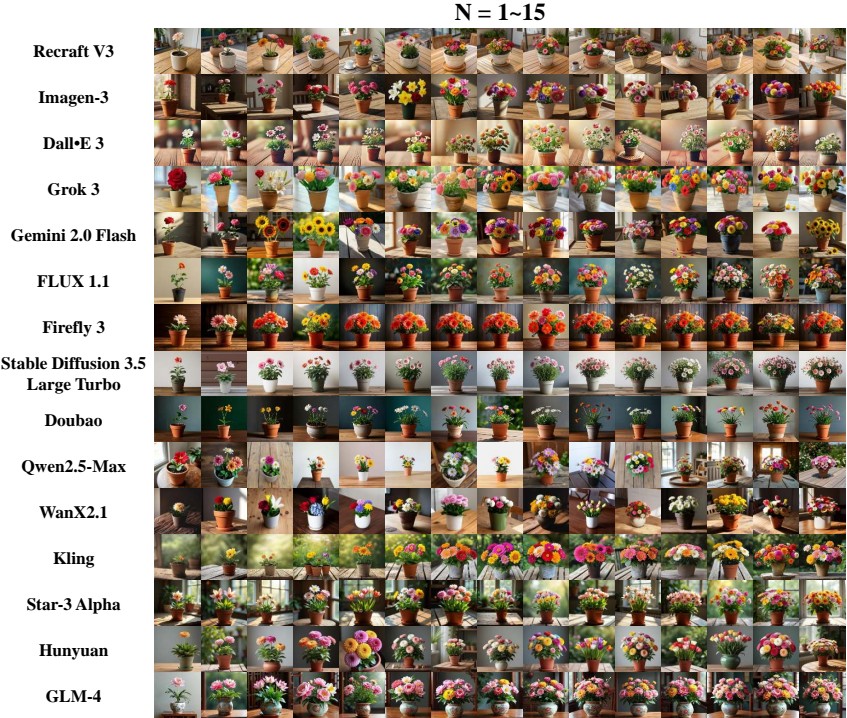

Figure 18: **Counting Flowers in Home Scene Results on 15 Models**. This figure presents the generation results of counting flowers. We use the Prompt:"A flowerpot with $N$ flowers on a wooden table.", where $N \in [1, 15]$ denotes the number of objects expected to be generated.

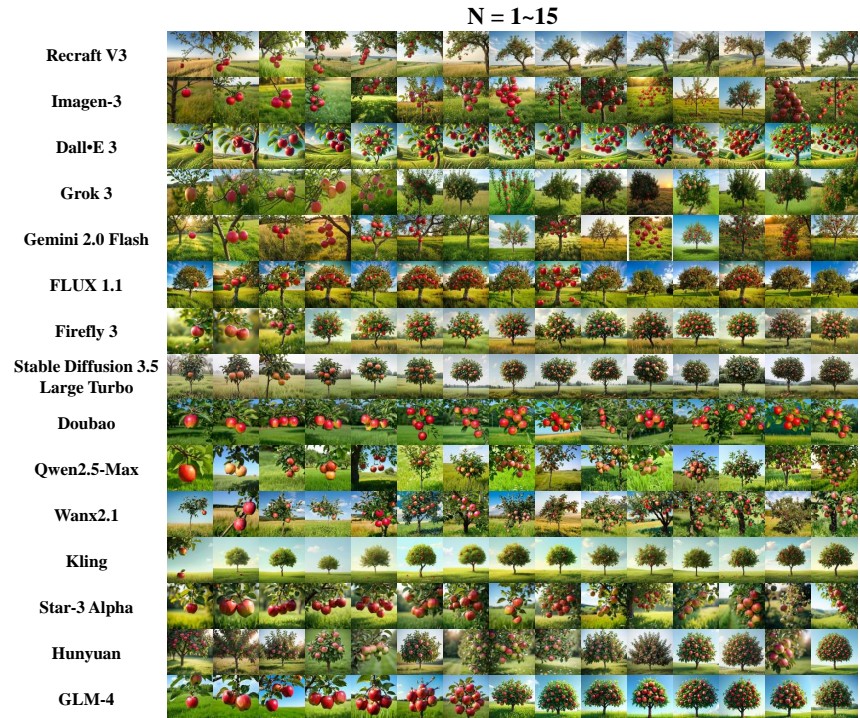

Figure 19: **Counting Apples in Nature Scene on 15 Models**. This figure presents the generation results of counting apples in a nature scene. We use the Prompt:"$N$ apples on an apple tree in a grassland", where $N \in [1, 15]$ denotes the number of objects expected to be generated.

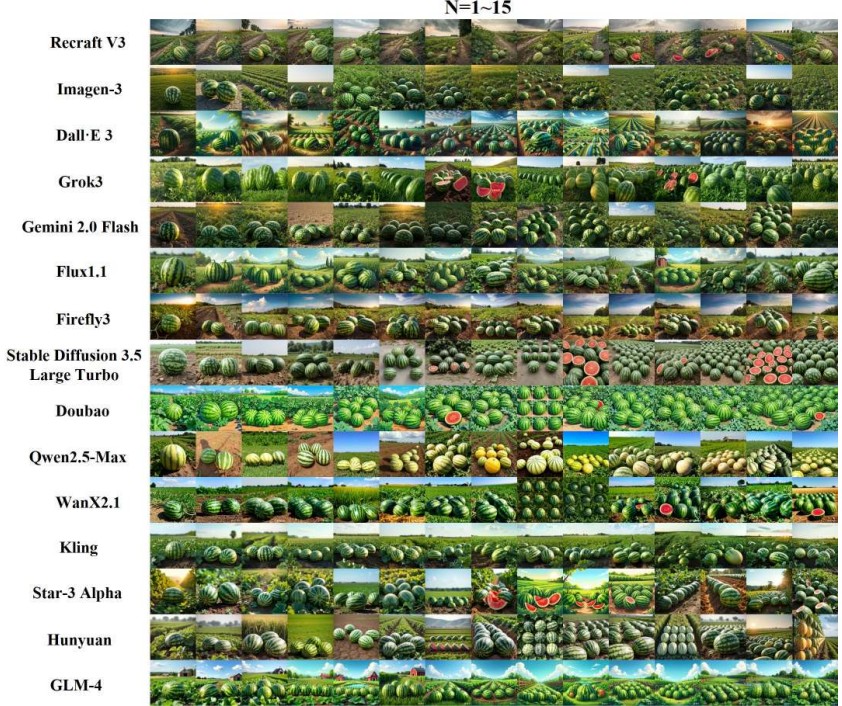

Figure 20: **Counting Watermelons in Nature Scene on 15 Models**. This figure presents the generation results of counting watermelons in a nature scene. We use the Prompt:"$N$ watermelon in a farmland", where $N \in [1, 15]$ denotes the number of objects expected to be generated.

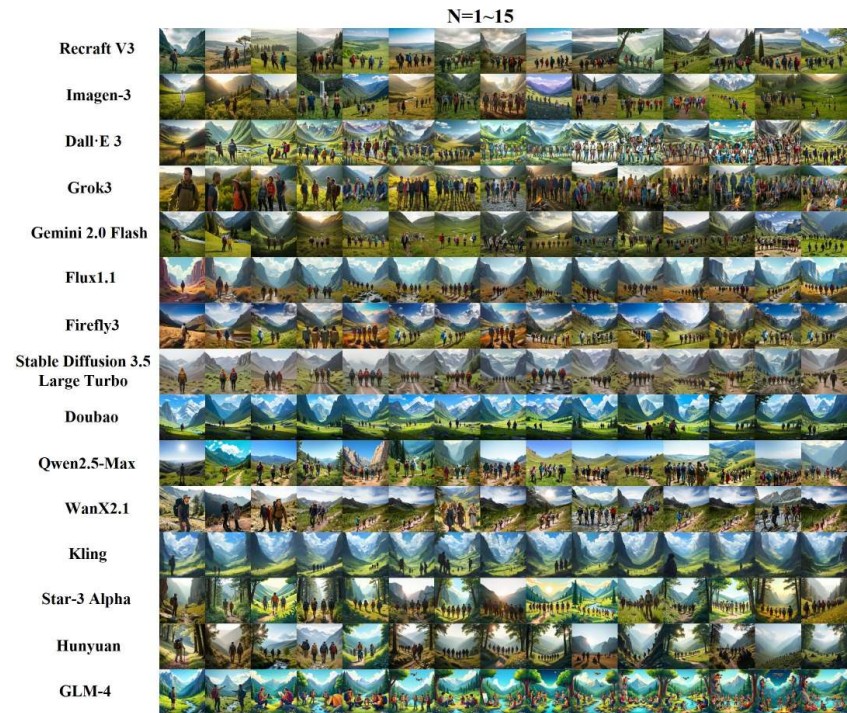

Figure 21: **Counting Humans in Nature Scene on 15 Models**. This figure presents the generation results of counting humans in a nature scene. We use the Prompt:"$N$ human travelers in a valley.", where $N \in [1, 15]$ denotes the number of objects expected to be generated.

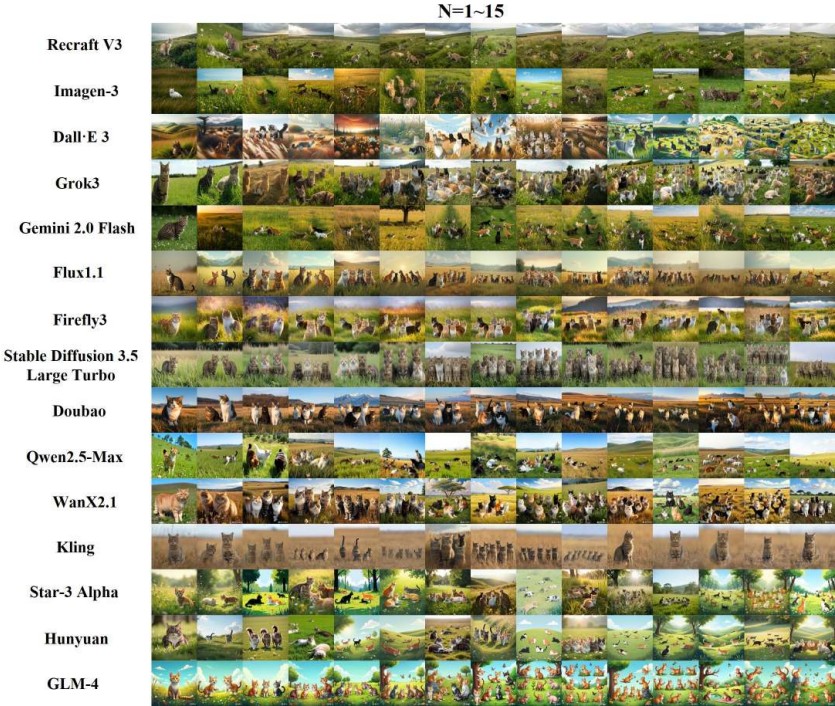

Figure 22: **Counting Cats in Nature Scene on 15 Models**. This figure presents the generation results of counting cats in a nature scene. We use the Prompt:"$N$ cats in a temperate grassland.", where $N \in [1, 15]$ denotes the number of objects expected to be generated.

**N = 1~15**

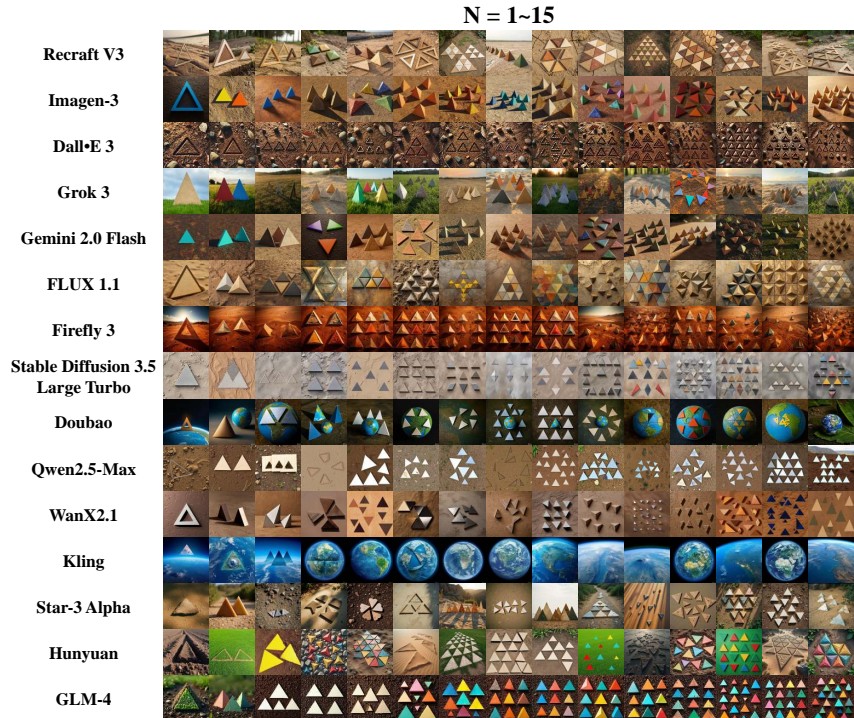

Figure 23: **Counting Triangles in Nature Scene on 15 Models**. This figure presents the generation results of counting triangles in a nature scene. We use the Prompt:"$N$ triangles on an earthy surface.", where $N \in [1, 15]$ denotes the number of objects expected to be generated.

**N = 1~15**

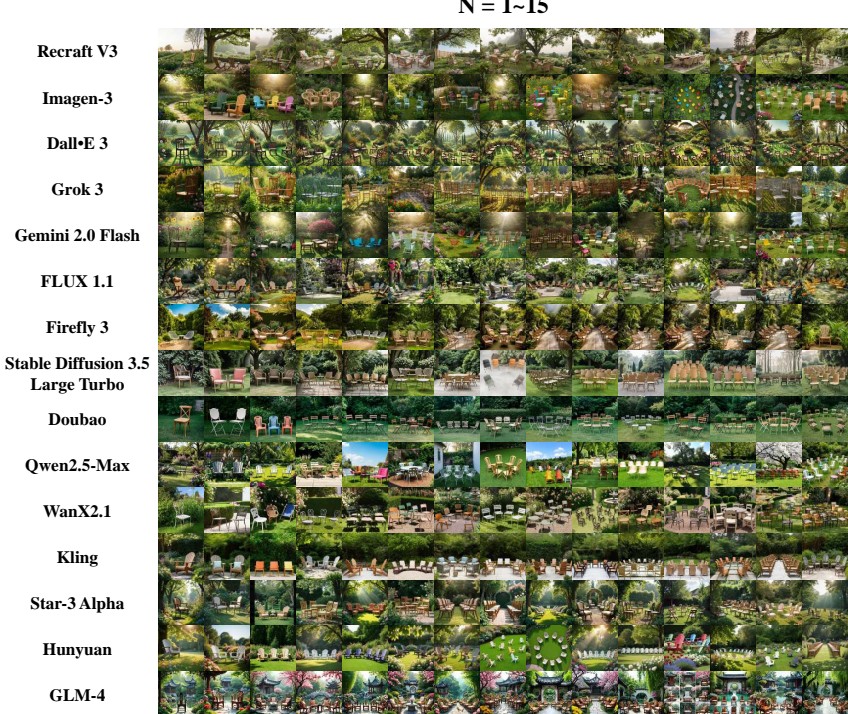

Figure 24: **Counting Chairs in Nature Scene on 15 Models**. This figure presents the generation results of counting chairs in a nature scene. We use the Prompt:"$N$ chairs in a garden.", where $N \in [1, 15]$ denotes the number of objects expected to be generated.

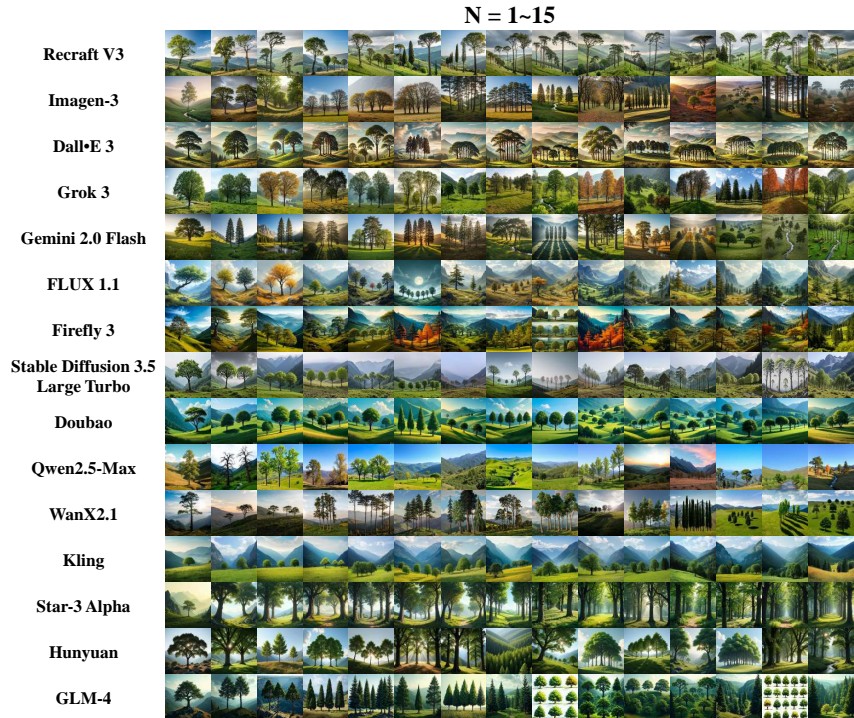

Figure 25: **Counting Trees in Nature Scene on 15 Models**. This figure presents the generation results of counting trees in a nature scene. We use the Prompt:"$N$ trees in a valley.", where $N \in [1, 15]$ denotes the number of objects expected to be generated.

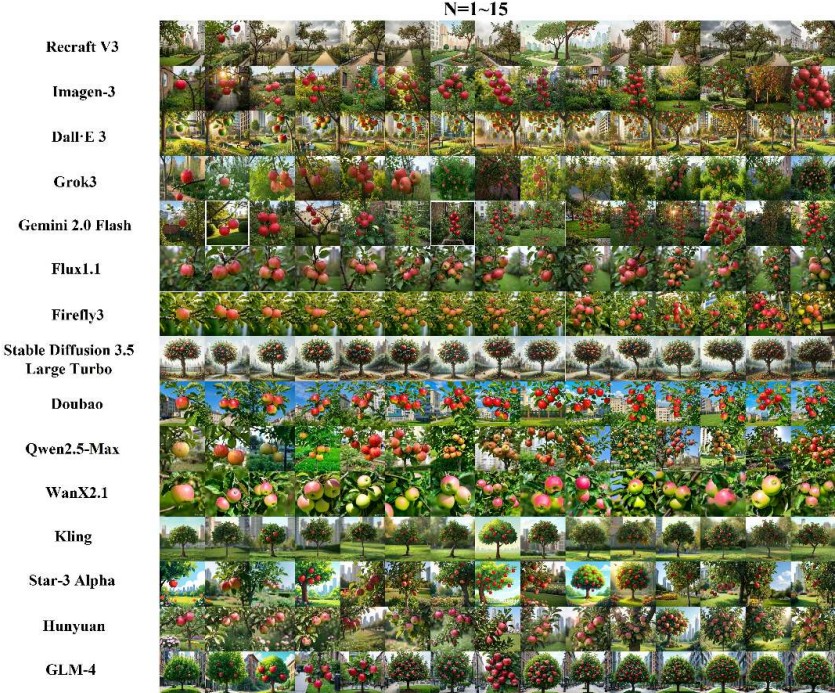

Figure 26: **Counting Apples in City Scene on 15 Models**. This figure presents the generation results of counting apples on apple trees. We use the Prompt:"$N$ apples on an apple tree in a city garden. ", where $N \in [1, 15]$ denotes the number of objects expected to be generated.

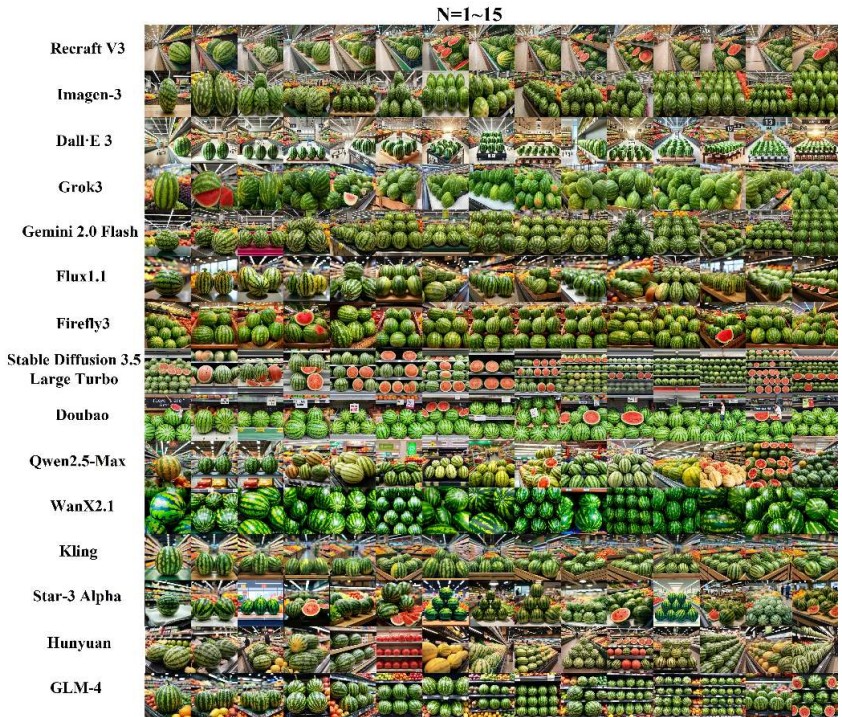

Figure 27: **Counting Watermelons in City Scene on 15 Models**. This figure presents the generation results of counting watermelons. We use the Prompt:"$N$ watermelons in a supermarket.", where $N \in [1, 15]$ denotes the number of objects expected to be generated.

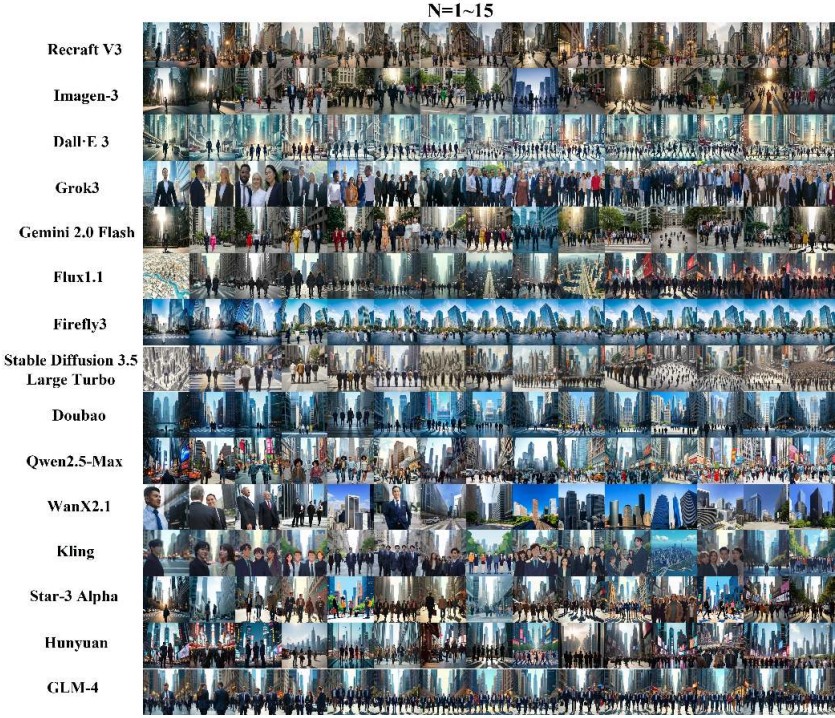

Figure 28: **Counting Humans in City Scene on 15 Models**. This figure presents the generation results of counting humans. We use the Prompt:"$N$ humans in a city central business district.", where $N \in [1, 15]$ denotes the number of objects expected to be generated.

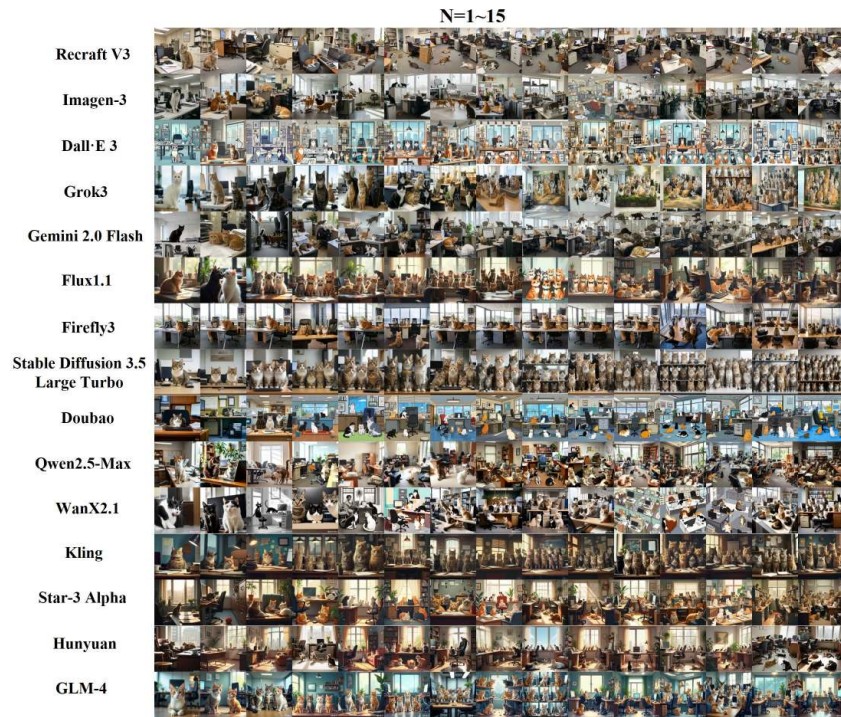

Figure 29: **Counting Cats in City Scene on 15 Models**. This figure presents the generation results of counting cats. We use the Prompt:"$N$ cats in an office.", where $N \in [1, 15]$ denotes the number of objects expected to be generated.

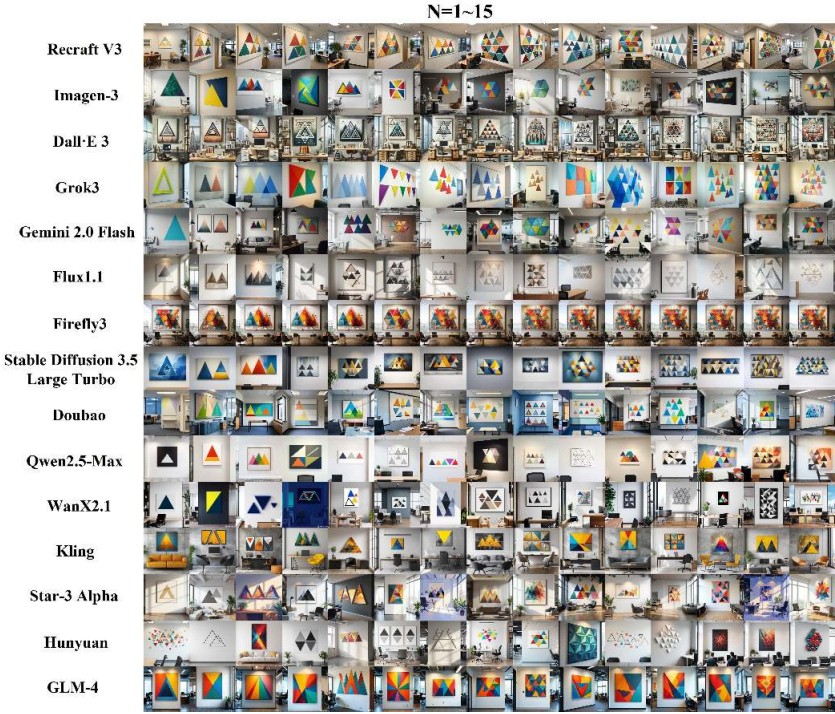

Figure 30: **Counting Triangles in City Scene on 15 Models**. This figure presents the generation results of counting triangles. We use the Prompt:"$N$ triangles in a painting on the wall of an office.", where $N \in [1, 15]$ denotes the number of objects expected to be generated.

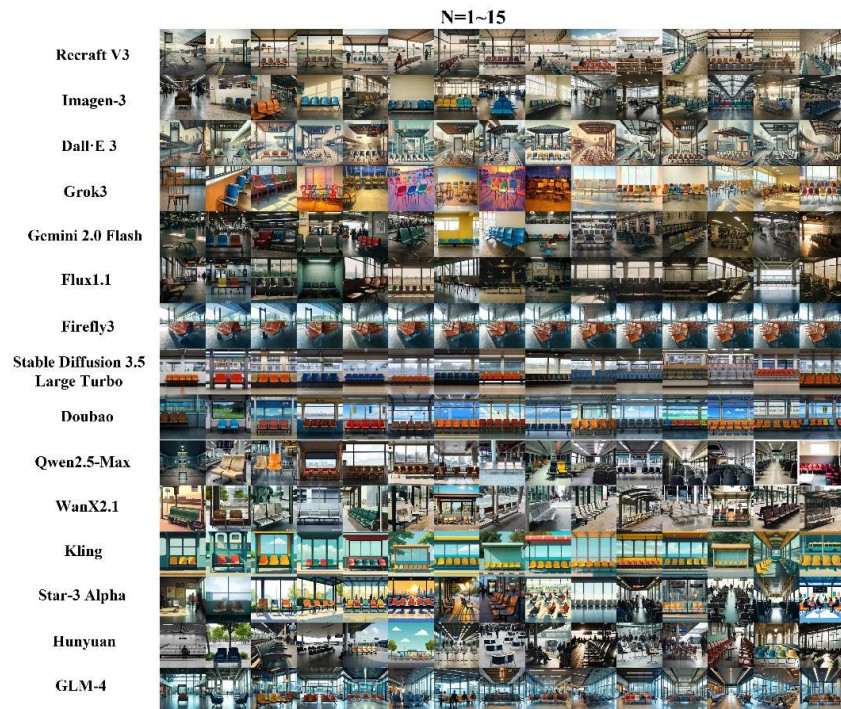

Figure 31: **Counting Chairs in City Scene on 15 Models**. This figure presents the generation results of counting chairs. We use the Prompt:"$N$ chairs in a bus station.", where $N \in [1, 15]$ denotes the number of objects expected to be generated.

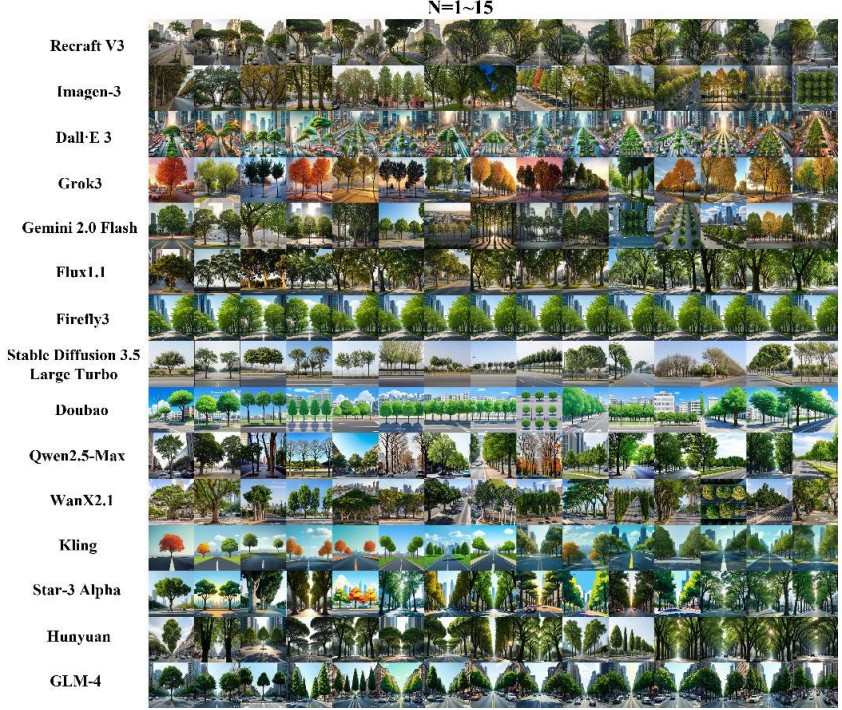

Figure 32: **Counting Trees in City Scene on 15 Models**. This figure presents the generation results of counting trees. We use the Prompt:"$N$ trees alongside a city's main road.", where $N \in [1, 15]$ denotes the number of objects expected to be generated.

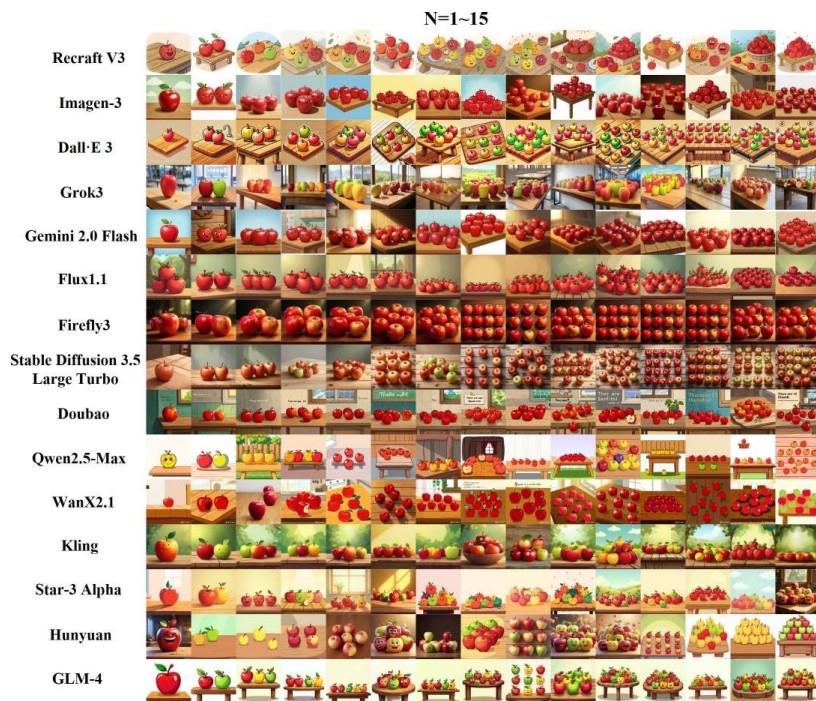

Figure 33: **Counting Apples With Cartoon Style on 15 Models**. This figure presents the generation results of counting apples on apple trees. We use the Prompt:"$N$ apple on a wooden table in a cartoon style.", where $N \in [1, 15]$ denotes the number of objects expected to be generated.

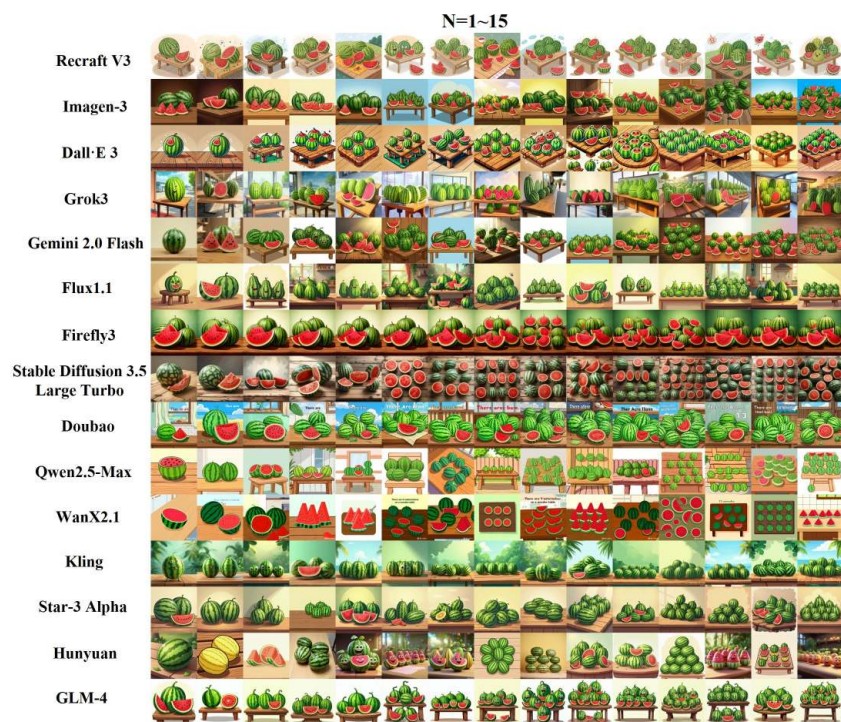

Figure 34: **Counting Watermelons With Cartoon Style on 15 Models**. This figure presents the generation results of counting watermelons. We use the Prompt:"$N$ watermelons on a wooden table in a cartoon style.", where $N \in [1, 15]$ denotes the number of objects expected to be generated.

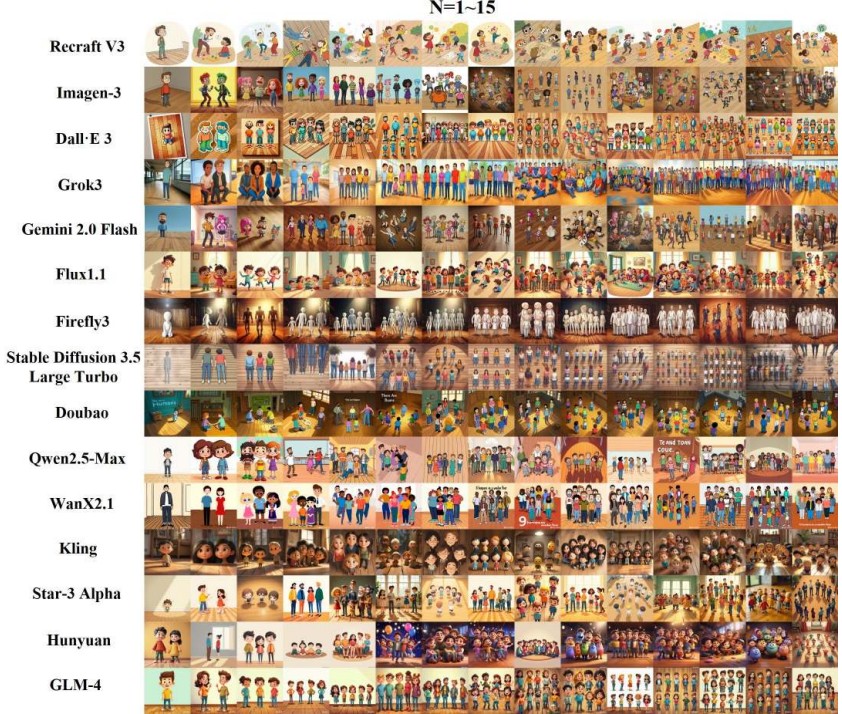

Figure 35: **Counting Humans With Cartoon Style on 15 Models**. This figure presents the generation results of counting humans. We use the Prompt:"$N$ humans on a wooden floor in a cartoon style.", where $N \in [1, 15]$ denotes the number of objects expected to be generated.

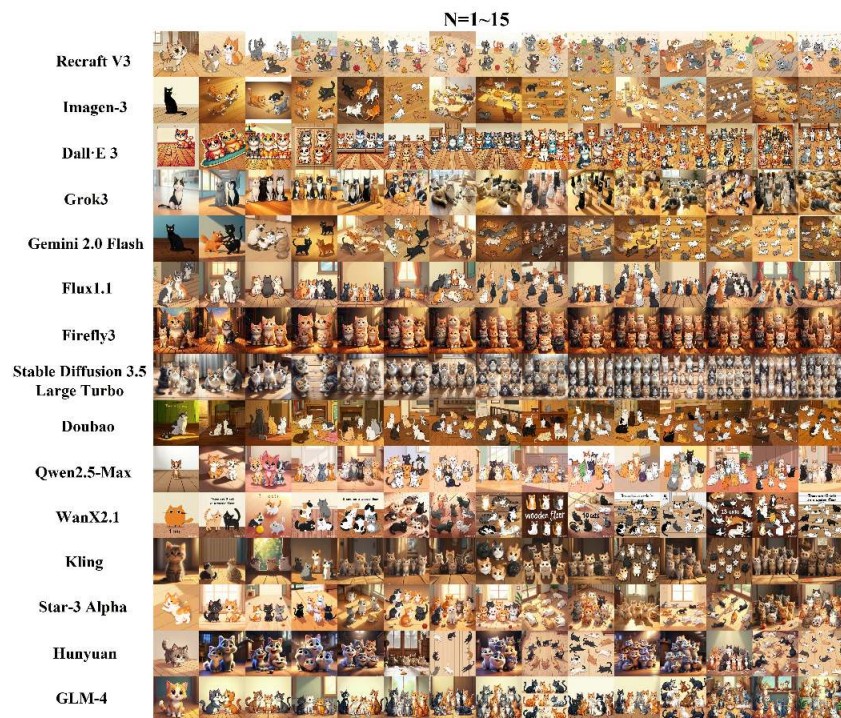

Figure 36: **Counting Cats With Cartoon Style on 15 Models**. This figure presents the generation results of counting cats. We use the Prompt:"$N$ cats on a wooden floor in a cartoon style.", where $N \in [1, 15]$ denotes the number of objects expected to be generated.

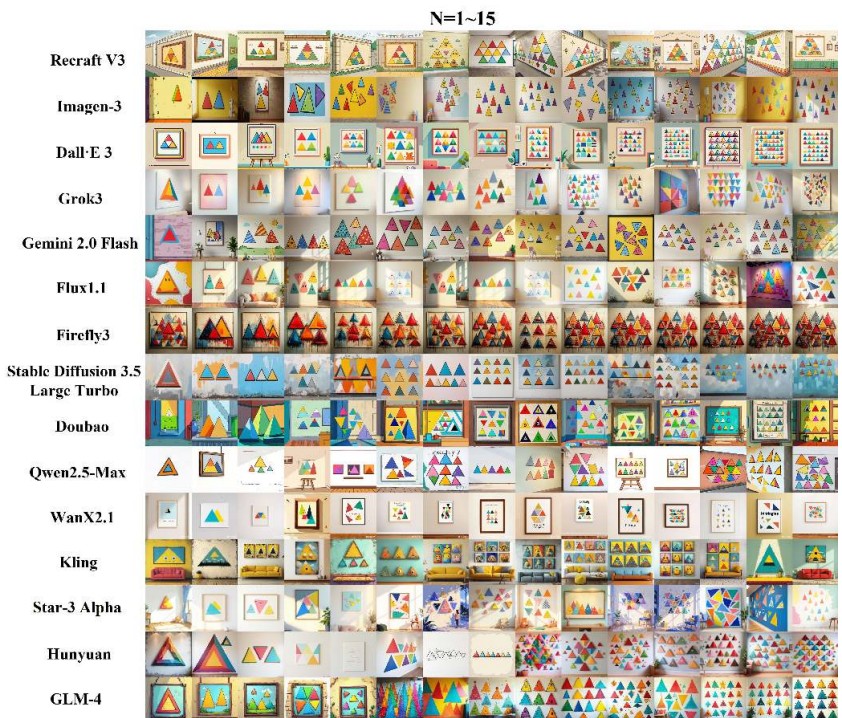

Figure 37: **Counting Triangles With Cartoon Style on 15 Models**. This figure presents the generation results of counting triangles. We use the Prompt:"$N$ triangles on a painting on a wall in a cartoon style.", where $N \in [1, 15]$ denotes the number of objects expected to be generated.

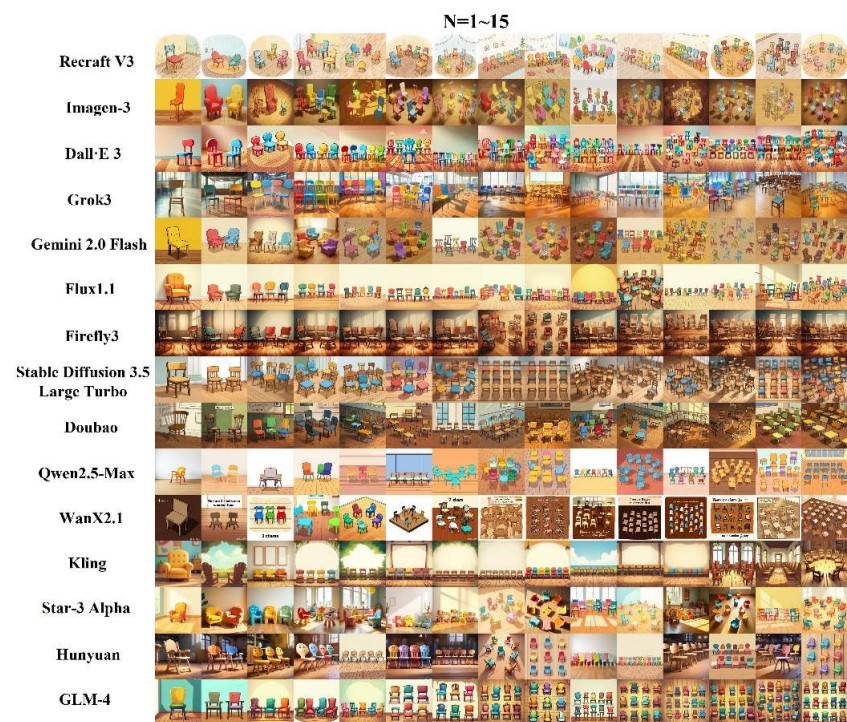

Figure 38: **Counting Chairs With Cartoon Style on 15 Models**. This figure presents the generation results of counting chairs. We use the Prompt:"$N$ chairs on a wooden floor in a cartoon style.", where $N \in [1, 15]$ denotes the number of objects expected to be generated.

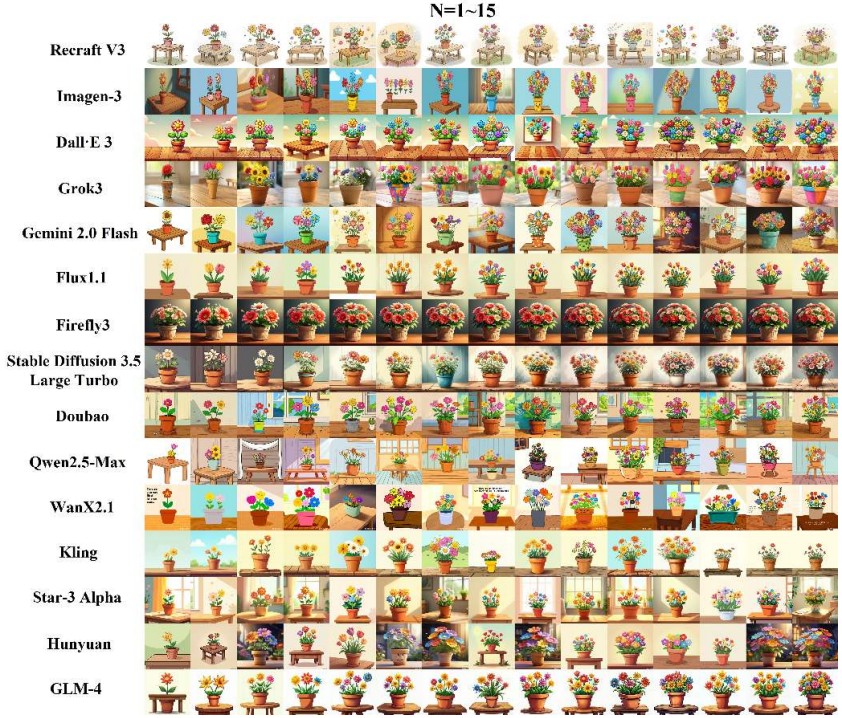

Figure 39: **Counting Flowers With Cartoon Style on 15 Models**. This figure presents the generation results of counting trees. We use the Prompt:"a flower pot with $N$ flowers on a wooden table in a cartoon style.", where $N \in [1, 15]$ denotes the number of objects expected to be generated.

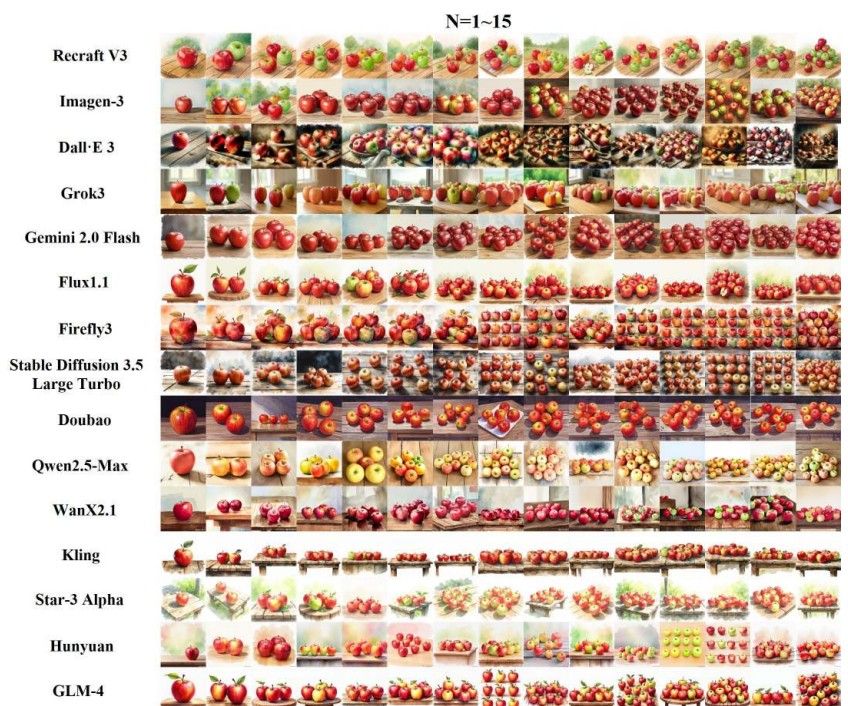

Figure 40: **Counting Apples with Watercolor style on 15 Models**. This figure presents the generation results of counting apples with watercolor style. We use the Prompt:"$N$ apples on a table in a watercolor style.", where $N \in [1, 15]$ denotes the number of objects expected to be generated.

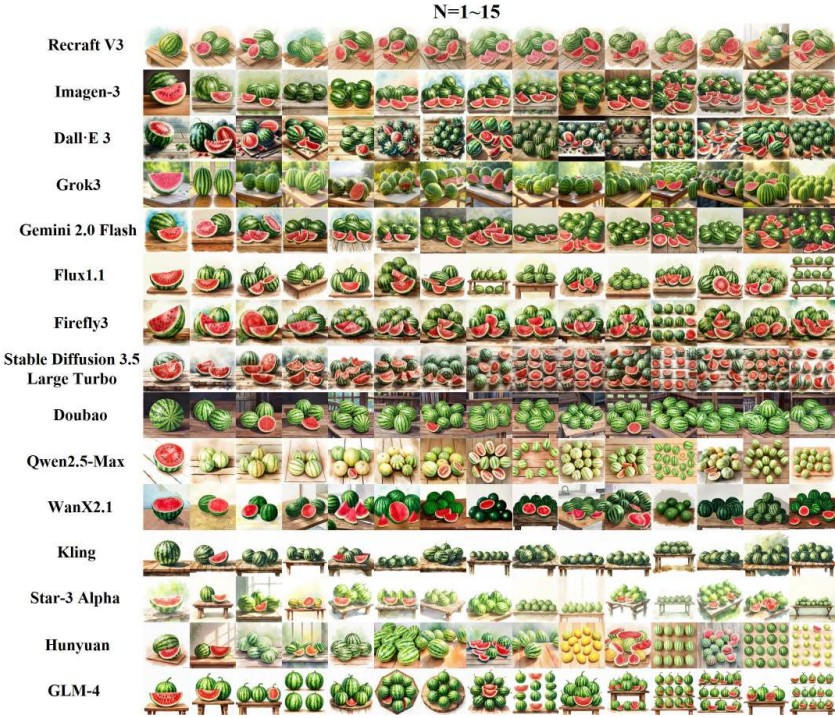

Figure 41: **Counting Watermelons with Watercolor style on 15 Models**. This figure presents the generation results of counting watermelons with watercolor style. We use the Prompt:"$N$ watermelons on a table in a watercolor style.", where $N \in [1, 15]$ denotes the number of objects expected to be generated.

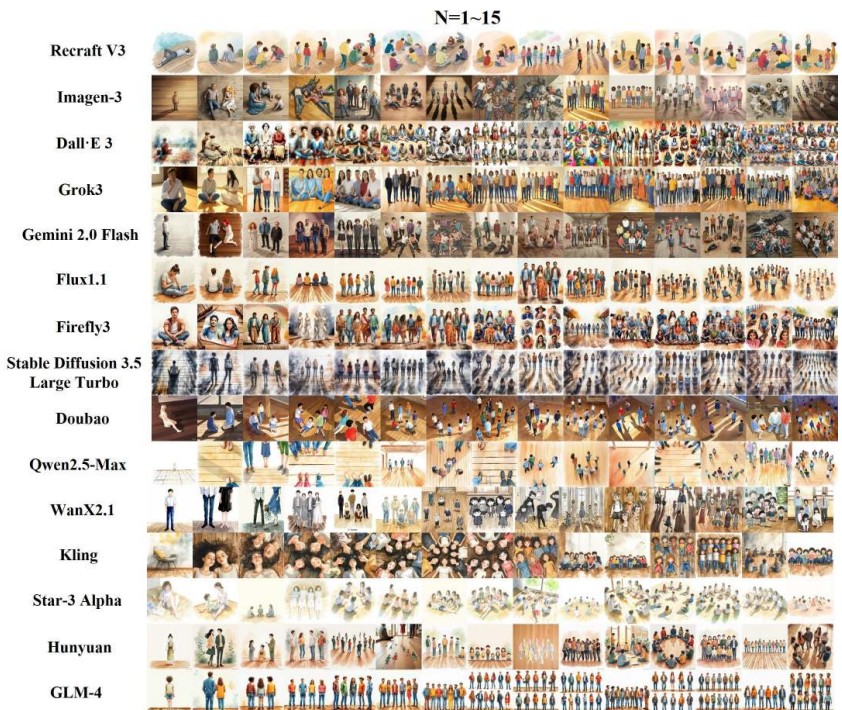

Figure 42: **Counting Humans with Watercolor style on 15 Models**. This figure presents the generation results of counting humans with watercolor style. We use the Prompt:"$N$ humans on a wooden in a watercolor style.", where $N \in [1, 15]$ denotes the number of objects expected to be generated.

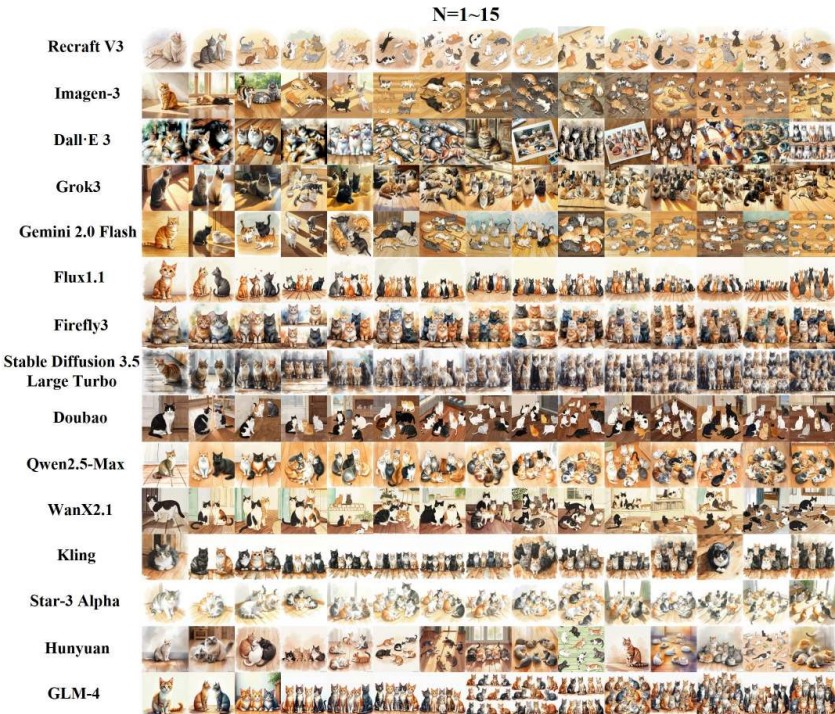

Figure 43: **Counting Cats with Watercolor style on 15 Models**. This figure presents the generation results of counting cats with watercolor style. We use the Prompt:"$N$ cats on a wooden in a watercolor style.", where $N \in [1, 15]$ denotes the number of objects expected to be generated.

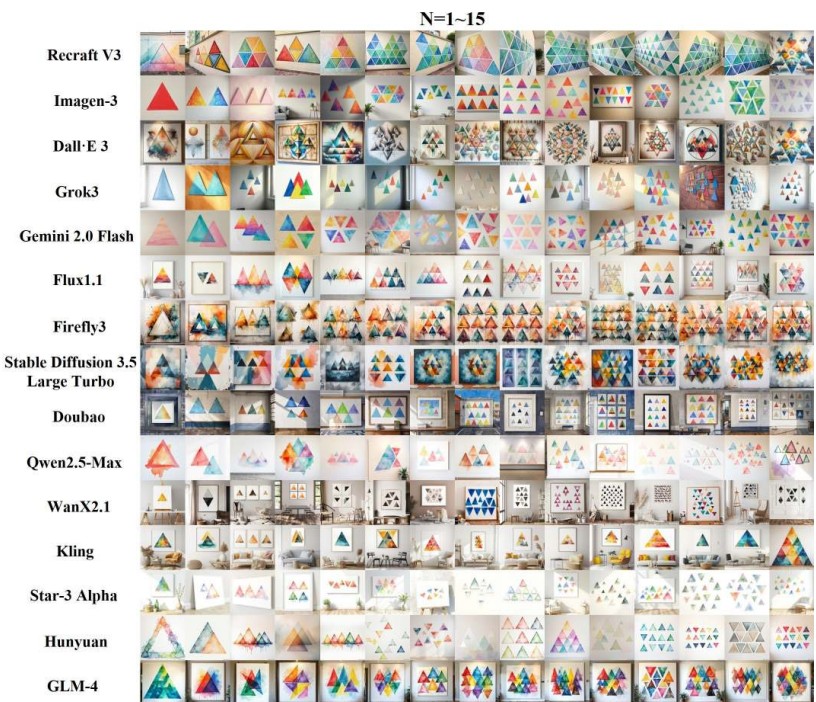

Figure 44: **Counting Triangles with Watercolor style on 15 Models**. This figure presents the generation results of counting triangles with watercolor style. We use the Prompt:"$N$ triangles on a painting on a wall in a watercolor style.", where $N \in [1, 15]$ denotes the number of objects expected to be generated.

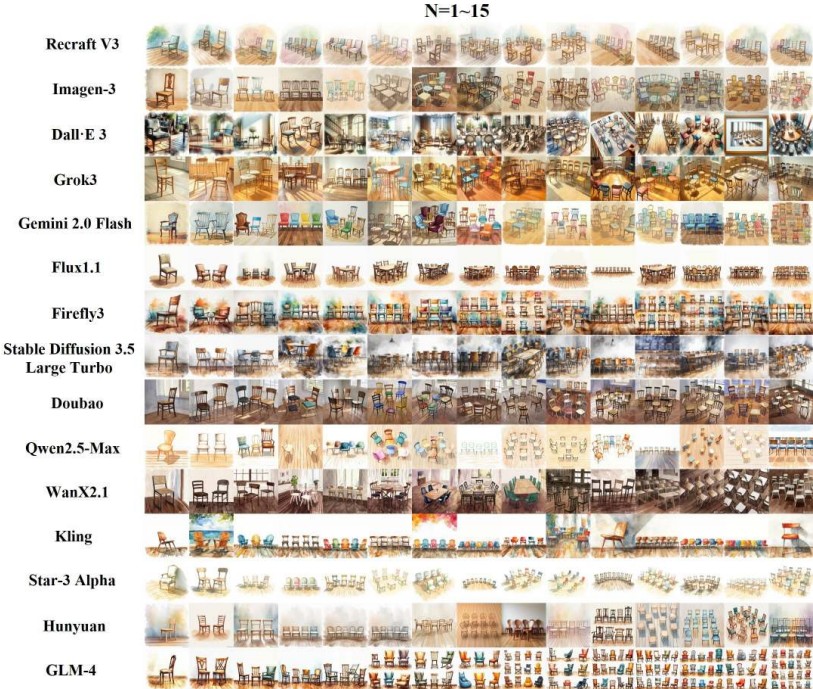

Figure 45: **Counting Chairs with Watercolor style on 15 Models**. This figure presents the generation results of counting chairs with watercolor style. We use the Prompt:"$N$ chairs on a wooden floor in a watercolor style.", where $N \in [1, 15]$ denotes the number of objects expected to be generated.

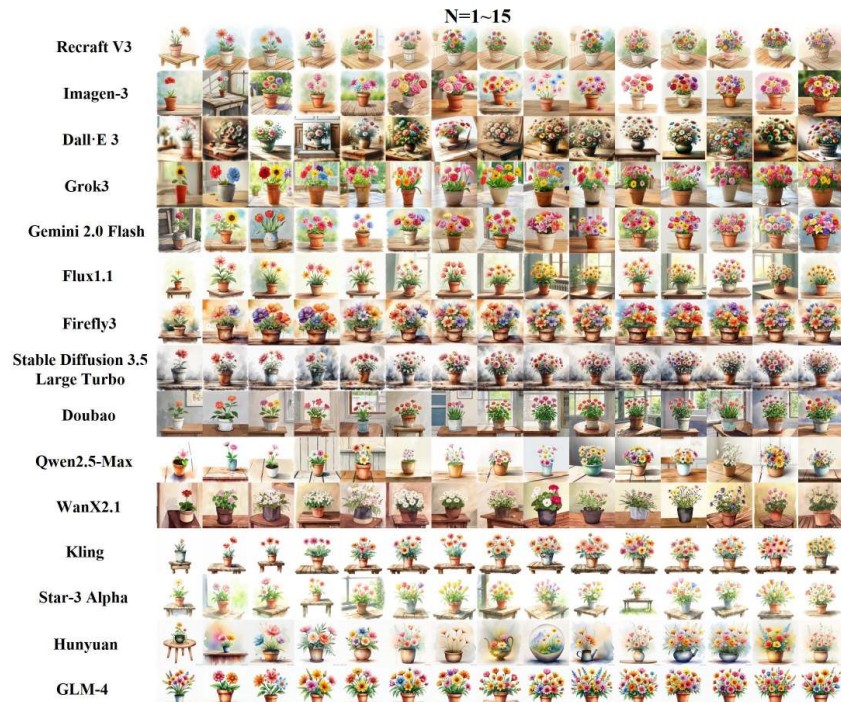

Figure 46: **Counting Flowers with Watercolor style on 15 Models**. This figure presents the generation results of counting flowers with watercolor style. We use the Prompt:" a flower pot with $N$ flowers on a wooden table in a watercolor style.", where $N \in [1, 15]$ denotes the number of objects expected to be generated.

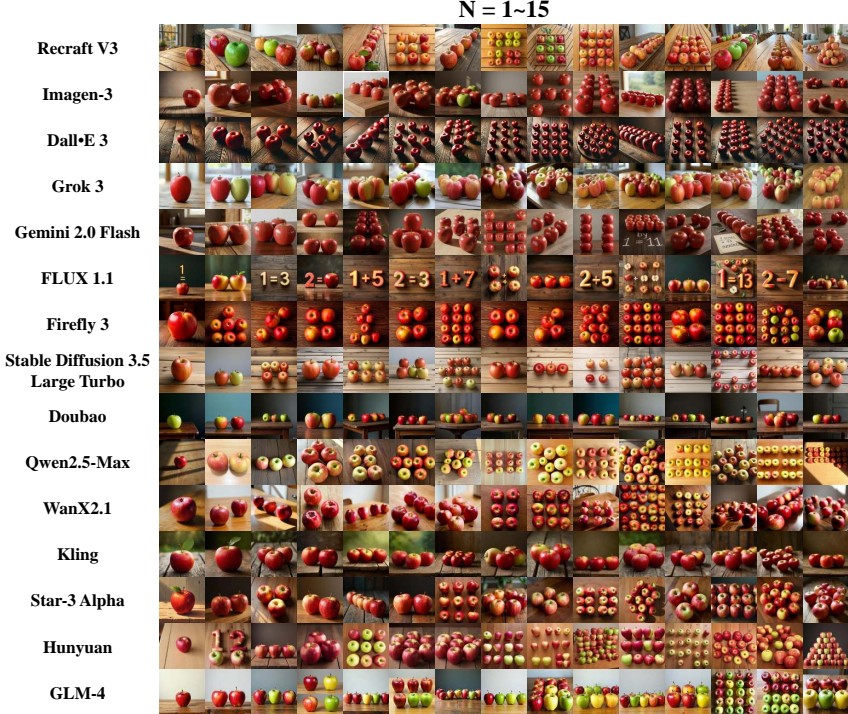

Figure 47: **Counting Apples with Multiplicative Decomposition on 15 Models**. This figure presents the generation results of counting apples with multiplicative decomposition. We use $r$ times $c$ to replace the $N$ in the Prompt:"$N$ apples on a wooden table."

**N = 1~15**

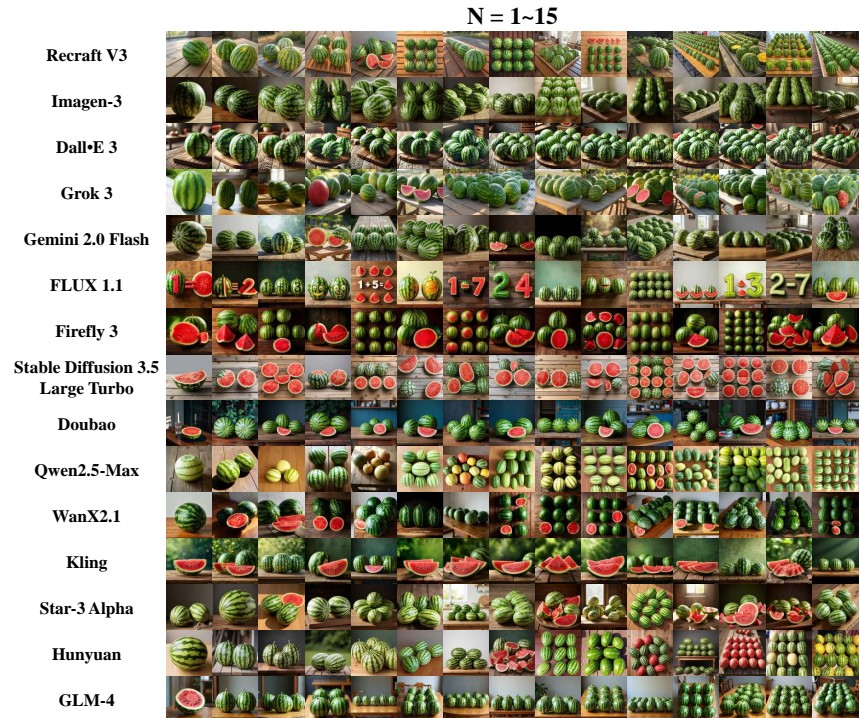

Figure 48: **Counting watermelons with Multiplicative Decomposition on 15 Models**. This figure presents the generation results of counting watermelons with multiplicative decomposition. We use $r$ times $c$ to replace the $N$ in the Prompt:"$N$ watermelons on a wooden table."

**N = 1~15**

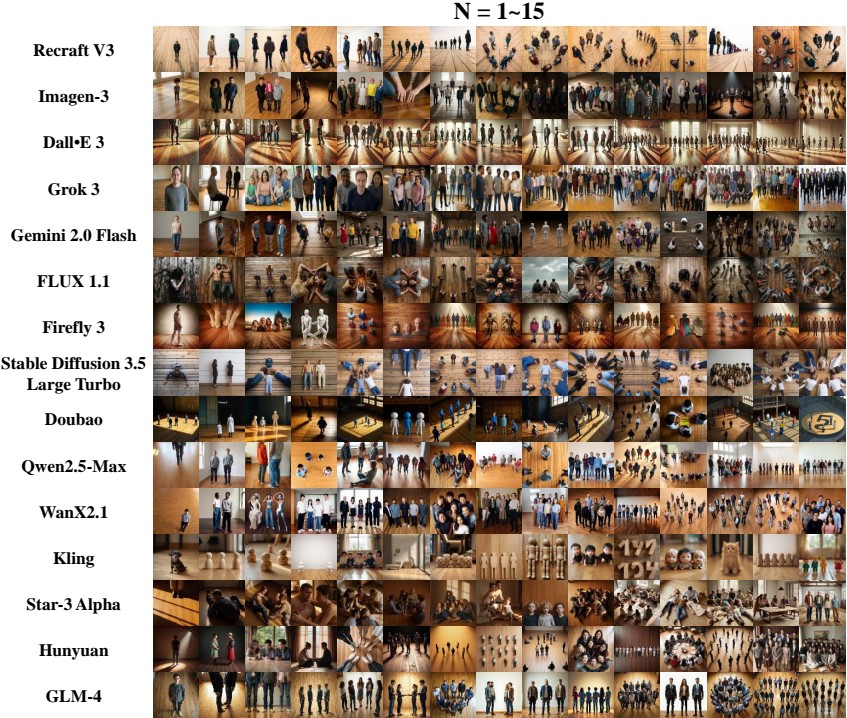

Figure 49: **Counting Trees with Multiplicative Decomposition on 15 Models**. This figure presents the generation results of counting humans with multiplicative decomposition. We use $r$ times $c$ to replace the $N$ in the Prompt:"$N$ humans on a wooden floor."

**N = 1~15**

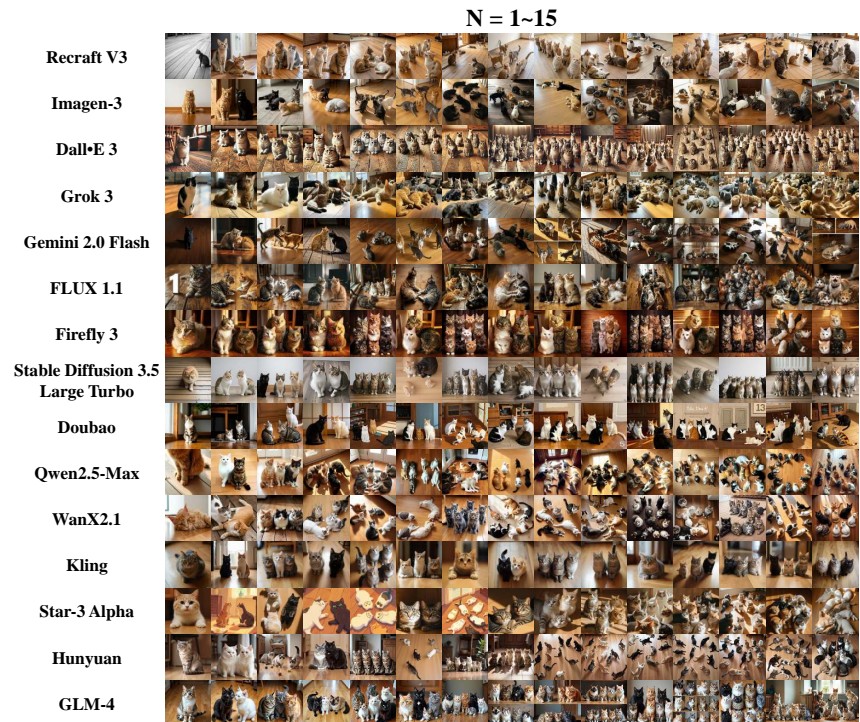

Figure 50: **Counting Cats with Multiplicative Decomposition on 15 Models**. This figure presents the generation results of counting cats with multiplicative decomposition. We use $r$ times $c$ to replace the $N$ in the Prompt:"$N$ cats on a wooden floor."

**N = 1~15**

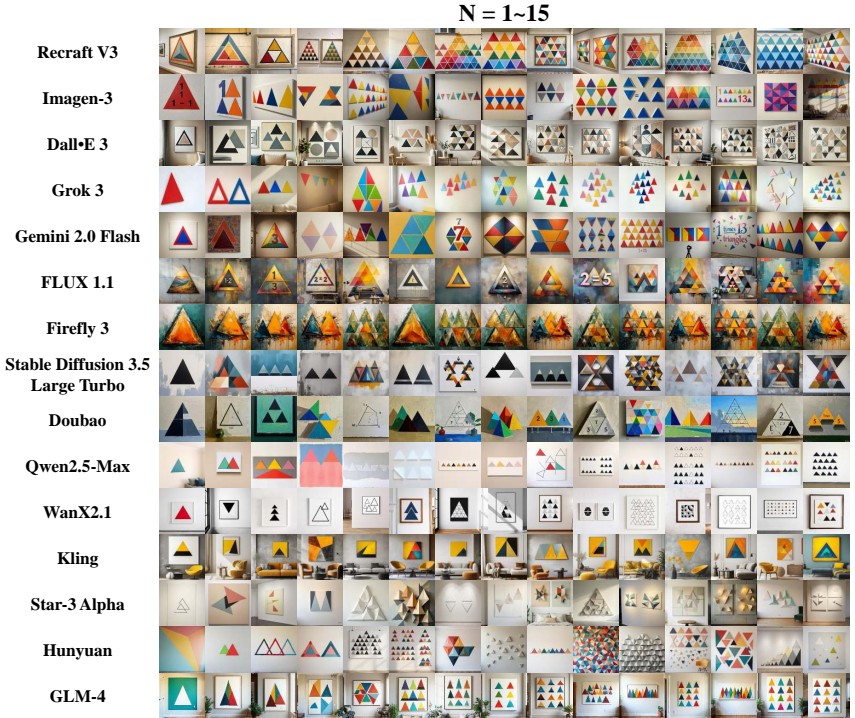

Figure 51: **Counting Triangles with Multiplicative Decomposition on 15 Models**. This figure presents the generation results of counting triangles with multiplicative decomposition. We use $r$ times $c$ to replace the $N$ in the Prompt:"$N$ triangles on a wooden floor."

**N = 1~15**

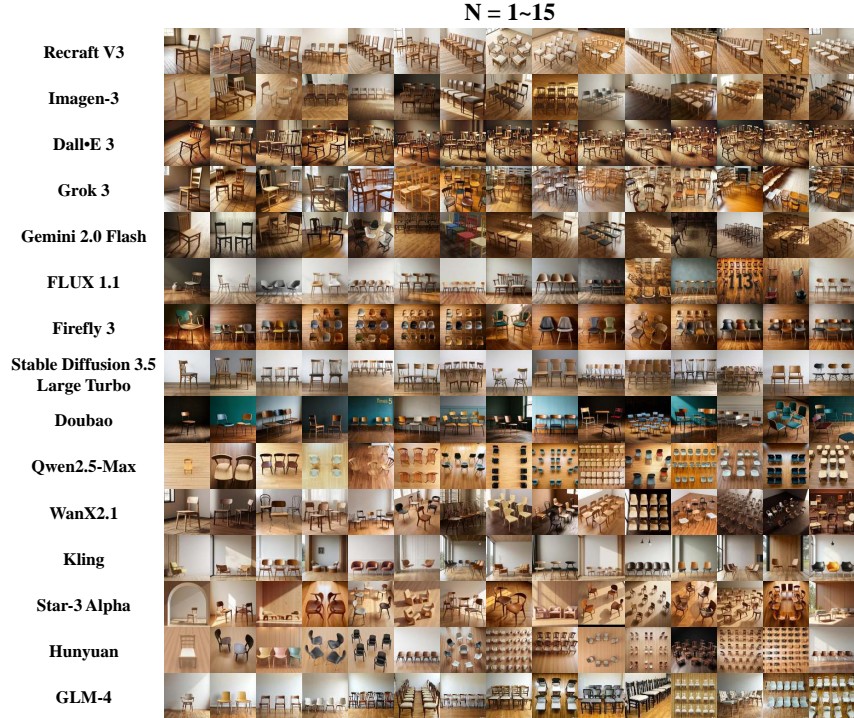

Figure 52: **Counting Chairs with Multiplicative Decomposition on 15 Models**. This figure presents the generation results of counting chairs with multiplicative decomposition. We use $r$ times $c$ to replace the $N$ in the Prompt:"$N$ chairs on a wooden floor."

**N = 1~15**

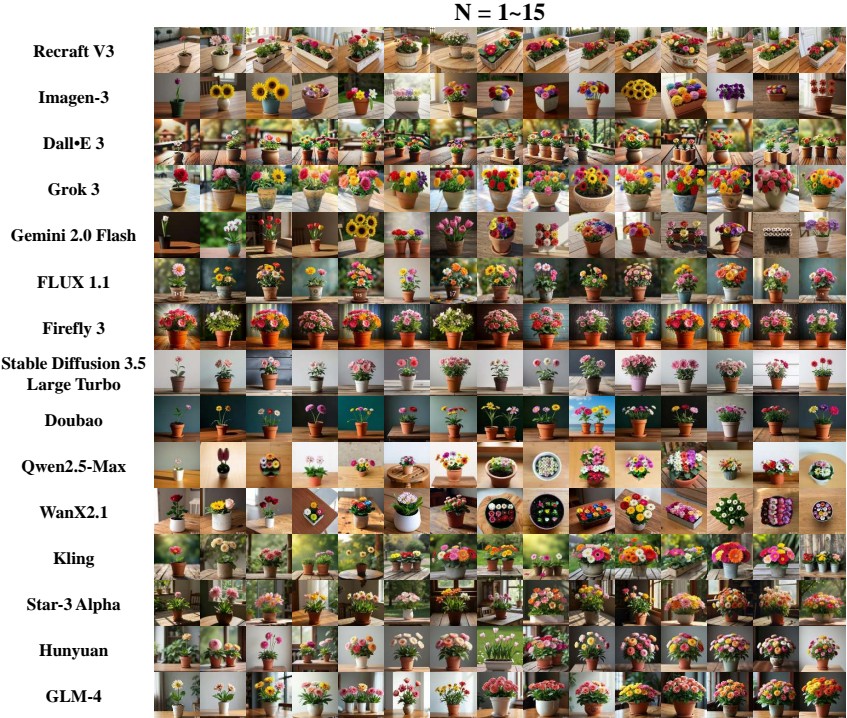

Figure 53: **Counting Flowers with Multiplicative Decomposition on 15 Models**. This figure presents the generation results of counting flowers with multiplicative decomposition. We use $r$ times $c$ to replace the $N$ in the Prompt:"A flower pot with $N$ flowers on a wooden table."

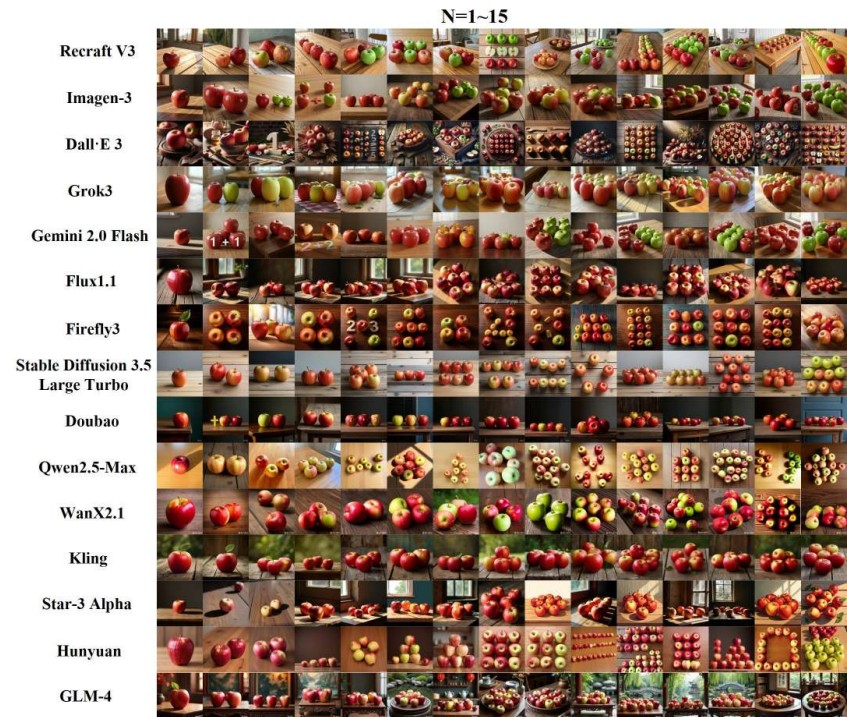

Figure 54: **Counting Apples with Additive Decomposition on 15 Models**. This figure presents the generation results of counting apples with additive decomposition. We use $\lfloor N/2 \rfloor$ plus $N - \lfloor N/2 \rfloor$ to replace the $N$ in the Prompt:"$N$ apples on a wooden table."

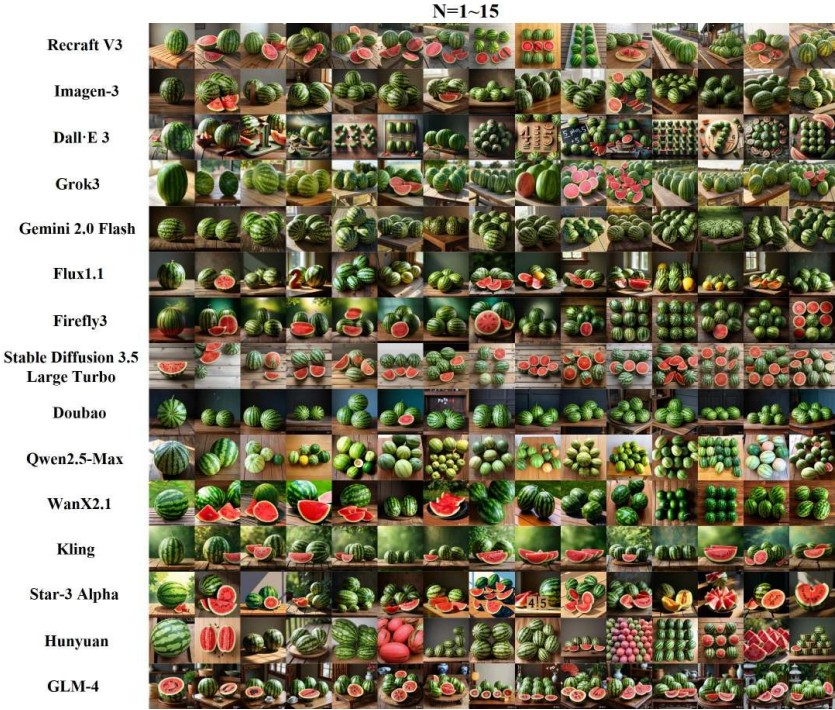

Figure 55: **Counting Watermelons with Additive Decomposition on 15 Models**. This figure presents the generation results of counting watermelons with additive decomposition. We use $\lfloor N/2 \rfloor$ plus $N - \lfloor N/2 \rfloor$ to replace the $N$ in the Prompt:"$N$ watermelons on a wooden table."

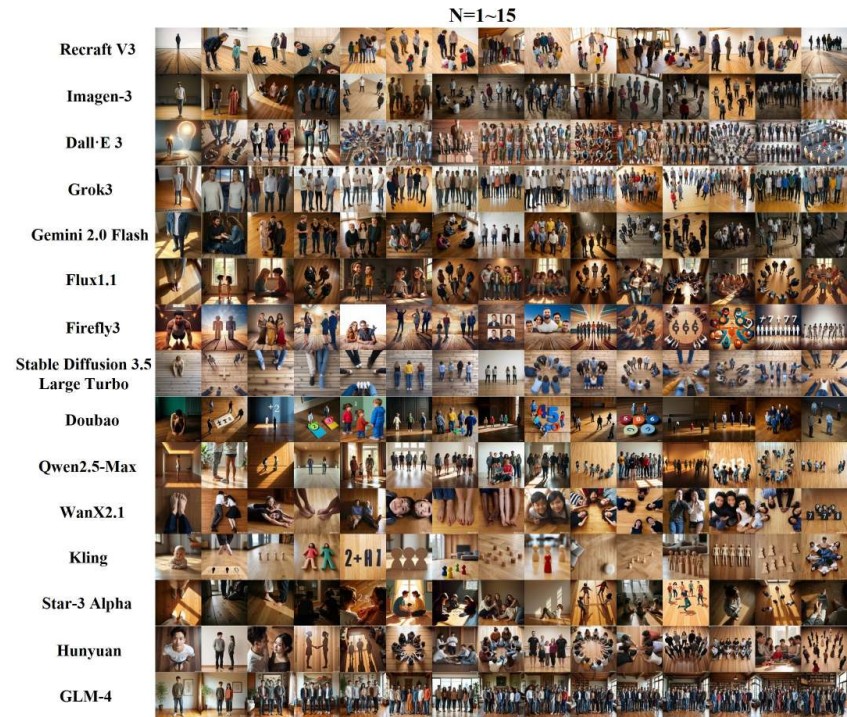

Figure 56: **Counting Humans with Additive Decomposition on 15 Models**. This figure presents the generation results of counting humans with additive decomposition. We use $\lfloor N/2 \rfloor$ plus $N - \lfloor N/2 \rfloor$ to replace the $N$ in the Prompt:"$N$ humans on a wooden floor."

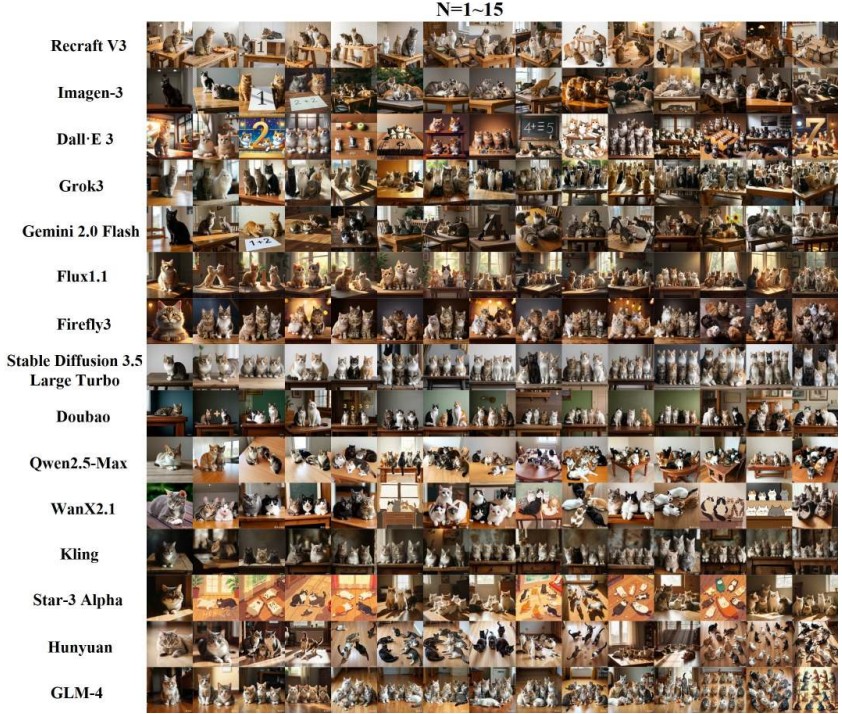

Figure 57: **Counting Cats with Additive Decomposition on 15 Models**. This figure presents the generation results of counting cats with additive decomposition. We use $\lfloor N/2 \rfloor$ plus $N - \lfloor N/2 \rfloor$ to replace the $N$ in the Prompt:"$N$ cats on a wooden table."

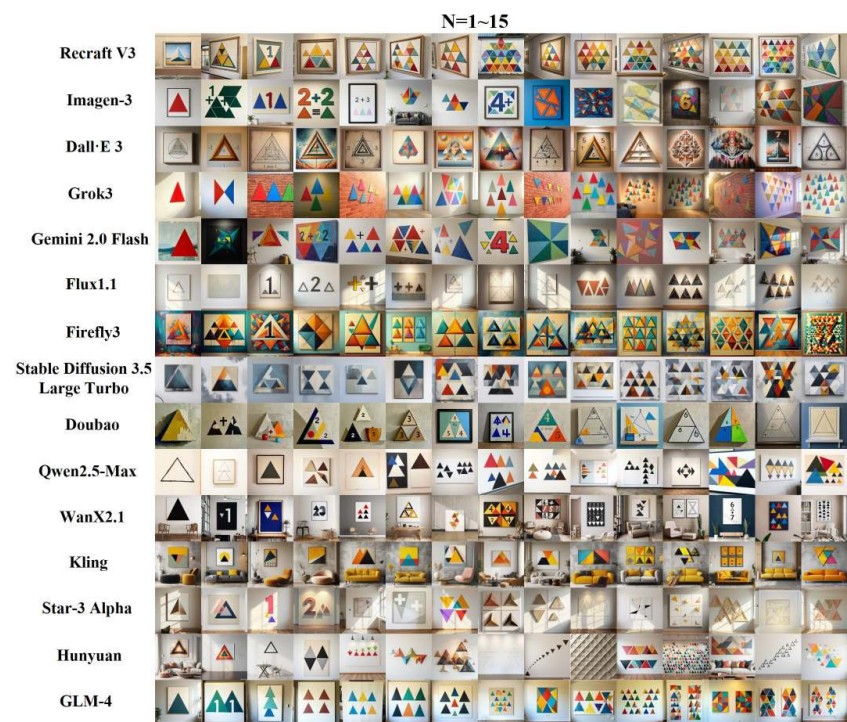

Figure 58: **Counting Triangles with Additive Decomposition on 15 Models**. This figure presents the generation results of counting triangles with additive decomposition. We use $\lfloor N/2 \rfloor$ plus $N - \lfloor N/2 \rfloor$ to replace the $N$ in the Prompt:"$N$ triangle on a painting on a wall."

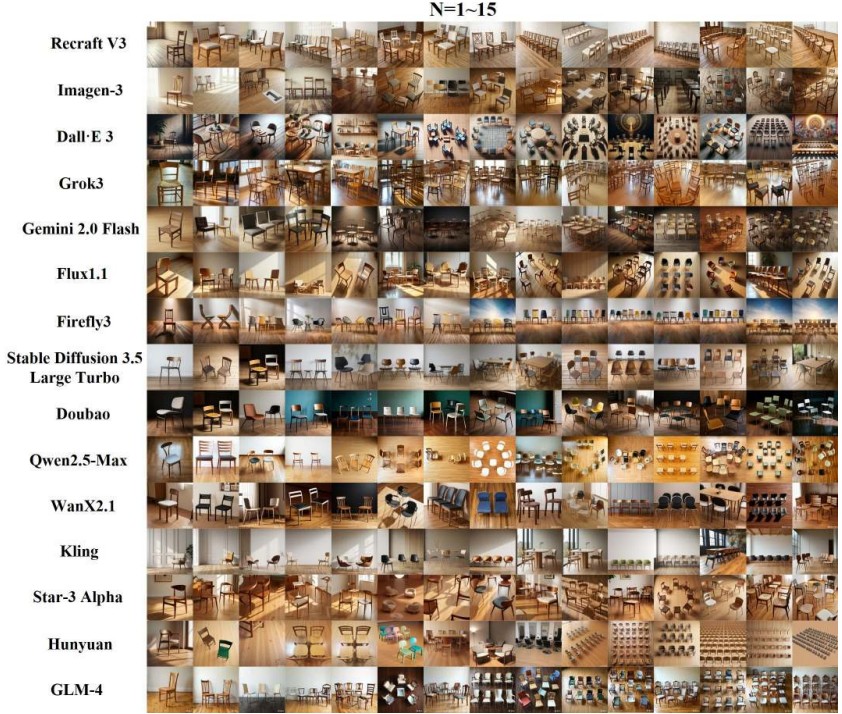

Figure 59: **Counting Chairs with Additive Decomposition on 15 Models**. This figure presents the generation results of counting chairs with additive decomposition. We use $\lfloor N/2 \rfloor$ plus $N - \lfloor N/2 \rfloor$ to replace the $N$ in the Prompt:"$N$ chairs on a wooden floor."

**N = 1~15**

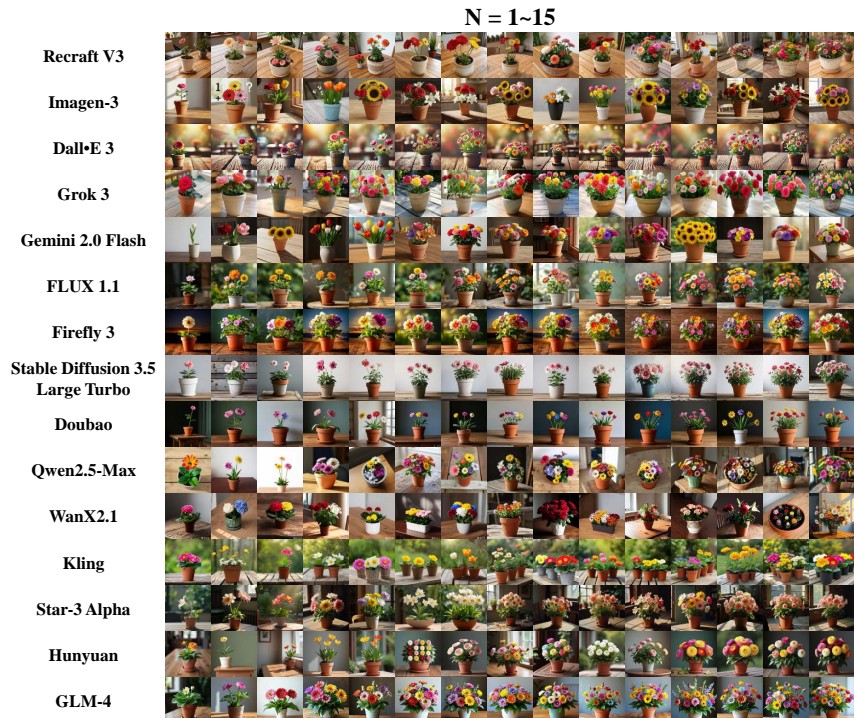

Figure 60: **Counting Flowers with Additive Decomposition on 15 Models**. This figure presents the generation results of counting flowers with additive decomposition. We use $\lfloor N/2 \rfloor$ plus $N - \lfloor N/2 \rfloor$ to replace the $N$ in the Prompt:"A flower pot with $N$ flowers on a wooden table."

**N = 1~15**

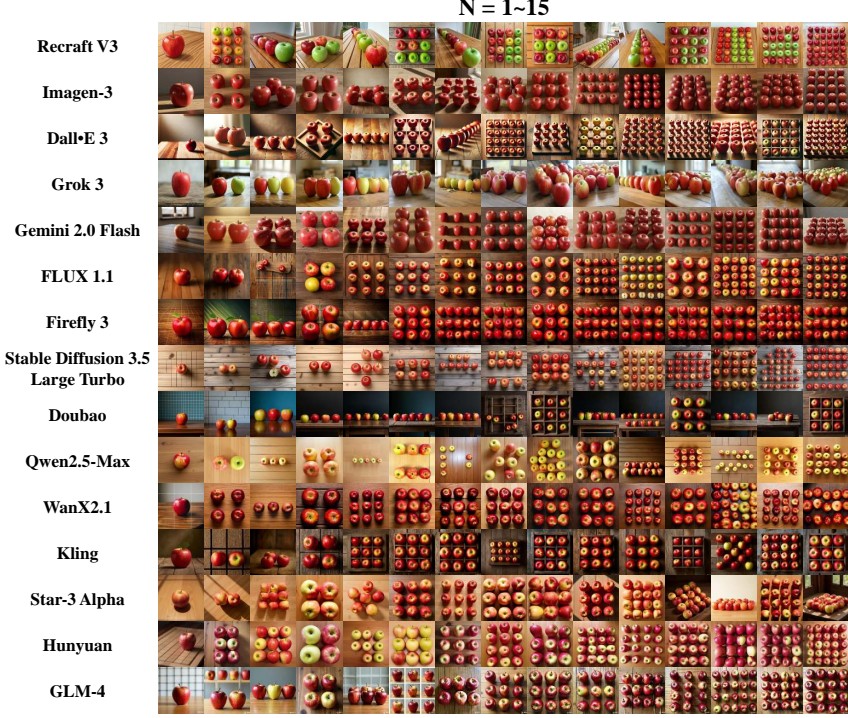

Figure 61: **Counting Apples with Grid Prior on 15 Models**. This figure presents the generation results of counting apples with grid prior. We use the prompt "$N$ apples on a wooden table, with $r$ row $c$ column grid", where $N \in [1, 15]$ denotes the number of objects expected to be generated and $N = r \times c$.

**N = 1~15**

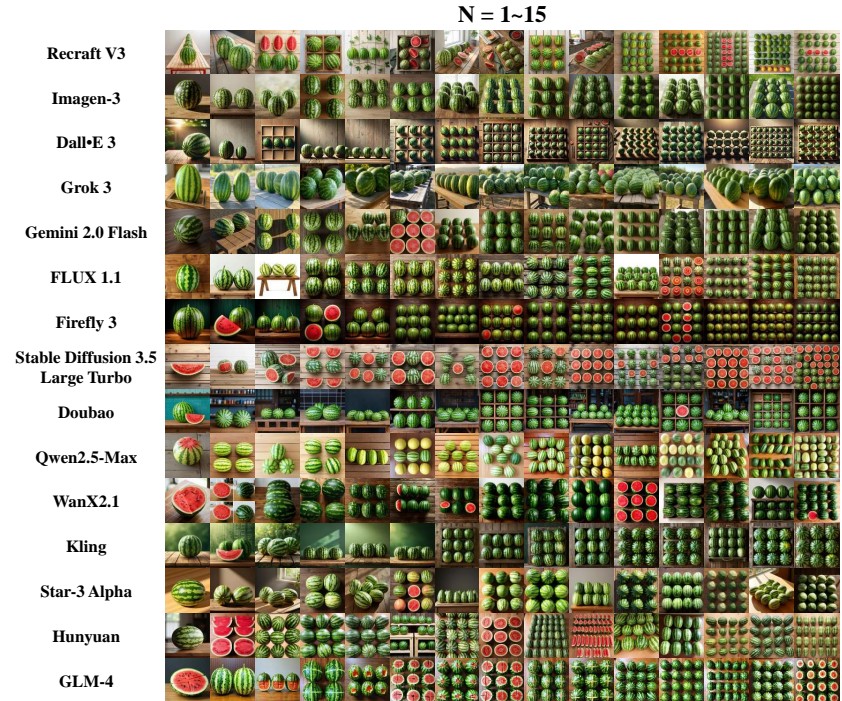

Figure 62: **Counting Watermelons with Grid Prior on 15 Models**. This figure presents the generation results of counting watermelons with grid prior. We use the prompt "$N$ watermelons on a wooden table, with $r$ row $c$ column grid", where $N \in [1, 15]$ denotes the number of objects expected to be generated and $N = r \times c$.

**N = 1~15**

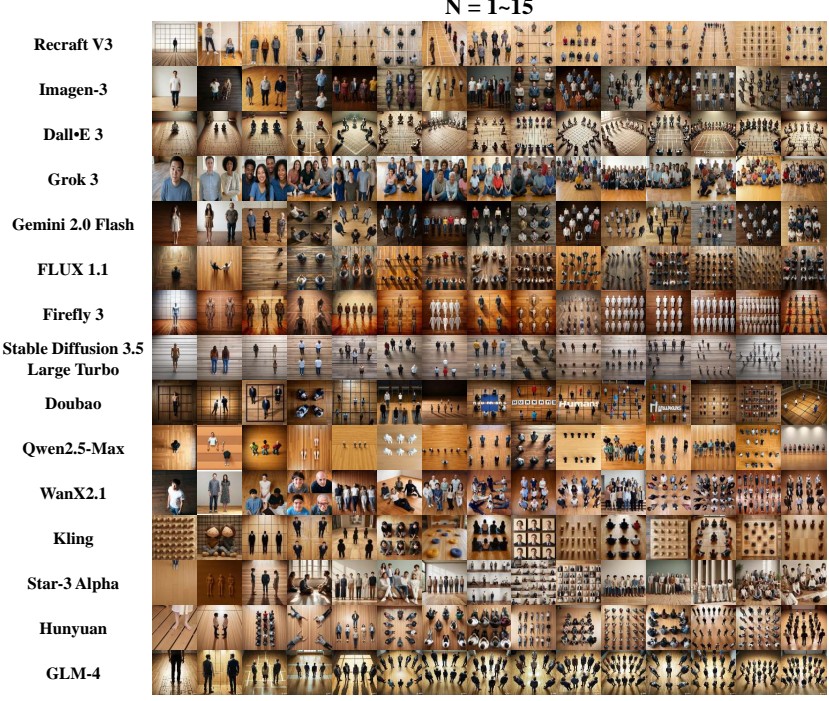

Figure 63: **Counting Humans with Grid Prior on 15 Models**. This figure presents the generation results of counting humans with grid prior. We use the prompt "$N$ humans on a wooden floor, with $r$ row $c$ column grid", where $N \in [1, 15]$ denotes the number of objects expected to be generated and $N = r \times c$.

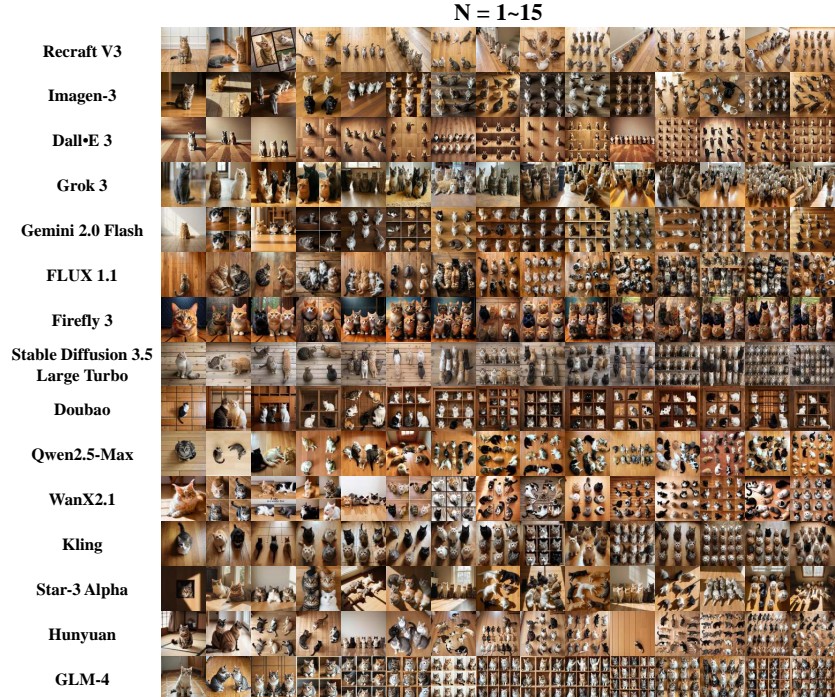

Figure 64: **Counting Cats with Grid Prior on 15 Models**. This figure presents the generation results of counting cats with grid prior. We use the prompt "$N$ cats on a wooden table, with $r$ row $c$ column grid", where $N \in [1, 15]$ denotes the number of objects expected to be generated and $N = r \times c$.

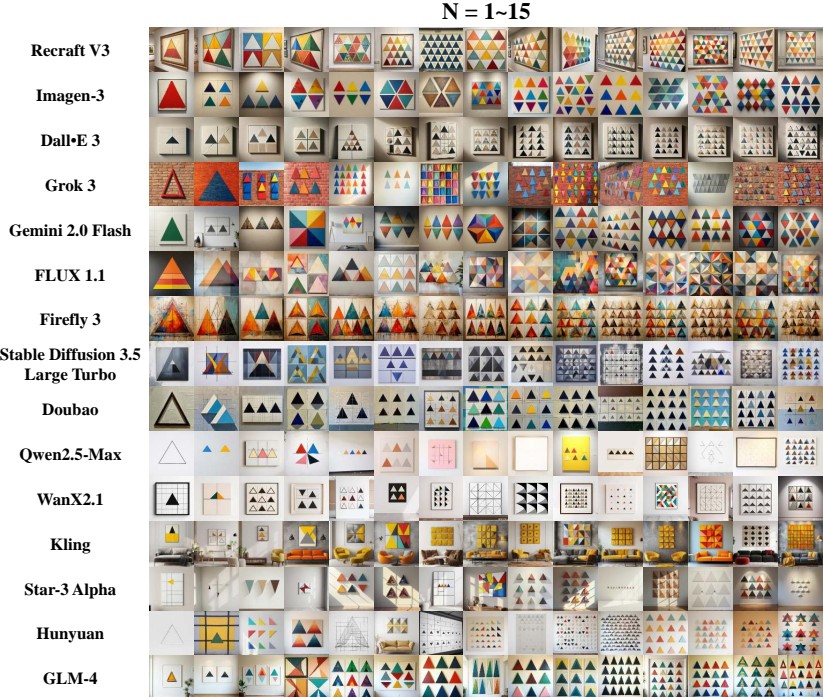

Figure 65: **Counting Triangles with Grid Prior on 15 Models**. This figure presents the generation results of counting triangles with grid prior. We use the prompt "$N$ triangles on a painting on a wall, with $r$ row $c$ column grid", where $N \in [1, 15]$ denotes the number of objects expected to be generated and $N = r \times c$.

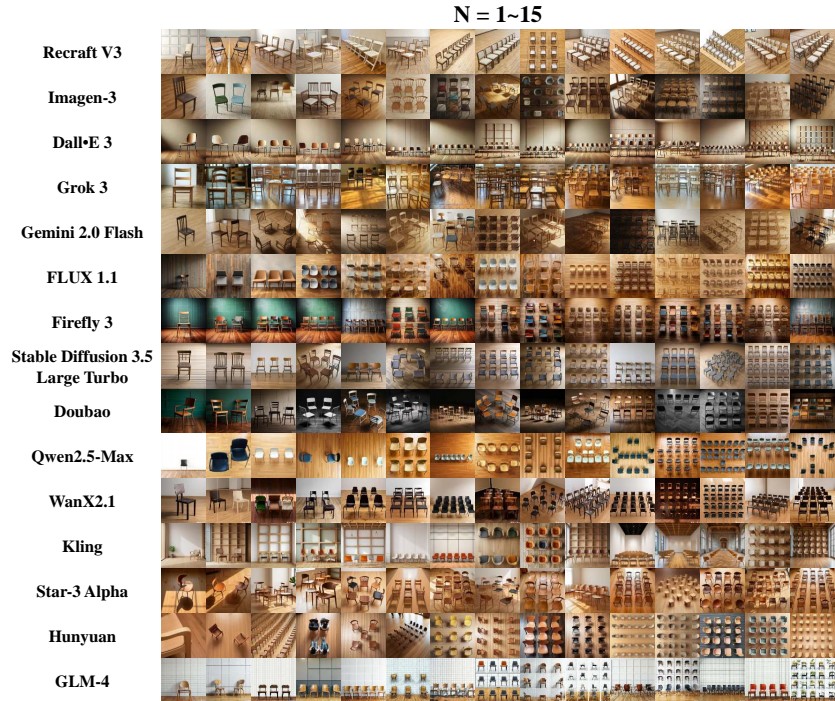

Figure 66: **Counting Chairs with Grid Prior on 15 Models**. This figure presents the generation results of counting chairs with grid prior. We use the prompt "$N$ chairs on a wooden floor, with $r$ row $c$ column grid", where $N \in [1, 15]$ denotes the number of objects expected to be generated and $N = r \times c$.

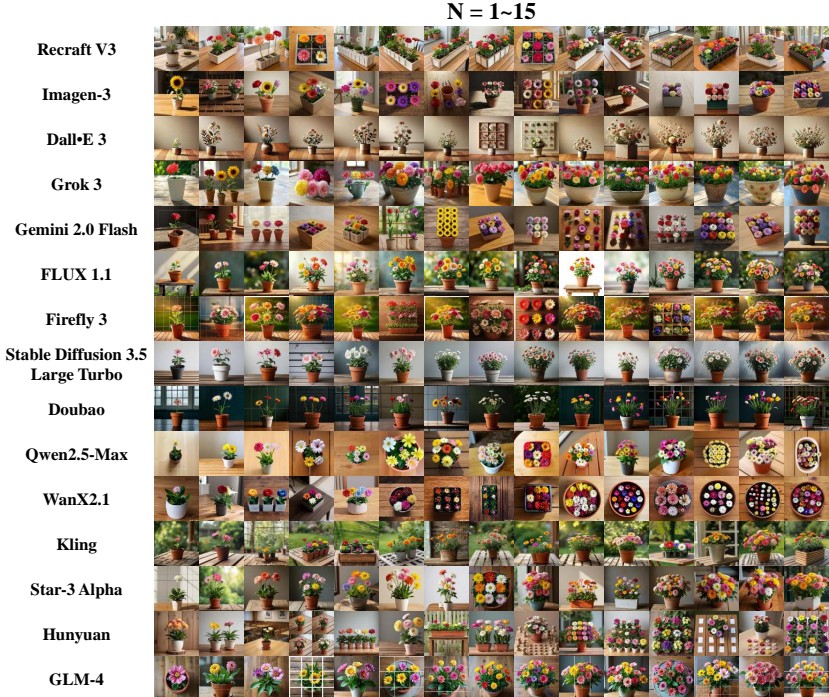

Figure 67: **Counting Flowers with Grid Prior on 15 Models**. This figure presents the generation results of counting flowers with grid prior. We use the prompt "A flower pot with $N$ flowers on a wooden table, with $r$ row $c$ column grid", where $N \in [1, 15]$ denotes the number of objects expected to be generated and $N = r \times c$.

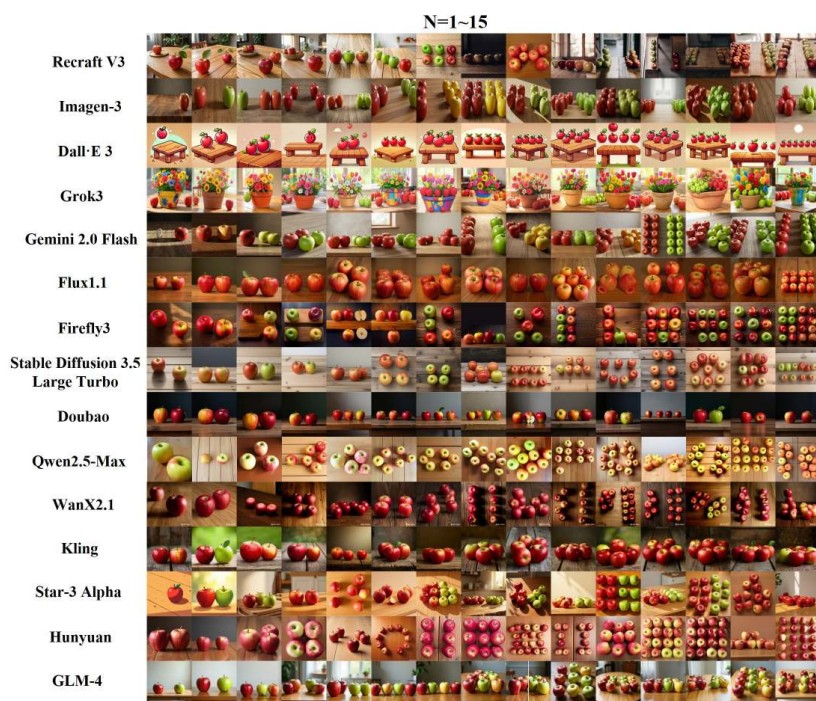

Figure 68: **Counting Apples with Position Guidance on 15 Models**. This figure presents the generation results of counting apples with position guidance. We use the prompt "$\lfloor N/2 \rfloor$ apples on the left, $N - \lfloor N/2 \rfloor$ apples on the right, on a wooden table", where $N \in [1, 15]$ denotes the number of objects expected to be generated.

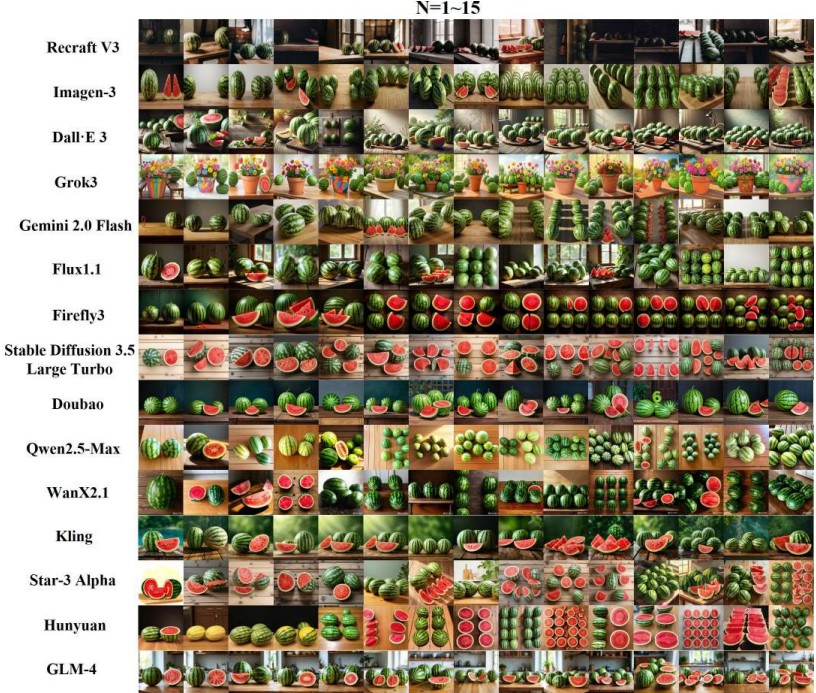

Figure 69: **Counting Watermelons with Position Guidance on 15 Models**. This figure presents the generation results of counting watermelons with position guidance. We use the prompt "$\lfloor N/2 \rfloor$ watermelons on the left, $N - \lfloor N/2 \rfloor$ watermelons on the right, on a wooden table", where $N \in [1, 15]$ denotes the number of objects expected to be generated.

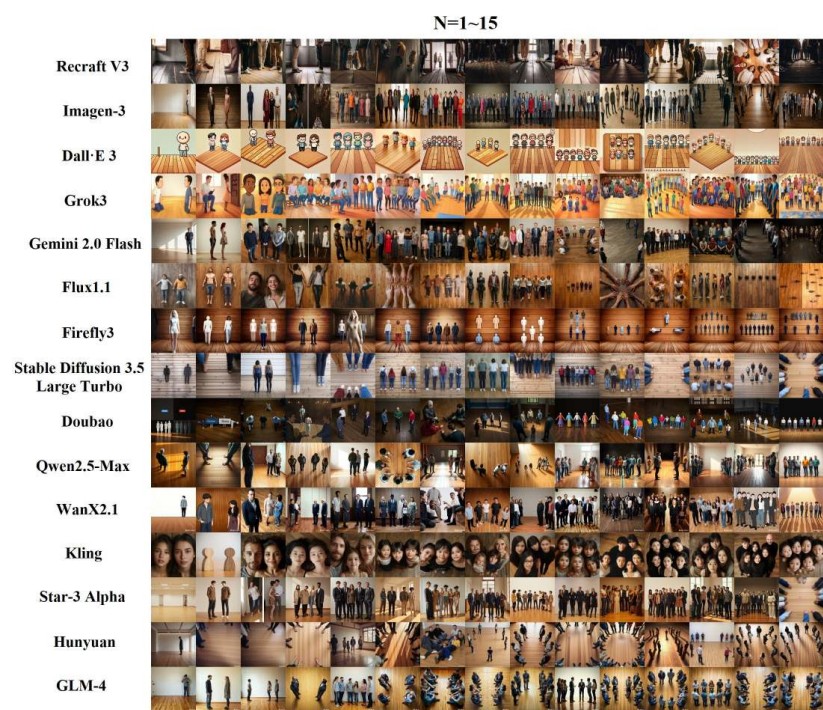

Figure 70: **Counting Humans with Position Guidance on 15 Models**. This figure presents the generation results of counting humans with position guidance. We use the prompt "$\lfloor N/2 \rfloor$ humans on the left, $N - \lfloor N/2 \rfloor$ humans on the right, on a wooden floor", where $N \in [1, 15]$ denotes the number of objects expected to be generated.

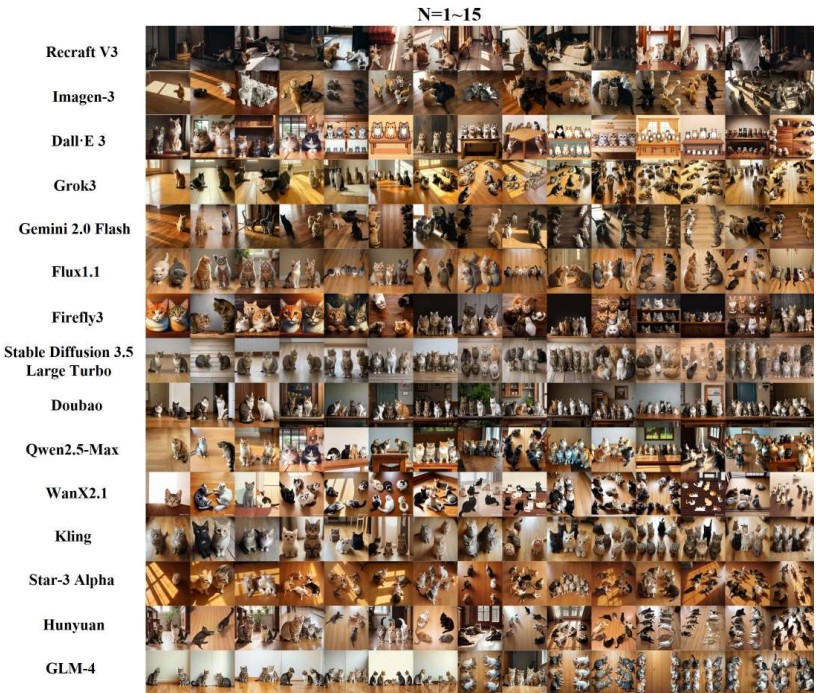

Figure 71: **Counting Cats with Position Guidance on 15 Models**. This figure presents the generation results of counting cats with position guidance. We use the prompt "$\lfloor N/2 \rfloor$ humans on the left, $N - \lfloor N/2 \rfloor$ humans on the right, on a wooden floor", where $N \in [1, 15]$ denotes the number of objects expected to be generated.

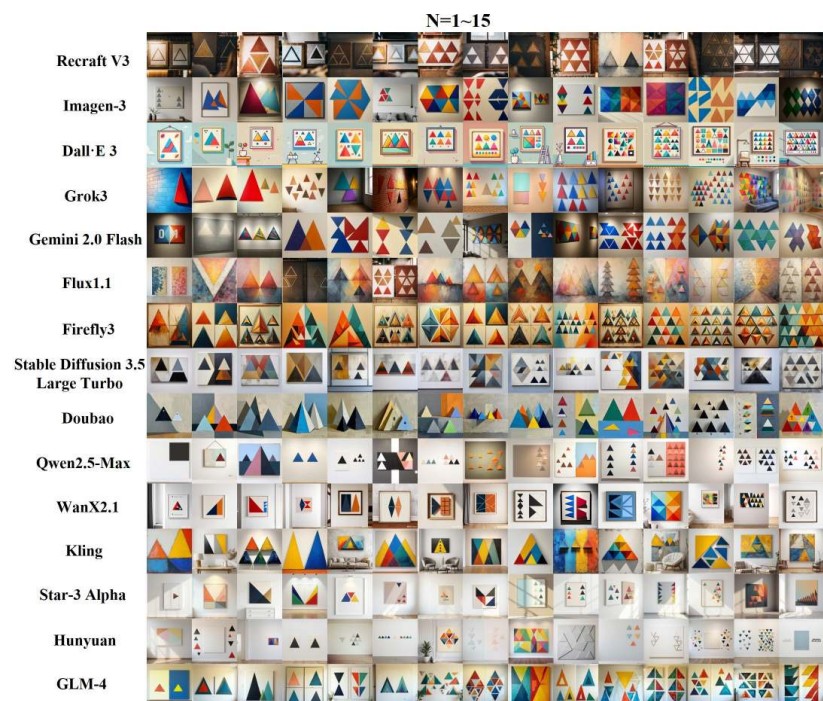

Figure 72: **Counting Triangles with Position Guidance on 15 Models**. This figure presents the generation results of counting triangles with position guidance. We use the prompt "$\lfloor N/2 \rfloor$ triangles on the left, $N - \lfloor N/2 \rfloor$ triangles on the right, on a painting on a wall", where $N \in [1, 15]$ denotes the number of objects expected to be generated.

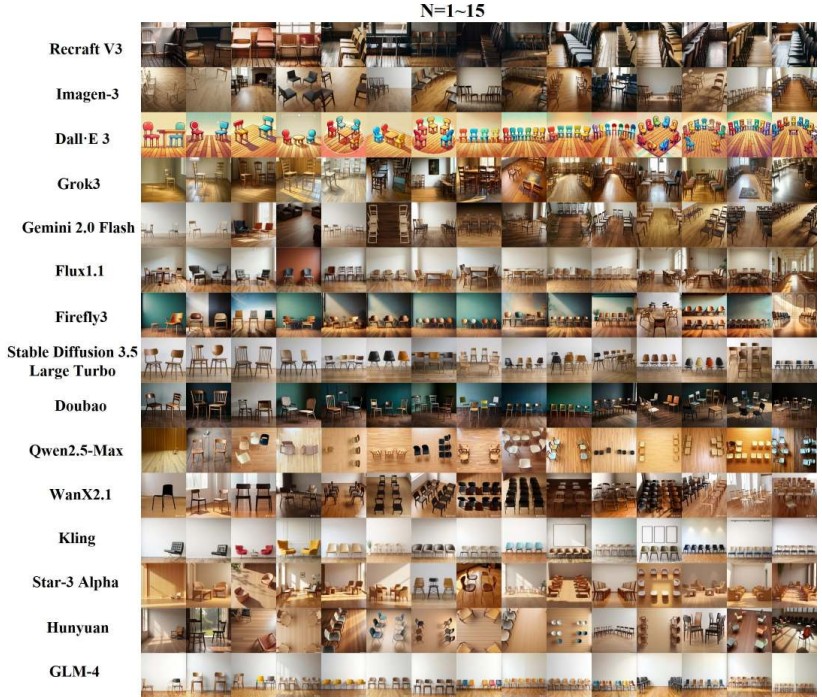

Figure 73: **Counting Chairs with Position Guidance on 15 Models**. This figure presents the generation results of counting chairs with position guidance. We use the prompt "$\lfloor N/2 \rfloor$ chairs on the left, $N - \lfloor N/2 \rfloor$ apples on the right, on a wooden floor", where $N \in [1, 15]$ denotes the number of objects expected to be generated.

N=1~15

Recraft V3

Imagen-3

Dall·E 3

Grok3

Gemini 2.0 Flash

Flux1.1

Firefly3

Stable Diffusion 3.5
Lager Turbo

Doubao

Qwen2.5-Max

WanX2.1

Kling

Star-3 Alpha

Hunyuan

GLM-4

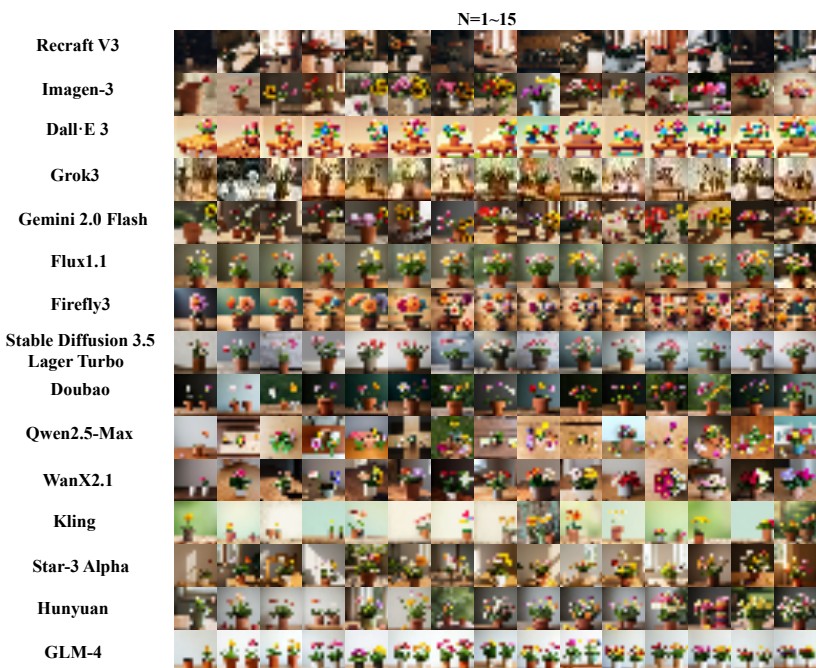

Figure 74: **Counting Flowers with Position Guidance on 15 Models**. This figure presents the generation results of counting flowers with position guidance. We use the prompt "$\lfloor N/2 \rfloor$ flowers on the left, $N - \lfloor N/2 \rfloor$ flowers on the right, on a wooden table", where $N \in [1, 15]$ denotes the number of objects expected to be generated.

## LLM USAGE DISCLOSURE

LLMs were used only to polish language, such as grammar and wording. These models did not contribute to idea creation or writing, and the authors take full responsibility for this paper's content.

