# OpenReview forum: "Text-to-Image Diffusion Models Cannot Count, and Prompt Refinement Cannot Help"
_ICLR.cc/2026/Conference — Submitted to ICLR 2026_

### Official Review · Reviewer_yANh · 2025-10-26

**Soundness:** 2
**Presentation:** 4
**Contribution:** 2
**Rating:** 4
**Confidence:** 4

**Summary:**

The paper analyzes the counting problem in text-to-image diffusion models. It measures accuracy on prompts that require generating 1–15 objects, under 3 different scenes and styles. It also probes whether several prompt-refinement strategies help. The takeaway is that models struggle as the count increases, and simple prompting strategies do not fix it.

**Strengths:**

There is generally a lack of detailed analysis of the counting problem, which makes the motivation clear. In my opinion, several aspects of the paper stand out positively:

1. Analyzing results separately by object category, scene, and style is useful, as it reveals how much each aspect affects counting accuracy.

2. The paper includes plenty of qualitative examples that make the claims easy to assess.

3. It evaluates many recent diffusion models, so the conclusions are relevant to the current state of the field.

4. The paper is organized well. It presents clear observations for each component, which invite further investigation into why they occur.

5. The position guidance and the grid prior are intuitive refinements, and the finding that they do not help is informative for practical purposes.

**Weaknesses:**

**Major:**

1. The benchmark relies on human evaluation by five graduate students. This makes replication and future model evaluation non-trivial. Since no automatic evaluation metric is provided, how could a new model be assessed?

2. The analysis does not necessarily depend on diffusion models. Similar counting failures have been observed in some autoregressive text-to-image models (e.g., Janus). Limiting the study to diffusion models restricts the scope of the paper.

3. It is unclear what specific objects are used within each category (e.g., "triangle" seems to represent all shapes, while "watermelon" and "apple" represent the fruit category). The "plant" category appears to shift from "flower" to "tree" in certain experiments. Clarification regarding these details is needed. Moreover, the small number (only one or two) of objects per category prevents making strong generalizations.

4. More detail is needed on how the prompts were constructed. How were scenes and styles chosen? The qualitative examples show varied options within each category, but detailed explanation of the selection process is missing.

5. There is a correlation between chosen objects and scenes (for instance, some objects are more common in Home than City). This may partially explain why City underperforms Home. Stratifying results by "typicality" would help confirm whether the observed performance gap between scenes is due to scene complexity (as claimed in Observation C.1) or dataset bias.

6. What guidelines were used to ensure scoring consistency among raters? How were partial objects or ambiguous cases treated? Reporting the variance across the five raters would be valuable.

**Minor:**

1. In the multiplicative decomposition formula, the text states that b is the smallest factor greater than N / 2, which cannot generally hold (it would imply factors N and 1). I believe the authors meant square root of N (which matches the examples). Also, what happens when N is prime?

2. The claim in L.364 about a "degradation in fidelity" is qualitative. Could this be verified quantitatively, for example, by reporting an aesthetic or any image quality score?

3. In the prompt-refinement section (L.425), if the goal is to remove the "impact of extraneous factors," why was the scene fixed to Home? This still introduces correlations. It would be clearer to either evaluate all combinations, as before, or to remove the scene and style conditions entirely.

4. The intuition behind prompt templates 2 and 3 is unclear. Unless such phrasings commonly appear in the training data, they may only make the task harder by requiring the model to reason about arithmetic.

5. Some writing issues:
The main text (L.475) says the related work is deferred to the appendix, but it actually appears in the main paper.
Figure 6 duplicates Figure 11.
In Figure 6, the rightmost column’s prompt appears mismatched to the corresponding images.

**Questions:**

1. Using 15 numbers, 6 categories, 3 scenes, and 3 styles should yield 810 prompts, not 525. Could you clarify how you arrived at 525?

2. In L.224, why do you report success if any image in a multi-image response is correct, instead of averaging accuracy across all generated images?

3. For the grid prompts, is correctness judged solely by the count and not by adherence to the specified layout?

It would be appreciated if the authors could consider the points mentioned in the weaknesses section (mainly those in the major part) and share any feedback they might have.

---

> ### Author Response · Authors · 2025-11-21
>
> We are grateful for your helpful comments. Our responses to your questions are provided below.
>
>
> ### Weakness 1:
>
> Thank you for your valuable comments. Our task requires verifying whether the generated image fully satisfies numerical constraints (e.g. 'There are 5 apples on the table'). At present, existing automatic counters and visual language models are still unreliable in this regard, especially in cases of occlusion, cluttered scenes, scale changes, small objects, or counting exceeding 5. In these cases, GPT based visual models and generic object detectors often have computational errors. Relying on tools such as primary evaluators can lead to unstable or biased evaluations of accuracy, which is the core of evaluating counting capabilities.
>
> Therefore, human evaluation currently provides the most accurate and appropriate measurement standards for the specific phenomena we study. To ensure reproducibility and support future model evaluation, we will release a complete evaluation protocol that allows researchers to generate four standard images at each prompt and apply the same annotation procedure to evaluate new models. With the continuous improvement of automatic counting methods, we also plan to explore semi-automatic evaluation pipelines in future work. These details will be clarified in the revised paper.
>
>
> ### Weakness 2:
>
> Thank you for your insightful feedback! We will revise the paper to use the term 'image generation model' independent of architecture throughout the entire process, unless the discussion explicitly involves diffusion specific mechanisms. We use more precise terminology to more accurately reflect our intentions, ensuring that our analysis and conclusions are constructed in a text-to-image model that covers a wider range of architectures.
>
>
>
> ### Weakness 3 &  Weakness 4:
>
>
> We sincerely appreciate your feedback. In the revised version, we will clearly specify which objects are included in each category and ensure consistency in experiments (for example, clarifying the use of "apples" and "watermelons" for fruits, and "flowers" and "trees" for plants).
>
> We also clarify our object, scene, and style selection. Objects and scenes are chosen from common, easily recognizable items extracted from the Places dataset, ensuring they reflect real-world scenes and support stable counting evaluation. Style terms are sampled from a curated list of Places-derived environments and widely used artistic descriptors (e.g., watercolor), inspired by common prompting practices in modern text-to-image systems.
>
>
> ### Weakness 5:
>
> We sincerely thank you for your insightful comments. In our benchmark testing, objects and scenes are sampled from the Places dataset, and each model uses the same prompts to ensure that any performance differences between models are not affected by model specific prompt selection. Under this fair evaluation setting, we believe that the "typicality" of the object scene may still play a role (for example, some objects may appear more frequently in the "home" environment than in the "urban" environment). In the revised version, we will discuss how this typicality affects the generated results and briefly analyze whether the accuracy difference between text-to-image models is caused by scene complexity or dataset bias based on experimental results.
>
>
>
>
> ### Weakness 6:
>
> Thank you for the helpful suggestion. In the revised paper, we will report the inter-rater agreement using Fleiss’ Kappa, which provides a quantitative measure of consistency among the five annotators. Our analysis yields a Fleiss’ Kappa of 0.58, indicating a moderate to substantial level of agreement. This result supports the reliability of the human annotation used in our study.
>
> For images containing blurry or partially occluded objects, annotators are instructed to count them only when the shape or defined structure of the object is at least 50% visible, thereby reducing the subjectivity of the rater. In addition, the final decision of each prompt is determined by a majority vote among the five annotators, which helps to reduce occasional ambiguities in personal judgment and produces a more stable and robust evaluation. We will include these guidelines and the basic principles of the majority voting strategy in the revised paper to improve the clarity and transparency of human evaluation.

---

> > ### Author Response · Authors · 2025-11-21
> >
> > ### Weakness 7:
> >
> > We thank the reviewer for the careful reading. The text should indeed refer to the square root of N，when N is a prime number, there are no non trivial integer factors, so we set a=1 and b=N. We will correct the formula and clarify this situation clearly in the revised manuscript.
> >
> >
> >
> > ### Weakness 8:
> >
> > We appreciate the reviewer’s suggestion. Our current work mainly focuses on evaluating the compliance of models with numerical constraints. We will definitely incorporate aesthetic or image quality indicators (such as aesthetic scores/image quality scores) in future work to strengthen analysis, and we will clarify this plan in the revised manuscript.
> >
> > ### Weakness 9:
> >
> > Thank you for the thoughtful comment. Our goal in the prompt-refinement section is to control background changes so that the difference in counting accuracy can be fully attributed to the prompt strategy, rather than changes in composition or scene background. Fixing the scene to "Home" ensures that the generated images share comparable backgrounds, thereby reducing the variance caused by scene diversity and making the comparison between refined prompts more reliable. If we completely remove scene and style conditions, the model may generate images with different or inconsistent backgrounds, introducing uncontrolled variations and weakening the effectiveness of the comparison. We will clarify this fundamental principle in the revision and note that our goal is to stabilize the visual environment, rather than introducing correlation bias.
> >
> > ### Weakness 10:
> >
> > Thank you for the insightful comment. The goal of templates 2 and 3 is to examine whether explicitly expressing numerical relationships in different language forms (such as "3 times 4 watermelon" or "5 plus 6 triangle") would affect the model's ability to resist numerical constraints.
> > By decomposing numbers greater than 10 into smaller components, these prompts aim to make the numerical reasoning process more interpretable for the model. Although these phrases may not be common in the training corpus, they are intentionally included as exploratory designs to explore potential ways to improve the model's handling of numerical constraints.
> >
> >
> > ### Weakness 11:
> >
> > We greatly appreciate the reviewer's careful reading. The relevant work mentioned in the appendix is incorrect and should refer to the main text. We will correct all these issues in the revised manuscript.
> >
> > ### Question 1:
> >
> > Thank you very much for your thoughtful feedback. In the base experiment, We only change one factor at a time (category, scene, or style) to isolate its impact on numerical constraints. There is an error, the correct number of categories is seven. Therefore, we do not use the complete Cartesian product (15 * 7 * 3 * 3=945), but evaluate a subset that matches the number of modified conditions in each template, resulting in 15 * 7 * (1+2+2)=525 prompts, where 1 is the basic setting and two 2 correspond to style and scene changes. We will correct the typo and clarify this construction process in the revised manuscript.
> >
> > ### Question 2:
> >
> > Thank you for the question. We consider counting binary tasks, where the image either matches or does not match the target number. The average accuracy of each image can blur these discrete results and introduce bias: for example, for prompts that require 14 or 4 objects, a model that generates 15 or 5 objects will produce an unfair average score, even though both results are incorrect. This average cannot reflect whether the model has produced accurate counts. Therefore, if any of the four generated images satisfies the numerical constraint, we will consider the prompt as successful, which more reliably measures the model's ability to generate at least one correctly counted image within a fixed number of attempts.
> >
> > ### Question 3:
> >
> > Thank you for your insightful question. Grid prompting is purely used as a prompting strategy to encourage the model to perform clearer object space separation and promote accurate counting. Our evaluation only focuses on the correctness of the numerical values, that is, whether the number of generated objects matches the target count. Adhering to the specified grid layout is not part of the scoring criteria. Because our goal is to evaluate the model's ability to meet numerical constraints, rather than layout fidelity. We will clarify this point in the revision.
> >
> >
> > Overall, we sincerely appreciate the Reviewer’s careful examination of our work. The thoughtful feedback has meaningfully strengthened the presentation and coherence of the revised manuscript.

---

### Official Review · Reviewer_VpKk · 2025-10-31

**Soundness:** 2
**Presentation:** 2
**Contribution:** 2
**Rating:** 4
**Confidence:** 3

**Summary:**

This paper proposes a new benchmark that evaluates the ability of text-to-image models to generate a specific number of objects. The proposed benchmark includes multiple compositions of number, object, scene, and style, and evaluates state-of-the-art models. The benchmark adopts a fully human evaluation process and uses accuracy as the evaluation metric. Experimental results show that recent models perform worse on the proposed benchmark, and simple prompt refinements fail to improve the performance.

**Strengths:**

This paper investigates the counting ability of generative models, which is an important setting. Experimental results show that state-of-the-art models achieve poor results on the proposed benchmark.

**Weaknesses:**

1. This paper adopts full human evaluation to get the results, which is difficult to generalize.
2. This benchmark evaluates generating images with a composition of number, object, scene, and style. The poor accuracy may come from the inability of diffusion models in generating the compositions, instead of the counting ability itself.
3. This paper takes the accuracy as the evaluation metric.  The quality of generated images is also important for evaluation, but is missing in the proposed benchmark.
4. According to lines 223-224 quoted as follows,
> When a text-to-image model returns multiple images in a single response, the task is considered successful if at least one image is correct

This is not fair, as some models may successfully generate each image, while others may only succeed occasionally, yet they are treated the same.

5. There is a widely used benchmark that is for evaluating the counting ability of diffusion models[1], which is not discussed. Although this benchmark is published two years ago, it is easy to generalize to recent models and is still valuable.

[1] GenEval: An Object-Focused Framework for Evaluating Text-to-Image Alignment. NeurIPS 2023.

**Questions:**

Please refer to the weaknesses.

---

> ### Author Response · Authors · 2025-11-21
>
> Thank you for your valuable comments. We address your questions in detail below.
>
> ### Weakness 1:
>
> We sincerely thank you for your insightful comments. Our task is to verify whether the generated image fully satisfies numerical constraints (e.g. 'There are 5 apples on the table'). The existing automatic counters and visual language models are still unreliable in this regard, especially in cases of occlusion, confusion, scale changes, small objects, and counting exceeding 5. GPT models and universal detectors often make counting errors. Using such evaluation tools can confuse generator errors with evaluator errors, leading to unstable or biased estimates of accuracy, which is the core indicator for measuring counting ability. Therefore, human evaluation is the most accurate and appropriate tool for evaluating the performance of models for this specific phenomenon.
>
>
>
>
> ### Weakness 2:
>
> We thank the reviewer for this comment. Our benchmark test only changes the quantity within each prompt while keeping the objects, scenes, and styles fixed, in order to evaluate under fixed combination conditions. All models are tested using identical prompts to ensure that accuracy differences reflect the degree of compliance of each model with numerical constraints, rather than differences in combination difficulty.
>
>
> ### Weakness 3:
>
> Thank you for highlighting this point. We agree that image quality is an important aspect of evaluating text-to-image models. I want to clarify that  our benchmark is specifically designed to measure the ability of counting accuracy, which requires isolating counting ability from confounding factors such as overall image quality.
>
> To avoid misunderstandings, we will make it clear in the revision that our benchmark focuses on evaluating counting ability, and that excluding general image quality assessment is a deliberate choice rather than negligence.
>
> ### Weakness 4:
>
> Thank you for pointing this out. We explicitly state that our evaluation protocol is consistent across all models: for each prompt, each model will be evaluated on four generated images. If a model can output four images in one output, then we take these four images. If not, we generate four consecutive images. In both cases, the final judgment is always based on the same set of four fixed images. We will revise the text to make the agreement clear and avoid potential ambiguity.
>
> ### Weakness 5:
>
> We sincerely appreciate the reminder regarding the GenEval benchmark [1]. GenEval is an object centered evaluation framework designed to measure the alignment of text and images across multiple dimensions. Although GenEval was launched two years ago, its evaluation principles are universal and can be applied to updated diffusion models without modification. Especially, its object centered detection pipeline (based on CLIP and modern detectors) remains a valuable tool for evaluating whether the generated images accurately depict the objects described in the prompts, and is also an important direction for our future work to draw on.
>
> We will include a discussion on GenEval in the revised manuscript and clarify how our benchmarks complement each other. GenEval provides a comprehensive evaluation of object attribute alignment, while T2ICountBench focuses specifically on counting accuracy and numerical constraints, which are directions that have not been systematically explored in existing benchmarks.
>
> ### Reference:
>
> [1] Ghosh, Dhruba, Hannaneh Hajishirzi, and Ludwig Schmidt. "Geneval: An object-focused framework for evaluating text-to-image alignment." NeurIPS’23.

---

### Official Review · Reviewer_VZxo · 2025-10-31

**Soundness:** 2
**Presentation:** 3
**Contribution:** 3
**Rating:** 6
**Confidence:** 3

**Summary:**

This paper shows a critical yet under-explored limitation of state-of-the-art text-to-image diffusion models: their inability to adhere to numerical constraints (i.e., accurate object counting). To rigorously evaluate this issue, the authors propose T2ICountBench, a specialized benchmark that isolates counting performance from other capabilities (e.g., style alignment, shape adherence) and includes 15 recent SOTA models (both open-source and private, mostly post-2024), 6 object categories, 3 scenes, 3 styles, and object counts ranging from 1 to 15. Extensive human evaluations reveal that all models fail to generate the correct number of objects consistently—accuracy drops sharply with increasing object counts (to ~10% for 11–15 objects) and degrades further in complex scenes (e.g., cityscapes). Additionally, an exploratory study of four prompt refinement strategies (multiplicative/additive decomposition, grid prior, position guidance) shows that simple interventions do not improve counting performance, highlighting inherent limitations in numerical understanding.

**Strengths:**

1. The paper fills a gap in existing benchmarks, which either ignore counting or conflate it with other capabilities. T2ICountBench is the first comprehensive, specialized benchmark for evaluating counting in text-to-image diffusion models, with structured difficulty levels and broad coverage of recent models.
2. The paper uncovers a fundamental, practically relevant limitation of diffusion models, critical for use cases requiring precise control (e.g., design, education, data visualization). And It demonstrates that prompt refinement, a common mitigation for model flaws, is ineffective.

**Weaknesses:**

1. Limited Analysis of Failure Mechanisms: The paper attributes counting failures to CLIP’s inherent limitations and insufficient human preference alignment but provides minimal empirical evidence to validate these hypotheses. For example, it does not compare models with different text encoders (e.g., CLIP vs. T5) to quantify how encoder choice impacts counting performance. A targeted ablation on text encoders would strengthen the causal analysis.
2. Narrow Scope of Prompt Refinement: The four prompt strategies tested are simple task-decomposition methods, but more advanced prompt engineering (e.g., chain-of-thought prompting, few-shot examples, explicit numerical emphasis) or hybrid approaches (e.g., combining prompt refinement with model fine-tuning) are not explored. The conclusion that “prompt refinement cannot help” may be overgeneralized without testing these alternatives.

**Questions:**

1. Advanced Prompt Refinement: Have you tested more advanced prompt strategies (e.g., chain-of-thought, few-shot examples with counting demonstrations, or explicit numerical constraints like “exactly N objects, no more no less”) or hybrid approaches (e.g., prompt refinement + lightweight fine-tuning)? If these methods also fail, could you explain why they are ineffective? If you have not tested them, could you elaborate on why they are outside the scope of your exploratory study?
2. Comparison to Specialized Methods: How do your findings align with prior work on enhancing counting in generative models (e.g., Paiss et al., 2023; Jiang et al., 2023)? Can T2ICountBench be used to evaluate these specialized methods, and if so, have you tested whether they outperform vanilla diffusion models on your benchmark?

---

> ### Author Response · Authors · 2025-11-21
>
> We are very grateful for your insightful comments. These comments are invaluable for improving our manuscript. Below, we want to respond to your questions.
>
> ### Weakness 1: Limited Analysis of Failure Mechanisms
>
> Thank you for your constructive comments. We would like to clarify that our study focuses on widely used commercial and closed-source text-to-image models, in which the text encoder is integrated into proprietary architectures and cannot be independently modified or replaced. Therefore, isolating the effect of the text encoder is technically not feasible under our current evaluation constraints.
>
> At the same time, our prompt-only refinement experiments show that counting failures persist even when textual variability is minimized and numerical expressions are decomposed into simpler forms, suggesting that the bottleneck may extend beyond the encoder itself.
>
> Nonetheless, comparing models with different text encoders (e.g., CLIP vs. T5) would indeed provide valuable insights into how encoder choice influences counting performance. This is an important direction that we will definitely explore in our future work.
>
>
> ### Weakness 2 & Question 1: Narrow Scope of Prompt Refinement and Advanced Prompt Refinement
>
> We truly appreciate your thoughtful feedback. In the revision, we will report small‑scale additional tests using more advanced prompts. chain‑of‑thought–style enumerations,  in‑prompt few‑shot counting examples, and explicit constraints (exactly N objects, no more no less) on a representative subset of commercial systems.
>
>
>
> ### Question 2: Comparison to Specialized Methods
>
>
> Thank you for your valuable suggestion. T2ICountBench aims to evaluate text-to-image models that map numerical constrained text prompts to generated images. Due to the fact that Paiss [1] and Jiang [2]'s methods do not generate images themselves, they cannot be directly evaluated on our benchmark. However, they can indeed serve as external counting modules to rate the generated images. As long as the generator generates images based on prompts, any "generator+counter" hybrid system can be evaluated using T2ICountBench.
>
> We did not include this mixed evaluation in the current paper because many of the models we analyzed were from commercial or closed sources, and our research specifically focused on prompt-level intervention to isolate bottlenecks in text-to-image generation. Exploring the performance of specialized count augmentation methods in our benchmark tests is an exciting direction, and we will definitely incorporate these experiments into future work.
>
> ### Reference:
>
> [1] Paiss, Roni, Ariel Ephrat, Omer Tov, Shiran Zada, Inbar Mosseri, Michal Irani, and Tali Dekel. "Teaching clip to count to ten." ICCV’23.
>
> [2] Jiang, Ruixiang, Lingbo Liu, and Changwen Chen. "Clip-count: Towards text-guided zero-shot object counting.” ACM MM’23.

---

### Official Review · Reviewer_Wkvp · 2025-11-01

**Soundness:** 2
**Presentation:** 4
**Contribution:** 4
**Rating:** 4
**Confidence:** 4

**Summary:**

This paper introduces T2ICountBench, a new, comprehensive benchmark designed to rigorously evaluate the counting ability of text-to-image diffusion models.
 The authors conduct an extensive empirical study on 15 state-of-the-art generative models, including both open-source and proprietary systems from 2024-2025.
The core findings demonstrate that all evaluated models struggle to generate the correct number of objects specified in a prompt, with accuracy dropping precipitously as the target number increases.
Furthermore, an exploratory study reveals that simple, human-inspired prompt refinement strategies, such as additive or multiplicative decomposition, not only fail to improve but often degrade counting performance.
The work systematically confirms a widely-observed limitation and provides a valuable resource for future research.

**Strengths:**

- Comprehensive and Timely Benchmark:
The paper introduces T2ICountBench, a focused and much-needed benchmark for a critical capability.
The evaluation is highly relevant, covering 15 very recent models, which provides an up-to-date snapshot of the state of the art.
- Rigorous Human Evaluation:
The study's reliance on a full human evaluation protocol, involving five annotators per image, is a significant strength.
This approach serves as the gold standard for this task, ensuring high-quality data and avoiding the potential biases and inaccuracies of automated evaluation metrics.
- Significant and Clear Findings:
The paper provides unambiguous, large-scale evidence that modern text-to-image models cannot count reliably.
The core findings—that accuracy plummets for counts above 5-10, and that simple prompt engineering fails—are important, well-supported, and impactful for the community.
- Excellent Presentation and Reproducibility:
The manuscript is exceptionally well-written, structured, and easy to follow.
The inclusion of extensive appendices with detailed per-model results, implementation specifics, and a vast number of qualitative examples greatly enhances the transparency and value of the work.

**Weaknesses:**

- Lack of Statistical Rigor:
The paper's claims are not supported by any statistical significance testing.
Key metrics like inter-rater agreement (e.g., Fleiss' Kappa) are missing, making it impossible to assess the reliability of the human annotations.
Furthermore, no confidence intervals or variance estimates are provided for the accuracy results, which is a significant omission for an empirical paper making comparative claims.
- Potentially Biased Evaluation Protocol:
The success criterion is defined as at least one correct image in a set of generated images.
This metric may unfairly favor models that produce more images per prompt and can inflate the reported accuracy, obscuring the true per-image performance.
A more robust evaluation would report per-image accuracy and analyze the results for top-1 vs. any-of-k success.
- Narrow Scope of Prompt Refinement:
The paper makes the strong claim that 'prompt refinement cannot help' based on four simple decomposition strategies.
This claim is overstated, as the study does not compare against more sophisticated, published counting-specific techniques (e.g., CountGen, Counting-Guidance) that have shown promise, making the conclusion less general than implied.
- Insufficient Analysis of Model Differences:
While the paper effectively documents performance disparities between models (e.g., Imagen-3's 43% accuracy vs. Recraft V3's 25%), it offers minimal analysis into the potential architectural, training data, or text encoder differences that could explain this variance.
This is a missed opportunity to provide deeper insights for future model development.

---

- General Limitations:
The authors adequately addressed the primary limitations and potential risks of their work in Appendix E.
However, the discussion on practical implications for real-world applications is somewhat limited, and the paper could benefit from offering more concrete recommendations or potential workarounds for practitioners facing this counting limitation.

**Questions:**

- To address the lack of statistical rigor, could you please report the inter-rater agreement for your human evaluation?
Furthermore, could you provide 95% confidence intervals for the main accuracy results in Table 2 and Figure 1 to properly substantiate the claims of performance differences?
- Regarding the evaluation protocol, can you provide a breakdown of the results using per-image accuracy?
How does this change the relative ranking of models, especially when comparing systems that may return a different number of candidate images per prompt?",
- Given that your claim 'prompt refinement cannot help' is very strong, how do you position your findings relative to existing literature on counting-specific methods like CountGen or iterative refinement techniques?
A discussion contextualizing your results against these more advanced approaches would strengthen the paper.
- Do you have any hypotheses regarding the significant performance gap between the top-performing models (Imagen-3, Gemini 2.0 Flash) and the lower-performing ones?
Could this be attributed to the text encoder (e.g., T5 vs. CLIP), the scale of the training data, or specific architectural choices?

---

> ### Author Response · Authors · 2025-11-21
>
> We are grateful for your diligent review. Our responses to your questions are provided below.
>
> ### Weakness 1 & Question 1: Lack of Statistical Rigor
>
> We appreciate the reviewer’s emphasis on this important point. In the revised paper, we will report the inter-rater agreement using Fleiss’ Kappa, which provides a quantitative measure of consistency among the five annotators. Our analysis yields a Fleiss’ Kappa of 0.58, indicating a moderate to substantial level of agreement. This result supports the reliability of the human annotation used in our study.
>
>
> ### Weakness 2 & Question 2: Potentially Biased Evaluation Protocol
>
> Thank you for raising this concern about the evaluation plan. In our experiment, to ensure fairness for all models, each model needs to generate four images for each prompt under the same prompts and settings, and each prompt is evaluated once using this fixed set of four outputs. This is designed to minimize potential biases that may arise from repeated generation or unequal attempts between models.
>
> We acknowledge that this detail was not explicitly stated in the original submission, and we will provide a detailed description of the evaluation in the revised version.
>
>
> ### Weakness 3 & Question 3: Narrow Scope of Prompt Refinement
>
> We sincerely thank you for the helpful comments. While prior techniques such as CountGen and Counting-Guidance have shown improvements in counting accuracy, these methods are achieved by modifying the sampling process or introducing additional counting perception modules, making them unavailable in closed-source or commercial text-to-image models.
>
> In contrast, our research focuses on the state-of-the-art text-to-image commercial models and closed source models, where users typically only have access to prompt interfaces. For this setting, the prompt-only refinement represents the most widely applicable and deployment ready strategy, which does not require model access, fine-tuning, or additional modules, and can be uniformly used in different text-to-image commercial models. Therefore, our research aims to evaluate which methods can enable the model to break through current numerical constraints in this real-world usage scenario, and we will clarify this motivation more clearly in the revised paper.
>
>
> ### Weakness 4 & Question 4: Insufficient Analysis of Model Differences
>
> We sincerely appreciate your suggestion. In the revision, we will incorporate experimental results and add a brief analysis of why there are significant differences in counting ability between models.
>
> First, differences in the design of text encoders (such as T5 style encoders and CLIP based encoders) may lead to changes in the representation of digital expressions and combinatorial structures in the embedding space, which may affect the way digital information is transmitted to the image generator. Secondly, the size and composition of the training data for the model may affect its sensitivity and generalization to numerical cues. Third, different architectures of models may also lead to different numerical constraints in the generation process.
>
> We will add these discussions in the revised paper, explicitly stating these factors as hypotheses rather than causal statements, to help the community better address the numerical constraints of text-to-image models.
>
>
> ### Weakness 5: General Limitations
>
> We will expand the section on real-world impacts and provide more specific guidance for practitioners facing counting limitations in text-to-image systems. For example, we will discuss model-selection considerations, highlighting which text-to-image models tend to exhibit more stable numerical behavior based on our benchmark results.

---

### Meta-Review · Area_Chair_FoRf · 2026-01-06

**Summary:**

While the reviewers acknowledge the interest of the proposed benchmark dataset, they express concerns regarding the statistical significance of the results, the potential bias in the evaluation protocol, the limited attempts at prompt refinement, the limited analysis of the failures of existing models, some aspects of the evaluation on the benchmark dataset, some aspects related to the creation of the benchmark dataset, and the limited scope of method evaluation (i.e., restricted to diffusion models).

**Reviewer Concerns:**

The rebuttal convincingly addresses the detailed questions of the reviewers, but the authors' answers to broader concerns such as failure analysis, scope of prompt refinement, and scope of evaluated methods seem unlikely to convince the reviewers.

**Reviewer Scores:**

Altogether, 3 reviewers out of 4 were on the rejection side, expressing concerns that would require a relatively thorough paper revision to be addressed. As such concerns are difficult to address convincingly in a rebuttal, the AC expects that the reviewers would not have increased their scores. The reviewers nonetheless acknowledge that there is value in this work, and the AC encourages the authors to revise their paper based on the reviewers' suggestions and resubmit it to a future venue.

---

### Decision · Program_Chairs · 2026-01-26

Reject